# `MADLAD-400`: A Multilingual And Document-Level Large Audited Dataset

**Sneha Kudugunta**[†][*]    **Isaac Caswell**[◇]    **Biao Zhang**[†]    **Xavier Garcia**[†]
**Derrick Xin**[†]    **Aditya Kusupati**[◇]    **Romi Stella**[†]    **Ankur Bapna**[†]    **Orhan Firat**[†]
[†]Google DeepMind    [◇]Google Research

## Abstract

We introduce `MADLAD-400`, a manually audited, general domain 3T token monolingual dataset based on CommonCrawl, spanning 419 languages. We discuss the limitations revealed by self-auditing `MADLAD-400`, and the role data auditing had in the dataset creation process. We then train and release a 10.7B-parameter multilingual machine translation model on 250 billion tokens covering over 450 languages using publicly available data, and find that it is competitive with models that are significantly larger, and report the results on different domains. In addition, we train a 8B-parameter language model, and assess the results on few-shot translation. We make the baseline models [2] available to the research community.

## 1 Introduction

The availability of large multilingual corpora has accelerated the progress of multilingual natural language processing (NLP) models [64, 16, 42, 8, 46]. However, most publicly available general-domain multilingual corpora contain 100-200 languages [64, 46, 2], with some datasets containing more languages in specific domains such as religious content [4], children's books [40] or dialects [3].

A common approach to creating such datasets is to mine language specific data from general web crawls such as CommonCrawl [52, 38, 63] to create datasets. We simply take this approach and scale it. We train a document-level LangID model on 498 languages to obtain CommonCrawl annotations at a document level and obtain a 5-trillion token, document-level monolingual dataset.

However, such web-scale corpora are known to be noisy and contain undesirable content [48, 43, 18], with their multilingual partitions often having their own specific issues such as unusable text, misaligned and mislabeled/ambiguously labeled data [35]. To mitigate this, we manually audit our data. Based on our findings, we discard 79 of the languages from our preliminary dataset, rename or combine several languages and apply additional preprocessing steps. Finally, to validate the efficacy of our dataset, we train multilingual machine translation models of various sizes up to 10.7B parameters, as well as an 8B decoder-only model, and then evaluate these models on highly multilingual translation evaluation sets.

In Section 2, we describe the creation and composition of `MADLAD-400`, and discuss the results of the audit. Then, in Section A.5, we describe the parallel data we collect using publicly available sources to train the multilingual machine translation models described in Section 4.1. In Section 4, we describe the training process of the multilingual machine translation models and 8B decoder-only model, and then evaluate these models on highly multilingual translation datasets. Finally, we discuss the limitations of this work and directions for future work.

---

[*]Correspondence: `{snehakudugunta,icaswell}@google.com`

[2]https://github.com/google-research/google-research/tree/master/madlad_400

37th Conference on Neural Information Processing Systems (NeurIPS 2023) Track on Datasets and Benchmarks.

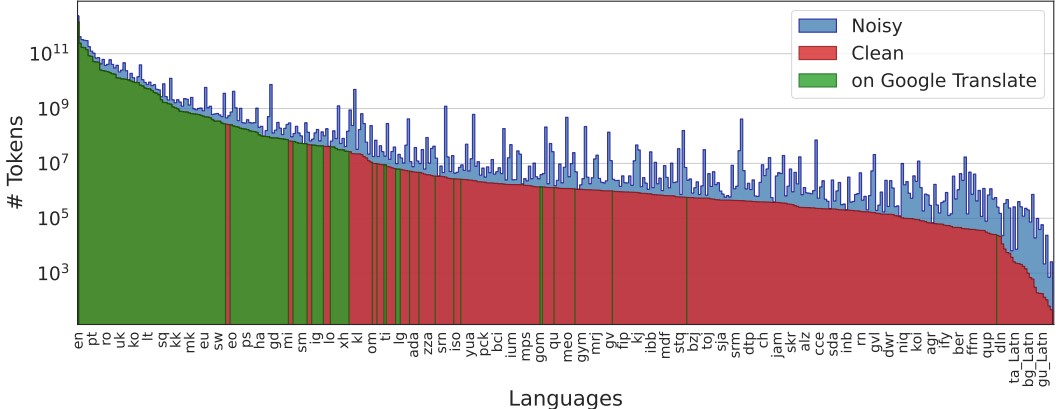

Figure 1: **Comparing the size of the noisy and clean monolingual datasets in `MADLAD-400`.** The difference is more noticeable on lower-resource languages, where noise effects are especially severe. For reference, languages supported by Google Translate are shaded in green. Note that, since this chart is in log scale, the difference in size is much greater than it may appear; for instance, for the lower-resource half of the dataset, the ratio is about $4\times$ on median.

## 2 `MADLAD-400`

The process we follow to create `MADLAD-400` is similar to that of other large-scale web corpora [12, 63, 2, 46]. First, we collect as large a dataset of unlabeled web text as possible. More specifically, we use all available snapshots of CommonCrawl[3] as of August 20, 2022. After some preliminary data cleaning, we use a highly multilingual LangID model to provide document-level annotations (Section 2.2). Finally, we conduct a self-audit (Section 2.4), or quality review, of this preliminary dataset partitioned by language, and design filters to remove noisy content. When appropriate, we correct language names and remove languages from the preliminary dataset. We note that building `MADLAD-400` was an iterative process, and that while we describe one major quality review in depth, we conducted several stages of filtering. To reflect this, we describe the preprocessing steps and improvements made in chronological order.

We release two version of this dataset: a 5 trillion token `noisy` dataset, which is the dataset obtained before applying document-level LangID and the final filters, and a 3 trillion token `clean` dataset, which has a variety of filters applied based on our self-audit, though it naturally has a fair amount of noise itself. Each dataset is released in both a document-level form and a sentence-level form. Some overall statistics for these dataset versions are given in Table 2, with a graph visualizing the distribution of sizes (number of tokens) across languages in Figure 1. The final version of `MADLAD-400` has 419 languages, with a varied geographic distribution, as seen in Table 1.

Table 1: Geographic distribution of languages in `MADLAD-400`.

| Continent | # Languages |
|---|---|
| Asia | 149 |
| Americas | 66 |
| Africa | 87 |
| Europe | 89 |
| Oceania | 26 |
| Constructed | 2 |

Table 2: Overall statistics of both the `noisy` and `clean` partitions of `MADLAD-400`.

| Dataset Version | # Documents | | # Sentences | | # Tokens | |
|---|---|---|---|---|---|---|
| | Total | Median | Total | Median | Total | Median |
| `MADLAD-400-noisy` | 7.8B | 27K | 150B | 240K | 5.0T | 7.1M |
| `MADLAD-400-clean` | 4.0B | 1.7K | 100B | 73K | 2.8T | 1.2M |

---

[3]https://commoncrawl.org/

## 2.1 Preliminary Filters

We carry out a few preliminary preprocessing steps on the web-crawled corpus: first, we deduplicate lines across documents [39]. Then, we filter out all pages that do not contain at least 3 lines of 200 or more characters (as done by Xue et al. [63]). We also use other commonly used filtering heuristics such as removing lines containing the word "Javascript" and removing pages that contain "lorem ipsum" and curly brackets "{" (as done by Raffel et al. [52]).

## 2.2 Language Identification (LangID)

We train a Semi-Supervised LangID model (SSLID) on 500 languages, following the recipe introduced by Caswell et al. [12]. We then filter the corpus on document-level LangID, which was taken to be the majority sentence-level LangID prediction. The resulting dataset is `MADLAD-400-noisy`. For the Additional details on these LangID models is in Appendix A.1.

## 2.3 Filtering Out Questionable Content

To assess the quality of this preliminary dataset, we inspected 20 sentences each from a subset of 30 languages in our dataset. Based on our observations, we introduced a score, `pct_questionable`. The `pct_questionable` score is simply the percentage of sentences in the input document that were "questionable". A sentence was considered questionable if any of the following were true:

1. **Document consistency:** Sentence-level LangID does not match the document-level LangID.
2. **List Case:** Over 50% percent of the tokens began in a capital letter (we apply this filter only if the sentence has at least 12 tokens.)
3. **Abnormal Lengths:** The sentence has under 20 characters or over 500 characters. We note that this is a bad heuristic for ideographic languages[4]).
4. **Technical Characters:** Over 20% of the characters in the sentence match `[0-9{}+/()>]`.
5. **Cursed Regexes:** The sentence matched a "cursed regex". These are a heuristic set of substrings and regexes that we found accounted for a significant amount of questionable content in the data samples we observed. They are described in depth in Appendix A.2.

We removed all documents with a `percent_questionable` score greater than 20%. Furthermore, we removed any document with under 5 sentences.

## 2.4 Self-Audit (Quality Review)

After filtering out generally lower-quality content with the approach described above, we performed a self-audit of every corpus in this dataset, following Kreutzer et al. [35]. The aim of our self-audit was to correct any remaining systematic issues by either applying additional filters, renaming/merging language codes, or completely removing the language from the dataset. Although we do not speak most of the 498 languages, we were able to give high-level comments on the general quality. For each language, we inspected a sample of 20 documents. This task was evenly divided between the first two authors based in part on which scripts they could read. We used the following guidelines:

- If dataset is mostly plausibly in-language text, we can keep it. For unknown languages, search the web for a few sentences and look at the website and URL for language clues.
- If dataset is noisy but the noise looks filterable, leave a note of how to filter it.
- If the dataset is very noisy and does not look possible to filter, mark it for removal.
- Optionally put note that may be helpful for downstream users, e.g. if dataset is 100% Bible.

We made the decision to include languages that looked noisy, but omit any language that was majority noise, or only had 20 or fewer docs. While this is not a high quality bar, we hope it still has the potential to be useful to the research community, given that foundation models have demonstrated the potential to learn distributions for very few exammples [11]. The motivation for not releasing "nonsense" or tiny datasets is to avoid giving a false sense of how multilingual the dataset is ("Representation washing"), as recommended by **Quality at a Glance** [35].

**Overall Results.** Of the 498 languages that we obtained LangID annotations for, we decided to omit 79 languages, bringing the final number of languages in `MADLAD-400` to 419. Based on

---

[4]http://www.grcdi.nl/dqglossary/ideographic%20language.html

the self-audit, we also expanded the filters (particularly the cursed regexes), and made changes as described in Sections 2.5 and 2.6. We details stats for these languages in Appendix Section A.4.

For transparency, we provide full results of the self-audit in Appendix A.4. In Table 3, we provide an overview of the issues surfaced through this self-audit. We find that a significant fraction of languages contain mostly or entirely religious documents, while other issues include misrendered text, pornographic content, and boilerplate.

Table 3: Summary of results of the audit on the preliminary dataset comprising of 498 languages. Note that there may be multiple issues with data in one language.

| # Languages... | |
| --- | --- |
| Audited | 498 |
| With significant amounts of Bible data | 141 |
| With significant amounts of JW data | 37 |
| With significant amounts of LDS data | 2 |
| With significant amounts of virama-based issues | 8 |
| With a significant number of short docs | 42 |
| With complaints about noise | 28 |
| With complaints about porn | 10 |
| With complaints about boilerplate | 15 |
| With a note to remove from the dataset | 77 |

## 2.5 Additional Filters

Based on the results of the self-audit, we apply three additional filters.

**Virama Filtering and Correction.** Many languages using Brahmic Abugida (South and Southeast Asian scripts like Devanagari, Khmer, etc.) use some variant on the virama [5] character. We found that such languages in `MADLAD-400-noisy` had incorrectly encoded viramas: for example, तुम्हारे was rendered as तुम ़ हारे, where the middle character is a detached virama. Therefore, for the languages `bn`, `my`, `pa`, `gu`, `or`, `ta`, `te`, `kn`, `ml`, `si`, `th`, `tl`, `mn`, `lo`, `bo`, `km`, `hi`, `mr`, `ne`, `gom`, `as`, `jv`, `dv`, `bho`, `dz`, `hne`, `ks_Deva`, `mag`, `mni`, `shn`, `yue`, `zh`, `ja`, `kjg`, `mnw`, `ksw`, `rki`, `mtr`, `mwr` and `xnr`, we did a special filtering/correction step — we removed all extraneous spaces before virama characters. We provide the pseudocode and list of virama characters in Appendix A.2.

**Zawgyi Encoded Data.** We found that languages using Myanmar script like `my` and `mnw` appeared to have the same issues with virama characters that still remained after applying the virama correction. This was because a large fraction of Myanmar script data on the internet is Zawgyi encoded data, which appears to have the rendering issues described above if rendered in Unicode. Therefore, we used an open-source Zawgyi detector [6] to convert the encoding of documents with more than a 50% probability of being Zawgyi encoded into standard Unicode encoding.

**Chinese-Specific Filters.** The Mandarin (`zh`) data in CommonCrawl had a particular issue with pornographic content. We combed through the data and developed a list of strings likely to be present in pornographic content, and filtered out all documents containing the strings in the blocklist. This resulted in a 17% reduction in the number of documents and a 56% reduction in file size. We list these strings in Appendix A.2.

## 2.6 Correcting Other Systematic Issues.

Based on various specific notes from the self-audit, we made a variety of changes. Five datasets were found to be in the wrong language, and were renamed or merged into the correct dataset. Six

---

[5] https://en.wikipedia.org/wiki/Virama
[6] https://github.com/google/myanmar-tools

languages that looked suspicious were run by native speakers of those or related languages, some of which were discarded, and some of which were merged into the correct dataset. Finally, we removed all languages with fewer than 20 documents. Details can be seen in Appendix A.3.

## 3 Parallel Data

To train the machine translation (MT) models described in Section 4.1, we also collect a dataset composed of publicly available datasets coming from various data sources. A full list of the data sources and associated language pairs are in Appendix A.5. The final dataset has 156 languages across 4.1B sentence pairs and 4124 language pairs total. In the rest of the paper, we refer to the input sentence to an MT model as the "source side" and the reference/output sentence as the "target side".

### 3.1 Filters

We describe the data preprocessing steps taken below. We find that a significant amount of data is filtered out, with the amount of data available 396 of 4.1k language pairs reducing by more than $40\%$.

**Deduplication.** We deduplicate sentence pairs that are an exact match on both the source and target.

**Virama Filtering and Correction/Zawgyi Encoded Data.** We observed the same issues described in Section 2.5, and used the same filters for sentence pairs where either the source language or target language belonged to the list of languages in Section 2.5.

**Unmatched Toxicity Filters.** We use the unmatched toxicity filters described by NLLBTeam et al. [46], but ultimately unusable for our purposes in most cases. For the languages ace, am, ar, az, bg, bm, bn, bs, cs, din, en, es, fa, fr, ga, gl, ha, hi, id, it, kk, ko, ml, ms, my, nl, no, nus, prs, ru, scn, sd, so, sv, tg, th, tt, ur, uz and zh, more than 3% of documents were marked as having unmatched toxicity. On closer inspection, we found that while zh and ko had a lot of pornographic content that was removed by the filtering process, most other languages removed sentences that had homonyms of non-toxic words. Similarly, languages like id, ur, tg, fa and no had data from Tanzil (Qur'an dataset), but the toxicity word lists contained words such as kafir, mercy and purity, that are not normally considered toxic content for our purpose of filtering the dataset using wordlists.

**Source-Target Filters.** We removed all sentences that have more than 75% overlap between the source and target side. To avoid filtering out valid entity translations, we only applied this filter on sentences longer than 5 tokens. In addition, we remove sentence pairs whose source length to target length ratio falls outside of $0.66 - 1.5$. We omitted this filter for the following, which are mainly non-whitespace languages: zh, ja, ko, km, my, lo, th, wuu, shn, zh_tw, zh_cn, iu, simple, dz, kr_Arab, din, nus and mi.

**Script Filters.** We removed all sentences that are less than 50% in-script for both the source and target language. For instance, if the sentence was supposed to be in kaa (Cyrillic script) but was 70% in the Latin script, we removed it.

### 3.2 Self-Audit (Quality Review)

Similar to the self-audit done for MADLAD-400, we conducted a review of the data sources that compose the parallel data we collected to verify the quality of this data. We collected 20 source-target pairs from each language, and assessed the data for the presence of offensive content, porn, and whether the data seemed to be of the correct language pair and whether the target sentence seemed to be a plausible translation. Since we did not have access to native speakers of all 157 languages, the latter was primarily based on guesses. In Appendix A.5 we provide full details of the instructions we provided to auditors, the results of the self-audit and any changes made the dataset.

### 3.3 A Note on Language Codes

As observed by Kreutzer et al. [35], the datasets used to create the parallel data (and `MADLAD-400`) use a variety of different language codes. We use the BCP-47 standard, which specifies the 2-letter ISO-693-1 code when applicable, and otherwise the ISO-693-3 code. Script tags and region tags are omitted when they are defined as the default value by CLDR [7], and otherwise included. For example, `ks` refers to Kashmiri in Nastaliq/Arabic script (CLDR default), whereas `ks_Deva` refers to Kashmiri in Devanagari. A detailed investigation of codes in `MADLAD-400` can be found in Appendix A.3.

### 3.4 Multiway Data

We create additional multiway data by applying the $n$-gram matching method ($n = 8$) from Freitag and Firat [22] to the processed dataset. Using this, and the publicly available 4.1k languages, we obtain 11.9B sentences across a total of 20742 language pairs. Full details may be found in Appendix A.7.

## 4 Experiments

We validate our data by training encoder-decoder machine translation models in Section 4.1 and decoder-only language models in Section 4.2, and test them on several translation benchmarks.

### 4.1 MT Models

We train models of various sizes: a 3B, 32-layer parameter model,[8] a 7.2B 48-layer parameter model and a 10.7B 32-layer parameter model. We share all parameters of the model across language pairs, and use a Sentence Piece Model [36] with 256k tokens shared on both the encoder and decoder side. Each input sentence has a <2xx> token prepended to the source sentence to indicate the target language [30].

We use both supervised parallel data with a machine translation objective and the monolingual `MADLAD-400` dataset with a MASS-style [57] objective to train this model. Each of these objectives is sampled with a 50% probability. Within each task, we use the recently introduced UniMax [15] sampling strategy to sample languages from our imbalanced dataset with a threshold of $N = 10$ epochs for any particular language. We list specific architecture and training details of these models in Appendix A.8.

### 4.2 Zero-shot Translation with Language Models

Given recent interest in the efficacy of unsupervised translation using large language models, we explore training language models solely on the monolingual data. We follow the same training schedule and model configurations from Garcia et al. [24]. In particular, we consider 8B decoder-only models, following the same model hyperparameters as previous work [14, 24]. We train these models using a variant of the UL2 objective [58] adapted for decoder-only models, and use the same configuration as previous work [24, 47]. We provide additional details in Appendix A.8.

### 4.3 Evaluation

We use the sacreBLEU [50] implementation of bleu[9] and chrf[10] as metrics. We evaluate our trained models on the following datasets:

**WMT.** We use the 15 WMT languages frequently used to evaluate multilingual machine translation models by Siddhant et al. [56], Kim et al. [33], Kudugunta et al. [37], NLLBTeam et al. [46]: `cs`, `de`, `es`, `fi`, `fr`, `gu`, `hi`, `kk`, `lv`, `lt`, `ro`, `rs`, `es`, `tr` and `zh`.

---

[7] https://cldr.unicode.org/

[8] Here and elsewhere, 'X-layer' means X encoder layers and also X decoder layers, for a total of 2X layers.

[9] BLEU+case.mixed+lang.<sl>-<tl>+ numrefs.1+smooth.exp+tok.<tok>+version.1.3.0, tok=zh if tl=zh and 13a otherwise.

[10] nrefs:1|case:mixed|eff:yes|nc:6|nw:0|space:no|version:2.3.1

Table 4: Evaluation scores on WMT (depicted as `<bleu>` / `<chrf>`) for the MT models and language models described in Section 4.1 and Section 4.2 compared against NLLB-54B.

| | NLLB | MT-3B | MT-7.2B | MT-10.7B | LM-8B | | | |
| | | | | | 0-shot | 1-shot | 5-shot | 10-shot |
|---|---|---|---|---|---|---|---|---|
| **xx2en** | 34.2 / 60.4 | 33.4 / 60.0 | 34.9 / 60.6 | **34.6 / 60.8** | 2.3 / 17.3 | 25.1 / 51.4 | 26.2 / 52.9 | 26.2 / 53.4 |
| **en2xx** | **31.1 / 58.0** | 28.2 / 55.4 | 29.3 / 56.2 | 29.0 / 56.2 | 1.0 / 10.3 | 18.7 / 43.5 | 18.8 / 44.5 | 19.3 / 45.5 |
| **Average** | **32.7 / 59.2** | 30.8 / 57.7 | 32.1 / 58.4 | 31.8 / 58.5 | 1.6 / 13.8 | 21.9 / 47.4 | 22.5 / 48.7 | 22.8 / 49.4 |

**Flores-200.** We evaluate on the languages in the Flores-200 dataset [46] that overlap with the languages available in either `MADLAD-400` or the parallel data described in Section A.5. We list these languages in Appendix A.9. For non-English-centric pairs, we evaluate on a 272 language pair subset of the 40k language pairs possible due to computational constraints. We evaluate on all language pairs possible using the following languages as either source or target language: `en`, `fr`, `cs`, `zh`, `et`, `mr`, `eu`, `cy`, `so`, `ckb`, `or`, `yo`, `ny`, `ti`, `ln`, `fon` and `ss`. We obtained this set of languages by selecting every $10^{th}$ language by number of tokens in MADLAD-400 (clean), starting with French (`fr`). Noticing that this had no Indian languages, we shifted `af` and `fo` (both close dialects of HRLS) down one index to `mr` and `or`, respectively. Finally, we noticed that this initial list had supervised and unsupervised languages, but didn't have a good representative of a "slightly supervised language", that is, one with a small but extant amount of parallel data. Therefore, we added `yo` to the list, which has the least parallel data of any supervised language. This resulting subset of languages also contains a nice variety of scripts: Latin, Chinese, Devanagari, Arabic, Odia, and Ethiopic scripts.

**NTREX.** We evaluate on the languages in the recently introduced NTREX dataset [20].

**Gatones.** Finally, we evaluate on the languages in GATONES, the in-house, 38-language eval set used in [8] and the GATITOS paper [31]. Again, we take the subset of languages overlapping with the languages available in either `MADLAD-400` or the parallel training data.

Table 5: Evaluation scores on Flores-200 (depicted as `<bleu>` / `<chrf>`) for the MT models and language models described in Section 4.1 and Section 4.2 compared against NLLB-54B. All metrics are computed with the sacrebleu reference implementation.

| | NLLB | MT-3B | MT-7.2B | MT-10.7B | LM-8B | | | |
| | | | | | 0-shot | 1-shot | 5-shot | 10-shot |
|---|---|---|---|---|---|---|---|---|
| **xx2en** | **35.5 / 59.6** | 29.7 / 54.4 | 30.9 / 55.4 | 31.9 / 56.4 | 2.0 / 13.3 | 20.5 / 44.1 | 22.3 / 46.9 | 22.4 / 47.6 |
| **en2xx** | **20.7 / 50.**1 | 17.3 / 44.1 | 17.8 / 44.7 | 18.6 / 45.7 | 0.4 / 5.7 | 8.1 / 26.7 | 8.7 / 29.0 | 8.7 / 28.8 |
| **Mean** | **28.2 / 54.9** | 23.5 / 49.2 | 24.4 / 50.0 | 25.3 / 51.1 | 1.2 / 9.6 | 14.3 / 35.5 | 15.6 / 38.0 | 15.6 / 38.2 |
| **xx2yy** | **13.7 / 40.5** | 8.8 / 31.2 | 8.4 / 30.9 | 10.1 / 34.0 | 0.3 / 4.1 | 4.0 / 16.1 | 4.4 / 17.3 | 4.2 / 17.1 |

### 4.3.1 Few-shot evaluation for language modeling

We perform few-shot prompting to evaluate the language model with the following prompt:

`[sl]`:$X_1$`\n[tl]`:$Y_1$`\n\n[sl]`:$X_2$`\n[tl]`:$Y_2$`\n\n...[sl]`:$X$`\n[tl]`:

where `[sl]` and `[tl]` denote the source and target language name (expressed in English. For example, when translating a sentence from `en` to `te`, we use `[sl]=English` and `[tl]=Telugu`), respectively. $X_\star$ and $Y_\star$ are demonstration examples used for prompting, and $X$ is the test input.

For each test example, we randomly sample demonstration examples, which is simple yet performs competitively with more complicated strategies [61, 67]. In particular, we randomly select examples from the dev split of each dataset. Since NTREX does not have a dev split, we randomly sample 1000 examples as the dev set and use the rest for test evaluation.

### 4.4 Results

In Tables 4 and 6 we present evaluation scores on the WMT datasets and NTREX datasets, which are evaluation sets in the news domain. We find that both the 7.2B parameter model and the 10B

Table 6: Evaluation scores on the recently introduced NTREX test set (depicted as `<bleu>` / `<chrf>`) for the MT models and language models described in Section 4.1 and Section 4.2 compared against unsupervised baselines [9]. Note that LM-8B is evaluated on a 50% split of the NTREX data and is not comparable to the MT-model evaluations.

| | Baziotis et al. [9] | MT-3B | MT-7.2B | MT-10.7B | LM-8B | | | |
|---|---|---|---|---|---|---|---|---|
| | | | | | 0-shot | 1-shot | 5-shot | 10-shot |
| **xx2en** | - / - | 30.6 / 54.5 | 32.7 / 56.2 | **33.6 / 57.6** | 3.2 / 17.3 | 20.4 / 43.8 | 23.8 / 48.2 | 24.4 / 49.0 |
| **en2xx** | - / - | 16.5 / 39.6 | 17.6 / **41.9** | **17.9 / 41.9** | 0.8 / 7.3 | 11.7 / 31.2 | 12.6 / 32.4 | 12.3 / 32.3 |
| **Average** | 19.8 / - | 23.5 / 47.0 | 25.1 / 49.0 | **25.7 / 49.7** | 2.0 / 12.3 | 16.0 / 37.4 | 18.1 / 40.2 | 18.3 / 40.6 |

parameter model is competitive with the significantly larger NLLB-54B model [46] on WMT. For the recent NTREX dataset, the only published results are small-scale results by Baziotis et al. [9].

In Table 5 we find that on the Flores-200 dataset, our model is within 3.8 chrf of the 54B parameter NLLB model, while on **xxyy** pairs the 10.7B models is behind by 6.5 chrf. This is likely due to a combination of factors, including using a significantly smaller model (5x smaller), domain differences [9, 8], and back-translated data [55]. Similarly, in Table 7, we find that the 10.7B parameter model is within 5.7 chrf of the scores reported by Bapna et al. [8]. Again, it is very difficult to compare their results to ours; their two largest advantages are 1) iterative back-translation, and 2) access to a much larger in-house text data. We note that the results we present are merely baselines to demonstrate the utility of `MADLAD-400`, and hope that future work builds upon these experiments by applying improved modeling techniques.

Across all evaluation datasets, we find that while results on few-shot translation using the 8B language model increase with an increasing number of demonstrations, these results are still significantly weaker than the results of models trained on supervised data. We present per-language pair results on all datasets in Appendix A.10.

Table 7: Evaluation scores on the GATONES test set used by Bapna et al. [8] (depicted as `<bleu>` / `<chrf>`) for the MT models and language models described in Section 4.1 and Section 4.2.

| | NTL (Bapna et al. [8]) | | MT-3B | MT-7.2B | MT-10.7B | LM-8B | | | |
|---|---|---|---|---|---|---|---|---|---|
| | 1.6B | 6.4B | | | | 0-shot | 1-shot | 5-shot | 10-shot |
| **xx2en** | - / 37.2 | - / **41.2** | 13.3 / 34.6 | **15.5** / 36.5 | 14.8 / 36.0 | 0.3 / 6.5 | 6.6 / 25.4 | 8.3 / 28.1 | 8.4 / 28.4 |
| **en2xx** | - / 28.5 | - / **33.1** | 4.5 / 23.9 | **5.5** / 26.4 | 5.4 / 26.2 | 0.2 / 4.2 | 1.7 / 10.5 | 1.7 / 9.9 | 1.8 / 9.4 |
| **Average** | - / 32.9 | - / **37.2** | 8.9 / 29.3 | **10.5** / 31.5 | 10.1 / 31.1 | 0.3 / 5.4 | 4.2 / 18.0 | 5.0 / 19.0 | 5.1 / 18.9 |

## 5 Related Work

Extensive work has been done to mine general purpose datasets for multilingual machine translation and language modeling. Xue et al. [63] introduce mC4, a general web domain corpus on 101 languages to train mT5, a pretrained language model for downstream NLP tasks. Similarly, Conneau et al. [16] introduce CC-100, later extended to CC100-XL by Lin et al. [42]. The OSCAR corpus [2] is also a mined dataset that supports 166 languages and the ROOTS corpus is a compiled dataset that contains 46 natural languages. Glot500-C [28] covers 511 languages: however, it is not clear how many of these languages comprise solely of religious texts. Bapna et al. [8] create an internal dataset on 1500+ languages, while NLLBTeam et al. [46] mine a dataset from CommonCrawl and ParaCrawl [19]. Recently, Leong et al. [40] created a 350+ language dataset from children's books.

In addition, there have been efforts to get better represented corpora and models for languages often underrepresented in general multilingual corpora: Serengeti [3] introduces a dataset and associated model trained on 517 African languages and language varieties, while IndicTrans2 [23] introduces a machine translated model for the 22 scheduled languages in India.

## 6 Limitations

While we used thorough self-audits to guide the creation of `MADLAD-400`, we note that most audits were conducted by non-speakers of the languages in `MADLAD-400`; as a result, many types of noise,

like machine-generated or disfluent content, could not be detected. Moreover, toxicity detectors, classifiers and filters that work reliably for all the 419 languages in `MADLAD-400` do not exist, limiting the extent to which we can clean and document [18, 7] the dataset. It is possible that issues still remain, so we encourage users to report issues that will be listed on the project github page. This paucity extends to the availability of multilingual evaluation sets for these languages - we could only evaluate our models on 204 of the languages in `MADLAD-400`. Additionally, even though decoder-only models are often evaluated on NLP tasks that are not necessarily machine translation [27, 6, 5], we did not conduct such evaluations - most available benchmarks cover only 30-50 languages of which most are not tail languages (which forms the focus of `MADLAD-400`). We instead leave this to future work. Finally, during our self-audit we noted the skew of data on the long tail towards specific domains such as religious texts. We hope that these limitations motivate the creation of more language-specific corpora not captured by web crawls, and the development of language-specific data cleaning tools and practices.

# 7   Conclusion

Through `MADLAD-400`, we introduce a highly multilingual, general web-domain, document-level text dataset. We perform a self-audit of this dataset for quality on samples of all 498 languages, develop filters, and remove spurious datasets, for a total of 419 languages in the release. We carefully describe the dataset creation process, laying out the iterations of audits and improvements upon the preliminary dataset along with observations that guided our decisions. We hope that this encourages creators of large-scale pretraining datasets both to put in their due diligence for manually inspecting and dealing with data, and also to describe and publicize the process in a level of detail that is reproducible and insightful for downstream users. This increased visibility into the dataset creation cycle can in turn improve model development and enable responsible data use [53]. Using `MADLAD-400`, we train and release large machine translation and general NLP models and evaluate them thoroughly. We hope that this further motivates work towards language technologies that are more inclusive of the rich language diversity housed by humanity.

# 8   Ethics Statement

Innovation in NLP technologies in English has been accelerated by training large scale deep learning models [17, 11] on massive web corpora [13, 68, 52]. However, on the long tail of written languages in the world there is a lack of high quality general data sources [32] that impede the progress of NLP tools for other languages. We hope that making an audited and cleaned corpus such as `MADLAD-400` available mitigates this issue. While we extensively cleaned `MADLAD-400`, the extent to which we can preprocess this data is limited by how not all languages have available tools for removing problematic content such as porn, toxic content, PII, copyrighted content or noise. We urge practitioners to carefully consider their target usecase before using `MADLAD-400`.

# Acknowledgements

We would like to thank Wolfgang Macherey, Zoubin Ghahramani and Orevaoghene Ahia for their helpful comments on the draft. We would also like to thank Subramanian Venkateswaran for debugging the virama rendering issues, and Ali Dabirmoghaddam for his insight on data samples of various languages in `MADLAD-400`.

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
