# A  Appendix

## A.1  LangID Details

Following *Language Id In the Wild* [12], we trained a Transformer-Base [60] Semi-Supervised LangId model (SSLID) on 498 languages. The training data is as described in *Language ID in the Wild*, with the differences that 1) training data is sampled to a temperature of T=3 to reduce over-triggering on low-resource languages; and 2) the data is supplemented with web-crawled data from the same paper (that has already been through the various filters described therein). The purpose of adding this data is to increase robustness to web-domain text, and possibly distill some of the filters used to create the web-crawl. The languages chosen for this model were roughly the top 498 by number of sentences in the dataset reported by *Language ID in the Wild*. The complete list may be seen in Table 8.

Table 8: BCP-47 codes, name, script and amount of data associated with all languages in `MADLAD-400`. The last 79 languages, with entries of "-", were languages that the LangID model was trained to detect, but was omitted after the self-audit.

| BCP-47 | Name | Script | docs (noisy) | docs (clean) | sents (noisy) | sents (clean) | chars (noisy) | chars (clean) |
|---|---|---|---|---|---|---|---|---|
| total | - | - | 7.8B | 4B | 148.4B | 105.5B | 33.3T | 18.3T |
| median | - | - | 21.6K | 1.5K | 202.3K | 61.8K | 63.6M | 8.5M |
| | | | | | | | | |
| en | English | Latn | 3.6B | 1.8B | 87.9B | 53.4B | 15T | 9T |
| ru | Russian | Cyrl | 823M | 402.5M | 823M | 12.4B | 3.1T | 1.8T |
| es | Spanish | Latn | 476.4M | 250.9M | 8.3B | 4.5B | 2.1T | 1.1T |
| fr | French | Latn | 384.2M | 218.9M | 7.9B | 5B | 2T | 1T |
| de | German | Latn | 478.8M | 225.1M | 11.5B | 6B | 2.2T | 1T |
| it | Italian | Latn | 238.9M | 126.4M | 4.5B | 2.5B | 1.2T | 553.1B |
| pt | Portuguese | Latn | 209.2M | 124.2M | 4B | 2.4B | 791.5B | 499.8B |
| pl | Polish | Latn | 145.1M | 90.9M | 3.3B | 2.4B | 505B | 356.4B |
| nl | Dutch | Latn | 134.5M | 86.6M | 134.5M | 2.3B | 698.5B | 334.5B |
| vi | Vietnamese | Latn | 92.8M | 55M | 1.6B | 1B | 342B | 228.8B |
| tr | Turkish | Latn | 107M | 56.4M | 107M | 1.2B | 328.8B | 198.9B |
| sv | Swedish | Latn | 65.2M | 35.2M | 65.2M | 1B | 422.6B | 153.7B |
| id | Indonesian | Latn | 120.9M | 38M | 2.2B | 747.5M | 443B | 148.3B |
| ro | Romanian | Latn | 60.8M | 35.4M | 60.8M | 746.4M | 244.1B | 148.2B |
| cs | Czech | Latn | 72.1M | 38.3M | 1.7B | 1B | 272.2B | 147.9B |
| zh | Mandarin Chinese | Hans | 29.3M | 19.9M | 492.3M | 298.8M | 333B | 142.3B |
| hu | Hungarian | Latn | 47.6M | 29.7M | 1.3B | 806.3M | 223.6B | 134.9B |
| ja | Japanese | Jpan | 23.3M | 21.8M | 326M | 321.6M | 133.3B | 132.2B |
| th | Thai | Thai | 19M | 17.4M | 19M | 385.8M | 118.6B | 117.6B |
| fi | Finnish | Latn | 35.8M | 20.4M | 1B | 650.3M | 202.2B | 101.1B |
| fa | Persian | Arab | 58.1M | 23.1M | 920.6M | 493.5M | 220.4B | 96.7B |
| uk | Ukrainian | Cyrl | 46.6M | 25M | 1B | 599.9M | 164.2B | 95.2B |
| da | Danish | Latn | 38.5M | 17.9M | 1.1B | 508M | 252B | 83.1B |
| el | Greek | Grek | 52.4M | 20.9M | 808M | 445.4M | 173.2B | 80.9B |
| no | Norwegian | Latn | 34.7M | 14.9M | 34.7M | 498.7M | 305.6B | 74.8B |
| bg | Bulgarian | Cyrl | 27.2M | 12.8M | 599.4M | 360.3M | 95.6B | 57.8B |
| sk | Slovak | Latn | 23.2M | 11.9M | 487.9M | 300.6M | 77.8B | 45.7B |
| ko | Korean | Kore | 19.7M | 12.7M | 628.6M | 471.8M | 65.9B | 43.8B |
| ar | Arabic | Arab | 67.6M | 12.4M | 876.6M | 182.6M | 243B | 43.2B |
| lt | Lithuanian | Latn | 15.3M | 8.7M | 374M | 256.9M | 58.6B | 41.3B |
| ca | Catalan | Latn | 17.9M | 9.5M | 258.6M | 153M | 56.5B | 34.6B |
| sl | Slovenian | Latn | 12M | 6.3M | 316M | 180M | 47.8B | 30.5B |
| he | Hebrew | Hebr | 14.1M | 7.2M | 302.2M | 196.8M | 54.9B | 30.5B |
| et | Estonian | Latn | 8.8M | 5.5M | 223.8M | 176.3M | 40.1B | 28.7B |
| lv | Latvian | Latn | 8.4M | 5M | 186.1M | 138.5M | 36.7B | 23.9B |
| hi | Hindi | Deva | 9.9M | 4.5M | 254.4M | 152M | 39.9B | 20.1B |
| sq | Albanian | Latn | 5.5M | 3.6M | 5.5M | 56.1M | 17B | 12.7B |
| ms | Malay | Latn | 14.1M | 2.3M | 14.1M | 55.2M | 58.8B | 12.5B |
| az | Azerbaijani | Latn | 5.2M | 3.3M | 90.3M | 70.9M | 16.3B | 11.9B |
| sr | Serbian | Cyrl | 4.7M | 2M | 4.7M | 64M | 18.6B | 11B |
| ta | Tamil | Taml | 5.6M | 2.6M | 122.5M | 81.9M | 19.2B | 10.6B |
| hr | Croatian | Latn | 23M | 2.8M | 476.6M | 53M | 85.1B | 9.6B |
| kk | Kazakh | Cyrl | 3.1M | 1.8M | 87.4M | 59.1M | 13.4B | 8.6B |
| is | Icelandic | Latn | 2.9M | 1.6M | 73.7M | 39.3M | 14.9B | 6.4B |
| ml | Malayalam | Mlym | 3.7M | 2.1M | 75M | 52M | 10.5B | 6.3B |
| mr | Marathi | Deva | 2.9M | 1.7M | 2.9M | 50M | 8.7B | 5.5B |
| te | Telugu | Telu | 2.5M | 1.7M | 59M | 46.4M | 7.4B | 5.1B |
| af | Afrikaans | Latn | 2.9M | 868.7K | 51.9M | 30M | 11.8B | 4.8B |
| gl | Galician | Latn | 4.2M | 1.3M | 45.3M | 18.8M | 15.6B | 4.8B |
| fil | Filipino | Latn | 4.2M | 901.5K | 67.4M | 19.2M | 14.6B | 4.7B |
| be | Belarusian | Cyrl | 2M | 1.1M | 48.8M | 31.3M | 7.2B | 4.6B |

| | | | | | | | |
|---|---|---|---|---|---|---|---|
| mk | Macedonian | Cyrl | 2.9M | 1.4M | 41.3M | 22.6M | 9.1B | 4.5B |
| eu | Basque | Latn | 2.1M | 1.2M | 41.7M | 24.8M | 6.9B | 4.3B |
| bn | Bengali | Beng | 4.3M | 1.1M | 151.2M | 38.6M | 16.8B | 4.3B |
| ka | Georgian | Geor | 3.1M | 936.5K | 53.7M | 26.6M | 10.3B | 3.8B |
| mn | Mongolian | Cyrl | 2.2M | 879.9K | 43.3M | 24M | 7.9B | 3.5B |
| bs | Bosnian | Cyrl | 12.9M | 1.4M | 163.6M | 9M | 39.5B | 3.3B |
| uz | Uzbek | Latn | 1.4M | 669.9K | 25.7M | 17.5M | 5.2B | 3.3B |
| ur | Urdu | Arab | 967.2K | 467.2K | 29M | 18.4M | 5.2B | 2.7B |
| sw | Swahili | Latn | 1.3M | 537.8K | 1.3M | 9.5M | 4.6B | 2.4B |
| yue | Cantonese | Hant | 465.9K | 309.3K | 2.8M | 2.4M | 2.4B | 2.3B |
| ne | Nepali | Deva | 876.4K | 453.7K | 876.4K | 20.4M | 3.9B | 2.2B |
| kn | Kannada | Knda | 1.6M | 657.8K | 32.9M | 19.2M | 4.6B | 2.2B |
| kaa | Kara-Kalpak | Cyrl | 1.1M | 586.4K | 19.8M | 13.3M | 3.8B | 2.2B |
| gu | Gujarati | Gujr | 1.3M | 659.7K | 28.9M | 18.1M | 3.9B | 2.1B |
| si | Sinhala | Sinh | 788K | 349.2K | 22.1M | 16M | 3.4B | 1.9B |
| cy | Welsh | Latn | 4.9M | 430.7K | 68.3M | 7.4M | 26.4B | 1.7B |
| eo | Esperanto | Latn | 1.4M | 260K | 33.9M | 9.3M | 5.5B | 1.7B |
| la | Latin | Latn | 2.9M | 319.2K | 85.7M | 13.8M | 8.2B | 1.5B |
| hy | Armenian | Armn | 2M | 397.5K | 31.1M | 9.9M | 8.1B | 1.5B |
| ky | Kyrghyz | Cyrl | 751.1K | 367.6K | 14.3M | 9.6M | 2.5B | 1.4B |
| tg | Tajik | Cyrl | 789.2K | 328.2K | 789.2K | 7.4M | 2.6B | 1.4B |
| ga | Irish | Latn | 5.3M | 286K | 31.7M | 6.9M | 30.6B | 1.4B |
| mt | Maltese | Latn | 1.2M | 265.4K | 1.2M | 5.6M | 3.2B | 1.3B |
| my | Myanmar (Burmese) | Mymr | 176.5K | 172.4K | 176.5K | 10.1M | 1.3B | 1.3B |
| km | Khmer | Khmr | 297.8K | 285.7K | 5M | 5M | 1.1B | 1.1B |
| tt | Tatar | Cyrl | 2.1M | 346.9K | 60.2M | 8.6M | 12.1B | 1B |
| so | Somali | Latn | 729.2K | 293.2K | 729.2K | 3.1M | 2.1B | 992.4M |
| ku | Kurdish (Kurmanji) | Latn | 671.9K | 218.9K | 10.7M | 4.9M | 2.1B | 849.9M |
| ps | Pashto | Arab | 429.9K | 252.9K | 5.1M | 3.6M | 1.4B | 848.9M |
| pa | Punjabi | Guru | 368.2K | 150.6K | 368.2K | 6M | 1.6B | 797.1M |
| rw | Kinyarwanda | Latn | 681.8K | 226.5K | 681.8K | 1.9M | 1.7B | 749.1M |
| lo | Lao | Laoo | 229.1K | 216K | 2.9M | 2.8M | 706.9M | 697.6M |
| ha | Hausa | Latn | 443.9K | 173.5K | 4.5M | 2.4M | 1.3B | 630.2M |
| dv | Dhivehi | Thaa | 264.4K | 167.2K | 4.3M | 3.5M | 877.3M | 603.1M |
| fy | W. Frisian | Latn | 1.7M | 210K | 12.1M | 3.7M | 3.7B | 592.3M |
| lb | Luxembourgish | Latn | 7.6M | 146K | 47.1M | 3.4M | 58.4B | 575.5M |
| ckb | Kurdish (Sorani) | Arab | 622.7K | 148.9K | 5.6M | 2.5M | 2.2B | 572.7M |
| mg | Malagasy | Latn | 295.2K | 115.4K | 4.5M | 2.6M | 1.3B | 548.5M |
| gd | Scottish Gaelic | Latn | 206K | 94.3K | 3.7M | 2.4M | 812M | 526M |
| am | Amharic | Ethi | 245.2K | 106.3K | 7.1M | 5.3M | 869.9M | 509M |
| ug | Uyghur | Arab | 227.1K | 106.5K | 4.5M | 3.1M | 998.5M | 504.6M |
| ht | Haitian Creole | Latn | 425.6K | 110.4K | 6.7M | 2.6M | 994.5M | 461.5M |
| grc | Ancient Greek | Grek | 364.8K | 70.7K | 13.7M | 2.8M | 2B | 417.8M |
| hmn | Hmong | Latn | 241.3K | 75.2K | 3.5M | 1.9M | 1.2B | 408.8M |
| sd | Sindhi | Arab | 115.6K | 65.9K | 115.6K | 2.4M | 561M | 380.4M |
| jv | Javanese | Latn | 999.5K | 69.5K | 13M | 2M | 2.3B | 376.1M |
| mi | Maori | Latn | 711.9K | 79.5K | 5.9M | 1.9M | 1.6B | 371.9M |
| tk | Turkmen | Latn | 180.2K | 82.5K | 180.2K | 1.8M | 575.2M | 369M |
| ceb | Cebuano | Latn | 617.5K | 66.2K | 6.7M | 1.6M | 1.5B | 357.7M |
| yi | Yiddish | Hebr | 160.6K | 64.9K | 3.3M | 1.9M | 838.4M | 352.6M |
| ba | Bashkir | Cyrl | 372.4K | 90.3K | 9.3M | 2.6M | 766.5M | 320.7M |
| fo | Faroese | Latn | 382.9K | 97.8K | 3.9M | 1.8M | 923.3M | 314.9M |
| or | Odia (Oriya) | Orya | 139.6K | 100.5K | 139.6K | 3.1M | 437.2M | 309.5M |
| xh | Xhosa | Latn | 310.9K | 53.7K | 2.9M | 1.4M | 749.5M | 287.3M |
| su | Sundanese | Latn | 336.6K | 55K | 336.6K | 1.6M | 967.2M | 286.7M |
| kl | Kalaallisut | Latn | 85.9K | 46K | 2.1M | 1.5M | 403.9M | 279.1M |
| ny | Chichewa | Latn | 181.6K | 52.2K | 181.6K | 1.5M | 611.2M | 277.5M |
| sm | Samoan | Latn | 137.8K | 52.6K | 1.9M | 1.3M | 607.9M | 276.3M |
| sn | Shona | Latn | 3.1M | 60.2K | 3.1M | 1.2M | 10.6B | 266M |
| co | Corsican | Latn | 546.7K | 55.4K | 6.1M | 1.3M | 1.1B | 265.5M |
| zu | Zulu | Latn | 372.3K | 53.8K | 3.8M | 1.2M | 1.2B | 257.4M |
| ig | Igbo | Latn | 130.4K | 54.4K | 2.1M | 1.4M | 846.1M | 251.4M |
| yo | Yoruba | Latn | 115K | 52.1K | 2M | 1.2M | 415.6M | 239M |
| pap | Papiamento | Latn | 259.1K | 54.5K | 259.1K | 1.4M | 1.4B | 229.9M |
| st | Sesotho | Latn | 96.8K | 40.4K | 96.8K | 1.1M | 381.5M | 226.9M |
| haw | Hawaiian | Latn | 310.4K | 45.7K | 7.1M | 1M | 892M | 214.2M |
| as | Assamese | Beng | 53.9K | 33.8K | 2.4M | 1.7M | 275.8M | 182.1M |
| oc | Occitan | Latn | 2.4M | 36.4K | 2.4M | 1.6M | 6.7B | 177.6M |
| cv | Chuvash | Cyrl | 599.4K | 47.3K | 12M | 1.6M | 1B | 168.9M |
| lus | Mizo | Latn | 91.5K | 36.4K | 1.4M | 863.5K | 298.3M | 167.3M |
| tet | Tetum | Latn | 291K | 40.4K | 1.9M | 475.7K | 1.6B | 152.3M |
| gsw | Swiss German | Latn | 7.6M | 42.7K | 64.5M | 1M | 42.3B | 149.2M |
| sah | Yakut | Cyrl | 1.3M | 29.2K | 1.3M | 1.2M | 2.2B | 148.2M |
| br | Breton | Latn | 705.4K | 33.2K | 7.8M | 731.7K | 3.7B | 125.4M |
| rm | Romansh | Latn | 238.1K | 33.8K | 238.1K | 603.4K | 391M | 100.2M |
| sa | Sanskrit | Deva | 154.3K | 7.1K | 154.3K | 1.1M | 512.5M | 88.8M |
| bo | Tibetan | Tibt | 6.2K | 6.2K | 1.1M | 1.1M | 88.7M | 88.7M |
| om | Oromo | Latn | 846.1K | 18.9K | 846.1K | 469.8K | 1.9B | 88.5M |
| se | N. Sami | Latn | 54.3K | 23.9K | 879.5K | 493.3K | 148.4M | 84.6M |
| ce | Chechen | Cyrl | 59.3K | 15K | 991.1K | 460.1K | 130.6M | 67.8M |
| cnh | Hakha Chin | Latn | 44.4K | 21.6K | 688.6K | 406.9K | 110.8M | 63M |

| | | | | | | | |
|---|---|---|---|---|---|---|---|
| ilo | Ilocano | Latn | 69.8K | 11.8K | 889.2K | 365.1K | 187.9M | 59.4M |
| hil | Hiligaynon | Latn | 126.8K | 10.6K | 1.1M | 379.7K | 293.5M | 57.2M |
| udm | Udmurt | Cyrl | 67.1K | 13.4K | 942.7K | 510.3K | 106M | 55.5M |
| os | Ossetian | Cyrl | 172.1K | 12.6K | 172.1K | 359.3K | 233.5M | 50.1M |
| lg | Luganda | Latn | 61.1K | 13K | 510.9K | 166.1K | 160.7M | 48M |
| ti | Tigrinya | Ethi | 20.8M | 7.3K | 20.8K | 481.3K | 95.4M | 44.6M |
| vec | Venetian | Latn | 1.1M | 11.1K | 10M | 209.7K | 1.8B | 43.8M |
| ts | Tsonga | Latn | 34.7K | 5.2K | 34.7K | 248.6K | 377.2M | 38.8M |
| tyv | Tuvinian | Cyrl | 61.6K | 9.1K | 596.6K | 268.3K | 80.2M | 38.5M |
| kbd | Kabardian | Cyrl | 154.7K | 7.5K | 1.4M | 257.2K | 321.4M | 36.8M |
| ee | Ewe | Latn | 14.1K | 4.5K | 353.6K | 246.7K | 67.9M | 32.8M |
| iba | Iban | Latn | 34K | 7.6K | 326.9K | 126.1K | 251.4M | 30.5M |
| av | Avar | Cyrl | 107.6K | 6.3K | 806.1K | 190.1K | 129M | 30.2M |
| kha | Khasi | Latn | 37.8K | 12.1K | 235.5K | 75.2K | 88.6M | 30.2M |
| to | Tonga (Tonga Islands) | Latn | 14.3K | 4.6K | 14.3K | 149K | 58.2M | 29.9M |
| tn | Tswana | Latn | 138.2K | 4.8K | 138.2K | 174.4K | 302.3M | 29.2M |
| nso | Sepedi | Latn | 376.2K | 4.4K | 376.2K | 188.4K | 2B | 28.2M |
| fj | Fijian | Latn | 17K | 4K | 410K | 164.1K | 67.7M | 28M |
| zza | Zaza | Latn | 370.1K | 6K | 3.3M | 229.2K | 617.3M | 26.3M |
| ak | Twi | Latn | 19.5K | 4.8K | 341.7K | 210.2K | 74.5M | 24.8M |
| ada | Adangme | Latn | 6.5K | 3.1K | 291.5K | 199.2K | 38.9M | 24.2M |
| otq | Querétaro Otomi | Latn | 17.6K | 5.6K | 17.6K | 114.8K | 65M | 23.4M |
| dz | Dzongkha | Tibt | 1.9K | 1.9K | 191.7K | 191.7K | 22.7M | 22.7M |
| bua | Buryat | Cyrl | 9.8K | 5.3K | 252K | 144.6K | 38M | 21.7M |
| cfm | Falam Chin | Latn | 9.1K | 4.9K | 199.6K | 128.6K | 32.9M | 21.5M |
| ln | Lingala | Latn | 94.7K | 3.3K | 718.7K | 139K | 291.8M | 21.5M |
| chm | Meadow Mari | Cyrl | 81.5K | 4.7K | 929.1K | 179.7K | 132.2M | 21.3M |
| gn | Guarani | Latn | 87.1K | 3.9K | 770.9K | 162.6K | 140.7M | 20.8M |
| krc | Karachay-Balkar | Cyrl | 359.5K | 4.8K | 2.3M | 153.9K | 369.5M | 20.7M |
| wa | Walloon | Latn | 70.6K | 2.8K | 1.5M | 127.2K | 198.8M | 20.4M |
| hif | Fiji Hindi | Latn | 702K | 2.4K | 7.9M | 124.7K | 9.1B | 19.1M |
| yua | Yucateco | Latn | 10.4K | 4K | 141.6K | 77.6K | 36.8M | 17.2M |
| srn | Sranan Tongo | Latn | 16.7K | 2.3K | 16.7K | 139.5K | 49.1M | 17M |
| war | Waray (Philippines) | Latn | 1M | 2.9K | 114M | 96.2K | 3.5B | 16.1M |
| rom | Romani | Latn | 22.9K | 4.2K | 22.9K | 76.1K | 59M | 15.9M |
| bik | Central Bikol | Latn | 44.8K | 3.1K | 376.7K | 77K | 102.3M | 15.7M |
| pam | Pampanga | Latn | 174.2K | 2.8K | 174.2K | 23.3K | 324M | 15.5M |
| sg | Sango | Latn | 4.2K | 2.1K | 154K | 117.9K | 22.6M | 15.5M |
| lu | Luba-Katanga | Latn | 10.6K | 1.4K | 316K | 112.1K | 54.2M | 15.4M |
| ady | Adyghe | Cyrl | 74.9K | 4.2K | 446.8K | 96.9K | 67.9M | 14.8M |
| kbp | Kabiyè | Latn | 5.9K | 3K | 247.9K | 128.3K | 30.8M | 14.6M |
| syr | Syriac | Syrc | 3.5K | 716 | 326.4K | 197.1K | 31.5M | 14M |
| ltg | Latgalian | Latn | 13.1K | 4.1K | 213.7K | 87.3K | 29.2M | 13.9M |
| myv | Erzya | Cyrl | 164.8K | 3.1K | 164.8K | 130K | 120.3M | 13.8M |
| iso | Isoko | Latn | 3.7K | 1.7K | 155.8K | 111.5K | 23M | 13.7M |
| kac | Kachin | Latn | 5.9K | 2.6K | 109.2K | 77.4K | 26.6M | 13.6M |
| bho | Bhojpuri | Deva | 13.6K | 4.1K | 306.2K | 118.5K | 37.6M | 13.4M |
| ay | Aymara | Latn | 8.1K | 2.5K | 196.7K | 83.8K | 34.5M | 13.1M |
| kum | Kumyk | Cyrl | 4.2K | 2.5K | 132.2K | 89.7K | 18.2M | 12.4M |
| qu | Quechua | Latn | 149.7K | 2.4K | 1M | 87K | 200.6M | 12.2M |
| za | Zhuang | Latn | 824.7K | 1.7K | 19.2M | 53.9K | 3B | 12.1M |
| pag | Pangasinan | Latn | 49.6K | 1.6K | 49.6K | 88.8K | 92.9M | 12M |
| ngu | Guerrero Nahuatl | Latn | 3.8K | 1.5K | 3.8K | 87.1K | 21.4M | 11.8M |
| ve | Venda | Latn | 3.8K | 1.9K | 97.8K | 79.4K | 19M | 11.7M |
| pck | Paite Chin | Latn | 8.9K | 1.3K | 8.9K | 69.7K | 39.8M | 11.5M |
| zap | Zapotec | Latn | 5.5K | 1.8K | 202.3K | 93.5K | 26.4M | 11.4M |
| tyz | Tày | Latn | 8K | 1.7K | 454.8K | 104.6K | 46.3M | 11.3M |
| hui | Huli | Latn | 2K | 1.7K | 80.1K | 74.7K | 11.8M | 10.9M |
| bbc | Batak Toba | Latn | 72.3K | 1.3K | 718.3K | 73.2K | 151.3M | 10.6M |
| tzo | Tzotzil | Latn | 2.8K | 1.4K | 100.4K | 75.7K | 15.9M | 10.6M |
| tiv | Tiv | Latn | 3.8K | 1.1K | 3.8K | 80.7K | 20.4M | 10.2M |
| ksd | Kuanua | Latn | 14.9K | 2K | 533K | 78.6K | 62.4M | 10M |
| gom | Goan Konkani | Deva | 4.6K | 2.1K | 178.3K | 108K | 19.8M | 10M |
| min | Minangkabau | Latn | 28.2K | 1.5K | 500.9K | 75.6K | 70.5M | 9.9M |
| ang | Old English | Latn | 66.5K | 803 | 1.8M | 86.7K | 193M | 9.8M |
| nhe | E. Huasteca Nahuatl | Latn | 3K | 1.7K | 3K | 57.7K | 15.6M | 9.8M |
| bgp | E. Baluchi | Latn | 355.7K | 2.4K | 5.6M | 43.3K | 1.1B | 9.8M |
| nzi | Nzima | Latn | 2.5K | 1.4K | 2.5K | 71.8K | 14.4M | 9.4M |
| nnb | Nande | Latn | 4.9K | 1.1K | 4.9K | 70.2K | 27.7M | 9.1M |
| nv | Navajo | Latn | 17.1K | 12.6K | 17.1K | 86.5K | 24.8M | 9.1M |
| zxx | Noise | - | 118.8K | 1.8K | 3.8M | 49.3K | 501K | 6.6K |
| bci | Baoulé | Latn | 7.4K | 1.3K | 124.8K | 87.1K | 32.8M | 9M |
| kv | Komi | Cyrl | 59.1K | 1.9K | 584.3K | 88.8K | 91.4M | 9M |
| new | Newari | Deva | 6.6K | 1.6K | 6.6K | 85K | 21.2M | 8.8M |
| mps | Dadibi | Latn | 2.7K | 1.2K | 132.8K | 71.9K | 16M | 8.7M |
| alt | S. Altai | Cyrl | 2.6K | 1.4K | 110.1K | 65.9K | 14.3M | 8.7M |
| meu | Motu | Latn | 5.9K | 1.7K | 232.1K | 72.6K | 27.2M | 8.6M |
| bew | Betawi | Latn | 311.1K | 2.7K | 10.4M | 58.4K | 1.4B | 8.5M |
| fon | Fon | Latn | 5.3K | 1.1K | 222.9K | 67.3K | 34M | 8.3M |
| iu | Inuktitut | Cans | 5.4K | 2.5K | 92.6K | 53.1K | 17.5M | 8.3M |
| abt | Ambulas | Latn | 1.6K | 1.3K | 122.7K | 110.3K | 9.6M | 8.2M |

| | | | | | | | |
|---|---|---|---|---|---|---|---|
| mgh | Makhuwa-Meetto | Latn | 5.5K | 1.2K | 151.8K | 61.2K | 24.1M | 8.2M |
| mnw | Mon | Mymr | 1.1K | 1.1K | 144.8K | 144.7K | 8.1M | 8.1M |
| tvl | Tuvalu | Latn | 2.3K | 933 | 72.9K | 53.6K | 12.6M | 8.1M |
| dov | Dombe | Latn | 3.5K | 923 | 129.8K | 56.7K | 20.7M | 8M |
| tlh | Klingon | Latn | 516.9K | 3.1K | 516.9K | 46.9K | 1.4B | 7.8M |
| ho | Hiri Motu | Latn | 2K | 1.5K | 57K | 47.8K | 12.3M | 7.8M |
| kw | Cornish | Latn | 176.9K | 2.3K | 1M | 51.6K | 327.8M | 7.7M |
| mrj | Hill Mari | Cyrl | 97.1K | 1.4K | 97.1K | 60.3K | 100.6M | 7.6M |
| meo | Kedah Malay | Latn | 790.7K | 4.7K | 16.5M | 39K | 3B | 7.5M |
| crh | Crimean Tatar | Cyrl | 5.1K | 1.2K | 170.9K | 61.8K | 18.8M | 7.5M |
| mbt | Matigsalug Manobo | Latn | 1.6K | 969 | 86K | 45.4K | 14.6M | 7.5M |
| emp | N. Emberá | Latn | 3.6K | 1.2K | 106.4K | 75.4K | 14.5M | 7.4M |
| ace | Achinese | Latn | 65.5K | 966 | 632.5K | 32.5K | 146.1M | 7.4M |
| ium | Iu Mien | Latn | 100.3K | 1.7K | 6.2M | 54.9K | 314M | 7.4M |
| mam | Mam | Latn | 23K | 1.5K | 446.3K | 52.9K | 70.4M | 7.2M |
| gym | Ngäbere | Latn | 1.5K | 820 | 73.7K | 49.6K | 10.3M | 6.9M |
| mai | Maithili | Deva | 54.3K | 1.2K | 1M | 60.2K | 156M | 6.8M |
| crs | Seselwa Creole French | Latn | 7.6K | 873 | 282.4K | 40.1K | 40.1M | 6.8M |
| pon | Pohnpeian | Latn | 5.7K | 1.5K | 167.8K | 48.7K | 18.3M | 6.7M |
| ubu | Umbu-Ungu | Latn | 2.2K | 846 | 113.5K | 47.5K | 15.9M | 6.7M |
| fip | Fipa | Latn | 3.7K | 729 | 165.6K | 49K | 25.7M | 6.6M |
| quc | K'iche' | Latn | 4.4K | 1.5K | 89.2K | 41.2K | 16.6M | 6.4M |
| gv | Manx | Latn | 501.9K | 1.6K | 18.8M | 26.9K | 933.1M | 6.2M |
| kj | Kuanyama | Latn | 112.2K | 2.1K | 881.8K | 22.6K | 339.6M | 6M |
| btx | Batak Karo | Latn | 3.1K | 1K | 81.7K | 43.9K | 13.1M | 5.9M |
| ape | Bukiyip | Latn | 7K | 814 | 147K | 56.1K | 71M | 5.8M |
| chk | Chuukese | Latn | 2.8K | 1.1K | 98.8K | 44K | 12M | 5.8M |
| rcf | Réunion Creole French | Latn | 21.6K | 2.6K | 21.6K | 50.5K | 30.2M | 5.7M |
| shn | Shan | Mymr | 889 | 788 | 46.4K | 46.2K | 5.7M | 5.7M |
| tzh | Tzeltal | Latn | 1.7K | 702 | 41.7K | 33.9K | 9.3M | 5.6M |
| mdf | Moksha | Cyrl | 71K | 1.6K | 394.7K | 45.1K | 65.8M | 5.5M |
| ppk | Uma | Latn | 2.6K | 1.1K | 85.8K | 34.9K | 13.2M | 5.5M |
| ss | Swati | Latn | 8.1K | 1.1K | 8.1K | 30.4K | 23.7M | 5.5M |
| gag | Gagauz | Latn | 33.9K | 1.6K | 491K | 37K | 84.9M | 5.2M |
| cab | Garifuna | Latn | 1.2K | 629 | 50.4K | 37.5K | 7.5M | 5.1M |
| kri | Krio | Latn | 39.1K | 786 | 271.2K | 38.8K | 86.4M | 5M |
| seh | Sena | Latn | 5.6K | 545 | 68.8K | 37.2K | 14.9M | 4.9M |
| ibb | Ibibio | Latn | 74.1K | 818 | 516.5K | 36.3K | 190.9M | 4.9M |
| tbz | Ditammari | Latn | 5.1K | 1.1K | 128.7K | 37.5K | 22M | 4.8M |
| bru | E. Bru | Latn | 3K | 1.1K | 89.7K | 48.2K | 12.9M | 4.8M |
| enq | Enga | Latn | 7.1K | 793 | 241.9K | 39.1K | 68.5M | 4.8M |
| ach | Acoli | Latn | 2K | 915 | 63K | 40.1K | 9M | 4.7M |
| cuk | San Blas Kuna | Latn | 4.1K | 899 | 76.5K | 34.3K | 24.7M | 4.6M |
| kmb | Kimbundu | Latn | 1.3K | 538 | 60.4K | 36.9K | 8.4M | 4.6M |
| wo | Wolof | Latn | 36.4K | 871 | 303.4K | 25.4K | 213.4M | 4.5M |
| kek | Kekchí | Latn | 3.2K | 782 | 70.4K | 38.4K | 13.6M | 4.4M |
| qub | Huallaga Huánuco Quechua | Latn | 972 | 705 | 61K | 51.1K | 5.9M | 4.4M |
| tab | Tabassaran | Cyrl | 7.8K | 1.2K | 226.4K | 26.8K | 33.7M | 4.4M |
| bts | Batak Simalungun | Latn | 3.2K | 869 | 109.1K | 29.1K | 20.8M | 4.2M |
| kos | Kosraean | Latn | 2.2K | 881 | 44.6K | 27.8K | 6.5M | 4.2M |
| rwo | Rawa | Latn | 938 | 572 | 938 | 45.5K | 5.1M | 4.2M |
| cak | Kaqchikel | Latn | 1.2K | 617 | 70.4K | 32.6K | 7.6M | 4.2M |
| tuc | Mutu | Latn | 3.5K | 635 | 193.2K | 50.3K | 17.2M | 4.1M |
| bum | Bulu | Latn | 4.7K | 559 | 103.8K | 36.5K | 18.8M | 4M |
| cjk | Chokwe | Latn | 3.6K | 586 | 144.1K | 24.1K | 22.5M | 3.9M |
| gil | Gilbertese | Latn | 3.9K | 586 | 151.5K | 24.1K | 24.1M | 3.9M |
| stq | Saterfriesisch | Latn | 111.9K | 809 | 111.9K | 27.7K | 243.1M | 3.8M |
| tsg | Tausug | Latn | 353.8K | 789 | 353.8K | 17.9K | 1.1B | 3.8M |
| quh | S. Bolivian Quechua | Latn | 1K | 501 | 42K | 29.9K | 5.8M | 3.7M |
| mak | Makasar | Latn | 1K | 555 | 32.5K | 20.4K | 6.1M | 3.7M |
| arn | Mapudungun | Latn | 2.4K | 593 | 64.5K | 26.2K | 10.2M | 3.7M |
| ban | Balinese | Latn | 8K | 637 | 150.9K | 16.3K | 35.4M | 3.6M |
| jiv | Shuar | Latn | 1.7K | 696 | 80.9K | 32K | 9.6M | 3.5M |
| sja | Epena | Latn | 1.3K | 527 | 67.7K | 24.9K | 7.7M | 3.4M |
| yap | Yapese | Latn | 1.9K | 638 | 37.6K | 19.5K | 6.9M | 3.3M |
| tcy | Tulu | Knda | 10.7K | 632 | 338.7K | 37.1K | 41.6M | 3.3M |
| toj | Tojolabal | Latn | 736 | 452 | 736 | 26.1K | 4.3M | 3.3M |
| twu | Termanu | Latn | 2.5K | 539 | 109.9K | 24.4K | 14.2M | 3.2M |
| xal | Kalmyk | Cyrl | 71.8K | 913 | 498.5K | 30.8K | 64.7M | 3.2M |
| amu | Guerrero Amuzgo | Latn | 1.8K | 511 | 72K | 25.2K | 9.6M | 3.2M |
| rmc | Carpathian Romani | Latn | 2.4K | 738 | 2.4K | 25.8K | 7.9M | 3.2M |
| hus | Huastec | Latn | 825 | 569 | 26.5K | 23.7K | 4.4M | 3.1M |
| nia | Nias | Latn | 2K | 408 | 2K | 25K | 11.3M | 3.1M |
| kjh | Khakas | Cyrl | 1.5K | 672 | 42.8K | 28.7K | 4.5M | 3.1M |
| bm | Bambara | Latn | 21.9K | 702 | 172.3K | 24.5K | 48.4M | 3M |
| guh | Guahibo | Latn | 1.9K | 331 | 104.9K | 28.4K | 11.2M | 3M |
| mas | Masai | Latn | 15.2K | 405 | 216.8K | 17.6K | 42.1M | 3M |
| acf | St Lucian Creole French | Latn | 4.9K | 730 | 81.9K | 24.6K | 11.6M | 3M |
| dtp | Kadazan Dusun | Latn | 4.6K | 1.3K | 51.2K | 7.9K | 12.7M | 3M |
| ksw | S'gaw Karen | Mymr | 560 | 536 | 16.1K | 16K | 2.9M | 2.9M |
| bzj | Belize Kriol English | Latn | 983 | 404 | 33.6K | 26.4K | 4.5M | 2.9M |

| din | Dinka | Latn | 128.4K | 611 | 885.8K | 23.6K | 210M | 2.9M |
|-----|-------|------|--------|-----|--------|-------|------|------|
| zne | Zande | Latn | 1.3K | 239 | 61.9K | 21.3K | 8.2M | 2.8M |
| mad | Madurese | Latn | 103.8K | 509 | 500.6K | 18.5K | 111.8M | 2.8M |
| msi | Sabah Malay | Latn | 686.7K | 1.9K | 686.7K | 22.6K | 2.6B | 2.7M |
| mag | Magahi | Deva | 631 | 138 | 62.6K | 22.1K | 10.7M | 2.6M |
| mkn | Kupang Malay | Latn | 956 | 402 | 33.1K | 25.4K | 3.4M | 2.6M |
| kg | Kongo | Latn | 4.7K | 365 | 85.5K | 21.7K | 16.6M | 2.6M |
| lhu | Lahu | Latn | 46K | 377 | 975K | 15.7K | 208.6M | 2.5M |
| ch | Chamorro | Latn | 12.9K | 449 | 147.5K | 16K | 63.5M | 2.5M |
| qvi | Imbabura H. Quichua | Latn | 1.2K | 266 | 48.4K | 19.3K | 6.5M | 2.3M |
| mh | Marshallese | Latn | 4.6K | 296 | 235.1K | 13K | 24.9M | 2.2M |
| djk | E. Maroon Creole | Latn | 560 | 246 | 30.9K | 24.4K | 3.7M | 2.2M |
| sus | Susu | Latn | 664 | 437 | 664 | 15.2K | 3.7M | 2.1M |
| mfe | Morisien | Latn | 7.5K | 320 | 198.8K | 18.2K | 26.9M | 2.1M |
| srm | Saramaccan | Latn | 847 | 227 | 847 | 17.3K | 6.3M | 2M |
| dyu | Dyula | Latn | 1.2K | 483 | 55.8K | 19.7K | 5.7M | 2M |
| ctu | Chol | Latn | 690 | 366 | 35.5K | 20.6K | 3.6M | 2M |
| gui | E. Bolivian Guaraní | Latn | 1.1K | 409 | 62.7K | 24.8K | 6.5M | 2M |
| pau | Palauan | Latn | 1.7K | 185 | 1.7K | 13.1K | 12.4M | 2M |
| inb | Inga | Latn | 387 | 343 | 17.3K | 17K | 2M | 1.9M |
| bi | Bislama | Latn | 71.9K | 311 | 308.5K | 13.6K | 132.4M | 1.9M |
| mni | Meiteilon (Manipuri) | Beng | 1.2K | 290 | 38.1K | 13.2K | 6.4M | 1.8M |
| guc | Wayuu | Latn | 537 | 214 | 22.9K | 12.5K | 3.4M | 1.8M |
| jam | Jamaican Creole English | Latn | 12.7K | 416 | 68.5K | 15.8K | 25.8M | 1.7M |
| wal | Wolaytta | Latn | 2.6K | 286 | 128K | 14K | 17M | 1.7M |
| jac | Popti' | Latn | 8.2K | 303 | 61.6K | 11.9K | 15.7M | 1.7M |
| bas | Basa (Cameroon) | Latn | 4.2K | 216 | 105.2K | 14.9K | 25.7M | 1.7M |
| gor | Gorontalo | Latn | 1.7K | 303 | 53.3K | 6.5K | 9.4M | 1.7M |
| skr | Saraiki | Arab | 3.8K | 107 | 279.3K | 17.1K | 32.2M | 1.7M |
| nyu | Nyungwe | Latn | 1.2K | 195 | 1.2K | 11K | 7.7M | 1.6M |
| noa | Woun Meu | Latn | 902 | 234 | 902 | 11.5K | 5.2M | 1.6M |
| sda | Toraja-Sa'dan | Latn | 1.6K | 317 | 43.2K | 6.2K | 15.8M | 1.6M |
| gub | Guajajára | Latn | 31.7K | 271 | 160.4K | 25K | 44.7M | 1.6M |
| nog | Nogai | Cyrl | 970 | 419 | 970 | 11K | 2.6M | 1.6M |
| cni | Asháninka | Latn | 1K | 261 | 46K | 14K | 5.9M | 1.6M |
| teo | Teso | Latn | 2.8K | 274 | 131.5K | 13.7K | 15.3M | 1.6M |
| tdx | Tandroy-Mahafaly Malagasy | Latn | 1.7K | 262 | 26.3K | 13.2K | 7M | 1.6M |
| sxn | Sangir | Latn | 587 | 197 | 587 | 9.9K | 3.4M | 1.5M |
| rki | Rakhine | Mymr | 331 | 251 | 331 | 7.8K | 1.6M | 1.5M |
| nr | South Ndebele | Latn | 10.7K | 246 | 10.7K | 11.3K | 49M | 1.5M |
| frp | Arpitan | Latn | 148K | 550 | 3.5M | 8.2K | 535.4M | 1.4M |
| alz | Alur | Latn | 2.2K | 195 | 59.3K | 12.2K | 7.9M | 1.4M |
| taj | E. Tamang | Deva | 146 | 65 | 21.6K | 14.3K | 2.3M | 1.4M |
| lrc | N. Luri | Arab | 42.4K | 587 | 351.9K | 9K | 85.3M | 1.4M |
| cce | Chopi | Latn | 847 | 116 | 23.2K | 11K | 3.3M | 1.3M |
| rn | Rundi | Latn | 8.2K | 323 | 8.2K | 11.1K | 33.2M | 1.3M |
| jvn | Caribbean Javanese | Latn | 1K | 213 | 36.2K | 7.8K | 5.3M | 1.2M |
| hvn | Sabu | Latn | 737 | 200 | 33.9K | 7K | 4.3M | 1.2M |
| nij | Ngaju | Latn | 1K | 183 | 1K | 9.2K | 4.7M | 1.2M |
| dwr | Dawro | Latn | 452 | 215 | 22.1K | 11.1K | 2.2M | 1.2M |
| izz | Izii | Latn | 423 | 237 | 21.7K | 14.5K | 2.1M | 1.1M |
| msm | Agusan Manobo | Latn | 520 | 177 | 520 | 8.6K | 2.5M | 1.1M |
| bus | Bokobaru | Latn | 467 | 322 | 21.4K | 12.1K | 2.1M | 1.1M |
| ktu | Kituba (DRC) | Latn | 3.3K | 144 | 115.5K | 7.8K | 18.5M | 1.1M |
| chr | Cherokee | Cher | 964 | 301 | 33.8K | 7.5K | 4.7M | 1M |
| maz | Central Mazahua | Latn | 585 | 170 | 21.3K | 8.2K | 2.9M | 951.7K |
| tzj | Tz'utujil | Latn | 471 | 136 | 11.1K | 7.3K | 1.9M | 884.2K |
| suz | Sunwar | Deva | 226 | 186 | 226 | 11.3K | 1M | 855.2K |
| knj | W. Kanjobal | Latn | 229 | 126 | 10.1K | 9.2K | 1.1M | 855K |
| bim | Bimoba | Latn | 410 | 40 | 31.1K | 6.3K | 3.2M | 793.4K |
| gvl | Gulay | Latn | 37.9K | 126 | 213K | 6.9K | 141M | 789.2K |
| bqc | Boko (Benin) | Latn | 275 | 228 | 9.8K | 8.2K | 997K | 788.4K |
| tca | Ticuna | Latn | 410 | 117 | 20K | 7.3K | 2.3M | 786K |
| pis | Pijin | Latn | 1.1K | 139 | 62K | 7.2K | 7.7M | 764K |
| prk | Parauk | Latn | 1.1K | 18 | 12.3K | 4.3K | 3.1M | 734.8K |
| laj | Lango (Uganda) | Latn | 6.5K | 144 | 61K | 6.4K | 15.8M | 730.5K |
| mel | Central Melanau | Latn | 119.3K | 103 | 878.4K | 3.7K | 315.2M | 729.6K |
| qxr | Cañar H. Quichua | Latn | 2.6K | 153 | 40.8K | 6.4K | 6.6M | 724K |
| niq | Nandi | Latn | 26.7K | 226 | 26.7K | 4.2K | 72.1M | 716.2K |
| ahk | Akha | Latn | 244 | 77 | 6.2K | 4.1K | 1.3M | 715.5K |
| shp | Shipibo-Conibo | Latn | 874 | 150 | 22.4K | 3.7K | 3.8M | 710.4K |
| hne | Chhattisgarhi | Deva | 3K | 146 | 118.4K | 4.3K | 12M | 697K |
| spp | Supyire Senoufo | Latn | 733 | 123 | 733 | 5.8K | 4.4M | 682.5K |
| koi | Komi-Permyak | Cyrl | 20.7K | 196 | 153.9K | 5K | 17.1M | 664.5K |
| krj | Kinaray-A | Latn | 1.5K | 96 | 54.6K | 3.8K | 7.6M | 616.5K |
| quf | Lambayeque Quechua | Latn | 522 | 86 | 8.4K | 5.2K | 1.5M | 609K |
| luz | S. Luri | Arab | 90.5K | 354 | 1.2M | 6.7K | 329.4M | 590.7K |
| agr | Aguaruna | Latn | 465 | 93 | 16.1K | 3.6K | 2.3M | 554.5K |
| tsc | Tswa | Latn | 12.6K | 82 | 12.6K | 4K | 23.4M | 521.3K |
| mqy | Manggarai | Latn | 69.3K | 119 | 309K | 2.5K | 78.9M | 506.5K |
| gof | Gofa | Ethi | 2.8K | 97 | 33.8K | 5.5K | 5.5M | 506K |

| gbm | Garhwali | Deva | 2.5K | 137 | 50.8K | 3.8K | 9.1M | 499.6K |
|-----|----------|------|------|-----|-------|------|------|--------|
| miq | Mískito | Latn | 236 | 45 | 6.4K | 3.5K | 1.2M | 485.6K |
| dje | Zarma | Latn | 913 | 100 | 40.2K | 3.7K | 4.7M | 480.7K |
| awa | Awadhi | Deva | 5.8K | 126 | 100.1K | 8.4K | 11.1M | 475K |
| bjj | Kanauji | Deva | 830 | 107 | 39.6K | 8K | 3.1M | 439.7K |
| qvz | N. Pastaza Quichua | Latn | 534 | 88 | 6.8K | 3.5K | 1.2M | 438.3K |
| sjp | Surjapuri | Deva | 19K | 31 | 498.2K | 2.9K | 94.3M | 430K |
| tll | Tetela | Latn | 200 | 37 | 200 | 2.7K | 2.2M | 409.8K |
| raj | Rajasthani | Deva | 1.8K | 40 | 1.8K | 5.7K | 7.1M | 405K |
| kjg | Khmu | Laoo | 113 | 84 | 3K | 2.9K | 408.5K | 399K |
| bgz | Banggai | Latn | 32K | 7 | 864.1K | 17K | 79.3M | 391.1K |
| quy | Ayacucho Quechua | Latn | 588 | 78 | 28.1K | 2.7K | 4.5M | 368.2K |
| cbk | Chavacano | Latn | 10.1K | 78 | 43.8K | 2K | 10.3M | 339.3K |
| akb | Batak Angkola | Latn | 1K | 71 | 21.3K | 408 | 5.2M | 337.8K |
| oj | Ojibwa | Cans | 2.5K | 135 | 2.5K | 1.6K | 9.6M | 337.1K |
| ify | Keley-I Kallahan | Latn | 611 | 79 | 19.8K | 2.8K | 2.6M | 334K |
| mey | Hassaniyya | Arab | 14.8K | 127 | 109.9K | 3K | 36.2M | 323.5K |
| ks | Kashmiri | Arab | 5.6K | 51 | 53.9K | 3.3K | 9.4M | 320.9K |
| cac | Chuj | Latn | 212 | 77 | 3.4K | 1.8K | 978.7K | 319.8K |
| brx | Bodo (India) | Deva | 322 | 62 | 5.3K | 2.4K | 1.1M | 304.4K |
| qup | S. Pastaza Quechua | Latn | 169 | 53 | 4.3K | 2.5K | 763.8K | 297.8K |
| syl | Sylheti | Beng | 5.9K | 61 | 5.9K | 4.3K | 21.5M | 293.1K |
| jax | Jambi Malay | Latn | 1.5M | 58 | 30M | 2.3K | 6.8B | 290.2K |
| ff | Fulfulde | Latn | 13.6K | 26 | 150K | 5K | 22.8M | 277.6K |
| ber | Tamazight (Tfng) | Tfng | 2.7K | 79 | 12.6K | 1.2K | 6.4M | 265.9K |
| tks | Takestani | Arab | 63.7K | 127 | 63.7K | 6.8K | 88.9M | 260.8K |
| trp | Kok Borok | Latn | 12.8K | 36 | 12.8K | 1.7K | 29.9M | 257.3K |
| mrw | Maranao | Latn | 11.3K | 29 | 11.3K | 1K | 27.8M | 257.2K |
| adh | Adhola | Latn | 2.6K | 87 | 107.2K | 1K | 14.5M | 254.9K |
| smt | Simte | Latn | 1.4K | 34 | 1.4K | 703 | 6.8M | 245.4K |
| srr | Serer | Latn | 41.1K | 91 | 41.1K | 2.3K | 63.6M | 240.6K |
| ffm | Maasina Fulfulde | Latn | 1.8K | 65 | 30.1K | 2K | 4.6M | 236.1K |
| qvc | Cajamarca Quechua | Latn | 3.4K | 27 | 14.6K | 2.2K | 5M | 233.7K |
| mtr | Mewari | Deva | 1.8K | 11 | 1.8K | 2.2K | 7.6M | 231.1K |
| ann | Obolo | Latn | 464 | 56 | 5K | 1.6K | 760.9K | 215.1K |
| kaa-Latn | Kara-Kalpak (Latn) | Latn | 375.2K | 61.2K | 3.6M | 1.3M | 1.5M | 209.5K |
| aa | Afar | Latn | 39.5K | 32 | 176.1K | 1.3K | 63.3M | 200K |
| noe | Nimadi | Deva | 2K | 22 | 2K | 2.2K | 13.8M | 195.3K |
| nut | Nung (Viet Nam) | Latn | 29K | 67 | 29K | 1.5K | 23.5M | 184.1K |
| gyn | Guyanese Creole English | Latn | 32.6K | 45 | 211.7K | 2.1K | 34.5M | 177.7K |
| kwi | Awa-Cuaiquer | Latn | 382 | 37 | 16.9K | 2.2K | 1.8M | 172.8K |
| xmm | Manado Malay | Latn | 24.5K | 58 | 218.8K | 1.2K | 48.7M | 171.3K |
| msb | Masbatenyo | Latn | 811 | 41 | 811 | 1K | 4.4M | 167.5K |
| el-Latn | Greek (Latn) | Latn | - | - | - | - | - | - |
| doi | Dogri | Deva | - | - | - | - | - | - |
| mtq | Muong | Latn | - | - | - | - | - | - |
| dln | Darlong | Latn | - | - | - | - | - | - |
| cyo | Cuyonon | Latn | - | - | - | - | - | - |
| abs | Ambonese Malay | Latn | - | - | - | - | - | - |
| hi-Latn | Hindi (Latn) | Latn | - | - | - | - | - | - |
| shu | Chadian Arabic | Arab | - | - | - | - | - | - |
| yaq | Yaqui | Latn | - | - | - | - | - | - |
| nyo | Nyoro | Latn | - | - | - | - | - | - |
| cgg | Chiga | Latn | - | - | - | - | - | - |
| sxu | Upper Saxon | Latn | - | - | - | - | - | - |
| mdh | Maguindanaon | Latn | - | - | - | - | - | - |
| rwr | Marwari (India) | Deva | - | - | - | - | - | - |
| xnr | Kangri | Deva | - | - | - | - | - | - |
| mui | Musi | Latn | - | - | - | - | - | - |
| skg | Sakalava Malagasy | Latn | - | - | - | - | - | - |
| ymm | Maay | Latn | - | - | - | - | - | - |
| ctd-Latn | Tedim Chin (Latn) | Latn | - | - | - | - | - | - |
| ayl | Libyan Arabic | Arab | - | - | - | - | - | - |
| kjb | Q'anjob'al | Latn | - | - | - | - | - | - |
| rhg-Latn | Rohingya (Latn) | Latn | - | - | - | - | - | - |
| bmm | N. Betsimisaraka Malagasy | Latn | - | - | - | - | - | - |
| azg | San Pedro Amuzgos Amuzgo | Latn | - | - | - | - | - | - |
| kfy | Kumaoni | Deva | - | - | - | - | - | - |
| bto | Rinconada Bikol | Latn | - | - | - | - | - | - |
| ja-Latn | Japanese (Latn) | Latn | - | - | - | - | - | - |
| mfb | Bangka | Latn | - | - | - | - | - | - |
| ru-Latn | Russian (Latn) | Latn | - | - | - | - | - | - |
| tuf | Central Tunebo | Latn | - | - | - | - | - | - |
| ctg | Chittagonian | Beng | - | - | - | - | - | - |
| pmy | Papuan Malay | Latn | - | - | - | - | - | - |
| xog | Soga | Latn | - | - | - | - | - | - |
| te-Latn | Telugu (Latn) | Latn | - | - | - | - | - | - |
| ber-Latn | Tamazight (Latn) | Latn | - | - | - | - | - | - |
| mdy | Male (Ethiopia) | Ethi | - | - | - | - | - | - |
| az-RU | Azerbaijani (Russia) | Cyrl | - | - | - | - | - | - |
| ta-Latn | Tamil (Latn) | Latn | - | - | - | - | - | - |

| BCP-47 | Name | Script | docs (noisy) | docs (clean) | sents (noisy) | sents (clean) | chars (noisy) | chars (clean) |
|---|---|---|---|---|---|---|---|---|
| clu | Caluyanun | Latn | - | - | - | - | - | - |
| tly-IR | Talysh (Iran) | Arab | - | - | - | - | - | - |
| ng | Ndonga | Latn | - | - | - | - | - | - |
| bzc | S. Betsimisaraka Malagasy | Latn | - | - | - | - | - | - |
| nan-Latn-TW | Min Nan Chinese (Latn) | Latn | - | - | - | - | - | - |
| ml-Latn | Malayalam (Latn) | Latn | - | - | - | - | - | - |
| max | North Moluccan Malay | Latn | - | - | - | - | - | - |
| ar-Latn | Arabic (Latn) | Latn | - | - | - | - | - | - |
| gom-Latn | Goan Konkani (Latn) | Latn | - | - | - | - | - | - |
| bg-Latn | Bulgarian (Latn) | Latn | - | - | - | - | - | - |
| nd | North Ndebele | Latn | - | - | - | - | - | - |
| zyj | Youjiang Zhuang | Latn | - | - | - | - | - | - |
| rkt | Rangpuri | Beng | - | - | - | - | - | - |
| kn-Latn | Kannada (Latn) | Latn | - | - | - | - | - | - |
| zh-Latn | Chinese (Latn) | Latn | - | - | - | - | - | - |
| el-CY | Greek (Cypress) | Grek | - | - | - | - | - | - |
| dcc | Deccan | Arab | - | - | - | - | - | - |
| bgc | Haryanvi | Deva | - | - | - | - | - | - |
| mwr | Marwari | Deva | - | - | - | - | - | - |
| vkt | Tenggarong Kutai Malay | Latn | - | - | - | - | - | - |
| cr-Latn | Cree (Latn) | Latn | - | - | - | - | - | - |
| apd-SD | Sudanese Arabic | Arab | - | - | - | - | - | - |
| trw | Torwali | Arab | - | - | - | - | - | - |
| bn-Latn | Bengali (Latn) | Latn | - | - | - | - | - | - |
| gu-Latn | Gujarati (Latn) | Latn | - | - | - | - | - | - |
| gju | Gujari | Arab | - | - | - | - | - | - |
| sat-Latn | Santali (Latn) | Latn | - | - | - | - | - | - |
| ndc-ZW | Ndau | Latn | - | - | - | - | - | - |
| kmz-Latn | Khorasani Turkish (Latn) | Latn | - | - | - | - | - | - |
| mr-Latn | Marathi (Latn) | Latn | - | - | - | - | - | - |
| en-Cyrl | English (Cyrl) | Cyrl | - | - | - | - | - | - |
| en-Arab | English (Arab) | Arab | - | - | - | - | - | - |
| ms-Arab | Malay (Jawi) | Arab | - | - | - | - | - | - |
| ms-Arab-BN | Malay (Jawi, Brunei) | Arab | - | - | - | - | - | - |
| bhb-Gujr | Bhili | Gujr | - | - | - | - | - | - |
| pa-Arab | Lahnda Punjabi (PK) | Arab | - | - | - | - | - | - |
| syl-Latn | Sylheti (Latn) | Latn | - | - | - | - | - | - |
| ff-Adlm | Fulah | Latn | - | - | - | - | - | - |
| pcm | Nigerian Pidgin | Latn | - | - | - | - | - | - |
| tpi | Tok Pisin | Latn | - | - | - | - | - | - |
| gjk | Kachi Koli | Arab | - | - | - | - | - | - |
| bfy | Bagheli | Deva | - | - | - | - | - | - |
| sgj | Surgujia | Deva | - | - | - | - | - | - |
| nyn | Nyankole | Latn | - | - | - | - | - | - |
| BCP-47 | Name | Script | docs (noisy) | docs (clean) | sents (noisy) | sents (clean) | chars (noisy) | chars (clean) |

## A.2 Filtering Details

**Cursed Substrings** Following is the list of cursed substrings that we used to filter the monolingual data. Here are a few general notes about these strings:

1. low quality sentences ending in the pipe character were very common. (Note: this was not Devanagari-script text using a Danda.)

2. The last few regexes are meant to match A N T S P E A K, *List Case*, and weirdly regular text (for instance, lists of shipping labels or country codes)

Here is the complete list of cursed substrings and cursed regexes, along with the function used for filtering:

```
# this implementation is for demonstration and not very efficient;
# to speed it up, use string inclusion (`in`) instead of regex for
# all but the last four, and for those use a compiled regex.
def is_cursed(s):
  return any(re.findall(curse, s) in s for curse in CURSED_SUBSTRINGS)

CURSED_SUBSTRINGS = [' \u2116', '\ufffd\ufffd\ufffd', '\\|\\s*$', ' nr\\.$',
'aute irure dolor ', ' sunt in culpa qui ', 'orem ipsum ', ' quis nostrud ',
' adipisicing ', ' dolore eu ', ' cupidatat ', 'autem vel eum', 'wisi enim ad',
```

```
' sex ', ' porn ', '\u9ec4\u8272\u7535\u5f71', 'mp3', 'ownload',
'Vol\\.', ' Ep\\.', 'Episode', ' \u0433\\.\\s*$', ' \u043a\u0433\\.\\s*$',
' \u0448\u0442\\.', 'Develop', 'Facebook', ' crusher ', ' xxx ',
' ... ... ... ... ... ... ... ... ...',
' .... .... .... .... .... .... .... .... ....',
' [^ ] [^ ] [^ ] [^ ] [^ ] [^ ] [^ ] [^ ] [^ ]',
', ..,,? ..,,? ..,,? ..,,?',
]
```

**Virama Correction**    Below is the virama substitution code:

```
_VIRAMA_CHARS = (
'\u094d\u09cd\u0a4d\u0acd\u0b4d\u0bcd\u0c4d\u0ccd\u0d3b'
'\u0d3c\u0d4d\u0dca\u0e3a\u0eba\u0f84\u1039\u103a\u1714'
'\u1734\u17d2\u1a60\u1b44\u1baa\u1bab\u1bf2\u1bf3\u2d7f'
'\ua806\ua82c\ua8c4\ua953\ua9c0\uaaf6\uabed\u10a3f\u11046'
'\u1107f\u110b9\u11133\u11134\u111c0\u11235\u112ea\u1134d'
'\u11442\u114c2\u115bf\u1163f\u116b6\u1172b\u11839\u1193d'
'\u1193e\u119e0\u11a34\u11a47\u11a99\u11c3f\u11d44\u11d45'
'\u11d97\u1031\u1057\u1058\u1059\u1056\u1060\u1062\u1068'
'\u1063\u1067\u1068\u1069\u105e\u105f\u1036\u103d\u102d'
'\u102f\u102e\u102d\u1030\u1033\u1034\u1035\u102f\u1032'
'\u102c\u103c\u103d\u103e\u102b\u1037\u1038\u25cc\u25cc'
'\u000a\u1071\u1072\u1073\u1074\u1082\u1083\u1084\u1085'
'\u1086\u1087\u1088\u1089\u108a\u108b\u108c\u108d\u108f'
'\u109a\u109b\u109c\u109d\ua9e5\uaa7b\uaa7c\uaa7d'
)

def remove_viramas(x: str) -> str:
  return '%s' % regex.sub(r' ([%s]) ' % _VIRAMA_CHARS, '\\1', x)
```

**Chinese Porn Filter**    Below is the Chinese porn filter list:

```
zh_pornsignals = [
'caoporn', 'caoprom', 'caopron', 'caoporen', 'caoponrn', 'caoponav', 'caopom',
'caoorn', '99re', 'dy888', 'caopro', 'hezyo', 're99', '4438x', 'zooskool',
'xfplay', '7tav', 'xxoo', 'xoxo', '52av', 'freexx', '91chinese', 'anquye',
'cao97', '538porm', '87fuli', '91pron', '91porn', '26uuu', '4438x', '182tv',
'kk4444', '777me', 'ae86', '91av', '720lu', 'yy6080', '6080yy', 'qqchub',
'paa97', 'aiai777', 'yy4480', 'videossexo', '91free',
'\u4e00\u7ea7\u7279\u9ec4\u5927\u7247',
'\u5077\u62cd\u4e45\u4e45\u56fd\u4ea7\u89c6\u9891',
'\u65e5\u672c\u6bdb\u7247\u514d\u8d39\u89c6\u9891\u89c2\u770b',
'\u4e45\u4e45\u514d\u8d39\u70ed\u5728\u7ebf\u7cbe\u54c1',
'\u9ad8\u6e05\u6bdb\u7247\u5728\u7ebf\u770b',
'\u65e5\u672c\u6bdb\u7247\u9ad8\u6e05\u514d\u8d39\u89c6\u9891',
'\u4e00\u7ea7\u9ec4\u8272\u5f55\u50cf\u5f71\u7247',
'\u4e9a\u6d32\u7537\u4eba\u5929\u5802',
'\u4e45\u4e45\u7cbe\u54c1\u89c6\u9891\u5728\u7ebf\u770b',
'\u81ea\u62cd\u533a\u5077\u62cd\u4e9a\u6d32\u89c6\u9891',
'\u4e9a\u6d32\u4eba\u6210\u89c6\u9891\u5728\u7ebf\u64ad\u653e',
'\u8272\u59d1\u5a18\u7efc\u5408\u7ad9',
'\u4e01\u9999\u4e94\u6708\u556a\u556a',
'\u5728\u7ebf\u89c6\u9891\u6210\u4eba\u793e\u533a',
'\u4e9a\u6d32\u4eba\u6210\u89c6\u9891\u5728\u7ebf\u64ad\u653e',
'\u4e45\u4e45\u56fd\u4ea7\u81ea\u5077\u62cd',
```

```
'\u4e00\u672c\u9053',
'\u5927\u9999\u8549\u65e0\u7801',
'\u9999\u6e2f\u7ecf\u5178\u4e09\u7ea7',
'\u4e9a\u6d32\u6210\u5728\u4eba\u7ebf\u514d\u8d39\u89c6\u9891',
'\u5929\u5929\u8272\u7efc\u5408\u7f51',
'\u5927\u9999\u8549\u4f0a\u4eba\u4e45\u8349',
'\u6b27\u7f8e\u4e00\u7ea7\u9ad8\u6e05\u7247',
'\u5929\u5929\u9c81\u591c\u591c\u556a\u89c6\u9891\u5728\u7ebf',
'\u514d\u8d39\u9ec4\u7247\u89c6\u9891\u5728\u7ebf\u89c2\u770b',
'\u52a0\u6bd4\u52d2\u4e45\u4e45\u7efc\u5408',
'\u4e45\u8349\u70ed\u4e45\u8349\u5728\u7ebf\u89c6\u9891',
'\u97e9\u56fd\u4e09\u7ea7\u7247\u5927\u5168\u5728\u7ebf\u89c2\u770b',
'\u9752\u9752\u8349\u5728\u7ebf\u89c6\u9891',
'\u7f8e\u56fd\u4e00\u7ea7\u6bdb\u7247',
'\u4e45\u8349\u5728\u7ebf\u798f\u5229\u8d44\u6e90',
'\u556a\u556a\u556a\u89c6\u9891\u5728\u7ebf\u89c2\u770b\u514d\u8d39',
'\u6210\u4eba\u798f\u5229\u89c6\u9891\u5728\u7ebf\u89c2\u770b',
'\u5a77\u5a77\u6211\u53bb\u4e5f',
'\u8001\u53f8\u673a\u5728\u7ebf\u56fd\u4ea7',
'\u4e45\u4e45\u6210\u4eba\u89c6\u9891',
'\u624b\u673a\u770b\u7247\u798f\u5229\u6c38\u4e45\u56fd\u4ea7',
'\u9ad8\u6e05\u56fd\u4ea7\u5077\u62cd\u5728\u7ebf',
'\u5927\u9999\u8549\u5728\u7ebf\u5f71\u9662',
'\u65e5\u672c\u9ad8\u6e05\u514d\u8d39\u4e00\u672c\u89c6\u9891',
'\u7537\u4eba\u7684\u5929\u5802\u4e1c\u4eac\u70ed',
'\u5f71\u97f3\u5148\u950b\u7537\u4eba\u8d44\u6e90',
'\u4e94\u6708\u5a77\u5a77\u5f00\u5fc3\u4e2d\u6587\u5b57\u5e55',
'\u4e9a\u6d32\u9999\u8549\u89c6\u9891\u5728\u7ebf\u64ad\u653e',
'\u5929\u5929\u556a\u4e45\u4e45\u7231\u89c6\u9891\u7cbe\u54c1',
'\u8d85\u78b0\u4e45\u4e45\u4eba\u4eba\u6478\u4eba\u4eba\u641e',
]
```

## A.3 Other issues fixed after the self-audit

**Consulting Language Speakers** For a few languages, we had strong suspicions that the text was noisy or spurious, but were unable to acertain the quality of the data. In these cases we asked a native speaker to audit the data. Based on their recommendations, we did the following:

1. zh, zh_Latn: This resulted in the special filters described below.
2. en_Arab, tly_IR: This data was found to boilerplate, so we removed this data.
3. fa, bho: No changes were made.

**Language Renames and Merges** For several languages, we found that (mostly by checking URLs) the corpora were in languages different from the LangID predictions. This led to the following changes:

1. dty renamed to zxx-xx-dtynoise, aka a "language" of noise. This is mainly mis-rendered PDFs and may have practical applications for denoising, or for decoding such garbled PDFs.
2. fan renamed to bum
3. cjk merged into the gil dataset
4. bjj merged into the awa dataset
5. ss-SZ renamed to ss – this was a result of inconsistent data labels.

## A.4 Monolingual Data Details

Notes from rounds 2 and 3 of the self-audit can be seen in Table 9. Some of these notes may refer to previous, less filtered versions of the data, especially those with a "r1" (meaning "round 1"). Some of them however do have some useful information about quirks of problems with the current dataset. The overall statistics of MADLAD-400 are in Table 8.

Table 9: Notes that we made about individual samples while auditing them. Some languages have notes from the earlier round of auditing in parentheses, e.g. '(r1: get this checked by Hindi speaker)'. Notes from Round 0, which were used to find cursed substrings, were not kept.

| BCP-47 | notes |
|---|---|
| aa | some pretty bad data but also some good data. filter on "Woo" (case sensitive) (r1: ok) |
| abs | all short nonsense remove (r1: ok) |
| abt | fine; bible (r1: ok) |
| ace | good; bible (r1: ok) |
| acf | good; bible (r1: ok) |
| ach | good; bible (r1: ok) |
| ada | good; bible; likely mixed with gaa (r1: ok but odd character usage LATIN CAPITAL LETTER OPEN O when it should be lower case in the middle of words) |
| adh | good; bible (r1: ok, lots bible) |
| ady | good (r1: ok but weird boilerplate) |
| af | good (r1: ok) |
| agr | good; bible (r1: ok; some AL in Arabic script) |
| ahk | good; bible; crazy diacritics (r1: ok but weird: lots of u748 MODIFIER LETTER VOICING) |
| ak | good; much but not all bible (r1: ok) |
| akb | good; bible (r1: empty) |
| alt | WAIT THIS IS AMAZING IT IS ACTUALLY ALTAI! e.g. from urls like https://altaicholmon.ru/2020/02/28/jarashty-la-jajaltany-jarkyndu-lekeri/ (r1: ok but there are just lots of numbers...not very clean) |
| alz | good; bible (r1: ok; bible) |
| am | good (r1: ok) |
| amu | good; bible; crazy diacritics (r1: empty) |
| ang | much noise but some good Old English in there! (r1: ok; wikipedia; one document that is just "lastfootwear.com" 1M times) |
| ann | good; all from wikimedia incubator (r1: ok) |
| apd-SD | terribly questionable; probably remove (r1: maybe ok, but looks like lots of template....maybe remove) |
| ape | good; bible (r1: remove) |
| ar | good (r1: ok) |
| ar-Latn | terrible, 0pct correct, remove (r1: remove) |
| arn | good; bible (r1: ok) |
| as | good (r1: ok) |
| av | good (r1: ok) |
| awa | OK; should be used with caution and suspicion (r1: remove) |
| awa | all bible in awadhi (awa). Renamed from bjj (r1: remove) |
| ay | good; mix of bible and other news sources (r1: ok but very noisy) |
| ayl | remove. not ayl. (r1: uh this is all Arabic with "homo" in English...remove?) |
| az | good (r1: ok) |
| az-RU | good; a lot of JW (r1: ok) |
| azg | 70pct short noise; 30pct good bible (r1: empty) |
| ba | ok (r1: ok) |
| ban | ok bible (r1: ok) |
| bas | ok; has some fun blog stuff! (r1: empty) |
| bbc | ok (r1: ok) |
| bci | ok bible (r1: ok; bible) |
| be | ok (r1: ok) |
| ber | ok great! (r1: ok; Mixed in French) |
| ber-Latn | ok (r1: ok) |
| bew | mostly blogs. i have no way of knowing if this is standard indonesian or not (r1: ok; noisy) |
| bfy | very bad. remove unless it looks better after filtering short docs; remove (r1: remove) |
| bg | ok (r1: ok) |
| bg-Latn | ok (r1: ok but questionable...Slavic speaker review needed) |
| bgc | super sketch. Remove unless short doc filter leaves some. remove (r1: very questionable....Hindi speaker review) |
| bgp | almost all ur-Latn. consider removing or renaming (r1: very questionable. Remove? mainly ur-Latn) |
| bgz | idk maybe ok but probably bad (r1: remove. Wow, this is amazing. It is in all sorts of languages – the only thing they share is that they each have like 500 question marks) |
| bhb-Gujr | bad. remove. all junk gu. (r1: remove; great noise?) |
| bho | mostly from anjoria.com. Ankur reviews and says that it looks like valid Bhojpuri for the most part (r1: questionable but ok?) |
| bi | good! fun! (r1: ok) |
| bik | ok. keep in mind the bik vs bcl issue. (r1: ok) |
| bim | good; bible (r1: empty) |
| bm | good (r1: ok but these headers are LONG) |
| bmm | terrible. filter on short and reevaluate (r1: remove) |
| bn | ok (r1: ok) |
| bn-Latn | ok (r1: ok) |
| bo | needs some serious script filtering. but there is some ok data in there. (r1: ok) |
| bqc | ok; bible (r1: ok but too short?) |
| br | ok after shortfilter (r1: ok) |
| bru | ok; bible (r1: ok) |
| brx | quite good! (r1: ok but questionable...Hindi speaker review?) |
| bs | good (r1: ok) |
| bto | bad; remove unless short filter keeps enough (r1: empty) |
| bts | ok; mostly bible (r1: ok) |
| btx | ok probably (r1: ok) |
| bua | ok (r1: ok) |
| bum | ok bible; but technically wrong language. Data is in Bulu, not Fang, though they are closely related, so ranamed from "fan" |

| | |
|---|---|
| bus | ok; bible; about 50bzc (r1: ok bible) |
| bzj | ok bible (r1: ok) |
| ca | ok (r1: ok i guess....but is it actually italian?) |
| cab | ok jw (r1: ok) |
| cac | ok bible (r1: ok bible) |
| cak | ok bible (r1: ok bible) |
| cbk | ok bible; not Spanish (r1: remove; all Spanish) |
| cce | ok jw (r1: empty) |
| ce | ok (r1: ok) |
| ceb | ok (r1: ok) |
| cfm | ok mostly from chinland.co (r1: ok) |
| cgg | rather noisy but potentialy ok. not sure if WL or not (r1: ok) |
| ch | ok; not sure about WL (r1: ok) |
| chk | ok bible (r1: ok bible) |
| chm | ok; fyi watch out for yandex translationese (r1: ok) |
| chr | ok bible (r1: ok bible) |
| ckb | ok (r1: ok) |
| clu | ok bible (r1: ok) |
| cnh | good, some local news! not sure if WL (r1: ok) |
| cni | ok; bible; lots of mixed in content in not,cob,cpc,arl (r1: ok) |
| co | ok;l i suspect lots of MT (r1: ok i guess?) |
| cr-Latn | noise and lorem ipsom. But some ok Cree text. (r1: mostly Lorem Ipsom. remove? Or release with note? there is some plausible stuff here too.) |
| crh | ok (r1: ok but review with russian speaker as it could be russian....) |
| crs | ok (r1: ok) |
| cs | ok (r1: ok) |
| ctd-Latn | ok; from some local news? (r1: ok) |
| ctg | probably terrible probably remove (r1: very questionable....remove?) |
| ctu | ok bible (r1: ok bible) |
| cuk | ok bible (r1: ok bible) |
| cv | good (r1: ok) |
| cy | ok after shortfilter; OK (r1: ok) |
| cyo | terrifying noise; remove (r1: empty) |
| da | ok (r1: ok) |
| dcc | remove (r1: empty) |
| de | ok (r1: ok) |
| din | ok after short doc filter (r1: ok but LONG headers uh oh) |
| dje | ok; mostly but not all bible (r1: ok bible) |
| djk | ok; bible+jw (r1: empty) |
| dln | ok bible (r1: empty) |
| doi | ok actually nice! (r1: sus; review by hindi speaker) |
| dov | ok bible + jw (r1: ok) |
| dtp | ok; mostly from www.newsabahtimes.com.my (r1: ok) |
| dv | good (r1: ok) |
| dwr | ok; bible; mixed script (r1: empty) |
| dyu | ok bible (r1: empty) |
| dz | ok; hidden parallel text; maybe actually bo; mainly buddhist (r1: ok; mixed dz-Latn) |
| ee | good; mostly religious (r1: ok bible) |
| el | ok (r1: ok) |
| el-CY | bad (r1: v suspicious; mainly comma lists or boilerplate; remove) |
| el-Latn | good; a lot of old content! (r1: ok) |
| emp | ok bible (r1: ok) |
| en | ok (r1: ok) |
| en-Arab | Ali reviewed; this is not good data. remove. (r1: idk review w/Arabic reader) |
| en-Cyrl | ok ... some fr-Cyrl too and maybe others (r1: OMG LOL yes ok) |
| enq | ok bible (r1: ok bible) |
| eo | ok; likely a lot of MT (r1: ok) |
| es | good (r1: ok) |
| et | ok (r1: ok) |
| eu | ok (r1: ok; lots of poetry?) |
| fa | consulted Ali; he says it's ok (r1: ok) |
| ff | ok after shortfilter (r1: some noise but some nice stuff! ok!) |
| ff-Adlm | good (r1: ok sweet) |
| ffm | ok bible; mixed fulfulde dialects; consider mergind with ff (r1: ok but idk the dialect) |
| fi | ok (r1: ok but lotsa headers) |
| fil | ok more bible than expected for such a major language (r1: ok pls note in release that this is the same as tl) |
| fip | ok jw ; but wrong language. mostly Mambwe-Lungu and Bemba, not Fipu (mgr+bem vs fip) (r1: ok bible) |
| fj | ok (r1: ok bible lotsa noise) |
| fo | good (r1: ok TODO check that this is not icelandic review) |
| fon | ok mostly jw but not all (r1: ok bible) |
| fr | ok (r1: ok) |
| frp | fair amount from wikipedia. (r1: remove; all noise + hashtags) |
| fy | ok plausible but i bet there is a lot of Dutch in there (r1: ok) |
| ga | ok some en noise (r1: ok) |
| gag | has 1-2 cyrillic examples with small amts of arabic script noise (r1: ok) |
| gbm | ok (r1: ok) |
| gd | ok (r1: ok; but barely) |
| gil | empty; but merged in data in "cjk" (r1: empty) |
| gil | this is all in gil (Kiribati). merged into "gil" (r1: empty) |
| gjk | empty remove (r1: empty) |
| gju | remove short boilerplate (r1: empty) |

| | |
|---|---|
| gl | ok (r1: ok) |
| gn | ok some broken characters some bible (r1: ok) |
| gof | ok some bible (r1: empty) |
| gom | ok (r1: ok) |
| gom-Latn | filter on really short boilerplate in en; some porn; after: ok very noisy ; some ok stuff ; release with disclaimer (r1: ok) |
| gor | ok bible (r1: ok) |
| grc | warning: this is likely polyphonic greek, not ancient greek (r1: ok but idk diff between ancient and modern greek) |
| gsw | wtf is happening here; keep with disclaimer; STILL BOILERPLATE (r1: ok but idk diff between gsw and de) |
| gu | ok (r1: ok some en boilerplate) |
| gu-Latn | filter short en boillerplate and repetitive sentences (r1: lots of social media pages and some porn) |
| gub | ok bible (r1: empty) |
| guc | ok bible (r1: ok) |
| guh | ok bible (r1: ok) |
| gui | ok bible (r1: ok) |
| gv | filter short repetitive sentenecs; still same but keep (r1: ok) |
| gvl | filter short boilerplate mostly bible (r1: ok) |
| gym | ok biblle (r1: ok) |
| gyn | remove boilerplate and porn (r1: remove) |
| ha | ok (r1: ok) |
| haw | ok scam tv products (r1: ok but filter u65533 REPLACEMENT CHARACTER) |
| hi | ok some porn (r1: ok but some en boilerplate) |
| hi-Latn | filter porn this is half porn (r1: ok but some hi and en) |
| hif | ok some en noise and religious (r1: ok it is in Latin) |
| hil | ok some en boilerplate (r1: ok) |
| hmn | ok (r1: ok) |
| hne | ok (r1: ok) |
| ho | ok (r1: ok but but split between wiki boilerplate and actual content) |
| hr | ok (r1: ok) |
| ht | ok (r1: ok) |
| hu | ok (r1: ok) |
| hui | ok some bible (r1: ok bible) |
| hus | ok bible (r1: some wiki boilerplate) |
| hvn | ok religioous text (r1: ok bible) |
| hy | ok (r1: ok) |
| iba | ok jw data (r1: ok) |
| ibb | ok bible and repeated @ (r1: ok but bible and some repeated lines) |
| id | ok (r1: ok) |
| ify | ok bible (r1: empty) |
| ig | ok (r1: ok) |
| ilo | ok some bible (r1: some repetitive content) |
| inb | ok bible (r1: remove; it's a single bible doc lol) |
| is | ok (r1: ok) |
| iso | ok jw (r1: ok) |
| it | ok (r1: ok) |
| iu | filter script some is en rest is iu script (r1: ok filter latin script) |
| ium | filter out zh (r1: remove mostly en) |
| iw | ok (r1: ok has some codemixing because of boilerplate) |
| izz | ok bible (r1: empty) |
| ja | ok a little en mixed in (r1: ok but some porn) |
| ja-Latn | remove maybe low quality short and repeated (r1: ok some noise that is manga pages in english) |
| jac | ok bible (r1: remove 'home loan' repeated over and over) |
| jam | ok bible (r1: ok) |
| jax | filter mostly text.medjugorje.ws boilerplate (r1: remove) |
| jiv | ok bible |
| jv | ok (r1: ok) |
| jvn | ok bible (r1: ok) |
| ka | ok (r1: ok) |
| kaa | ok (FYI cyrllic) (r1: ok) |
| kaa-Latn | ok urls are .ru or .kz (r1: ok) |
| kac | ok (r1: ok) |
| kbd | ok many .ru (r1: ok some repetitive text and en noise) |
| kbp | not sure if right script wiki says latin (r1: ok) |
| kek | ok jw bible (r1: ok bible) |
| kfy | filter virama issue (r1: ok) |
| kg | ok bible jw (r1: ok) |
| kha | ok (r1: ok some repetitive boilerplate) |
| kj | ok (r1: filter english out) |
| kjb | ok bible (r1: empty) |
| kjg | ok bible (r1: empty) |
| kjh | ok .ru domain (r1: ok) |
| kk | ok (r1: ok) |
| kl | ok (r1: ok) |
| km | ook (r1: ok) |
| kmb | ok bible jw (r1: ok) |
| kmz-Latn | ok soome ar script noise (r1: ok) |
| kn | ok (r1: ok) |
| kn-Latn | filter en noise of karnataka govt websites (r1: filter porn there is too much porn and repetitive content) |
| knj | ok bible (r1: empty) |
| ko | ok (r1: ok) |
| koi | ok (r1: ok) |
| kos | ok lds bible (r1: ok bible) |

| | |
|---|---|
| krc | ok (r1: ok some repetitive content) |
| kri | ok boilerplate noise bible jw (r1: remove repetitive) |
| ks | ok shorter docs (r1: ok) |
| ksd | ok bible (r1: ok bible) |
| ksw | ok bible (r1: ok) |
| ktu | ok bible jw (r1: ok) |
| ku | ok (r1: ok) |
| kum | ok (r1: ok) |
| kv | ok a lil boilerplate vibes (r1: ok) |
| kw | ok short boilerplate bible wiki; ok some porn (r1: ok filter english) |
| kwi | ok bible (r1: ok) |
| ky | ok (r1: ok) |
| la | ok some broken chars |
| laj | ok bible |
| lb | ok shorter text; ok AFTER |
| lg | ok lot of www.bukedde.co.ug in this |
| lhu | ok bible |
| ln | ok bible jw |
| lo | ok many entities in latin script |
| lrc | ok |
| lt | ok |
| ltg | ok mostly www.lakuga.lv |
| lu | ok jw |
| lus | ok |
| luz | terrible; remove |
| lv | ok |
| mad | remove mostly short text |
| mag | ok fix virama issue |
| mai | ok mild amounts of en noise |
| mak | ok bible |
| mam | ok bible jw |
| mas | ok some amount of bible |
| max | remove short some ru |
| maz | ok bible jw |
| mbt | ok bible |
| mdf | ok some short docs |
| mdh | filter porn short text and repetitive boilerplate |
| mdy | ok bible |
| mel | remove noisy en |
| meo | ok mostly blogs |
| meu | ok bible |
| mey | mostly short and noisy borderline |
| mfb | remove short boilerplate |
| mfe | ok mostly bible maybe some french creole short doc noise |
| mg | ok some bible jw |
| mgh | ok bible jw |
| mh | ok jw lds |
| mi | ok |
| min | ok mostly wiki and bible |
| miq | ok |
| mk | ok |
| mkn | ok bible |
| ml | ok |
| ml-Latn | ok some short docs |
| mn | ok |
| mni | ok |
| mnw | remove en noise and boilerplate |
| mps | ok bible |
| mqy | bible remove short docs |
| mr | ok fix virama |
| mr-Latn | remove mostly porn and short docs |
| mrj | remove short docs; ok |
| mrw | ok remove short docs |
| ms | ok |
| ms-Arab | ok mostly utusanmelayu website |
| ms-Arab-BN | ok not sure if same as ms-Arab |
| msb | ok bible |
| msi | ok filter short docs |
| msm | ok bible |
| mt | ok |
| mtq | remove short doc repetitive |
| mtr | ok fix virama remove en noise |
| mui | remove short docs |
| mwr | filter short docs fix virama |
| my | filter noise and en fix virama |
| myv | maybe has .ru urls |
| nan-Latn-TW | ok |
| nd | ok |
| ndc-ZW | ok |
| ne | ok |
| new | ok |

| | |
|---|---|
| ng | ok |
| ngu | ok |
| nhe | ok |
| nia | ok |
| nij | ok |
| niq | ok |
| nl | ok |
| nnb | ok |
| no | ok |
| noa | ok |
| noe | ok |
| nog | ok |
| nr | ok |
| nso | ok |
| nut | ok |
| nv | ok |
| ny | ok |
| nyn | ok |
| nyo | ok |
| nyu | ok |
| nzi | ok |
| oc | ok |
| oj | ok |
| om | ok |
| or | ok |
| os | ok |
| otq | ok |
| pa | ok |
| pa-Arab | ok |
| pag | bible |
| pam | remove |
| pap | ok |
| pau | ok |
| pck | ok |
| pcm | ok |
| pis | bible |
| pl | ok |
| pmy | remove |
| pon | bible |
| ppk | bible |
| prk | ok |
| ps | ok |
| pt | ok |
| qu | ok |
| qub | bible |
| quc | bible |
| quf | bible |
| quh | bible |
| qup | bible |
| quy | bible |
| qvc | bible |
| qvi | bible |
| qvz | bible |
| qxr | bible |
| raj | ok |
| rcf | ok |
| rhg-Latn | remove |
| rki | ok |
| rkt | ok |
| rm | ok |
| rmc | ok |
| rn | bible |
| ro | ok |
| rom | bible |
| ru | ok |
| ru-Latn | ok |
| rw | ok |
| rwo | bible |
| rwr | remove |
| sa | ok |
| sah | ok |
| sat-Latn | good! all from local news sources |
| sd | good |
| sda | ok bible |
| se | good |
| seh | ok jw |
| sg | ok jw |
| sgj | remove |
| shn | mostly English boilerplate. filter by latin text before releasing |
| shp | ok bible |
| shu | quite questionable. prob remove |

| | |
|---|---|
| si | good |
| sja | ok bibe |
| sjp | terible; probably remove; check again after short filter |
| sk | ok |
| skg | terrible; remove |
| skr | ok; some pnb mixed in |
| sl | ok |
| sm | ok |
| smt | ok bible but lots of different bibles! |
| sn | ok |
| so | good |
| spp | ok bible |
| sq | good |
| sr | ok |
| srm | ok; bible + jw |
| srn | ok bible + jw |
| srr | remove; englishboilerplate |
| ss | good mix of data ; renamed from "ss" |
| st | ok |
| stq | ok i think ? |
| su | good |
| sus | hella sus jk ok bible |
| suz | ok bible |
| sv | ok |
| sw | ok |
| sxn | ok bible ; also wild diacritics |
| sxu | rvisit after shortfilter |
| syl | idk maybe ok ? |
| syl-Latn | revist or remove after shortfilter |
| syr | good; practictitioners should keep dialect in mind. |
| ta | ok |
| ta-Latn | good text .... but pornographic, like all Indic-Latn datasets |
| tab | idk plausibly ok |
| taj | ok bible |
| tbz | good mostly bible but not all |
| tca | ok bible + jw |
| tcy | good; mostly wikipedia; likely some konkani mixed in |
| tdx | ok jw |
| te | ok a lot of weirdly low quality looking content like commerce |
| te-Latn | great good text....but all pornographic stories + blogs, like all Indic-Latn text |
| teo | ok bible |
| tet | good ; actually a lot of fun data! |
| tg | good |
| th | ok |
| ti | ok; poor tigray |
| tiv | ok jw |
| tk | ok; a few weird docs |
| tks | ok bible but again i think some mixed dialects |
| tlh | ok, but why tf are there websites in klingon? all MT ? |
| tll | ok jw |
| tly-IR | deeply sus; remove after shortfilter |
| tn | good |
| to | good ; news bible government |
| toj | ok jw |
| tpi | empy |
| tr | ok |
| trp | good ; lots of random stuff |
| trw | sus; remove |
| ts | good |
| tsc | ok |
| tsg | much noise but some good data too! |
| tt | good plus some nonunicode misrendered PDF |
| tuc | ok bible |
| tuf | ok bible |
| tvl | ok jw |
| twu | ok bible, but also i think it's lots of mixed similar dialects |
| tyv | ok fun stuff plus some russian noise i think |
| tyz | ok bible bu again i think some mixed dialects |
| tzh | ok jw |
| tzj | ok bible |
| tzo | ok bible + jw |
| ubu | ok bible |
| udm | ok |
| ug | ok |
| uk | ok |
| ur | ok |
| uz | ok some cyrllic noise |
| ve | ok mostly bible jw |
| vec | very noisy has wiki from other langs and .it websites so not sure if vec |
| vi | ok |
| vkt | 1 doc remove |

| | |
|---|---|
| wa | ok lots of wiki stuff |
| wal | ok bible + jw |
| war | ok but v sus. Pls filter out wikipedia |
| wo | ok; mostly bible. PS i have found that Wolof web-crawled data is often bad, so pls give an extra look if you want |
| xal | ok has .ru sites though |
| xh | ok |
| xmm | very noisy lots of dj tiktok and peppa pig repeated |
| xnr | ok maybe fix virama though it seems fine |
| xog | ok bible and stories |
| yap | ok |
| yaq | remove |
| yi | ok |
| ymm | remove |
| yo | ok |
| yua | ok |
| yue | pretty low quality; mostly not Canto |
| za | revisit after shortfilter |
| zap | ok JW. PS pls note that at least some Zapotec speakers view it as one language, not as a million dialects like ISO does |
| zh | mixed simplified and trad; also much porn |
| zh-Latn | revisit after shortfilter |
| zne | ok jw |
| zu | good |
| zxx-xx-dtynoise | BEAUTIFUL NOISE rename but keep as beautiful xample. (was called "dty") |
| zyj | deeply bad data .. revisit after shortfilter |
| zza | good ; also pls note that the Zazaki community is often super into NLP for Zazaki |

## A.5 Parallel Data Details

To create the dataset described in Section , we use the data sources described in Table 10. After preprocessing, we obtain a dataset with a total of 157 different languages and 4.1B sentence pairs that vary from en-es with 280.3M sentence pairs to zu-en with 7959 sentence pairs. The list of language pairs along with the associated data count is available along with the model checkpoints.

Table 10: The various data sources used to create the parallel data used to train our MT models with the number of available languages and language pairs. (*for NewsCommentary v14 we only use Kazakh (kk) data)

| Dataset | Version | # Language Pairs | # Languages |
|---|---|---|---|
| Europarl [34] | v7 | 40 | 21 |
| | v9 | 12 | 8 |
| Paracrawl [19] | - | 64 | 41 |
| TED57 [65] | - | 114 | 58 |
| Tanzil [59] | - | 1560 | 40 |
| NewsCommentary [1] | v10 | 6 | 3 |
| | v14* | 2 | 1 |
| | v16 | 200 | 15 |
| Wikimatrix [54] | - | 1620 | 85 |
| Wikititles [1] | v2 | 12 | 14 |
| OPUS100 [66] | - | 198 | 100 |
| SETimes [59] | - | 90 | 10 |
| UNv1.0 [69] | - | 20 | 5 |
| Autshumato [25] | - | 10 | 5 |
| PMIndia [26] | v1 | 13 | 13 |
| CVIT [49] | MKB (v0), PIB (v1.3) | 110 | 11 |
| Inuktitut [29] | - | 2 | 1 |
| NLPC [21] | - | 2 | 1 |
| JESC [51] | - | 2 | 1 |
| KFTT [45] | v1.0 | 2 | 1 |
| ASPEC [44] | - | 2 | 1 |

## A.6 Language Codes

The specifics of the language code changes described in Section 4.1 that we made are as follows:

Table 11: Preprocessed multiway data divided by target language in decreasing order of number of # Sentence Pairs.

| Language (Code) | # Sentence Pairs | Language (Code) | # Sentence Pairs |
|---|---|---|---|
| English (en) | 6825073150 | Malagasy (mg) | 1190715 |
| Spanish (es) | 630962081 | Punjabi (pa) | 1122414 |
| German (de) | 542301841 | Hausa (ha) | 952163 |
| French (fr) | 540506439 | Latin (la) | 882038 |
| Portuguese (pt) | 355008897 | Inuktitut (iu) | 841459 |
| Italian (it) | 283297447 | Myanmar (Burmese) (my) | 759196 |
| Dutch (nl) | 272264696 | Walloon (wa) | 706400 |
| Swedish (sv) | 183450145 | Uzbek (uz) | 695915 |
| Czech (cs) | 166419080 | Luxembourgish (lb) | 645610 |
| Polish (pl) | 154989834 | Assamese (as) | 623321 |
| Mandarin Chinese (zh) | 147268699 | Pashto (ps) | 606852 |
| Danish (da) | 145764833 | Armenian (hy) | 603397 |
| Hungarian (hu) | 136980377 | Sindhi (sd) | 597211 |
| Russian (ru) | 134568973 | Northern Sami (se) | 585304 |
| Romanian (ro) | 117227121 | Bashkir (ba) | 551617 |
| Slovak (sk) | 102889505 | Amharic (am) | 548828 |
| Finnish (fi) | 100468712 | Somali (so) | 543643 |
| Greek (el) | 97855862 | Dhivehi (dv) | 543145 |
| Arabic (ar) | 83364997 | Kurdish (Kurmanji) (ku) | 483477 |
| Bulgarian (bg) | 68424658 | French (Canada) (fr_CA) | 472160 |
| Lithuanian (lt) | 66562681 | Odia (Oriya) (or) | 465494 |
| Slovenian (sl) | 65876962 | Faroese (fo) | 434288 |
| Latvian (lv) | 59061871 | Kannada (kn) | 417255 |
| Norwegian (no) | 57783339 | Kinyarwanda (rw) | 386133 |
| Estonian (et) | 47403866 | Wu Chinese (wuu) | 369891 |
| Japanese (ja) | 42480849 | Lombard (lmo) | 366147 |
| Korean (ko) | 33987656 | Egyptian Arabic (arz) | 353218 |
| Catalan (ca) | 28627098 | Uyghur (ug) | 337582 |
| Croatian (hr) | 28475191 | Limburgan (li) | 304279 |
| Turkish (tr) | 26508188 | Aragonese (an) | 251004 |
| Ukrainian (uk) | 26258530 | Sicilian (scn) | 249682 |
| Icelandic (is) | 24073502 | Dzongkha (dz) | 248172 |
| Persian (fa) | 23237370 | Meiteilon (Manipuri) (mni) | 242426 |
| Indonesian (id) | 19860783 | Maori (mi) | 234833 |
| Vietnamese (vi) | 18803243 | Nuer (nus) | 231194 |
| Hebrew (he) | 17171755 | Magahi (mag) | 230839 |
| Macedonian (mk) | 15900226 | Bhojpuri (bho) | 230378 |
| Irish (ga) | 14939080 | Shan (shn) | 229967 |
| Galician (gl) | 14305858 | Friulian (fur) | 228928 |
| Serbian (sr) | 13989182 | Kashmiri (ks) | 228580 |
| Albanian (sq) | 13904626 | Sardinian (sc) | 227585 |
| Basque (eu) | 13755514 | Kanuri (kr_Arab) | 227549 |
| Maltese (mt) | 12583626 | Dinka (din) | 227233 |
| Esperanto (eo) | 11431440 | Venetian (vec) | 225065 |
| Hindi (hi) | 11044239 | Chhattisgarhi (hne) | 224085 |
| Bosnian (bs) | 10906114 | Ligurian (lij) | 222513 |
| Serbo-Croatian (sh) | 10592606 | Central Atlas Tamazight (tzm) | 222365 |
| Bengali (bn) | 8394639 | Tamasheq (taq_Tfng) | 220907 |
| Sinhala (si) | 7363378 | Dari (prs) | 220899 |
| Thai (th) | 6623168 | Banjar (bjn) | 220753 |
| Malayalam (ml) | 5995504 | Achinese (ace_Arab) | 220651 |
| Tamil (ta) | 5552495 | Tamasheq (taq) | 219708 |
| Marathi (mr) | 5517596 | Banjar (bjn_Arab) | 219020 |
| Telugu (te) | 5379834 | Achinese (ace) | 217062 |
| Malay (ms) | 5241831 | Bambara (bm) | 216698 |
| Filipino (fil) | 4532364 | Balinese (ban) | 215965 |
| Urdu (ur) | 4526509 | Moroccan Arabic (ary) | 214226 |
| Taiwanese Mandarin (zh_Hant) | 3782804 | Nigerian Fulfulde (fuv) | 209659 |
| Swahili (sw) | 3220515 | Silesian (szl) | 209224 |
| Nepali (ne) | 2833740 | Kashmiri (ks_Deva) | 208575 |
| Norwegian Nynorsk (nn) | 2726466 | Buginese (bug) | 207209 |
| Azerbaijani (az) | 2353731 | Guarani (gn) | 201241 |
| Kazakh (kk) | 2153356 | Latgalian (ltg) | 200302 |
| Tajik (tg) | 2005810 | Crimean Tatar (crh_Latn) | 192342 |
| Low German (nds) | 1942746 | Kanuri (kr) | 191037 |
| Occitan (oc) | 1934958 | Scottish Gaelic (gd) | 162882 |
| Georgian (ka) | 1882623 | Bavarian (bar) | 159419 |
| Belarusian (be) | 1808663 | Javanese (jv) | 144974 |
| Tatar (tt) | 1763087 | Mongolian (mn) | 132599 |
| Western Frisian (fy) | 1500410 | Zulu (zu) | 130485 |
| Afrikaans (af) | 1459139 | Kyrghyz (ky) | 128832 |
| Breton (br) | 1431191 | Low German (Netherlands) (ndsNL) | 113101 |
| English (simple) (en_xx_simple) | 1413074 | Mirandese (mwl) | 96829 |
| Khmer (km) | 1330590 | Ido (io) | 72811 |
| Gujarati (gu) | 1309569 | Igbo (ig) | 41296 |
| Cebuano (ceb) | 1255728 | Turkmen (tk) | 40650 |
| Welsh (cy) | 1229401 | Yiddish (yi) | 37128 |
| Xhosa (xh) | 1219701 | Yoruba (yo) | 26589 |
| | | **Total # Sentence Pairs** | 11944961985 |

1. We use `fil` for Filipino/Tagalog, not `tl`

2. We use `ak` for Twi/Akan, rather than `tw`. This includes Fante.

3. Unfortunately, we use the macro code `chm` for Meadow Mari (instead of the correct `mhr`), and `mrj` for Hill Mari

4. By convention, we use `no` for Norwegian Bokmål, whereas some resources use `nb`

5. By convention we use `ps` for Pashto instead of `pbt` (Southern Pashto)

6. By convention, we use `ms` for Standard Malay, not `zlm`

7. By convention, we use `sq` for Albanian, and don't distinguish dialects like Gheg (`aln`) and Tosk (`als`)

8. We use `ber` as the code for Tamazight, after consultation with Tamazight speakers opining that the dialect distinctions are not significant. Other resources use the individual codes like `tzm` and `kab`.

9. We use the macrocode `qu` for Quechua. In practice, this seems usually to be a mix of the Ayacucho and Cusco dialects. Other resources, like NLLB, may use the dialect code, e.g. `quy` for Ayacucho Chanka. The same is true for a few other macro codes, like `ff` (Macro code for Fulfulde, whereas other sources may use e.g. `fuv`.)

10. Really, there are notes that can be made about almost any code, from the well-accepted conventions like `zh` for Mandarin, to many dialectical notes, like which variant of Hmong really is the `hmn` data? But The above ones are made specifically for ones where we are aware of other datasources floating out there that use different conventions.

### A.7 Multiway Data Details

On creating the multiway data described in Section 3.4, we obtain a dataset with 11.9B sentence pairs across 19.7k language pairs. In Table 11, we list the combined number of sentence pairs for each target language.

### A.8 Model Training Details

**MT Model Training**  We train models of various sizes: a 3B, 32-layer parameter model,[11] a 7.2B 48-layer parameter model and a 10.7B 32-layer parameter model. We describe the specifics of the model architecture in Table 12.

We share all parameters of the model across language pairs, and use a Sentence Piece Model (SPM) [36] with 256k tokens shared on both the encoder and decoder side. We train the SPM model on upto 1M sentence samples of the sentences in the sentence-level version of `MADLAD-400`, supplemented by data from the languages in the parallel data used to train MT models when not available in `MADLAD-400` with a temperature of $T = 100$ and a character coverage of $99.9995\%$.

Each input sentence has a `<2xx>` token prepended to the source sentence to indicate the target language [30]. We use both supervised parallel data with a machine translation objective and the monolingual `MADLAD-400` dataset with a MASS-style [57] objective to train this model. Each of these objectives is sampled with a 50% probability. Within each task, we use the recently introduced UniMax [15] sampling strategy to sample languages from our imbalanced dataset with a threshold of $N = 10$ epochs for any particular language.

We used a square root learning rate decay schedule over the total number of training steps, starting at 0.01 and ending at X, as well as the AdaFactor optimizer with `factorized=False` and 10k warmup steps. We note that for the 10.7B model we use a dropout probability of $p = 0.2$ instead of $p = 0.1$ in order to mitigate overfitting to low-resource languages.

**Language Model Training**  We follow the same training schedule and model configurations from Garcia et al. [24]. In particular, we consider 8B decoder-only models, following the same model hyperparameters as previous work [14, 24]. We train these models using a variant of the UL2 objective [58] adapted for decoder-only models, and use the same configuration as previous work [24, 47]. We point the reader to these papers for a detailed overview of the training process, and include basic architectural details in Table 12. We use the same SPM trained for the MT models.

---

[11]Here and elsewhere, 'X-layer' means X encoder layers and also X decoder layers, for a total of 2X layers.

Table 12: Architecture and training details for the various models we train in this work using `MADLAD-400`.

| Model | SPM Size | Training Objective | # Attn Heads | # Enc. Layers | # Dec. Layers | Model Dim | Hidden Dim | Batch Size | Max. Seq. Length | Training Steps |
|---|---|---|---|---|---|---|---|---|---|---|
| 3B MT Model | | MT+MASS | 16 | 32 | 32 | 1024 | 8192 | 4096 | 256 | 1M |
| 7.2B MT Model | 256k | MT+MASS | 16 | 32 | 32 | 2048 | 8192 | 4096 | 256 | 500k |
| 10.7B MT Model | | MT+MASS | 32 | 48 | 48 | 2048 | 16384 | 4096 | 256 | 250k |
| 8B LM | | UL2 | 16 | N/A | 32 | 4096 | 16384 | 1024 | 1024 | 500k |

Table 13: Languages on which we evaluate the trained models for each multilingual evaluation set.

| Datasets | Languages Evaluated |
|---|---|
| WMT | cs, de, es, fi, fr, gu, hi, kk, lv, lt, ro, rs, es, tr, zh |
| Flores-200 | ac_Arab, ace, af, am, ar, arz, as, awa, ay, az, ba, ban, be, ber, bg, bho, bjn_Arab, bjn, bm, bn, bo, bs, bug, ca, ceb, ckb, crh_Latn, cs, cy, da, de, din, dyu, dz, ee, el, eo, es, et, eu, fa, ff, fi, fil, fj, fo, fon, fr, fur, ga, gd, gl, gn, gu, ha, hi, hne, hr, ht, hu, hy, id, ig, ilo, is, it, he, ja, jv, ka, kac, kbp, kg, kk, km, kmb, kn, ko, kr_Arab, kr, ks_Deva, ks, ky, lb, lg, li, lij, lmo, ln, lo, lt, ltg, lus, lv, mag, mai, mg, mi, min, mk, ml, mn, mni, mr, ms, mt, my, ne, nl, nn, no, nso, nus, ny, oc, om, or, pa, pag, pap, pl, ps, pt, quy, rn, ro, ru, rw, sa, sc, scn, sd, sg, shn, si, sk, sl, sm, sn, so, sq, sr, ss, st, su, sv, sw, szl, ta, taq_Tfng, taq, te, tg, th, ti, tk, tn, tr, ts, tt, ug, uk, ur, vec, vi, war, wo, xh, yi, yo, zh_Hant, zh, zu |
| NTREX | af, am, ar, az, ba, be, bem, bg, bn, bo, bs, ca, ckb, cs, cy, da, de, dv, dz, ee, el, en_GB, en_IN, es, es_MX, et, eu, fa, fa_AF, ff, fi, fil, fj, fo, fr, fr_CA, ga, gl, gu, ha, hi, hmn, hr, hu, hy, id, ig, is, it, iw, ja, ka, kk, km, kn, ko, ku, ky, lb, lo, lt, lv, mey, mg, mi, mk, ml, mn, mr, ms, mt, my, nd, ne, nl, nn, no, nso, ny, om, pa, pl, ps, pt, pt_PT, ro, ru, rw, sd, shi, si, sk, sl, sm, sn, so, sq, sr, sr_Latn, ss, sv, sw, ta, te, tg, th, ti, tk, tn, to, tr, tt, ty, ug, uk, ur, uz, ve, vi, wo, xh, yo, yue, zh, zh_Hant, zu |
| Gatones | ady, ak, as, av, ay, ba, ban, bbc, bci, ber_Latn, bew, bho, bm, bo, ce, chr, ckb, cv, doi, dv, dyu, dz, ee, ff, gn, gom, ilo, iso, kl, kri, lg, ln, lus, mad, mai, meo, min, mni, nso, om, or, qu, quc, rw, sa, sg, skr, ti, tiv, tk, ts, tt, ug, wo, yua, zza |

We use a square root learning rate decay schedule over 500k steps, starting at 0.01 and ending at 0.001, as well as the AdaFactor optimizer with `factorized=False` with 10k warmup steps. We describe the evaluation setup in Section 4.3.1.

## A.9 Languages Evaluated

In Table 13, we list the languages for which we evaluate the models trained as described in Sections 4.1 and 4.2.

## A.10 Results Details

In Tables 14, 18, 15, 16 and 17 we list the WMT, NTREX, Gatones, Flores-200 and Flores-200 (direct pairs) chrf and SacreBLEU scores respectively by language pair along with the model checkpoints.

Table 14: Evaluation scores on WMT (depicted as `<bleu>` / `<chrf>`) for the MT models and language models described in Section 4.1 and Section 4.2 compared against NLLB-54B.

| | NLLB | MT-3B | MT-7.2B | MT-10.7B | LM-8B | | | |
|---|---|---|---|---|---|---|---|---|
| | | | | | 0-shot | 1-shot | 5-shot | 10-shot |
| csen | 33.6 / 58.8 | 34.8 / 59.9 | 35.7 / 60.5 | 35.3 / 60.4 | 3 / 22.2 | 27.6 / 53.7 | 26.7 / 51.8 | 28.2 / 54 |
| deen | 37.7 / 62.4 | 37.3 / 62.5 | 38.1 / 63 | 38 / 63 | 3.3 / 23.6 | 26 / 51.9 | 29.7 / 55.2 | 30.1 / 55.5 |
| esen | 37.6 / 61.7 | 38.1 / 62.4 | 38.5 / 62.6 | 38.5 / 62.7 | 2.2 / 19 | 30 / 55.2 | 31.5 / 57.4 | 32.3 / 58.4 |
| eten | 34.7 / 61.1 | 36.2 / 61.9 | 37.4 / 62.8 | 37.4 / 62.9 | 4.7 / 24.7 | 26.4 / 52.8 | 27.9 / 55 | 27.7 / 54.5 |
| fien | 28.8 / 55.7 | 32.8 / 59.1 | 33.5 / 59.4 | 33.3 / 59.6 | 1.6 / 16.8 | 27.1 / 53.5 | 26.2 / 52 | 26.3 / 52.6 |
| fren | 41.9 / 65.7 | 42.2 / 65.6 | 42.7 / 65.9 | 42.4 / 65.8 | 2 / 18.6 | 28.9 / 54.8 | 35.1 / 60.7 | 35.6 / 61.1 |
| guen | 31.2 / 58.9 | 30.4 / 58.2 | 29.9 / 58 | 29.9 / 57.9 | 1 / 13 | 22.2 / 50.7 | 21.9 / 51.2 | 24.1 / 53.2 |
| hien | 37.4 / 64.2 | 33.5 / 61.3 | 36.1 / 62.7 | 35.5 / 62.4 | 1.2 / 10.2 | 23.4 / 52.3 | 23.4 / 54.5 | 24.4 / 54.5 |
| kken | 30.2 / 58.4 | 12.3 / 48.1 | 22.9 / 49.8 | 19.9 / 52.1 | 2.4 / 14.2 | 15.3 / 41.2 | 11.5 / 40.9 | 11.9 / 41.6 |
| lten | 29.7 / 58.8 | 34.5 / 60.9 | 35.3 / 61.4 | 35.1 / 61.5 | 3 / 21.4 | 24.1 / 51.1 | 26.1 / 52.8 | 28.1 / 54.9 |
| lven | 24.8 / 53.1 | 27.7 / 55.6 | 28.1 / 56 | 28.3 / 56.1 | 1.3 / 15.7 | 21.4 / 49.3 | 21.4 / 49.2 | 22 / 50.1 |
| roen | 43.4 / 66.4 | 43.6 / 66.6 | 44.4 / 67 | 44.4 / 67.1 | 4.1 / 24.8 | 34.9 / 58.9 | 36.1 / 59.9 | 36.2 / 60.5 |
| ruen | 39.9 / 63.8 | 39.9 / 64.4 | 40.7 / 64.9 | 40.5 / 64.9 | 2.1 / 16.4 | 32.7 / 56.9 | 34.1 / 58.5 | 27.4 / 53.7 |
| tren | 34.3 / 60.5 | 33.2 / 58.9 | 34.3 / 59.7 | 34.1 / 59.8 | 1.6 / 16.8 | 21.7 / 48.1 | 21.7 / 47.7 | 22.1 / 48.7 |
| zhen | 28.5 / 56.4 | 24.9 / 55.6 | 26.2 / 56 | 25.7 / 56.2 | 0.4 / 1.6 | 15.3 / 40 | 19.3 / 46.2 | 16.9 / 47 |
| encs | 25.2 / 53.2 | 26.2 / 53.8 | 27 / 54.2 | 26.8 / 54.2 | 0.7 / 10.3 | 18.5 / 45.2 | 19.3 / 48 | 19.6 / 47.9 |
| ende | 33 / 62 | 33.7 / 62.5 | 35 / 63.2 | 35 / 63.2 | 1.6 / 16.2 | 19.7 / 48.1 | 21.9 / 53 | 21.9 / 54.3 |
| enes | 37.2 / 61.1 | 37.3 / 61.1 | 37.7 / 61.3 | 37.4 / 61.3 | 0.9 / 13.9 | 28.4 / 53.7 | 30.2 / 55.6 | 30.9 / 56.3 |
| enet | 27 / 59.3 | 27.6 / 60.4 | 28.3 / 61 | 28.4 / 61.1 | 0.5 / 10.7 | 20.1 / 52.5 | 19.9 / 51.2 | 20.8 / 53.4 |
| enfi | 27.7 / 61.7 | 28.3 / 60 | 29 / 60.4 | 27.8 / 60.6 | 0.5 / 10.5 | 18.1 / 49 | 19.8 / 51.6 | 20.2 / 51.9 |
| enfr | 44.2 / 67.7 | 44.8 / 67.2 | 45.3 / 67.6 | 45.6 / 67.7 | 1 / 13.7 | 32.8 / 58.6 | 32.1 / 57.6 | 32.7 / 59.5 |
| engu | 17.6 / 50 | 18.6 / 51.3 | 19.2 / 52.1 | 18.8 / 52 | 0.1 / 0.3 | 0.2 / 1.2 | 0.1 / 1.4 | 0.1 / 1.6 |
| enhi | 26 / 53.5 | 23.4 / 50.1 | 24.2 / 50.8 | 24.5 / 51.1 | 1.5 / 10.6 | 16.9 / 42.6 | 15.7 / 40.5 | 16.7 / 42.7 |
| enkk | 34.8 / 65.2 | 7.6 / 44.2 | 12.9 / 47 | 10.7 / 47.1 | 0.2 / 5.2 | 6.1 / 37.6 | 6.2 / 37.7 | 5.7 / 38.9 |
| enlt | 37 / 66.9 | 26.5 / 59 | 27.2 / 59.5 | 27 / 59.6 | 1.1 / 14.2 | 20.6 / 53.3 | 20.9 / 53.6 | 20.1 / 53.1 |
| enlv | 21.3 / 53.6 | 23.9 / 55.8 | 25.2 / 56.6 | 24.8 / 56.4 | 1 / 12.4 | 18.3 / 48.5 | 19.7 / 52 | 19.8 / 51.6 |
| enro | 33.4 / 60.4 | 35.5 / 61.4 | 35.5 / 61.5 | 35.6 / 61.7 | 3.7 / 21 | 29.6 / 56.5 | 25.3 / 54 | 29.5 / 56.4 |
| enru | 44.8 / 67.4 | 32.9 / 57.8 | 33.6 / 58.4 | 33.6 / 58.3 | 0.6 / 2.5 | 20.4 / 44 | 22.3 / 47 | 23.8 / 49.4 |
| entr | 23.3 / 58.3 | 24.2 / 57.2 | 26.1 / 58.3 | 25.9 / 58.1 | 0.5 / 10.9 | 11.1 / 42.2 | 11 / 46.4 | 8.6 / 46.1 |
| enzh | 33.9 / 30.1 | 32.6 / 29.8 | 33.3 / 30.9 | 33.6 / 31.1 | 0.9 / 2.7 | 20.2 / 18.9 | 16.9 / 18.1 | 19 / 19 |
| **xx2en** | 34.2 / 60.4 | 33.4 / 60.0 | 34.9 / 60.6 | **34.6 / 60.8** | 2.3 / 17.3 | 25.1 / 51.4 | 26.2 / 52.9 | 26.2 / 53.4 |
| **en2xx** | **31.1 / 58.0** | 28.2 / 55.4 | 29.3 / 56.2 | 29.0 / 56.2 | 1.0 / 10.3 | 18.7 / 43.5 | 18.8 / 44.5 | 19.3 / 45.5 |
| **Average** | **32.7 / 59.2** | 30.8 / 57.7 | 32.1 / 58.4 | 31.8 / 58.5 | 1.6 / 13.8 | 21.9 / 47.4 | 22.5 / 48.7 | 22.8 / 49.4 |

Table 15: Evaluation scores on the GATONES test set used by Bapna et al. [8] (depicted as `<bleu>` / `<chrf>`) for the MT models and language models described in Section 4.1 and Section 4.2.

| | NTL (Bapna et al. [8]) | | MT-3B | MT-7.2B | MT-10.7B | LM-8B | | | |
|---|---|---|---|---|---|---|---|---|---|
| | 1.6B | 6.4B | | | | 0-shot | 1-shot | 5-shot | 10-shot |
| ady_en | - / 53.7 | - / 54.8 | 32.7 / 58.8 | 34.2 / 59.8 | 33.9 / 60.1 | 1.5 / 12.4 | 18.9 / 50.2 | 25 / 54.2 | 21.1 / 52.1 |
| ak_en | - / 31.7 | - / 36.3 | 6.9 / 27 | 10.4 / 29.9 | 9.3 / 29 | 0.1 / 5.5 | 3.6 / 15.5 | 1.8 / 18.6 | 1.9 / 16.6 |
| as_en | - / 52.9 | - / 58.6 | 30.6 / 55.6 | 32.1 / 56.8 | 33.8 / 58 | 1 / 9.7 | 17.4 / 44.5 | 19.4 / 47.4 | 21.1 / 48.6 |
| av_en | - / 48.1 | - / 48.1 | 19.7 / 47.8 | 21.3 / 50.6 | 21.5 / 50.9 | 0.3 / 6.1 | 11.6 / 32.5 | 10.3 / 40.2 | 11.1 / 40.6 |
| ay_en | - / 32.3 | - / 30.8 | 9.1 / 28.7 | 11.3 / 29 | 11.3 / 30.1 | 0.2 / 5.6 | 1.4 / 13.7 | 5.9 / 23.1 | 5.5 / 22.6 |
| ba_en | - / 42.3 | - / 45.8 | 4.5 / 23.3 | 5.4 / 24 | 6.9 / 27.2 | 0.2 / 2.9 | 8.7 / 30.3 | 7.6 / 35 | 7.7 / 34.6 |
| ban_en | - / 32.2 | - / 35.4 | 10.1 / 30.9 | 10.3 / 31.2 | 10.9 / 32.4 | 0.2 / 6.6 | 2.2 / 21.2 | 4.7 / 24.6 | 4.7 / 22.7 |
| bbc_en | - / 40.9 | - / 44 | 12.4 / 36.6 | 12.1 / 36.7 | 13.3 / 38 | 0.2 / 6.6 | 5.6 / 27.5 | 7.7 / 29.4 | 7.9 / 28.4 |
| bci_en | - / 11.6 | - / 10.8 | 2.7 / 19.7 | 3.6 / 20.4 | 3.9 / 21.6 | 0 / 4 | 0.6 / 16.8 | 0.5 / 9.5 | 1.9 / 15.3 |
| bew_en | - / 49.8 | - / 51.8 | 25.3 / 50.1 | 26.6 / 51.4 | 26.7 / 51.7 | 0.7 / 10.8 | 20.3 / 44.8 | 23.8 / 49.2 | 21.3 / 48.3 |
| bho_en | - / -10000 | - / -10000 | 28.6 / 55.8 | 25.6 / 52.9 | 29.3 / 56.3 | 0.6 / 7.4 | 17.7 / 45.6 | 20 / 49.1 | 21.6 / 49.7 |
| bm_en | - / 27.6 | - / 36.3 | 11.9 / 32.8 | 10.6 / 31.7 | 12.6 / 33.9 | 0.1 / 4.2 | 0.1 / 11.5 | 1.2 / 11 | 1.2 / 14.4 |
| ce_en | - / 44.6 | - / 44.9 | 11.4 / 36 | 12.5 / 37 | 13.5 / 37.3 | 0.6 / 7.5 | 11.4 / 36.4 | 9.4 / 33.2 | 15.1 / 41.3 |
| chr_en | - / 16.3 | - / 15.6 | 1.1 / 5.9 | 0.6 / 8 | 0.9 / 7.2 | 0 / 0.6 | 0.2 / 12.8 | 0.3 / 12.6 | 0.1 / 10 |
| cv_en | - / 39.9 | - / 46.3 | 19 / 42.8 | 21.2 / 44.9 | 22.9 / 46.6 | 0.3 / 5.7 | 7.5 / 24.6 | 13 / 38 | 12.2 / 36.3 |
| doi_en | - / 57.4 | - / 63.1 | 15.3 / 43.1 | 16 / 42.9 | 18 / 45 | 0.2 / 4.7 | 6.9 / 31.9 | 8.8 / 35.2 | 6.1 / 36.4 |
| dv_en | - / 37 | - / 45.2 | 20.1 / 45.3 | 21.4 / 47.1 | 22.5 / 47.7 | 0.8 / 8.9 | 11.4 / 35.5 | 13.2 / 38.1 | 14.6 / 41.5 |
| dyu_en | - / 23.9 | - / 23.5 | 7.6 / 28 | 6.7 / 27.7 | 8.6 / 30 | 0 / 4.6 | 2 / 14.1 | 2.3 / 16.2 | 1.3 / 15.1 |
| ee_en | - / 29.1 | - / 33.5 | 8.6 / 25.8 | 11.1 / 28.9 | 10.4 / 28.3 | 0.1 / 6 | 2 / 17.6 | 2.1 / 18.9 | 3.1 / 16.4 |
| en_ady | - / 21.8 | - / 28.2 | 0.6 / 8.8 | 2 / 14.2 | 1.4 / 11.6 | 0.1 / 0.6 | 2.1 / 18.9 | 0.4 / 11.2 | 1.7 / 28.6 |
| en_ak | - / 32.1 | - / 34.3 | 2.2 / 17.9 | 4.1 / 21.9 | 3.9 / 21.2 | 0.3 / 5.9 | 0.1 / 3.5 | 0.1 / 4.5 | 0 / 3.1 |
| en_as | - / 36.3 | - / 36.7 | 8.4 / 36.7 | 8.4 / 36.9 | 8.7 / 38 | 0.1 / 1.8 | 0.7 / 17.6 | 0.7 / 18.4 | 0.9 / 19.1 |
| en_av | - / 26.1 | - / 28.1 | 1.7 / 17.5 | 1.8 / 21.5 | 1.9 / 20.4 | 0.1 / 0.5 | 0.1 / 4.5 | 0.5 / 15.8 | 0.1 / 5.9 |
| en_ay | - / 30 | - / 30.5 | 1.4 / 17 | 2.4 / 21.7 | 2.1 / 21.6 | 0.2 / 6.2 | 0.5 / 9.3 | 0.2 / 4.7 | 0 / 3.4 |
| en_ba | - / 37.6 | - / 38.4 | 2.3 / 22.1 | 3.6 / 25.3 | 4 / 26.8 | 0.1 / 0.9 | 0.2 / 8.8 | 0.2 / 9.3 | 0.1 / 5.7 |

| | | | | | | | | |
|---|---|---|---|---|---|---|---|---|
| en_ban | - / 31.8 | - / 33.1 | 3.9 / 30.8 | 3.7 / 30.9 | 4 / 31.5 | 0.2 / 8.5 | 0.7 / 14.1 | 0.1 / 4.4 | 0.1 / 8.5 |
| en_bbc | - / 34.9 | - / 35.4 | 1.9 / 26.8 | 2.6 / 28.8 | 3.1 / 29.5 | 0.1 / 4.4 | 0.4 / 12.5 | 0.1 / 8.1 | 0.1 / 7.1 |
| en_bci | - / 14.1 | - / 14.7 | 1.5 / 12.7 | 2 / 13.8 | 2.4 / 14.9 | 0.1 / 4.8 | 0 / 2.2 | 1.1 / 8.8 | 0 / 1.7 |
| en_bew | - / 45.7 | - / 46 | 7.9 / 35.6 | 12.6 / 44.9 | 12.7 / 45.5 | 0.1 / 6.2 | 7.5 / 35.2 | 10.4 / 41.9 | 11.7 / 43.9 |
| en_bho | - / 0 | - / 40.6 | 12.8 / 39.5 | 12 / 38.4 | 12.7 / 40.1 | 0.2 / 3.8 | 2.9 / 24.6 | 4.1 / 28.8 | 5.8 / 30.5 |
| en_bm | - / 28.7 | - / 34.3 | 6.5 / 28.5 | 6.1 / 27.2 | 7.1 / 30.5 | 0.2 / 5.5 | 0.1 / 4.8 | 0 / 2.6 | 0 / 2.1 |
| en_ce | - / 23.5 | - / 23.7 | 3.4 / 9 | 7.5 / 22.9 | 1.9 / 15.2 | 0.2 / 1.3 | 0.8 / 8.7 | 0 / 1.2 | 0.1 / 3.5 |
| en_ckb | - / 0 | - / 41.6 | 7.5 / 38.7 | 7.9 / 38.4 | 6.7 / 35.4 | 0.1 / 0.6 | 0.2 / 8.2 | 0.2 / 9 | 0.2 / 11.1 |
| en_cv | - / 28.1 | - / 32.1 | 3.1 / 25.6 | 3.1 / 25.6 | 3.1 / 25.8 | 0.1 / 0.9 | 0.1 / 5 | 0.1 / 5.5 | 0 / 4.5 |
| en_doi | - / 29.9 | - / 25.5 | 0.7 / 0.8 | 0.7 / 0.8 | 0.7 / 0.8 | 0.1 / 0.2 | 0.3 / 10.1 | 0.3 / 11 | 0.1 / 5.9 |
| en_dv | - / 43.2 | - / 43.7 | 2.9 / 38 | 3.5 / 40.2 | 3.8 / 40.8 | 0.1 / 0.4 | 0.6 / 0.4 | 0.6 / 0.5 | 0.5 / 0.4 |
| en_dyu | - / 12.7 | - / 20.4 | 1.9 / 19.4 | 2.3 / 19.6 | 2.2 / 20.1 | 0.1 / 4 | 0.4 / 11.2 | 0 / 4.6 | 0 / 1.5 |
| en_ee | - / 35.9 | - / 39.2 | 4.1 / 23.7 | 6.1 / 27.1 | 6 / 26.6 | 0 / 4.7 | 0.1 / 4.2 | 0 / 2.9 | 0 / 2.1 |
| en_ff | - / 31.1 | - / 32.3 | 3 / 21.4 | 3.3 / 21.3 | 3.7 / 22.9 | 0.1 / 5.5 | 0 / 12 | 0 / 1.8 | 0 / 3.3 |
| en_gn | - / 24.3 | - / 32.2 | 6.3 / 30.8 | 6.4 / 30.3 | 6.4 / 31.4 | 0.2 / 6.7 | 0.2 / 14.6 | 0.1 / 4.4 | 0 / 2.5 |
| en_gom | - / 1.1 | - / 39.1 | 1.2 / 20.1 | 2.1 / 23.5 | 4.3 / 28.2 | 0.1 / 0.7 | 0.6 / 15.3 | 0.5 / 16.8 | 1.2 / 20 |
| en_ilo | - / 49.7 | - / 52.4 | 13.9 / 37.8 | 16.2 / 41.3 | 16.9 / 42.8 | 0.3 / 7.1 | 2.5 / 23.6 | 3.4 / 25.7 | 2.1 / 24.8 |
| en_iso | - / 16.5 | - / 30.5 | 2.1 / 16.3 | 2.4 / 18.2 | 3.1 / 17.4 | 0.2 / 4.3 | 0 / 3.2 | 0.1 / 7.2 | 0 / 4.2 |
| en_kl | - / 0 | - / 23.1 | 2.6 / 29.5 | 3.6 / 33 | 3.4 / 33.6 | 0.2 / 4.6 | 0.4 / 9 | 0.1 / 4.7 | 0 / 4.8 |
| en_kri | - / 32.7 | - / 34.9 | 2.6 / 22.2 | 2.3 / 18.2 | 3.4 / 22.7 | 0.3 / 5.9 | 0.7 / 11.3 | 0.2 / 8.7 | 0.7 / 16.4 |
| en_lg | - / 35.9 | - / 38 | 1.7 / 23.6 | 1.6 / 25.8 | 2 / 27.4 | 0.1 / 5.3 | 0 / 3.6 | 0 / 3.6 | 0 / 2.6 |
| en_ln | - / 12.7 | - / 33.9 | 0.3 / 11.7 | 0.4 / 13.6 | 0.6 / 15.2 | 0 / 5.8 | 0 / 2.7 | 0 / 2.9 | 0 / 1.7 |
| en_lus | - / 16.9 | - / 39.3 | 7.7 / 31.1 | 10.6 / 35.6 | 10.2 / 34.7 | 0.2 / 5.6 | 0.1 / 4.8 | 0.1 / 4.7 | 0.1 / 4.2 |
| en_mad | - / 29.8 | - / 30.2 | 2.1 / 23.8 | 3.3 / 25.9 | 4 / 25.5 | 0.1 / 6.8 | 0.1 / 4.6 | 0.1 / 6.4 | 0.1 / 3.2 |
| en_mai | - / 34.1 | - / 37.6 | 12.7 / 42.4 | 13.2 / 41.2 | 14 / 43.2 | 0.1 / 0.8 | 1.6 / 18.2 | 3.4 / 28.9 | 3.4 / 29.5 |
| en_meo | - / 50.9 | - / 50.7 | 39.3 / 63 | 46.3 / 68.9 | 44.1 / 67.6 | 2.2 / 13.2 | 47.4 / 70.4 | 46.5 / 70.3 | 48.3 / 70.8 |
| en_min | - / 51.8 | - / 56.1 | 10.7 / 44.7 | 12.9 / 47.2 | 10 / 45 | 0.1 / 7.3 | 5.7 / 35.9 | 4.7 / 34.7 | 6.3 / 39.9 |
| en_mni | - / 35.7 | - / 38.2 | 7.1 / 38.9 | 7.9 / 39.4 | 7.9 / 40.6 | 0.1 / 0.3 | 0.1 / 1.7 | 0 / 1.7 | 0 / 0.6 |
| en_nso | - / 45 | - / 41.6 | 11.1 / 34.4 | 14.4 / 38.2 | 13.4 / 37.3 | 0.2 / 4.5 | 0.1 / 5.9 | 0.1 / 5.6 | 0.1 / 5 |
| en_om | - / 36 | - / 39.1 | 1 / 20 | 1.7 / 26.9 | 1.9 / 26.4 | 0.1 / 5.8 | 0 / 4.4 | 0 / 2.6 | 0 / 2.4 |
| en_qu | - / 29.8 | - / 33.1 | 1.5 / 19.5 | 2.3 / 21.1 | 1 / 22.4 | 0 / 6 | 0.1 / 6.7 | 0 / 2.2 | 0 / 1.1 |
| en_quc | - / 22.8 | - / 22.9 | 0.5 / 10.1 | 1 / 12.7 | 1.4 / 14.2 | 0.2 / 5.9 | 0 / 5.7 | 0 / 3.2 | 0 / 4.5 |
| en_sa | - / 26.9 | - / 28.4 | 1.3 / 24 | 1.5 / 26.9 | 1.8 / 27.8 | 0 / 1.2 | 0 / 5.3 | 0 / 5.7 | 0 / 6.2 |
| en_sg | - / 12.7 | - / 20.7 | 0.4 / 18.5 | 0.5 / 18.2 | 0.5 / 18.5 | 0 / 4.7 | 0 / 1.3 | 0.1 / 22.2 | 0 / 1.3 |
| en_skr | - / 32.8 | - / 31.3 | 1.3 / 21.9 | 1.5 / 24.9 | 1.9 / 25 | 0 / 0.1 | 0.1 / 7.9 | 0 / 6 | 0 / 4.8 |
| en_ti | - / 19.9 | - / 21.1 | 1.3 / 12.8 | 1.5 / 15.6 | 1.7 / 15.8 | 0.1 / 0.3 | 0 / 0.6 | 0 / 0.6 | 0 / 0.4 |
| en_tiv | - / 13.7 | - / 23 | 1 / 16.2 | 1.3 / 18.5 | 1 / 17.9 | 0.2 / 4.7 | 0.4 / 9.3 | 0 / 2.3 | 0 / 3.7 |
| en_ts | - / 17.4 | - / 45.5 | 6.4 / 33.6 | 8.9 / 37.4 | 7.4 / 35.7 | 0.1 / 5.2 | 0.1 / 4.1 | 0.1 / 5.8 | 0.1 / 5 |
| en_yua | - / 30.6 | - / 31.6 | 3.5 / 19 | 1.4 / 17.2 | 3 / 19.5 | 0.2 / 6.2 | 0.6 / 9.9 | 0.1 / 4 | 0.1 / 7 |
| en_zza | - / 22.1 | - / 22.6 | 1.7 / 19.8 | 1.1 / 18 | 1.3 / 17.6 | 0.1 / 3.6 | 0 / 8.9 | 0 / 2.6 | 0 / 1.6 |
| ff_en | - / 33.7 | - / 41.2 | 7.8 / 26.8 | 8.3 / 26.5 | 9.4 / 29.2 | 0.1 / 4.4 | 0.4 / 12.5 | 1.1 / 15.4 | 1.3 / 13.6 |
| gn_en | - / 26.7 | - / 38.9 | 12.9 / 35.5 | 12.7 / 35.2 | 13.7 / 37 | 0.2 / 6 | 1.5 / 16.7 | 3.9 / 22.2 | 2.6 / 19.8 |
| gom_en | - / 50.7 | - / 55.5 | 20.5 / 45.7 | 20.4 / 45 | 21 / 45.7 | 0.1 / 1.2 | 10.5 / 33.2 | 16.6 / 42.1 | 15.7 / 41 |
| ilo_en | - / 50.8 | - / 43.4 | 17.3 / 44.2 | 28.6 / 52.7 | 29.6 / 52.8 | 0.2 / 5.7 | 13.7 / 38.9 | 20.1 / 42.7 | 18 / 42.1 |
| iso_en | - / 17.9 | - / 29.4 | 5.8 / 24.3 | 7.7 / 25.3 | 8.2 / 26.1 | 0.1 / 4.5 | 1.1 / 16 | 1.6 / 14.4 | 2.3 / 17.2 |
| kri_en | - / 48.7 | - / 56.5 | 13.2 / 37 | 15.6 / 39.5 | 17.3 / 41.5 | 0.1 / 6 | 10.5 / 31.3 | 7.5 / 31.1 | 10.5 / 33.2 |
| lg_en | - / 32.6 | - / 37.9 | 6.3 / 24.1 | 8.9 / 26.7 | 8.2 / 27.1 | 0 / 3.9 | 1.9 / 18.9 | 2.5 / 18.3 | 3.3 / 17.3 |
| ln_en | - / 28.2 | - / 30.4 | 0.8 / 13.1 | 0.9 / 14.2 | 0.8 / 14.8 | 0.1 / 5.1 | 0.3 / 15.2 | 1 / 15 | 0.9 / 12.7 |
| lus_en | - / 22.6 | - / 34.5 | 13 / 35.7 | 18.4 / 40.9 | 17.9 / 41.2 | 0.6 / 12 | 7.5 / 31 | 12.1 / 33.6 | 10.8 / 33.4 |
| mad_en | - / 44.9 | - / 50.5 | 8.5 / 34 | 10.4 / 34.3 | 11.9 / 36.7 | 0.1 / 5.9 | 3.8 / 22.5 | 3.9 / 24.9 | 5 / 26.8 |
| mai_en | - / 57.8 | - / 61.6 | 30.9 / 57 | 29.2 / 55.7 | 33.3 / 58.8 | 0.9 / 9.3 | 15.3 / 45.1 | 20.5 / 50.1 | 20.3 / 49.2 |
| meo_en | - / 66.1 | - / 67.9 | 44.4 / 66.4 | 46.2 / 67.8 | 45.5 / 67.5 | 0.9 / 9.4 | 38.7 / 61.5 | 39.6 / 63.3 | 41.1 / 63.7 |
| min_en | - / 58.8 | - / 62.4 | 26.2 / 52.4 | 25.3 / 50.8 | 29 / 55.1 | 0.4 / 7.7 | 16.7 / 39.3 | 20.9 / 49.1 | 21.7 / 49.5 |
| mni_en | - / 54.8 | - / 56.4 | 32.2 / 55.9 | 31 / 54.9 | 33.4 / 57 | 0 / 1.9 | 1.2 / 17.5 | 2.6 / 20.4 | 2.7 / 20.5 |
| nso_en | - / 45.8 | - / 51.3 | 17.6 / 40.2 | 21.1 / 43.2 | 21.1 / 43.4 | 0.2 / 6.1 | 5.7 / 23.9 | 7.6 / 28.7 | 6.1 / 27.3 |
| om_en | - / 32.1 | - / 38.1 | 5.7 / 25.8 | 9.4 / 30.6 | 9.4 / 31.3 | 0.2 / 6.2 | 0.8 / 18.9 | 2.8 / 19 | 2.3 / 20.2 |
| qu_en | - / 29.9 | - / 32.5 | 6.7 / 27.2 | 7.3 / 27.6 | 7.5 / 28.2 | 0.1 / 6.1 | 2.5 / 18.1 | 1.8 / 13.7 | 2.3 / 18.8 |
| quc_en | - / 24.4 | - / 26.7 | 1.9 / 17.6 | 2 / 15.9 | 2.8 / 19.3 | 0.1 / 7.1 | 1.3 / 11.4 | 1.6 / 13.9 | 0.9 / 12.3 |
| sa_en | - / 43.5 | - / 46.3 | 15.1 / 39 | 17.3 / 40.4 | 14.5 / 37.9 | 2.6 / 18.1 | 7 / 35.8 | 7.4 / 34.8 | 9.8 / 37.3 |
| sg_en | - / 13.4 | - / 13.8 | 0.5 / 12.5 | 0.6 / 13.5 | 0.7 / 14.1 | 0 / 3.2 | 0 / 17 | 0.2 / 9.1 | 0.2 / 12.1 |
| skr_en | - / 44.4 | - / 48.4 | 9 / 35.5 | 12.5 / 38.8 | 12.5 / 39.1 | 0.3 / 8.5 | 6.5 / 28.5 | 9 / 33.2 | 8.5 / 34.3 |
| ti_en | - / 37.9 | - / 44.2 | 11.7 / 34 | 15.2 / 38.2 | 15.2 / 38.2 | 0.3 / 7.1 | 5.3 / 22.3 | 7.1 / 27.4 | 7 / 26.3 |
| tiv_en | - / 14.1 | - / 15 | 2.7 / 14 | 2.9 / 14.8 | 3 / 15.4 | 0.1 / 3.8 | 0.9 / 12.3 | 1.1 / 11.8 | 0.9 / 12.2 |
| ts_en | - / 25.5 | - / 43 | 14.3 / 35.7 | 18.5 / 40.5 | 17.8 / 40 | 0.3 / 7.5 | 1 / 16.3 | 7.6 / 25.9 | 8.7 / 27.8 |
| yua_en | - / 34.6 | - / 40.7 | 6 / 27 | 7.1 / 28.3 | 9.7 / 30.9 | 0.2 / 6.9 | 0.8 / 11.3 | 3.3 / 20.2 | 4.4 / 19.2 |
| zza_en | - / 23.4 | - / 23.2 | 4.6 / 22.4 | 5.1 / 22.8 | 5.2 / 24.2 | 0.2 / 10.6 | 1.9 / 18.9 | 3.3 / 21.3 | 2.3 / 20.6 |
| **xx2en** | - / 37.2 | - / **41.2** | 13.3 / 34.6 | **15.5** / 36.5 | 14.8 / 36.0 | 0.3 / 6.5 | 6.6 / 25.4 | 8.3 / 28.1 | 8.4 / 28.4 |
| **en2xx** | - / 28.5 | - / **33.1** | 4.5 / 23.9 | **5.5** / 26.4 | 5.4 / 26.2 | 0.2 / 4.2 | 1.7 / 10.5 | 1.7 / 9.9 | 1.8 / 9.4 |
| **Average** | - / 32.9 | - / **37.2** | 8.9 / 29.3 | **10.5** / 31.5 | 10.1 / 31.1 | 0.3 / 5.4 | 4.2 / 18.0 | 5.0 / 19.0 | 5.1 / 18.9 |

Table 16: Evaluation scores on en-centric Flores-200 pairs (depicted as `<bleu>` / `<chrf>`) for the MT models and language models described in Section 4.1 and Section 4.2 compared against NLLB-54B. All metrics are computed with the sacrebleu reference implementation.

| | NLLB | MT-3B | MT-7.2B | MT-10.7B | LM-8B | | | |
|---|---|---|---|---|---|---|---|---|
| | | | | | 0-shot | 1-shot | 5-shot | 10-shot |
| en_ace_Arab | 0.4 / 21.1 | 1.4 / 26.6 | 1.4 / 25.3 | 1.3 / 26.9 | 0 / 0.1 | 0 / 0.5 | 0 / 2.9 | 0 / 2.2 |
| en_ace | 9.3 / 41.4 | 8.3 / 39.6 | 6.5 / 36.6 | 7.6 / 39.1 | 0.1 / 3.7 | 0.9 / 15.3 | 0.8 / 21.1 | 0.7 / 22.4 |
| en_acm | 10.3 / 35.4 | 0.6 / 0.8 | 0.4 / 0.8 | 0.6 / 0.8 | 0 / 0.4 | 4.8 / 34.6 | 8.3 / 43.3 | 6.1 / 41.1 |
| en_acq | 13.6 / 46.8 | 0.6 / 0.8 | 0.4 / 0.7 | 0.6 / 0.8 | 0.1 / 1.2 | 5.9 / 36.1 | 10.1 / 43.5 | 9.7 / 44 |
| en_aeb | 12 / 42.1 | 0.6 / 0.8 | 0.6 / 0.8 | 0.6 / 0.8 | 0.1 / 1.2 | 1.4 / 20.3 | 4.1 / 35.4 | 4.9 / 37.2 |
| en_af | 38.4 / 66.6 | 38.8 / 66.1 | 40.9 / 67.6 | 40.4 / 67.3 | 0.7 / 10.5 | 31.2 / 62.2 | 36.9 / 64.1 | 38.6 / 64.3 |
| en_ajp | 20.5 / 56 | 0.7 / 1.3 | 0.6 / 1.3 | 0.7 / 1.3 | 0 / 1 | 9.5 / 41.9 | 9.3 / 43.1 | 9.7 / 44.5 |
| en_am | 15 / 42.2 | 8.5 / 31.2 | 8.8 / 32.2 | 9.1 / 32.5 | 0 / 0.2 | 0.1 / 0.6 | 0.2 / 0.7 | 0.2 / 0.7 |
| en_apc | 20.4 / 54.9 | 0.8 / 1.2 | 0.6 / 1.2 | 0.7 / 1.1 | 0 / 1 | 6 / 37.7 | 5.5 / 39 | 4.4 / 40 |
| #N/A en_ar_MA | 11.7 / 43.5 | 0.4 / 0.9 | 0.3 / 0.8 | 0.3 / 0.8 | 0 / 1.1 | 0.2 / 5.5 | 1 / 24.7 | 0.2 / 15.3 |
| en_ar | 29.7 / 60.6 | 27.3 / 57.9 | 28.2 / 58.4 | 27.7 / 58.2 | 0.1 / 1.2 | 12.6 / 44 | 17.5 / 50.2 | 18 / 50.6 |
| en_ars | 23.2 / 54.2 | 0.4 / 0.6 | 0.3 / 0.6 | 0.4 / 0.6 | 0.1 / 1.1 | 11.6 / 42.3 | 15.1 / 48.3 | 16.2 / 49.4 |
| en_arz | 16.7 / 50.9 | 16.6 / 48.7 | 15.1 / 47 | 16.4 / 48.9 | 0.1 / 1.2 | 2.6 / 27 | 3.4 / 34.8 | 6 / 38.6 |
| en_as | 7.9 / 40.3 | 7.3 / 37.3 | 8.1 / 38.9 | 8 / 39.2 | 0 / 0.9 | 0.2 / 12.9 | 0.7 / 21.8 | 0.6 / 21.3 |
| en_ast | 29.8 / 59.6 | 1.1 / 2.1 | 2 / 13.6 | 1.1 / 2 | 0.2 / 7.8 | 6.4 / 38.4 | 9.7 / 44.6 | 11.2 / 45.9 |
| en_awa | 18.5 / 50.7 | 10.3 / 43.8 | 9.5 / 42.3 | 10.2 / 44.4 | 0 / 0.3 | 0.8 / 16.5 | 7.2 / 38 | 5 / 38.6 |
| en_ay | 4 / 34.5 | 0.8 / 16.7 | 1.1 / 19.5 | 1.4 / 20.9 | 0 / 2.6 | 0 / 2 | 0 / 7 | 0 / 4.1 |
| en_az | 14.1 / 47.1 | 12.5 / 44.2 | 13.5 / 45.3 | 13.9 / 46.4 | 0.1 / 5.4 | 8.4 / 38.8 | 9.7 / 42.8 | 10.6 / 43.7 |
| en_azb | 1.3 / 27.8 | 0.2 / 0.6 | 0.2 / 0.6 | 0.2 / 0.6 | 0 / 0.1 | 0 / 0.8 | 0 / 6.9 | 0 / 5.6 |
| en_ba | 18.7 / 51 | 5.9 / 28.7 | 7.8 / 31.7 | 10.4 / 36.8 | 0 / 0.3 | 0.3 / 10.8 | 0.3 / 19.8 | 0.2 / 16.4 |
| en_ban | 15.3 / 48.4 | 13.1 / 45.5 | 11.1 / 42.4 | 13.4 / 45.9 | 0.1 / 5.5 | 0.4 / 10.3 | 0.7 / 21.7 | 1.7 / 30.4 |
| en_be | 14.4 / 45.7 | 12.3 / 40.5 | 13.5 / 42.1 | 13.6 / 42.8 | 0.1 / 2.6 | 11.5 / 40.9 | 11.5 / 41.5 | 11.1 / 41.6 |
| en_bem | 10.5 / 42.3 | 1.6 / 2.4 | 2.7 / 17.7 | 1.6 / 2.7 | 0.2 / 4.4 | 0 / 4.9 | 0.1 / 10.1 | 0.1 / 7.9 |
| en_ber | 8.1 / 34.3 | 6.6 / 31 | 6.2 / 30.6 | 7.5 / 33.4 | 0 / 0.2 | 0 / 0.5 | 0 / 2 | 0 / 1 |
| en_bg | 41.6 / 67.1 | 43.5 / 68.9 | 43.9 / 69.2 | 44.1 / 69.3 | 0.7 / 5.7 | 32.2 / 60.2 | 33 / 61.4 | 33.4 / 62.2 |
| en_bho | 17.3 / 45.2 | 13.4 / 41 | 11.7 / 38.8 | 13.8 / 42.1 | 0.1 / 2.1 | 1.4 / 19.5 | 4.6 / 31.7 | 4 / 31.2 |
| en_bjn_Arab | 1.1 / 20.3 | 3.3 / 30.5 | 2.9 / 28.6 | 3.6 / 31 | 0 / 0 | 0 / 1.1 | 0 / 4.4 | 0 / 1.5 |
| en_bjn | 19.2 / 51.8 | 15.8 / 49.6 | 13.3 / 45.6 | 15.8 / 49.8 | 0 / 2.4 | 1 / 17.2 | 4.9 / 36.3 | 3 / 34.9 |
| en_bm | 6.8 / 32.9 | 4.9 / 27.8 | 4.3 / 26.8 | 5.7 / 30.1 | 0 / 3.1 | 0 / 6.3 | 0 / 11.2 | 0 / 7.8 |
| en_bn | 19.4 / 54.2 | 14 / 47.2 | 14.8 / 48.2 | 14.8 / 48.3 | 0.1 / 3.7 | 4.5 / 35 | 8.2 / 44.4 | 7.1 / 44.3 |
| en_bo | 0.8 / 38.6 | 13.8 / 26.4 | 15.9 / 29.4 | 15.9 / 30.2 | 0 / 0.1 | 0 / 0 | 0 / 1.4 | 0 / 1 |
| en_bs | 33.2 / 61.4 | 33.4 / 61.9 | 34 / 62.3 | 34.6 / 62.9 | 0.3 / 5.9 | 24 / 53.6 | 27.6 / 56.9 | 26.9 / 57.5 |
| en_bug | 6.5 / 38.3 | 5.8 / 36.6 | 5.3 / 34.7 | 5.8 / 37.2 | 0.1 / 4.3 | 0 / 2.2 | 0 / 6.7 | 0.1 / 8.4 |
| en_ca | 43.9 / 66.8 | 45.8 / 67.5 | 46.4 / 68 | 46.7 / 68.4 | 0.9 / 11.3 | 35.9 / 60.7 | 36.3 / 61.8 | 37.1 / 61.9 |
| en_ceb | 30.3 / 59.6 | 7.4 / 26.5 | 8.3 / 27.6 | 9.4 / 29.4 | 0.2 / 3.3 | 9.1 / 39.8 | 9.3 / 39.4 | 10.3 / 40.7 |
| en_cjk | 2.7 / 28.3 | 1.3 / 2.4 | 1.2 / 16.6 | 1.4 / 3.8 | 0 / 3.3 | 0 / 1.3 | 0.1 / 6.2 | 0 / 6.2 |
| en_ckb | 13.2 / 52.5 | 9.2 / 44.9 | 9.8 / 45 | 9.4 / 44.4 | 0 / 0.1 | 0.1 / 5.6 | 0.2 / 14 | 0.1 / 13.5 |
| en_crh_Latn | 16.8 / 51 | 13.2 / 47.3 | 10.9 / 44.1 | 13.8 / 48.4 | 0.1 / 3.6 | 0.4 / 15.2 | 1.1 / 27.5 | 1.3 / 28.2 |
| en_cs | 33.6 / 59.8 | 34.8 / 61.3 | 34.7 / 61.1 | 35.6 / 61.5 | 0.2 / 5.6 | 22.5 / 52.1 | 26.2 / 54.5 | 25.9 / 54.6 |
| en_cy | 51.6 / 72.5 | 41.7 / 62.4 | 45.7 / 66.4 | 44.7 / 65.6 | 0.1 / 5.1 | 1.5 / 15.2 | 8.8 / 38.4 | 4.9 / 35.6 |
| en_da | 44.5 / 68.4 | 48 / 71.1 | 47.9 / 71.1 | 48.3 / 71.3 | 0.6 / 8.5 | 40.3 / 65.3 | 41 / 65.4 | 42.1 / 66.8 |
| en_de | 39.6 / 65.2 | 41.5 / 67.1 | 42 / 67.5 | 42.2 / 67.4 | 0.5 / 10.1 | 27.6 / 56.3 | 28.4 / 57.4 | 29.1 / 58 |
| en_din | 3.8 / 26.8 | 3.4 / 25.1 | 3.2 / 24.7 | 3.3 / 25.9 | 0.1 / 3.4 | 0 / 2.9 | 0 / 10.6 | 0 / 0.9 |
| en_dyu | 1.7 / 19.8 | 2 / 21 | 2.3 / 21.1 | 2.1 / 22.1 | 0 / 2.7 | 0 / 2.7 | 0 / 7.4 | 0 / 5.8 |
| en_dz | 0.6 / 45.4 | 34.2 / 42.5 | 32.3 / 41 | 37 / 45.1 | 0 / 0 | 0 / 1 | 0 / 3.4 | 0 / 2.4 |
| en_ee | 13.2 / 41.9 | 5.2 / 26.9 | 6.8 / 29.2 | 5.9 / 28.4 | 0 / 3.2 | 0.1 / 3.4 | 0.1 / 7.5 | 0.1 / 5.9 |
| en_el | 28 / 54.2 | 29.2 / 55.9 | 29 / 55.8 | 29.5 / 56.3 | 0.3 / 2.4 | 19 / 46.5 | 21.5 / 48.8 | 20.9 / 48.9 |
| en_eo | 35.3 / 63.9 | 33.6 / 62.6 | 34.1 / 63 | 34.3 / 63.7 | 0.2 / 5.4 | 5.1 / 32.4 | 12.6 / 47.2 | 9.4 / 44.8 |
| en_es | 28.1 / 56 | 28 / 56.4 | 28.4 / 56.6 | 28.5 / 56.7 | 0.3 / 8.9 | 22.6 / 50.6 | 23.6 / 52.1 | 23.7 / 52.2 |
| en_et | 27.1 / 59.9 | 29 / 61.3 | 28.5 / 61.2 | 29.3 / 61.8 | 0.2 / 6 | 20.4 / 54.1 | 22.7 / 55.5 | 23.1 / 55.5 |
| en_eu | 17.2 / 54.9 | 18.3 / 55.5 | 18.6 / 55.7 | 19.3 / 57.2 | 0 / 5.5 | 8.3 / 45.5 | 12.5 / 51.1 | 12.5 / 51.3 |
| en_fa_AF | 27.8 / 55.4 | 0.5 / 1 | 0.4 / 0.9 | 0.5 / 0.9 | 0 / 0.1 | 14.7 / 44.4 | 21.6 / 48.4 | 19.4 / 48.4 |
| en_fa | 24.9 / 53.5 | 24.8 / 51.2 | 25.3 / 51.4 | 25.1 / 51.7 | 0.1 / 0.8 | 15.3 / 44.7 | 20.7 / 49.5 | 20.1 / 49.2 |
| en_ff | 3.9 / 0 | 2.2 / 23.4 | 2.5 / 24 | 2.5 / 25.1 | 0 / 3.6 | 0 / 20 | 0 / 3.8 | 0 / 7.8 |
| en_fi | 26.2 / 59.4 | 26.9 / 61.7 | 27.3 / 62 | 27.7 / 62.6 | 0.1 / 7.2 | 19.1 / 52.5 | 20 / 53.9 | 19.9 / 54.2 |
| en_fil | 36.5 / 62.3 | 28.6 / 56.5 | 28.8 / 56.3 | 28.2 / 56.5 | 0.3 / 6.4 | 29 / 55.6 | 31.1 / 58.3 | 31.1 / 57.9 |
| en_fj | 20.1 / 48.8 | 9 / 32.5 | 11.1 / 35.6 | 10.4 / 34 | 0.1 / 4 | 0.1 / 13.9 | 0.1 / 9.4 | 0.1 / 8.3 |
| en_fo | 26.3 / 51.8 | 16.9 / 40 | 18.5 / 41.5 | 17.7 / 41.2 | 0.1 / 7.3 | 5.6 / 29.4 | 6.1 / 31.1 | 7.7 / 33.2 |
| en_fon | 3.1 / 23.5 | 1.6 / 15.6 | 1.9 / 16 | 1.6 / 15.9 | 0 / 0.3 | 0 / 1 | 0 / 1.9 | 0 / 1.6 |
| en_fr | 51.4 / 71.4 | 52.5 / 72.1 | 53.4 / 72.5 | 53.2 / 72.5 | 0.4 / 9.2 | 36.1 / 61.4 | 39.1 / 63.5 | 38.8 / 63.2 |
| en_fur | 34 / 58.7 | 28.8 / 55.8 | 24.7 / 51.4 | 28.8 / 56.1 | 0.1 / 5.6 | 0.7 / 18.1 | 3.1 / 28.8 | 3 / 30.5 |
| en_ga | 34.4 / 60.4 | 32.8 / 59.7 | 33.2 / 60 | 33.4 / 60.2 | 0.1 / 6.7 | 0.6 / 9.2 | 1.7 / 21.4 | 2 / 22.9 |
| en_gd | 21.6 / 53.6 | 19.7 / 48 | 23.2 / 51.8 | 21.3 / 50.2 | 0.1 / 2.3 | 0.2 / 6 | 0.3 / 14.9 | 0.2 / 12.8 |
| en_gl | 36 / 61.9 | 37.1 / 63.1 | 37.1 / 63 | 37.4 / 63.5 | 0.2 / 8 | 29.8 / 57.4 | 31.5 / 58.8 | 31.5 / 58.9 |
| en_gn | 9.8 / 40.4 | 11.6 / 40.9 | 10.9 / 38.9 | 12.5 / 41.8 | 0.1 / 3.9 | 0.1 / 5.6 | 0.3 / 15.2 | 0.1 / 9 |
| en_gu | 25 / 56.6 | 18.7 / 50.3 | 20.1 / 51.4 | 20.4 / 52.1 | 0 / 0.2 | 0.2 / 0.8 | 0.1 / 2 | 0 / 1.4 |
| en_ha | 28.8 / 56 | 16.9 / 44.4 | 18.9 / 45.3 | 18.7 / 47.1 | 0.1 / 2.6 | 0.2 / 6.2 | 0.2 / 12 | 0.3 / 13.9 |
| en_hi | 34.6 / 59.3 | 30.4 / 54.1 | 31 / 54.6 | 32.2 / 55.8 | 0.2 / 4.1 | 17.9 / 46.5 | 24.7 / 49.7 | 22.9 / 49.7 |
| en_hne | 27.1 / 57.5 | 19.9 / 51.2 | 17.8 / 49 | 20.4 / 52.1 | 0 / 0.2 | 1.7 / 23.8 | 2.4 / 31.2 | 2.9 / 33.2 |
| en_hr | 31.9 / 60.1 | 32.3 / 61.5 | 32.1 / 61 | 32.8 / 61.6 | 0.9 / 12.5 | 24.7 / 54.4 | 25.2 / 55.9 | 25.9 / 56.8 |
| en_ht | 25.1 / 53.9 | 18.8 / 47.7 | 23.5 / 52.1 | 22.5 / 51.3 | 0.1 / 4.7 | 0.9 / 15.9 | 4.1 / 30.2 | 3.3 / 28.8 |
| en_hu | 27.6 / 58.7 | 29.1 / 60 | 28.9 / 59.9 | 29.4 / 60.4 | 0.2 / 6.3 | 19 / 49.8 | 18.8 / 50.9 | 20.2 / 52.1 |

| | | | | | | | | |
|---|---|---|---|---|---|---|---|---|
| en_hy | 21.4 / 57.8 | 15.9 / 47.2 | 19.5 / 53.7 | 20.1 / 53.9 | 0 / 0.2 | 0.5 / 0.9 | 0.5 / 0.7 | 0.5 / 0.7 |
| en_id | 46.6 / 70.6 | 43.5 / 68.5 | 44.5 / 69 | 44.7 / 69.5 | 0.1 / 5.8 | 34.1 / 62.4 | 38.6 / 65.5 | 39 / 65.4 |
| en_ig | 16.9 / 43.6 | 13.1 / 34.6 | 16.1 / 38.7 | 16.2 / 38.5 | 0.1 / 3.4 | 0.2 / 8.6 | 0.2 / 8.7 | 0.1 / 6.5 |
| en_ilo | 25.6 / 55.8 | 15.6 / 42.4 | 17.7 / 45.7 | 17.9 / 45.8 | 0.1 / 4.5 | 3.7 / 29.6 | 3.5 / 29.6 | 1.9 / 28.3 |
| en_is | 25.3 / 52.8 | 25.9 / 53.5 | 27.5 / 54.2 | 27.3 / 54.5 | 0.8 / 11.7 | 24.1 / 51 | 24.3 / 51.7 | 24 / 51.6 |
| en_it | 31.8 / 59.9 | 32.4 / 60.3 | 32.2 / 60.3 | 32.7 / 60.5 | 0.4 / 10.8 | 24.5 / 54.2 | 24.8 / 53.7 | 25.1 / 54.6 |
| en_he | 34.7 / 62.6 | 30.5 / 59.3 | 32.5 / 61.1 | 31.4 / 60.6 | 0.1 / 0.9 | 15.7 / 46.7 | 19.9 / 50.6 | 19.7 / 50.9 |
| en_ja | 0 / 35 | 8.8 / 34.3 | 9.3 / 35.5 | 9.4 / 35.9 | 0 / 1.6 | 0.2 / 29.3 | 0 / 30.5 | 0 / 30.7 |
| en_jv | 28.6 / 57.2 | 12.1 / 37.9 | 14.5 / 42.2 | 16.4 / 44.4 | 0.1 / 5.1 | 2.7 / 25.4 | 8.1 / 39.1 | 9.3 / 40 |
| en_ka | 15 / 52.6 | 2.8 / 17.4 | 2.2 / 14.8 | 2.3 / 15.8 | 0.2 / 5.7 | 10.9 / 46.4 | 10.7 / 48.7 | 10 / 48.6 |
| en_kab | 10 / 37.9 | 0.7 / 1.4 | 0.9 / 12.4 | 0.7 / 1.9 | 0 / 3.4 | 0.1 / 9.6 | 0 / 9.6 | 0 / 5.9 |
| en_kac | 12 / 39.1 | 2.9 / 22.8 | 3.6 / 23.9 | 4 / 25.3 | 0.1 / 3.6 | 0.3 / 16.1 | 0.1 / 8.8 | 0 / 7.6 |
| en_kam | 4.3 / 28.8 | 1.2 / 2.3 | 2 / 17.7 | 1.4 / 3.6 | 0.1 / 4.5 | 0.7 / 10.3 | 0 / 4.9 | 0.1 / 5.7 |
| en_kbp | 6.9 / 30.4 | 1.4 / 14.7 | 2.3 / 16.4 | 2.1 / 16.9 | 0.1 / 2.3 | 0.1 / 5 | 0.1 / 5.5 | 0 / 3.3 |
| en_kea | 18.2 / 44.9 | 1.5 / 3.1 | 1.8 / 10 | 1.5 / 3.1 | 0.5 / 8.7 | 1.1 / 23.8 | 0.5 / 19 | 2.1 / 28.6 |
| en_kg | 16.6 / 48.2 | 0.8 / 14.7 | 1 / 16.3 | 1.4 / 19 | 0.3 / 7.8 | 0.1 / 7.9 | 0.1 / 6.9 | 0 / 6.3 |
| en_ki | 11.6 / 40.1 | 1.2 / 2.2 | 2.5 / 15.5 | 1.2 / 1.8 | 0.4 / 7.3 | 0.3 / 8.3 | 0.1 / 6.5 | 0.1 / 6 |
| en_kk | 22.2 / 56 | 17.1 / 50.2 | 18.7 / 52.2 | 18.8 / 53.1 | 0.5 / 7.7 | 13.4 / 48.7 | 12.6 / 50 | 10.4 / 49.7 |
| en_km | 3.7 / 45 | 5.3 / 40.6 | 5.2 / 40.7 | 5.4 / 41.4 | 0.1 / 0.7 | 0.9 / 29.2 | 0.2 / 28.5 | 0.1 / 24.7 |
| en_kmb | 2.9 / 28.3 | 1.7 / 19.8 | 1.8 / 19.1 | 1.4 / 19.4 | 0.2 / 6 | 0.2 / 8 | 0 / 8.4 | 0 / 4.9 |
| en_kn | 21.3 / 58.2 | 11.9 / 49.2 | 13.8 / 50.9 | 13.3 / 51.3 | 0.1 / 0.4 | 0.1 / 1.6 | 0 / 0.8 | 0 / 0.4 |
| en_ko | 14.3 / 38 | 14.7 / 36.2 | 14.9 / 36.7 | 15.1 / 37.1 | 0.3 / 3.2 | 5.5 / 24.3 | 4.9 / 25.5 | 4 / 25.6 |
| en_kr_Arab | 0.3 / 12.7 | 0.5 / 20.8 | 0.5 / 20.5 | 0.5 / 22.2 | 0.1 / 0.5 | 0 / 4 | 0 / 4.1 | 0 / 2.1 |
| en_kr | 3.9 / 31.5 | 3.8 / 28.4 | 3.3 / 27.2 | 4 / 29.9 | 0.3 / 6.3 | 0 / 11.4 | 0.1 / 13.3 | 0 / 4.4 |
| en_ks_Deva | 1.8 / 20.4 | 1.9 / 21.2 | 1.6 / 19.8 | 1.9 / 21.6 | 0.2 / 1.2 | 0.2 / 8.3 | 0 / 4.3 | 0 / 3.9 |
| en_ks | 6.7 / 38.9 | 6.6 / 35.6 | 5.8 / 32.9 | 7.2 / 36.5 | 0 / 0.3 | 0.1 / 4.7 | 0 / 3.6 | 0 / 2.9 |
| en_ku | 12.2 / 42.3 | 0.3 / 0.5 | 0.3 / 0.7 | 0.5 / 0.8 | 0.2 / 6.4 | 0.5 / 14 | 0.4 / 15.4 | 0.4 / 15.8 |
| en_ky | 13.7 / 49.1 | 10.9 / 44.7 | 13.4 / 48.1 | 12.3 / 46.9 | 0.2 / 1.7 | 1.5 / 25.9 | 1 / 27.8 | 0.6 / 26 |
| en_lb | 27.8 / 59.1 | 14.7 / 38 | 16 / 38.2 | 16.9 / 41.3 | 0.4 / 10.6 | 3.3 / 34.1 | 3.5 / 34.9 | 4 / 35.7 |
| en_lg | 9.9 / 44.7 | 3.5 / 27 | 4 / 30.3 | 3.7 / 29.6 | 0.3 / 7.4 | 0.4 / 13.9 | 0.2 / 11.5 | 0.1 / 10.9 |
| en_li | 19 / 51.6 | 18.4 / 51.1 | 15.6 / 47.1 | 18.4 / 51.5 | 0.8 / 12.5 | 2.9 / 30.5 | 5.9 / 38.9 | 6.5 / 39.8 |
| en_lij | 28.7 / 56.1 | 29.5 / 55.2 | 24.9 / 50.3 | 29.5 / 55.4 | 0.5 / 10.6 | 1.9 / 25.6 | 3.5 / 32.1 | 2.4 / 32 |
| en_lmo | 8 / 38.7 | 6.7 / 32.2 | 5 / 27.9 | 6.7 / 32.9 | 0.4 / 9.1 | 1.2 / 19.6 | 1 / 21.8 | 2.3 / 26.9 |
| en_ln | 19.1 / 50.8 | 2.1 / 18.1 | 2.3 / 18.8 | 2.7 / 20.8 | 0.4 / 9.4 | 0.7 / 11.4 | 0.1 / 9.3 | 0.2 / 10.6 |
| en_lo | 6.8 / 53.4 | 9.9 / 47.1 | 9.8 / 46.8 | 9.7 / 50.5 | 0.2 / 1 | 1.9 / 35.2 | 1.4 / 38 | 1.3 / 39.5 |
| en_lt | 27.3 / 58.1 | 29 / 60.1 | 29.1 / 60.5 | 30.1 / 61.1 | 1.5 / 13.8 | 22 / 54.3 | 21.9 / 54.2 | 22.5 / 54.4 |
| en_ltg | 26 / 56.8 | 25.3 / 57.4 | 21.5 / 53.3 | 26.3 / 58.9 | 0.3 / 8.7 | 3.2 / 29.9 | 2.5 / 29.5 | 3.5 / 31.2 |
| en_lua | 6.4 / 39.4 | 1.4 / 2.2 | 2.1 / 12 | 1.5 / 2.6 | 0.3 / 5.7 | 0.3 / 12.6 | 0.3 / 11.6 | 0.1 / 8.1 |
| en_luo | 11.7 / 41.7 | 1.1 / 2.7 | 2.4 / 18.3 | 1.1 / 2.7 | 0.2 / 2.9 | 0.1 / 20.8 | 0.1 / 11.6 | 0 / 5.4 |
| en_lus | 11.3 / 40.9 | 9.5 / 33.7 | 11.7 / 37.4 | 11.3 / 36.7 | 0.4 / 7.4 | 0.5 / 11.5 | 0.2 / 11 | 0.1 / 7.4 |
| en_lv | 28.6 / 57.9 | 32.8 / 61.8 | 32.7 / 61.9 | 32.9 / 62 | 1.4 / 11.8 | 25.2 / 54.9 | 26.1 / 56.1 | 26.2 / 56.4 |
| en_mag | 32.2 / 61.2 | 26.2 / 55.3 | 22.5 / 51.6 | 26.8 / 56.3 | 0 / 0.3 | 11.6 / 41.4 | 6.7 / 41 | 6 / 40 |
| en_mai | 16.6 / 50.7 | 8.3 / 42.5 | 12.2 / 45.3 | 10.3 / 44.3 | 0.1 / 0.9 | 4.8 / 35.9 | 3.3 / 34.7 | 3.4 / 35.6 |
| en_mg | 17.7 / 54.2 | 17 / 50.8 | 18.5 / 53.1 | 18 / 52.8 | 0.3 / 7.1 | 1.7 / 20.1 | 0.5 / 17 | 0.2 / 13.5 |
| en_mi | 18.9 / 45.9 | 18.6 / 44.1 | 16.8 / 42 | 19.1 / 45.2 | 0.4 / 7.9 | 0.5 / 11.5 | 0.4 / 12.1 | 0.3 / 10.8 |
| #N/A en_min | 23.4 / 55.6 | 12.4 / 44.9 | 14.1 / 45.6 | 10 / 43.2 | 0.2 / 6.5 | 7.6 / 38.5 | 8 / 39.2 | 5.6 / 37.7 |
| en_mk | 35.6 / 63.2 | 36.6 / 64.8 | 37.7 / 65.2 | 37.4 / 65 | 1 / 3.4 | 30.8 / 59.6 | 30.5 / 59.8 | 31.2 / 60.3 |
| en_ml | 17.1 / 57.3 | 9.2 / 47.1 | 9.4 / 47 | 9.8 / 49.2 | 0.5 / 8.6 | 5.8 / 42.8 | 6.4 / 46.1 | 5.6 / 47.7 |
| en_mn | 14.8 / 47.7 | 12.2 / 44.4 | 14.8 / 47.8 | 14.7 / 47.7 | 0.1 / 0.7 | 0.6 / 0.8 | 0.6 / 6 | 0.3 / 2.4 |
| en_mni | 7.8 / 44.3 | 6.6 / 38 | 7 / 39.2 | 7.4 / 40.4 | 0.1 / 0.2 | 0 / 4.3 | 0 / 4.8 | 0 / 3 |
| en_mos | 4.5 / 26.4 | 0.9 / 2.3 | 1.2 / 16.6 | 0.8 / 2.2 | 0.2 / 6.1 | 0.3 / 14.2 | 0 / 6.5 | 0 / 3.5 |
| en_mr | 17.6 / 52.1 | 10 / 39.1 | 9.9 / 37.7 | 11.8 / 42 | 0.1 / 0.9 | 10.7 / 42.8 | 8.6 / 43.1 | 8 / 44 |
| en_ms | 42.2 / 68.5 | 37.3 / 64.8 | 39.3 / 66.2 | 38.1 / 65.7 | 1.3 / 13.9 | 35.8 / 65 | 37.2 / 65.6 | 36.2 / 65.4 |
| en_mt | 35.6 / 70.8 | 51.7 / 71.7 | 52.7 / 72.2 | 52.5 / 72.2 | 0.4 / 8.9 | 5.2 / 34.3 | 5.4 / 38.5 | 2.9 / 36.7 |
| en_my | 2.9 / 40.1 | 3.3 / 40.5 | 3.1 / 39.1 | 3.5 / 38.3 | 0 / 0.2 | 0.1 / 5.7 | 0 / 9.9 | 0.1 / 12.9 |
| en_ne | 16.7 / 49.7 | 10.3 / 41.8 | 10.4 / 42.1 | 11.7 / 45.2 | 0.4 / 1.9 | 3.3 / 34.8 | 2.6 / 34.4 | 2.1 / 34.9 |
| en_nl | 27.5 / 57.8 | 28.6 / 59.3 | 28.7 / 59.4 | 28.8 / 59.4 | 1.6 / 15.9 | 22.5 / 53.7 | 22.4 / 54.3 | 23.2 / 54.6 |
| en_nn | 28.1 / 56.1 | 33.6 / 61.3 | 34.6 / 61.6 | 34.3 / 61.8 | 3 / 15.2 | 27.6 / 56.1 | 29.2 / 57.2 | 29.8 / 57.6 |
| en_no | 32.9 / 61.2 | 34.7 / 62.8 | 34.4 / 62.4 | 34.5 / 62.8 | 2.5 / 15.8 | 29.1 / 57.5 | 30.8 / 59.3 | 31.1 / 59.8 |
| en_nso | 23.7 / 53.3 | 12.8 / 37.7 | 15.7 / 41.1 | 14.5 / 40.5 | 0.3 / 4.5 | 0.6 / 11.8 | 0.2 / 10 | 0.2 / 9.1 |
| en_nus | 6.5 / 31.5 | 4.8 / 29.3 | 3.9 / 27.7 | 5 / 30.2 | 0.2 / 4.3 | 0 / 2.7 | 0 / 2.1 | 0 / 0.7 |
| en_ny | 13.6 / 48.7 | 8.1 / 38.6 | 11.5 / 44.9 | 10.6 / 43.5 | 0.4 / 8.1 | 0.7 / 16.4 | 0.4 / 15.7 | 0.4 / 17.2 |
| en_oc | 34 / 61.1 | 38.1 / 63 | 39.6 / 64.3 | 39.7 / 64.7 | 0.4 / 10.5 | 6.9 / 38.6 | 7.8 / 40.1 | 8.6 / 41.1 |
| en_om | 5.4 / 43.3 | 1.1 / 22.8 | 1.7 / 29.3 | 1.8 / 29.1 | 0.1 / 7.4 | 0.1 / 10.6 | 0 / 7.9 | 0 / 6.3 |
| en_or | 15.1 / 50.4 | 12.4 / 46.7 | 13.3 / 47.5 | 13.1 / 47.8 | 0.1 / 0.2 | 0.4 / 0.9 | 0.3 / 0.8 | 0.3 / 0.8 |
| en_pa | 24.5 / 51 | 20.7 / 45.7 | 21.4 / 46.5 | 21.1 / 46.5 | 0.1 / 0.3 | 0.9 / 6 | 0.3 / 1.7 | 0.2 / 1.8 |
| en_pag | 17.3 / 49.6 | 7.1 / 35.4 | 10.2 / 38.4 | 11.1 / 39.2 | 0.6 / 7.4 | 0.7 / 20.9 | 4.3 / 30.7 | 4.9 / 32.2 |
| en_pap | 37.8 / 61.5 | 15.5 / 46.5 | 20 / 49.4 | 16 / 47 | 0.6 / 11.4 | 2.7 / 28.5 | 5.7 / 34.6 | 6.7 / 36.6 |
| en_pl | 22.2 / 52.1 | 23.3 / 53.7 | 23.6 / 53.8 | 23.7 / 53.9 | 1 / 11.6 | 16.7 / 46.7 | 16.2 / 47.9 | 17.5 / 48.3 |
| en_ps | 15.5 / 40.8 | 15.1 / 38.7 | 15.3 / 38.8 | 15.4 / 39.1 | 0.1 / 0.9 | 0.4 / 12.1 | 0.2 / 10 | 0.1 / 8.9 |
| en_pt | 47.9 / 69.6 | 49.9 / 71.5 | 51 / 72 | 51.4 / 72.3 | 3 / 17.8 | 38.8 / 63.5 | 41 / 65.1 | 40.7 / 65.4 |
| en_quy | 3.6 / 31.2 | 1.5 / 22.8 | 2 / 18.6 | 1.3 / 25.5 | 0.2 / 8.7 | 0.3 / 14.5 | 0.1 / 8.8 | 0 / 6.7 |
| en_rn | 12.9 / 47.3 | 5 / 31.8 | 5.9 / 32.6 | 5.9 / 30.7 | 0.3 / 5.5 | 0.5 / 13.9 | 0.5 / 15.5 | 0.2 / 12.9 |
| en_ro | 38.4 / 63.6 | 42 / 66.2 | 42.3 / 66.2 | 42.4 / 66.5 | 4.4 / 20 | 32.6 / 58.1 | 34.8 / 60.5 | 35.4 / 60.9 |
| en_ru | 32.3 / 59 | 32.2 / 59.6 | 33 / 60.1 | 32.5 / 59.8 | 1.1 / 4.4 | 22.5 / 49.4 | 23.9 / 52.4 | 25.4 / 53.2 |
| en_rw | 21.1 / 53.8 | 1.8 / 14.6 | 0.3 / 9.6 | 0.9 / 11.9 | 0.5 / 6.9 | 0.8 / 15.7 | 0.5 / 15.5 | 0.5 / 17 |
| en_sa | 1.7 / 31.6 | 1.6 / 27 | 1.8 / 30.5 | 2 / 31.7 | 0.1 / 0.9 | 0.1 / 11.6 | 0.1 / 15.2 | 0.1 / 14.7 |
| #N/A en_sc | 30 / 58.1 | 24.2 / 52.8 | 19.3 / 47.4 | 24.5 / 53.3 | 0.5 / 10.3 | 2.7 / 27 | 0.8 / 21.4 | 1.1 / 23.8 |
| en_scn | 17.7 / 50.5 | 15.3 / 48.5 | 13.8 / 45.3 | 15.6 / 49.1 | 0.5 / 9.3 | 1.3 / 22.6 | 1.5 / 26.3 | 1.8 / 28 |
| en_sd | 23 / 50.1 | 19.8 / 43.7 | 17.5 / 42 | 18.4 / 42.5 | 0.1 / 0.5 | 0.8 / 6 | 0.7 / 13.7 | 0.4 / 10 |
| en_sg | 9 / 38 | 6 / 30.1 | 5.8 / 29.2 | 6.4 / 30.7 | 0.2 / 8.3 | 0.2 / 10.1 | 0.2 / 10.6 | 0.1 / 5.3 |

| | | | | | | | |
|---|---|---|---|---|---|---|---|
| en_shn | 5.5 / 42 | 5.4 / 37.5 | 5 / 37.3 | 5.6 / 39.6 | 0.2 / 1.2 | 0.1 / 3.1 | 0 / 2.6 | 0 / 3.6 |
| en_si | 14.7 / 48 | 13.5 / 46 | 13.4 / 45.2 | 14.2 / 47.6 | 0 / 0.4 | 0.5 / 1 | 0 / 0.7 | 0.1 / 0.8 |
| en_sk | 35.2 / 61.6 | 35.7 / 62.5 | 36.3 / 62.9 | 36.8 / 63.1 | 1.5 / 13.9 | 27.7 / 55.4 | 28.6 / 56 | 26.5 / 55.8 |
| en_sl | 31.7 / 58.7 | 32.3 / 59.9 | 32.9 / 60.2 | 33.3 / 60.6 | 0.8 / 12.7 | 23.7 / 52.4 | 25.1 / 54.1 | 24.8 / 53.3 |
| en_sm | 26.8 / 50.9 | 17.9 / 43.6 | 20.8 / 46.5 | 20.5 / 46.8 | 0.5 / 7.4 | 0.6 / 14 | 0.3 / 11.4 | 0.2 / 8.9 |
| en_sn | 12.6 / 48.8 | 7 / 38.2 | 9.3 / 42.3 | 8.8 / 41.8 | 0.5 / 7 | 0.8 / 16.6 | 0.4 / 17 | 0.3 / 16 |
| en_so | 12.6 / 47.3 | 9.4 / 41.8 | 10.3 / 43.4 | 10.2 / 43 | 0.2 / 6.8 | 0.7 / 15.3 | 0.2 / 12.9 | 0.2 / 13.2 |
| en_sq | 33.2 / 60.6 | 29.8 / 58.6 | 30.9 / 59.5 | 30.4 / 59.3 | 1.2 / 11.7 | 28.5 / 56.8 | 28.7 / 56.6 | 29 / 57.7 |
| en_sr | 35.9 / 62.1 | 34.9 / 61.6 | 35.8 / 62.4 | 36.2 / 62.7 | 0.5 / 2 | 29.2 / 57.4 | 29.9 / 58.4 | 30.4 / 58.1 |
| en_ss | 10.7 / 49.2 | 4.1 / 33.4 | 5.1 / 34.7 | 5.3 / 36 | 0.5 / 7.2 | 0.5 / 14.9 | 0.1 / 11.5 | 0.1 / 9.5 |
| en_st | 18.9 / 49.2 | 9.6 / 34.7 | 15.4 / 43.1 | 14.1 / 41.6 | 0.6 / 8.7 | 1 / 15.3 | 0.5 / 13.3 | 0.3 / 11.9 |
| en_su | 15.9 / 48.3 | 14.4 / 45.5 | 18.7 / 50.6 | 18.5 / 50.4 | 0.5 / 9 | 5.8 / 35.5 | 7 / 37.9 | 7.8 / 38.3 |
| en_sv | 44.2 / 68.1 | 47 / 70.8 | 47.2 / 70.8 | 47 / 70.8 | 2.5 / 15.8 | 37.7 / 64.2 | 39.4 / 65 | 39.1 / 65.3 |
| en_sw | 32.6 / 61.2 | 29.3 / 57.6 | 30.4 / 58.1 | 30.4 / 58.5 | 0.3 / 9 | 1.5 / 21.1 | 1.4 / 22.8 | 1.3 / 25.7 |
| en_szl | 27 / 56.7 | 24.9 / 56.1 | 21.6 / 51.8 | 26.2 / 56.7 | 0.5 / 9.3 | 6.9 / 36.7 | 4.9 / 35.1 | 4.3 / 35.8 |
| en_ta | 19.8 / 59.4 | 12.4 / 51.4 | 13 / 52.6 | 13.9 / 54.1 | 0.4 / 10.7 | 8.6 / 47.8 | 7.8 / 49.2 | 8.1 / 49.5 |
| en_taq_Tfng | 0.7 / 19.3 | 0.6 / 21.2 | 0.5 / 20.4 | 0.7 / 21.7 | 0.1 / 0.4 | 0 / 7.8 | 0 / 2 | 0 / 1.4 |
| en_taq | 4.2 / 25.9 | 3 / 24.6 | 3.3 / 24.7 | 3.6 / 25.7 | 0.2 / 8 | 0.1 / 4.7 | 0 / 2.8 | 0 / 4.7 |
| en_te | 24.8 / 60.1 | 16.7 / 51.4 | 17.6 / 52.1 | 17.5 / 52.7 | 0.7 / 9 | 10.9 / 45.9 | 8.3 / 47.5 | 10.5 / 49.6 |
| en_tg | 24 / 54 | 6.1 / 26.6 | 9.2 / 31.3 | 10.2 / 32.6 | 0.1 / 0.6 | 1.4 / 21 | 0.8 / 22.6 | 0.9 / 24.4 |
| en_th | 6 / 51.4 | 8 / 44.1 | 7.6 / 43.1 | 8 / 44.3 | 0.1 / 10 | 7.1 / 48.5 | 7.5 / 48.7 | 4 / 49.2 |
| en_ti | 6 / 27.9 | 1.8 / 15.7 | 2.5 / 18.6 | 2.5 / 18.9 | 0.1 / 0.3 | 0 / 1 | 0 / 0.8 | 0 / 0.6 |
| en_tk | 13.7 / 46.2 | 13.9 / 44.5 | 17.5 / 50.7 | 17.3 / 49.9 | 0.2 / 6.9 | 0.8 / 19.9 | 0.9 / 22.9 | 0.7 / 22.6 |
| en_tn | 23 / 51 | 10.6 / 34.8 | 14 / 38.6 | 12.2 / 37.7 | 0.5 / 8.2 | 0.6 / 14.5 | 0.3 / 11.5 | 0.2 / 10 |
| en_tpi | 18.3 / 42.8 | 1.1 / 2.7 | 3.4 / 22.2 | 1 / 2.3 | 0.6 / 5.4 | 0.7 / 16.7 | 1.4 / 22.8 | 0.3 / 11.3 |
| en_tr | 30.1 / 61.8 | 29.9 / 60.5 | 31 / 61.2 | 30.5 / 61.4 | 1.2 / 14.4 | 16.9 / 51 | 15 / 51.6 | 15.7 / 52.1 |
| en_ts | 23.3 / 53.2 | 9.1 / 34.7 | 12 / 38.1 | 10.3 / 37 | 0.5 / 7.8 | 0.3 / 6.9 | 0.2 / 10.5 | 0.2 / 10.3 |
| en_tt | 18.4 / 50.6 | 7 / 31.5 | 10.2 / 36.5 | 13.1 / 42.4 | 0.1 / 1 | 1.8 / 25.1 | 0.5 / 21.7 | 0.5 / 23.4 |
| en_tum | 10.5 / 38.5 | 1.4 / 2.2 | 2.2 / 15.3 | 1.5 / 2.5 | 0.4 / 7.4 | 0.2 / 14.8 | 0.4 / 13.8 | 0.1 / 10 |
| en_ug | 13.8 / 50.6 | 5.9 / 33.4 | 6.9 / 34.2 | 8.8 / 39.8 | 0 / 0.3 | 0.2 / 1.9 | 0.1 / 1.2 | 0.1 / 1.6 |
| en_uk | 31.5 / 59.1 | 30.6 / 58.9 | 30.9 / 59.4 | 31.1 / 59.4 | 1.6 / 7.8 | 23.3 / 52.3 | 24.4 / 53.5 | 23.5 / 53.4 |
| en_umb | 2.8 / 31.2 | 1.6 / 2.9 | 1.4 / 10.5 | 1.4 / 2.5 | 0.2 / 6.1 | 0.1 / 11.5 | 0 / 5.6 | 0 / 4.6 |
| en_ur | 23.7 / 51.1 | 18.8 / 44 | 20.1 / 44.8 | 20.3 / 45.4 | 0.1 / 0.8 | 7.7 / 33.4 | 6.1 / 34.5 | 2.9 / 30.3 |
| en_uz | 17.6 / 55.7 | 0.6 / 1.3 | 0.5 / 1.1 | 0.6 / 1.1 | 0.1 / 6.8 | 2.5 / 30.3 | 3.5 / 38.5 | 4.6 / 43.9 |
| en_vec | 22.8 / 54.8 | 25.1 / 56.4 | 22 / 52.2 | 25.2 / 56.7 | 0.4 / 12.8 | 3.5 / 29.6 | 5.6 / 37.7 | 7.6 / 41.2 |
| en_vi | 42.3 / 59.6 | 39.1 / 57.2 | 40.5 / 58.2 | 40.5 / 58.6 | 2.2 / 10.3 | 35.2 / 54 | 35.7 / 54.7 | 34 / 54.4 |
| en_war | 31.7 / 59.3 | 9.8 / 31.1 | 18.5 / 44.9 | 13.7 / 44 | 0.7 / 7.5 | 5.9 / 33.5 | 8.7 / 38.8 | 10.4 / 40.3 |
| en_wo | 6.8 / 31.8 | 1.2 / 13.2 | 1 / 12.6 | 1.4 / 15.4 | 0.3 / 7.7 | 0.1 / 5.5 | 0.1 / 7 | 0 / 3.5 |
| en_xh | 15.2 / 54.7 | 11.3 / 47.4 | 12.5 / 50.1 | 12.5 / 49.2 | 0.2 / 7.7 | 0.4 / 15.4 | 0.3 / 16.1 | 0.2 / 16.2 |
| en_yi | 11.9 / 41.8 | 8.7 / 34.8 | 6.6 / 36.3 | 7.9 / 32.5 | 0.2 / 0.7 | 0.8 / 13.6 | 0.9 / 17.6 | 0.5 / 16.3 |
| en_yo | 6 / 27.4 | 2.6 / 19.7 | 2.7 / 20.3 | 2.9 / 20.7 | 0.3 / 6.3 | 0.1 / 10.6 | 0.1 / 5.2 | 0 / 3.2 |
| en_yue | 2 / 20.8 | 0.3 / 2.2 | 0.2 / 2 | 0.3 / 2 | 0.1 / 2.5 | 0.1 / 24.7 | 1.2 / 25.5 | 0.8 / 25.3 |
| en_zh_Hant | 15.4 / 17.6 | 5.5 / 30.4 | 6.3 / 31.8 | 36.5 / 32.1 | 0.1 / 3 | 1.2 / 21.4 | 3.6 / 22.3 | 1.3 / 21.9 |
| en_zh | 31.2 / 29.3 | 5.8 / 35.7 | 6.1 / 36.6 | 41.3 / 36.5 | 2.9 / 3.3 | 29.2 / 26.2 | 30.1 / 26.7 | 29.8 / 26 |
| en_zu | 19.7 / 58.8 | 11.2 / 44.7 | 13.8 / 48.6 | 13.4 / 47.6 | 0.3 / 7.1 | 0.4 / 13.9 | 0.3 / 17 | 0.2 / 17 |
| ace_Arab_en | 13.6 / 37.9 | 17.1 / 43.5 | 16.6 / 42.7 | 19.1 / 45.8 | 0 / 0.7 | 0.8 / 13.6 | 1.5 / 16.5 | 1.6 / 18.5 |
| ace_en | 31.3 / 53.9 | 23.6 / 49 | 21.1 / 46.9 | 25 / 50.8 | 0 / 3 | 3.3 / 22.9 | 8.5 / 32.8 | 6.8 / 33.7 |
| acm_en | 40.1 / 65 | 34.2 / 60.8 | 35.8 / 61.7 | 35.2 / 61.6 | 0.1 / 1.6 | 25.9 / 52.3 | 28.4 / 55.3 | 28 / 55.1 |
| acq_en | 42.4 / 66.7 | 36.5 / 62.5 | 37.3 / 62.9 | 37.4 / 63.3 | 0.1 / 2 | 23.5 / 51.4 | 30.1 / 57.1 | 29.5 / 57.1 |
| aeb_en | 36 / 61.4 | 28.5 / 55.5 | 30.4 / 57.1 | 30.1 / 56.9 | 0.1 / 2.4 | 19 / 44.5 | 22.4 / 49 | 24.3 / 51.3 |
| af_en | 59.2 / 77.1 | 58.7 / 76.8 | 60.1 / 77.9 | 60 / 77.7 | 2.4 / 18 | 51.7 / 71.5 | 54.3 / 74 | 54.9 / 74 |
| ajp_en | 46.6 / 68.9 | 38.1 / 62.8 | 37.7 / 62.4 | 39.6 / 64 | 0.1 / 1.5 | 32.7 / 58.3 | 32.7 / 58 | 31.2 / 58.7 |
| am_en | 36.5 / 61.8 | 32.6 / 57.9 | 36.9 / 61.1 | 35.9 / 60.6 | 0.2 / 4.8 | 20.3 / 47.1 | 22.7 / 49.6 | 22.9 / 50 |
| apc_en | 42.5 / 66.7 | 34.1 / 60.6 | 34.7 / 60.7 | 36.2 / 62.1 | 0.1 / 2 | 26.2 / 52.1 | 28.7 / 55 | 29.3 / 56.3 |
| ar_MA_en | 32.2 / 0 | 23 / 50.3 | 24.5 / 51.2 | 25.1 / 51.7 | 0 / 1.4 | 15.3 / 43.4 | 19.3 / 45.5 | 18.1 / 46.8 |
| ar_en | 45.3 / 68.6 | 41.7 / 66.4 | 44.4 / 68.1 | 43.1 / 67.5 | 0.3 / 4.9 | 32.9 / 58.1 | 34.9 / 60.7 | 35.5 / 60.6 |
| ars_en | 43.8 / 67.8 | 40.3 / 65.4 | 42.7 / 66.7 | 41.7 / 66.5 | 0.1 / 2 | 28.3 / 52.4 | 34.7 / 59.9 | 34.5 / 60 |
| arz_en | 37.2 / 62.7 | 30.2 / 57.5 | 29.1 / 56.5 | 31.7 / 58.9 | 0.1 / 1.5 | 24.7 / 51.1 | 25 / 52.9 | 25.3 / 53.4 |
| as_en | 33.9 / 59.8 | 29.5 / 55.7 | 31.8 / 57.4 | 31.2 / 57.3 | 0.1 / 1.6 | 14.7 / 43.3 | 19.4 / 47.4 | 19.4 / 47.9 |
| ast_en | 43 / 66.4 | 37.3 / 62.5 | 39.7 / 64.2 | 39.8 / 64.5 | 1.2 / 13.3 | 26.2 / 56.1 | 29.2 / 57 | 31.6 / 58.3 |
| awa_en | 42.8 / 67.4 | 34.6 / 61.6 | 33.5 / 60.2 | 36.6 / 62.9 | 0.1 / 2.5 | 19.8 / 47.2 | 23.9 / 53.4 | 23.8 / 53.8 |
| ay_en | 12.1 / 33.7 | 6.1 / 27.4 | 7.9 / 28.4 | 8.3 / 29.6 | 0 / 2.3 | 0.5 / 16.4 | 3 / 18.8 | 2.8 / 19.1 |
| az_en | 27.2 / 56.5 | 19.7 / 47.7 | 20.7 / 47.2 | 25.1 / 52.7 | 1.3 / 15.7 | 18.8 / 47.5 | 20.2 / 49.5 | 20 / 48.7 |
| azb_en | 20.8 / 47.7 | 4.8 / 30.2 | 6.7 / 31.7 | 6.3 / 32 | 0.1 / 3.4 | 2.1 / 23.3 | 4.2 / 31.2 | 5.4 / 30.1 |
| ba_en | 33.6 / 60.1 | 6.7 / 25.7 | 8.2 / 26.8 | 10 / 29.2 | 0.1 / 2.9 | 13.1 / 36.1 | 19.7 / 46.7 | 19.3 / 46.8 |
| ban_en | 38.2 / 62.3 | 29 / 55.6 | 25.5 / 52.4 | 30.3 / 57 | 0.2 / 5.4 | 8.8 / 34.8 | 15.9 / 43.9 | 16.7 / 43.9 |
| be_en | 23.7 / 55.4 | 17.9 / 49.3 | 17.6 / 49.4 | 21.6 / 52.9 | 0.8 / 11.5 | 18.4 / 48.3 | 19.6 / 50.9 | 19.6 / 51.5 |
| bem_en | 29.6 / 52.4 | 7.4 / 30.4 | 8.6 / 31.5 | 10 / 33.2 | 0 / 2.5 | 1.2 / 14.7 | 5.4 / 22.2 | 4 / 21.9 |
| ber_en | 19.7 / 0 | 16.6 / 41.6 | 14.6 / 39.9 | 17.6 / 43.1 | 0.1 / 1.7 | 0.1 / 14.8 | 0.5 / 13.7 | 0.2 / 11.9 |
| bg_en | 43.1 / 68.3 | 42.2 / 67.8 | 43.6 / 68.9 | 43.2 / 68.5 | 1.8 / 15.5 | 34.1 / 62.7 | 36.8 / 64.1 | 37.8 / 64.5 |
| bho_en | 34.6 / 60.7 | 28.5 / 56 | 25.5 / 53.5 | 28.9 / 56.8 | 0.1 / 2.4 | 16.5 / 43.8 | 19.6 / 48.5 | 17 / 48 |
| bjn_Arab_en | 19.2 / 43.3 | 21.7 / 47.2 | 20.3 / 46.1 | 23.8 / 49.8 | 0 / 0.9 | 0.5 / 15.3 | 1.7 / 18.1 | 1.8 / 18.1 |
| bjn_en | 40.1 / 63.3 | 30.2 / 55.9 | 26.1 / 52.3 | 32 / 57.5 | 0.1 / 3.5 | 10.9 / 40.4 | 17.6 / 44.4 | 20.3 / 45.5 |
| bm_en | 19.6 / 42.2 | 14.1 / 37.1 | 12.9 / 36.3 | 15.5 / 39.4 | 0 / 2 | 1.7 / 12.8 | 1.9 / 17.8 | 2.3 / 18.8 |
| bn_en | 38.7 / 64.2 | 34.9 / 60.6 | 37 / 62.5 | 36.2 / 61.8 | 0.2 / 2.7 | 21.8 / 50.7 | 25.2 / 53.9 | 26.3 / 54.8 |
| bo_en | 15.6 / 40.9 | 12.6 / 38.1 | 12.6 / 38.2 | 14.6 / 41.1 | 0 / 1 | 1.9 / 20.5 | 5.9 / 29.6 | 5.6 / 29 |
| bs_en | 45.2 / 68.8 | 44.1 / 68.3 | 45 / 69.1 | 44.9 / 69.2 | 1.5 / 13.7 | 36.2 / 62.3 | 38.5 / 64.5 | 39 / 65 |
| bug_en | 23.1 / 48.4 | 17.7 / 43.6 | 15.7 / 42 | 18.9 / 45.3 | 0.1 / 3.3 | 1.8 / 18.7 | 5.6 / 28 | 5.9 / 27.6 |
| ca_en | 49 / 71.1 | 48.5 / 71 | 50.3 / 72.2 | 50 / 72 | 2.2 / 16.1 | 41.8 / 66.6 | 42.5 / 67.1 | 43 / 67.2 |
| ceb_en | 45.6 / 66.8 | 34.8 / 60.2 | 34.1 / 59.4 | 36.6 / 61.2 | 0.3 / 5.7 | 31.9 / 56.4 | 32.2 / 57.2 | 33.6 / 58.5 |
| cjk_en | 11.3 / 32.7 | 4.8 / 23.8 | 5 / 24.7 | 5.6 / 26.5 | 0 / 2.3 | 1.2 / 16.2 | 2 / 17.7 | 2.3 / 18.8 |

| | | | | | | | | |
|---|---|---|---|---|---|---|---|---|
| ckb_en | 37.5 / 61.6 | 28.3 / 53.1 | 34.4 / 59 | 34.2 / 59 | 0.2 / 4.4 | 18 / 42.1 | 22.5 / 48.6 | 22.8 / 49.5 |
| crh_Latn_en | 35.9 / 61.8 | 29.2 / 56.8 | 25 / 53.1 | 29.1 / 57 | 0.1 / 4.8 | 16.1 / 40.3 | 18.8 / 45.2 | 17.8 / 46.6 |
| cs_en | 41.2 / 66.4 | 41.3 / 66.6 | 42.3 / 67.3 | 42.3 / 67.3 | 2.1 / 18.1 | 34.7 / 61.4 | 35.8 / 62.5 | 36.1 / 62.9 |
| cy_en | 60 / 77.1 | 57.5 / 75.6 | 60.6 / 77.7 | 60.8 / 77.7 | 1.2 / 12.5 | 43.9 / 65.5 | 48.5 / 69.2 | 47.5 / 68.6 |
| da_en | 49.5 / 71.2 | 50.1 / 72.3 | 51.8 / 73.3 | 50.8 / 72.8 | 1.6 / 15.1 | 44.5 / 68.2 | 44.1 / 68.9 | 44.7 / 68.9 |
| de_en | 45.8 / 69.2 | 45.5 / 69.5 | 46.4 / 70.1 | 46.5 / 70 | 1.6 / 15.9 | 37.8 / 63.3 | 39.5 / 64.9 | 38 / 64.6 |
| din_en | 12.7 / 32.9 | 10.3 / 31.1 | 9.5 / 31.1 | 11 / 32.8 | 0 / 2.2 | 1.2 / 18.8 | 1.6 / 16.3 | 2.4 / 19.3 |
| dyu_en | 8.5 / 30 | 7.9 / 31.5 | 7.9 / 31.3 | 8.9 / 33.5 | 0 / 2.3 | 1.1 / 15.3 | 1.9 / 17.4 | 2.3 / 18.2 |
| dz_en | 17 / 44 | 12.9 / 40.4 | 12 / 39.2 | 13.7 / 41.6 | 0 / 0.8 | 0.7 / 16.4 | 2 / 22.4 | 1.9 / 22.7 |
| ee_en | 17.8 / 42 | 10.3 / 33.3 | 14.3 / 37.4 | 13.3 / 36.6 | 0.1 / 4.7 | 2.5 / 17.5 | 5 / 23.2 | 4.4 / 24.3 |
| el_en | 40 / 64.9 | 37.8 / 63.3 | 38.9 / 64.2 | 38.8 / 64 | 1.3 / 12.8 | 32.3 / 58.4 | 32.9 / 59.7 | 33.1 / 60 |
| eo_en | 47.8 / 70.1 | 45.8 / 68.5 | 47.2 / 69.4 | 47.1 / 69.5 | 9.8 / 31 | 38.3 / 63.3 | 40.4 / 64.7 | 40.9 / 65.3 |
| es_en | 33.8 / 61.4 | 31.5 / 61 | 32.2 / 61.3 | 32.5 / 61.4 | 4.5 / 23.9 | 26.4 / 56 | 27.6 / 57.8 | 27.7 / 57.9 |
| et_en | 38.8 / 64.6 | 39.4 / 64.9 | 40.3 / 65.6 | 39.6 / 65.5 | 7.9 / 28 | 31.8 / 59.2 | 32.5 / 59.8 | 32.6 / 59.8 |
| eu_en | 34.4 / 61.2 | 32.6 / 58.6 | 34.4 / 60 | 34.4 / 60.4 | 4.4 / 23 | 25.5 / 53.8 | 24.7 / 52.9 | 26.2 / 54.5 |
| fa_AF_en | 41.3 / 0 | 37.9 / 63 | 38.7 / 63.2 | 39.4 / 64.1 | 2.2 / 11.6 | 28.7 / 53.7 | 31.1 / 56.4 | 32.4 / 57.8 |
| fa_en | 40.9 / 65.6 | 37.7 / 62.9 | 40.3 / 64.8 | 39.4 / 64.3 | 3.9 / 16.4 | 31.1 / 57.6 | 30.8 / 57.2 | 32.3 / 58.7 |
| ff_en | 13.5 / 0 | 10.5 / 33.2 | 10.1 / 33.1 | 12.4 / 35.8 | 0.2 / 10.5 | 2.6 / 15.7 | 2.5 / 16.9 | 2.4 / 18.2 |
| fi_en | 36.9 / 63 | 36.1 / 62.6 | 36.7 / 63 | 37.1 / 63.3 | 3.3 / 22.9 | 27.9 / 55.8 | 29.8 / 57.4 | 30.5 / 58.3 |
| fil_en | 51.5 / 71.3 | 47.1 / 68.2 | 49.5 / 70 | 49.6 / 70.1 | 2.5 / 17.9 | 40.4 / 63.9 | 40.7 / 64.4 | 39.7 / 64.6 |
| fj_en | 21.6 / 45.4 | 12.2 / 36 | 14.6 / 39.2 | 14.5 / 38.8 | 0.3 / 10.8 | 5.1 / 24.8 | 5.2 / 24.1 | 4.3 / 26.2 |
| fo_en | 37.1 / 59.5 | 27.5 / 52.8 | 31 / 56.1 | 36.7 / 59.9 | 6.1 / 26 | 30.1 / 54.4 | 32.7 / 56.5 | 33.9 / 57.4 |
| fon_en | 12.9 / 35.4 | 3.9 / 25.1 | 5.6 / 26.3 | 6.4 / 27.8 | 0.1 / 7.9 | 0.8 / 12.6 | 1.5 / 17.7 | 1.8 / 18.3 |
| fr_en | 47.9 / 70.1 | 46.1 / 69.5 | 47.2 / 70.3 | 47.1 / 70.1 | 2.8 / 21.5 | 39.1 / 64.5 | 39.9 / 64.9 | 40 / 65.1 |
| fur_en | 45.3 / 68.6 | 36.9 / 63 | 30.8 / 58 | 37.4 / 63.4 | 3.6 / 23.8 | 24.2 / 50.3 | 24.2 / 53.4 | 23 / 54.7 |
| ga_en | 45.9 / 68.1 | 43.4 / 67 | 45.8 / 69 | 45.9 / 68.8 | 3.3 / 21.1 | 29.4 / 53.8 | 31.9 / 57.4 | 34.2 / 59.4 |
| gd_en | 36.8 / 60.9 | 33.4 / 58.9 | 37.3 / 61.9 | 37.5 / 61.6 | 2.1 / 18.1 | 21.7 / 46.8 | 22.8 / 49.1 | 22.1 / 49.6 |
| gl_en | 45.6 / 68.9 | 44.1 / 68.5 | 46 / 69.7 | 45.8 / 69.6 | 5.3 / 19.6 | 35.9 / 62.7 | 36.6 / 63.8 | 36.9 / 64.4 |
| gn_en | 27.7 / 52 | 19.5 / 45 | 18.2 / 44 | 20.8 / 46.9 | 0.6 / 11.9 | 5 / 26.3 | 8 / 30.8 | 8.1 / 29.9 |
| gu_en | 44.6 / 68.4 | 39.7 / 64.9 | 41.9 / 66.1 | 41.2 / 65.5 | 2.1 / 13.4 | 25.4 / 54.1 | 28 / 56.7 | 28.7 / 57.2 |
| ha_en | 36.4 / 58.7 | 31.8 / 54.5 | 35.9 / 57.9 | 35.4 / 57.6 | 0.8 / 15 | 17.2 / 40.1 | 21.6 / 44.8 | 21.1 / 45.7 |
| hi_en | 44.4 / 68.3 | 40.4 / 65.1 | 42.9 / 66.6 | 41.7 / 65.9 | 2.9 / 12.9 | 26.1 / 53.9 | 30.9 / 58.8 | 32.8 / 60.1 |
| hne_en | 52.2 / 73.8 | 40.1 / 65.5 | 35.1 / 61.7 | 41.9 / 66.9 | 0.5 / 4.9 | 22.4 / 52.1 | 21.5 / 53.8 | 23.2 / 54.5 |
| hr_en | 38.7 / 64.4 | 39 / 64.8 | 39.9 / 65.4 | 39.6 / 65.3 | 6.4 / 25.6 | 33.8 / 60.3 | 34.9 / 61 | 35.8 / 62 |
| ht_en | 39.9 / 64 | 39.3 / 63.5 | 40.5 / 64.3 | 40.9 / 65.1 | 0.9 / 10.9 | 26.9 / 52.6 | 29.4 / 55 | 29.4 / 56.5 |
| hu_en | 37.3 / 63.7 | 37.5 / 63.9 | 38.2 / 64.4 | 38.1 / 64.5 | 3.6 / 22.9 | 29.1 / 57.1 | 29.6 / 58 | 29.7 / 58 |
| hy_en | 42.3 / 67 | 33.5 / 59.4 | 40.7 / 65.2 | 40.1 / 64.8 | 2.5 / 13.6 | 27 / 54 | 31.1 / 58.5 | 31.6 / 59.7 |
| id_en | 46.5 / 69 | 42.6 / 66.6 | 45.3 / 68.4 | 44.1 / 67.7 | 2.4 / 16.1 | 38.4 / 63.4 | 38.8 / 63.6 | 37.7 / 63.1 |
| ig_en | 33.2 / 55.9 | 25 / 48.5 | 29.7 / 52.8 | 29.2 / 52.7 | 0.5 / 11.9 | 10.8 / 31.5 | 11.3 / 32.3 | 9.6 / 36.1 |
| ilo_en | 41 / 63.6 | 24.2 / 50.6 | 33.3 / 57.3 | 32.5 / 57 | 0.6 / 12.6 | 19.8 / 44.5 | 19.8 / 44 | 20.6 / 47.2 |
| is_en | 34.9 / 59.3 | 36.9 / 61 | 38 / 62.3 | 37.7 / 62 | 7.2 / 28.5 | 31.8 / 57.2 | 33 / 58.7 | 32 / 57.8 |
| it_en | 36.6 / 63.3 | 34.7 / 63 | 35.4 / 63.6 | 35.6 / 63.6 | 4.7 / 24 | 29.9 / 57.9 | 29.4 / 59.5 | 29.8 / 59.7 |
| he_en | 46.5 / 69 | 43.3 / 65.9 | 46.4 / 68.4 | 45.5 / 68.1 | 4.1 / 13.9 | 38.3 / 63 | 39.7 / 64 | 38.9 / 63.9 |
| ja_en | 30.3 / 58.2 | 27.6 / 55.7 | 28.8 / 55.9 | 28.9 / 56.9 | 0.1 / 0.6 | 19.3 / 49.8 | 19.6 / 48.8 | 20.3 / 49.2 |
| jv_en | 43.6 / 65.2 | 25.9 / 51.9 | 29.9 / 55.4 | 29.7 / 54.8 | 1.7 / 14.5 | 27.4 / 52.3 | 28.3 / 52.8 | 29.2 / 53.6 |
| ka_en | 31.4 / 59.5 | 21.5 / 48.9 | 21.3 / 48.8 | 28.7 / 55.6 | 1.9 / 9.9 | 22.4 / 50.9 | 24.4 / 53.6 | 25.2 / 54.7 |
| kab_en | 27.5 / 50.2 | 6.6 / 28.2 | 8.2 / 30.6 | 8.3 / 31.1 | 0.3 / 10.4 | 1.8 / 16.8 | 3.9 / 19.3 | 2.9 / 19.5 |
| kac_en | 18.6 / 43.7 | 6.3 / 30.1 | 9.2 / 32.4 | 10.1 / 34.1 | 0.2 / 8.2 | 1.4 / 13 | 3.2 / 20 | 2.9 / 20.2 |
| kam_en | 13.7 / 35.7 | 6.6 / 26.8 | 7.4 / 27.5 | 8.8 / 29.6 | 0.2 / 8 | 2.3 / 18.1 | 2.9 / 20.5 | 3.4 / 20.5 |
| kbp_en | 13.1 / 35.7 | 7.9 / 30.8 | 9.5 / 32.6 | 9.7 / 32.8 | 0.2 / 8.6 | 1 / 9.9 | 2.3 / 20.6 | 2.3 / 21.3 |
| kea_en | 48.1 / 70.1 | 34.5 / 60.5 | 33.6 / 59.1 | 36 / 61.7 | 0.6 / 10 | 16.8 / 41.2 | 23.3 / 49 | 23.6 / 50.5 |
| kg_en | 21.1 / 44.6 | 6.9 / 30.6 | 8.7 / 32.2 | 9.3 / 33.3 | 0.1 / 5.2 | 2 / 18.1 | 3.4 / 20.1 | 2.8 / 21.5 |
| ki_en | 25.6 / 49.5 | 7.6 / 30.6 | 8.7 / 30.6 | 9.5 / 32.5 | 0.1 / 8 | 2.5 / 18.4 | 3.3 / 19.3 | 2.6 / 20.7 |
| kk_en | 36 / 61.9 | 32.2 / 58.7 | 33.8 / 59.4 | 33.6 / 59.6 | 3.6 / 14.3 | 24.8 / 52.2 | 26.9 / 54.4 | 26.3 / 54.1 |
| km_en | 35.6 / 60.6 | 33.7 / 59.6 | 36.5 / 61.3 | 35.6 / 60.9 | 1.5 / 13.1 | 24.5 / 52.4 | 25.4 / 53.3 | 26.8 / 54.5 |
| kmb_en | 13.7 / 35.7 | 5.7 / 27 | 6.5 / 27.6 | 7.9 / 29.3 | 0.1 / 8.9 | 2.2 / 15.3 | 2.9 / 18.5 | 2 / 18.3 |
| kn_en | 36.9 / 63.1 | 31 / 58.3 | 32.8 / 59 | 32.6 / 59.4 | 0.7 / 6.8 | 17.6 / 45.5 | 20.9 / 50.6 | 22.9 / 52.3 |
| ko_en | 31.2 / 59 | 30.8 / 58.6 | 33.1 / 60.2 | 32.5 / 59.9 | 0.7 / 2.9 | 20.6 / 49 | 23.4 / 52 | 22 / 52.3 |
| kr_Arab_en | 3.2 / 21.2 | 5.4 / 24.1 | 5.2 / 24.1 | 6.9 / 26.2 | 0.1 / 5.5 | 0.6 / 12.1 | 1.2 / 16 | 0.8 / 14.5 |
| kr_en | 16.1 / 38.7 | 13.1 / 36.9 | 12.3 / 36.5 | 14.4 / 39 | 0.2 / 11.6 | 1.6 / 17.4 | 3.5 / 18.9 | 2.9 / 19.5 |
| ks_Deva_en | 27 / 52.7 | 21.5 / 48.2 | 19.8 / 46.5 | 23.3 / 50.4 | 0.3 / 5.3 | 4.2 / 27 | 3.5 / 26.3 | 4.7 / 27.7 |
| ks_en | 36.7 / 62.2 | 26.4 / 54 | 25.8 / 53.4 | 29.8 / 56.6 | 0.4 / 6.6 | 5.4 / 28.9 | 7.3 / 30.4 | 4.2 / 28.9 |
| ku_en | 29.8 / 54.6 | 29.1 / 54.2 | 30.8 / 55.5 | 32.2 / 56.8 | 1.8 / 16 | 16.7 / 40.9 | 21.6 / 47.4 | 21.6 / 49.1 |
| ky_en | 24.1 / 52.5 | 20.2 / 47.4 | 26.6 / 54.4 | 24.8 / 52.4 | 1.7 / 12.8 | 14.2 / 41.2 | 17.6 / 46.1 | 16.6 / 45.4 |
| lb_en | 49 / 71.2 | 19.1 / 47.2 | 21.3 / 49 | 23.1 / 50.8 | 3.5 / 20.6 | 35.3 / 60.6 | 38.2 / 63 | 36.8 / 63.2 |
| lg_en | 25.3 / 48.2 | 13.2 / 34.9 | 16.5 / 38.7 | 15.8 / 38.4 | 0.3 / 10.9 | 5.7 / 23.2 | 5.4 / 25.5 | 6.1 / 27.2 |
| li_en | 44.6 / 67.1 | 35.4 / 60.6 | 29.5 / 55.9 | 35.8 / 61.3 | 4.9 / 27.4 | 25.9 / 51.4 | 27.5 / 52.9 | 28.5 / 53.3 |
| lij_en | 49.2 / 70.8 | 40.6 / 65.5 | 33.6 / 60.2 | 40.3 / 65.3 | 4 / 24.5 | 26 / 53.5 | 26.2 / 54.7 | 24.4 / 55.2 |
| lmo_en | 42.9 / 66.3 | 34.8 / 60.3 | 30.2 / 56.8 | 35.7 / 61.3 | 2.2 / 20.6 | 20.6 / 47.6 | 22.2 / 52 | 23 / 50.9 |
| ln_en | 28.9 / 52.1 | 5.7 / 27.4 | 5.8 / 27.2 | 6.8 / 29.5 | 0.3 / 11.3 | 6.6 / 27.3 | 6.6 / 25.5 | 6.8 / 29.9 |
| lo_en | 39.5 / 63.4 | 35.1 / 60.1 | 37.7 / 61.9 | 38.4 / 63.2 | 1.1 / 10.2 | 26.1 / 52.8 | 28.2 / 55.3 | 28.9 / 56.9 |
| lt_en | 35.5 / 61.1 | 35.1 / 61.4 | 36.8 / 62.2 | 36.2 / 62.1 | 4.5 / 21.9 | 28.9 / 56.4 | 30.1 / 57.3 | 30.1 / 57.5 |
| ltg_en | 40.4 / 65.3 | 35.3 / 62 | 30.3 / 57.8 | 36.5 / 62.9 | 4.4 / 26 | 25.8 / 52.4 | 25.6 / 52.1 | 26.6 / 53.4 |
| lua_en | 19.9 / 43.1 | 6.3 / 27.8 | 6.7 / 28 | 7.9 / 30 | 0.2 / 7.9 | 4.3 / 21.3 | 4.2 / 21.9 | 4.3 / 23.8 |
| luo_en | 22 / 45.4 | 4.1 / 23.6 | 4.4 / 24.3 | 5.4 / 26.4 | 0.2 / 9 | 1.6 / 19.4 | 2.8 / 19.1 | 2.9 / 18.8 |
| lus_en | 18.4 / 43.6 | 13.1 / 37.7 | 18.5 / 43.2 | 18.4 / 43.8 | 1.1 / 15.1 | 9.8 / 32.1 | 9.9 / 31.9 | 10.9 / 34.4 |
| lv_en | 37.1 / 62.9 | 39 / 64.7 | 40.7 / 65.7 | 40.1 / 65.5 | 3.4 / 20 | 32.9 / 60.1 | 33.3 / 60.2 | 33.3 / 60.8 |
| mag_en | 51.6 / 73.7 | 40 / 65.8 | 35.8 / 62.5 | 42.1 / 67.3 | 0.5 / 4.6 | 24.2 / 53.5 | 26.6 / 55.7 | 26.8 / 57.2 |
| mai_en | 46.7 / 70.2 | 35.4 / 62.4 | 34.1 / 60.9 | 38.1 / 64.4 | 1 / 7.4 | 21.5 / 50.4 | 23.3 / 52.4 | 25 / 54.9 |
| mg_en | 35.1 / 59.1 | 30.4 / 55 | 33.8 / 57.7 | 34 / 58 | 1.7 / 18.5 | 15.6 / 39.5 | 19.1 / 44.7 | 20 / 45.9 |
| mi_en | 31 / 54.1 | 20.8 / 45.2 | 18 / 43.1 | 21.6 / 46.8 | 1.9 / 18.7 | 10.1 / 32 | 13.4 / 39.1 | 12.6 / 39.3 |

| | | | | | | | | |
|---|---|---|---|---|---|---|---|---|
| #N/A min_en | 41.7 / 64 | 27.3 / 52.9 | 25 / 51.2 | 30.6 / 55.9 | 0.6 / 12.5 | 19.4 / 43.4 | 23.4 / 47.7 | 21.7 / 48.7 |
| mk_en | 45.4 / 68.7 | 44.7 / 68.4 | 45.9 / 69.1 | 46 / 69.2 | 7.7 / 28 | 37.7 / 62.8 | 39.9 / 64.9 | 39.7 / 64.9 |
| ml_en | 39.1 / 64.9 | 33 / 59.4 | 34.3 / 59.9 | 35 / 61.1 | 0.9 / 4.8 | 19.2 / 47 | 23.8 / 52.6 | 23.9 / 53 |
| mn_en | 28.9 / 56.3 | 28.7 / 55.9 | 32.2 / 58.9 | 31.9 / 58.8 | 0.8 / 8 | 18.5 / 45.5 | 20.5 / 48 | 20.1 / 47.9 |
| mni_en | 27.8 / 53.5 | 23.8 / 51.1 | 22.6 / 49.5 | 24.1 / 51.2 | 0.1 / 2.2 | 1.9 / 20.5 | 3.3 / 22.3 | 2.2 / 21 |
| mos_en | 12.2 / 34.3 | 3.4 / 23 | 3.4 / 22.4 | 3.7 / 24.1 | 0.3 / 9.9 | 1.4 / 15 | 1.4 / 18 | 1.6 / 17.5 |
| mr_en | 40.3 / 65.8 | 35.2 / 61.3 | 36.3 / 61.6 | 36.9 / 62.4 | 1.7 / 7.8 | 24.8 / 53.7 | 27.8 / 56.3 | 28 / 56.4 |
| ms_en | 47.7 / 69.7 | 43.9 / 67.1 | 46 / 68.7 | 45.7 / 68.6 | 2.7 / 18.7 | 38.5 / 63.4 | 39.9 / 64.6 | 40.3 / 64.8 |
| mt_en | 59.2 / 77.6 | 57.6 / 77.1 | 59.2 / 78.1 | 59.4 / 78.4 | 4.6 / 22 | 44.8 / 67.3 | 46.1 / 68.8 | 47 / 70.2 |
| my_en | 31.5 / 58.4 | 28.7 / 55.1 | 30.9 / 56.7 | 31 / 57.2 | 0.2 / 4.2 | 16.1 / 45.6 | 16 / 43.7 | 17.7 / 45.9 |
| ne_en | 44.5 / 68.8 | 39.8 / 65.6 | 41 / 66.2 | 41.8 / 67.2 | 1.1 / 7.5 | 25 / 54.4 | 28.2 / 56.7 | 27.1 / 55.7 |
| nl_en | 34.4 / 61.2 | 34.3 / 61.2 | 34.3 / 61.3 | 34.3 / 61.4 | 6.5 / 28.8 | 27.1 / 55.2 | 28.8 / 58.1 | 29.1 / 57.7 |
| nn_en | 46.4 / 68.8 | 47 / 69.7 | 48.3 / 70.5 | 47.8 / 70.4 | 4.8 / 26.9 | 42 / 65.8 | 42.3 / 66.5 | 43.1 / 66.9 |
| no_en | 45 / 68 | 45.1 / 68.4 | 46.6 / 69.5 | 45.9 / 69 | 5.5 / 26.2 | 40.5 / 65 | 40.7 / 65.2 | 39.9 / 65.7 |
| nso_en | 42.6 / 63 | 26.5 / 48.3 | 31.8 / 53.2 | 31.9 / 53.2 | 0.4 / 11.4 | 9.1 / 26.9 | 11.3 / 33.8 | 12.7 / 36.1 |
| nus_en | 18.6 / 41.2 | 12.9 / 36.1 | 11.9 / 35.2 | 14.8 / 38.3 | 0.1 / 6.9 | 0.5 / 11.9 | 1 / 16.1 | 0.6 / 16.3 |
| ny_en | 29.2 / 52.9 | 24.3 / 47.2 | 27.2 / 50.1 | 27.2 / 50.5 | 0.6 / 12 | 8.4 / 27.7 | 12.2 / 35.4 | 13.9 / 37.2 |
| oc_en | 58.4 / 77.2 | 55 / 75.4 | 58.9 / 77.6 | 57.4 / 76.8 | 7.8 / 27.7 | 42.5 / 66.5 | 43.8 / 67.3 | 42.9 / 68.2 |
| om_en | 27.2 / 52.1 | 11.1 / 35.6 | 17.3 / 42.1 | 17 / 42.6 | 0.4 / 10.5 | 4.1 / 22.4 | 5 / 27.1 | 5.9 / 29 |
| or_en | 41.6 / 66.2 | 34.9 / 61 | 36.1 / 61.5 | 36.1 / 61.7 | 1 / 8.6 | 19.1 / 47.4 | 20.6 / 49.1 | 22.5 / 50.7 |
| pa_en | 44.8 / 67.9 | 39 / 63.7 | 42 / 65.8 | 41.1 / 65.2 | 1.5 / 11.5 | 24.9 / 52.8 | 26.3 / 53.7 | 27.2 / 55.2 |
| pag_en | 27.9 / 52.3 | 22 / 46.4 | 22.9 / 46.2 | 24.4 / 48.5 | 0.5 / 11 | 17.6 / 40.7 | 17.9 / 42.2 | 17.7 / 42.8 |
| pap_en | 53 / 73.4 | 48.2 / 70.8 | 49.7 / 71.7 | 53.4 / 73.9 | 1.4 / 15.8 | 33.1 / 58.1 | 37.4 / 61.9 | 37.6 / 62.8 |
| pl_en | 32.1 / 59.5 | 30.5 / 58.6 | 31.8 / 59.4 | 31.5 / 59.3 | 8.5 / 32.3 | 25.7 / 54.6 | 26.2 / 55.4 | 26 / 55.4 |
| ps_en | 36.6 / 61.8 | 30.1 / 57.1 | 30.6 / 57.2 | 32.6 / 59.1 | 1.2 / 11.8 | 15.7 / 39.4 | 22.9 / 49 | 23.2 / 50.7 |
| pt_en | 51.7 / 72.7 | 50.7 / 72.7 | 52.3 / 73.6 | 51.7 / 73.2 | 5.5 / 24 | 43.4 / 67.8 | 44.2 / 68.4 | 44.3 / 68.3 |
| quy_en | 13.6 / 36.6 | 8.7 / 31.5 | 10.3 / 32.7 | 11.4 / 34.1 | 0.3 / 8.5 | 2.8 / 23.2 | 2.7 / 21.7 | 3.5 / 22.8 |
| rn_en | 27.4 / 51.2 | 19.9 / 43.2 | 23.5 / 46.6 | 23.6 / 46.9 | 0.2 / 7.1 | 10.1 / 30.6 | 12.3 / 35.4 | 11.6 / 35.3 |
| ro_en | 46.9 / 70.4 | 45.1 / 69.6 | 46.2 / 70.3 | 46.1 / 70.2 | 6.3 / 27.3 | 37.6 / 62.9 | 38.2 / 64.9 | 39 / 64.9 |
| ru_en | 38.5 / 63.7 | 37.5 / 63.1 | 38.8 / 64.1 | 38.3 / 63.8 | 3.2 / 15.8 | 29.3 / 55.7 | 31.1 / 58.9 | 30.5 / 59 |
| rw_en | 34.8 / 57.4 | 29.1 / 52.1 | 33.5 / 55.9 | 33.7 / 56.2 | 1.9 / 18.2 | 19.8 / 43.2 | 19.1 / 42.5 | 17.3 / 44.9 |
| sa_en | 26.1 / 52.9 | 20.7 / 47.4 | 21.5 / 48 | 16.7 / 41.8 | 1.3 / 12.3 | 11.8 / 37.9 | 13.6 / 42.3 | 11.4 / 42.5 |
| #N/A sc_en | 47.8 / 69.3 | 37.1 / 62.1 | 30.1 / 56.6 | 36.9 / 62.3 | 3 / 22.8 | 18.4 / 45.6 | 20.6 / 48 | 20.5 / 49.2 |
| scn_en | 41.3 / 64.5 | 33.5 / 59.8 | 27.4 / 54.8 | 33.3 / 59.6 | 3.8 / 24.5 | 24.3 / 50.5 | 26.7 / 53.6 | 25.1 / 53.9 |
| sd_en | 45 / 67.9 | 39 / 62.9 | 42.2 / 65.6 | 41.7 / 65 | 1.1 / 9.8 | 24.2 / 50.5 | 25.3 / 51.8 | 26.4 / 53 |
| sg_en | 14.4 / 36.7 | 8.2 / 31.1 | 9 / 31.7 | 10 / 32.8 | 0.2 / 9.7 | 3.5 / 19.2 | 2.7 / 19.3 | 2.7 / 22.1 |
| shn_en | 26.5 / 51.5 | 18.6 / 45.3 | 17.3 / 43.5 | 20.2 / 47 | 0.1 / 3.3 | 1.9 / 17.3 | 2 / 18.4 | 1.8 / 17.4 |
| si_en | 37.9 / 63.7 | 34.1 / 60.4 | 36 / 61.7 | 35.9 / 62.2 | 0.7 / 6.3 | 17.2 / 45.6 | 21.6 / 50.2 | 21.9 / 50.6 |
| sk_en | 41.1 / 66.1 | 41.4 / 66.9 | 42.2 / 67.5 | 42.3 / 67.7 | 7.3 / 29.5 | 34.3 / 61.4 | 35.7 / 62.5 | 35.5 / 62.4 |
| sl_en | 36.5 / 63 | 37.1 / 63.2 | 37.8 / 63.9 | 37.9 / 63.9 | 4.8 / 25.4 | 31.5 / 58.8 | 32.1 / 59.4 | 31.7 / 59.1 |
| sm_en | 35.8 / 58.3 | 28 / 52.3 | 31.6 / 55.9 | 33.4 / 56.4 | 0.7 / 13.1 | 14.2 / 36.3 | 18.1 / 42 | 15.4 / 41.8 |
| sn_en | 28.9 / 51.9 | 23.7 / 46.6 | 27.2 / 49.9 | 26.9 / 49.9 | 0.6 / 11.4 | 12.4 / 33.2 | 13.1 / 37.5 | 14.7 / 37.7 |
| so_en | 30.2 / 54.4 | 27.8 / 51.6 | 30.8 / 54.6 | 31.2 / 55 | 1 / 11.6 | 17.5 / 40.6 | 21.1 / 45.1 | 20.1 / 45.7 |
| sq_en | 44.9 / 68.4 | 41.6 / 66.9 | 43.7 / 68.2 | 43.5 / 68 | 11.3 / 34.5 | 35.8 / 62 | 36.5 / 62.4 | 37.1 / 63.4 |
| sr_en | 46.3 / 69.4 | 44.1 / 68 | 46.3 / 69.4 | 45.7 / 69.2 | 6.1 / 22.2 | 38.7 / 63.8 | 39.1 / 65.4 | 40.7 / 66 |
| ss_en | 29.2 / 52.3 | 21.8 / 43.8 | 25.3 / 47.5 | 25 / 47.7 | 0.4 / 11 | 6.9 / 25.1 | 8.2 / 29.2 | 8.9 / 32.3 |
| st_en | 41.2 / 62.9 | 27.9 / 51.2 | 34.2 / 57 | 34 / 56.8 | 0.6 / 12 | 10.1 / 30.2 | 15.5 / 38.9 | 13.2 / 40.1 |
| su_en | 40.2 / 63.6 | 34.3 / 59.1 | 32.6 / 58.1 | 35.8 / 60.7 | 1.8 / 16.4 | 26.4 / 51.8 | 26.6 / 52.3 | 29.2 / 54.9 |
| sv_en | 49.4 / 71.2 | 49.8 / 71.6 | 50.4 / 72.1 | 50.4 / 72.2 | 7.6 / 30.5 | 44.7 / 67.8 | 45.3 / 68.1 | 45.5 / 68.4 |
| sw_en | 46.2 / 67.4 | 43.3 / 64.9 | 46.5 / 67.1 | 46.1 / 66.9 | 1.8 / 15.3 | 30.8 / 54.3 | 34.3 / 57.9 | 36 / 59.5 |
| szl_en | 45.8 / 69 | 39.4 / 64.4 | 34 / 60.4 | 40.3 / 65.2 | 6.9 / 29 | 29.6 / 54.9 | 29.2 / 55.6 | 28.8 / 56.2 |
| ta_en | 36.8 / 62.8 | 30.9 / 57.4 | 32.3 / 58 | 33.1 / 59.3 | 1 / 5.2 | 22.9 / 51 | 22.7 / 50.8 | 24.1 / 52.4 |
| taq_Tfng_en | 8.8 / 30.1 | 4.9 / 27.3 | 6.7 / 28.8 | 6.5 / 29.6 | 0.1 / 5.4 | 0.2 / 12.9 | 0.4 / 14.7 | 0.3 / 13.7 |
| taq_en | 13.4 / 35.8 | 10.2 / 32.8 | 9.6 / 32.6 | 11.5 / 34.7 | 0.3 / 9.4 | 1.1 / 18.7 | 2.1 / 18.6 | 3.3 / 20.1 |
| te_en | 43.6 / 67.3 | 37.4 / 62.4 | 38.2 / 62.5 | 39.5 / 63.9 | 1.1 / 3.3 | 23.8 / 51.5 | 27.5 / 55.9 | 27.2 / 55.4 |
| tg_en | 36.5 / 61.7 | 13.7 / 38.2 | 24.8 / 50.5 | 22.5 / 47.3 | 2.3 / 16.1 | 23.4 / 49.9 | 26.5 / 54.7 | 27.3 / 54.3 |
| th_en | 33 / 60 | 31.4 / 57.4 | 33.3 / 59.2 | 33.3 / 59.6 | 2.1 / 6.7 | 24.1 / 53.1 | 24.1 / 54.7 | 24.8 / 54.5 |
| ti_en | 26.4 / 52.4 | 15.2 / 41.5 | 19.8 / 46 | 20.6 / 47.1 | 0.6 / 10.2 | 8 / 29.8 | 8.8 / 34.6 | 9.3 / 35 |
| tk_en | 34.9 / 61.2 | 31.3 / 57.7 | 33 / 58.9 | 33.5 / 58.8 | 2.4 / 18.5 | 19.1 / 45.6 | 20 / 47 | 20.2 / 49.2 |
| tn_en | 29.7 / 53.4 | 18.7 / 42.6 | 21.9 / 46.7 | 23.4 / 46.8 | 0.3 / 10.8 | 7.4 / 25.9 | 9.9 / 31.8 | 7.4 / 31.1 |
| tpi_en | 30.4 / 54.6 | 10.7 / 37.8 | 10.5 / 37.3 | 14.2 / 42.2 | 0.9 / 15.8 | 7.3 / 31.1 | 8.3 / 28.7 | 8.9 / 29.3 |
| tr_en | 41.5 / 66.1 | 39.1 / 63.8 | 41.4 / 65.5 | 40.9 / 65.3 | 3.4 / 20.4 | 25 / 51.8 | 28.1 / 55.5 | 29.1 / 56.7 |
| ts_en | 32.1 / 54.9 | 19.6 / 42.3 | 24.3 / 47.3 | 25.4 / 47.7 | 0.4 / 10.3 | 7.4 / 25.9 | 8.8 / 28.1 | 7.8 / 30.7 |
| tt_en | 31.8 / 58.3 | 10.4 / 30.4 | 10.3 / 29.4 | 17.5 / 38.5 | 1.2 / 9.9 | 16.5 / 42.1 | 21.3 / 48.5 | 21.3 / 48.5 |
| tum_en | 20.5 / 44.9 | 12.2 / 37 | 15.2 / 40 | 15.9 / 40.5 | 0.2 / 6.9 | 7.2 / 27.2 | 7 / 27.3 | 7.2 / 28.7 |
| ug_en | 27 / 54.6 | 8.6 / 29 | 14.1 / 36 | 10.9 / 30.6 | 1.6 / 14.4 | 13.4 / 39.4 | 17.3 / 44.9 | 17.3 / 44.9 |
| uk_en | 42.4 / 66.7 | 42.1 / 66.4 | 43 / 66.8 | 42.5 / 66.7 | 6.6 / 24.8 | 34.6 / 60.5 | 35.9 / 61.7 | 34.5 / 62 |
| umb_en | 11.4 / 33.3 | 4.4 / 24.2 | 4.7 / 24.2 | 5.5 / 25.9 | 0.2 / 9.6 | 1.1 / 17.2 | 2.3 / 18.3 | 1.6 / 17.1 |
| ur_en | 39.6 / 64.8 | 33.3 / 59.6 | 36.3 / 62.1 | 35.9 / 61.7 | 1.3 / 17.4 | 24.8 / 52.8 | 26.7 / 54.1 | 28.7 / 56.2 |
| uz_en | 35.2 / 61.6 | 33.9 / 59 | 37.2 / 62.1 | 37.8 / 63.2 | 1.8 / 15.6 | 23.5 / 51.2 | 24.6 / 52.3 | 24.9 / 52.9 |
| vec_en | 45.4 / 68 | 39.4 / 64.6 | 33.4 / 59.9 | 39.4 / 64.7 | 4 / 23.6 | 31 / 56.5 | 34.1 / 60 | 35.1 / 60.2 |
| vi_en | 40 / 64.2 | 37.4 / 62.5 | 39.2 / 63.9 | 39.3 / 63.8 | 4.1 / 19.7 | 28 / 53.5 | 31.7 / 57.3 | 32.1 / 57.9 |
| war_en | 51 / 71.1 | 32.1 / 58.6 | 32.5 / 58.8 | 35.9 / 60.9 | 1.1 / 15.5 | 25 / 50 | 27.4 / 55.3 | 31.8 / 56.4 |
| wo_en | 19.8 / 42.9 | 6.6 / 28.4 | 6.7 / 29.3 | 7.1 / 29.8 | 0.3 / 9 | 1.8 / 16.9 | 3 / 18.2 | 3.2 / 19.7 |
| xh_en | 39.7 / 61.1 | 32.1 / 54.3 | 36.5 / 58.2 | 35.5 / 57.7 | 1 / 15.2 | 17.2 / 39.3 | 21 / 43.9 | 21.1 / 44 |
| yi_en | 53.6 / 72.4 | 38.8 / 62.6 | 48.8 / 69 | 45.5 / 67.4 | 1.2 / 8.5 | 34.2 / 56 | 37.6 / 59.6 | 39.1 / 62.2 |
| yo_en | 23.9 / 47.9 | 11.9 / 34.9 | 17.4 / 40.8 | 17.8 / 40.7 | 0.4 / 12.5 | 5.1 / 23.5 | 4 / 24.1 | 6.3 / 26 |
| yue_en | 30.8 / 58.5 | 30.4 / 57.9 | 31.7 / 58.7 | 32 / 59.3 | 0.3 / 1.3 | 21.5 / 49.9 | 22.5 / 51.3 | 23.3 / 52.5 |
| zh_Hant_en | 28.8 / 56.2 | 28.1 / 55.7 | 29.6 / 56.8 | 30 / 57.5 | 0.6 / 2.7 | 19.6 / 47.3 | 21.2 / 49.6 | 21.6 / 50.5 |
| zh_en | 31.9 / 59.5 | 29 / 57.9 | 30.4 / 59 | 30.5 / 59.2 | 0.3 / 1.9 | 21.8 / 50.2 | 22.5 / 51.5 | 23.4 / 52.1 |

| | | | | | | | | |
|---|---|---|---|---|---|---|---|---|
| zu_en | 41.3 / 62.7 | 33 / 55.1 | 37.7 / 59.1 | 37.1 / 58.9 | 0.8 / 13.4 | 17.5 / 40 | 20.3 / 44 | 20.2 / 45.5 |
| **xx2en** | **35.5 / 59.6** | 29.7 / 54.4 | 30.9 / 55.4 | 31.9 / 56.4 | 2.0 / 13.3 | 20.5 / 44.1 | 22.3 / 46.9 | 22.4 / 47.6 |
| **en2xx** | **20.7 / 50.**1 | 17.3 / 44.1 | 17.8 / 44.7 | 18.6 / 45.7 | 0.4 / 5.7 | 8.1 / 26.7 | 8.7 / 29.0 | 8.7 / 28.8 |
| **Mean** | **28.2 / 54.9** | 23.5 / 49.2 | 24.4 / 50.0 | 25.3 / 51.1 | 1.2 / 9.6 | 14.3 / 35.5 | 15.6 / 38.0 | 15.6 / 38.2 |
| **xx2yy** | **13.7 / 40.5** | 8.8 / 31.2 | 8.4 / 30.9 | 10.1 / 34.0 | 0.3 / 4.1 | 4.0 / 16.1 | 4.4 / 17.3 | 4.2 / 17.1 |

Table 17: Evaluation scores on Flores-200 direct pairs (depicted as `<bleu>` / `<chrf>`) for the MT models and language models described in Section 4.1 and Section 4.2 compared against NLLB-54B. All metrics are computed with the sacrebleu reference implementation.

| | NLLB | MT-3B | MT-7.2B | MT-10.7B | LM-8B | | | |
|---|---|---|---|---|---|---|---|---|
| | | | | | 0-shot | 1-shot | 5-shot | 10-shot |
| ckb_cs | 20.7 / 47.3 | 15.4 / 39.4 | 18.5 / 44.9 | 19.1 / 45.1 | 0 / 0.8 | 8.1 / 30.2 | 10.4 / 35.5 | 9.7 / 36.6 |
| ckb_cy | 26.9 / 53.6 | 19.3 / 41.4 | 23.6 / 47 | 23.9 / 47.8 | 0 / 0.7 | 0.3 / 8.8 | 0.7 / 15.1 | 0.3 / 12.4 |
| ckb_en | 37.5 / 61.6 | 28.3 / 53.1 | 34.4 / 58.9 | 34.2 / 59 | 1 / 9.7 | 17.9 / 42.5 | 22.4 / 48.4 | 22.7 / 49.4 |
| ckb_et | 16.4 / 48.6 | 13 / 40.8 | 15.3 / 45.6 | 16.3 / 47 | 0 / 1 | 1.9 / 12.9 | 6.7 / 36.6 | 7.3 / 38.2 |
| ckb_eu | 11.3 / 47.4 | 7.2 / 38.6 | 10 / 44.2 | 11.5 / 46 | 0 / 0.5 | 1.3 / 18.4 | 1.9 / 30.8 | 2.1 / 33.5 |
| ckb_fon | 2.1 / 19.9 | 1 / 14.7 | 0.4 / 10.8 | 1 / 14.4 | 0 / 0.2 | 0.3 / 8.5 | 0 / 1.8 | 0 / 3.2 |
| ckb_fr | 31.2 / 56.9 | 23.8 / 45.4 | 30.5 / 53.5 | 28.7 / 51.9 | 0.1 / 2.5 | 13.5 / 38.4 | 18.7 / 45.2 | 18.8 / 45.6 |
| ckb_ln | 15.5 / 47.5 | 1.3 / 16 | 0.5 / 14.4 | 1.2 / 17.4 | 0 / 0.3 | 0.1 / 6.2 | 0.1 / 10.3 | 0.2 / 11.4 |
| ckb_mr | 11 / 43.3 | 3.3 / 22.5 | 1.4 / 22.4 | 4.7 / 27.9 | 0 / 0.4 | 0.6 / 4.7 | 0.6 / 21.5 | 0.8 / 24.8 |
| ckb_ny | 10.4 / 44.7 | 3.5 / 29.7 | 2.7 / 31 | 6.8 / 35.9 | 0.1 / 0.4 | 0.3 / 13.1 | 0.1 / 9.9 | 0.1 / 8.5 |
| ckb_or | 8.6 / 41.8 | 7 / 36 | 7.3 / 38.5 | 8.5 / 40.1 | 0 / 0.1 | 0 / 0.2 | 0 / 0.3 | 0 / 0.3 |
| ckb_so | 8.8 / 42.2 | 5 / 32.4 | 4.1 / 34.4 | 6.9 / 37.2 | 0 / 0.3 | 0.3 / 6.6 | 0.1 / 5.8 | 0.1 / 7.4 |
| ckb_ss | 6.5 / 44.3 | 2 / 27.4 | 0.9 / 23.5 | 3.1 / 30.9 | 0 / 0.2 | 0.1 / 7.7 | 0.1 / 9.8 | 0 / 9.3 |
| ckb_ti | 4 / 24.2 | 1.1 / 12.8 | 0.3 / 10.5 | 1.7 / 16 | 0 / 0.1 | 0 / 0.4 | 0 / 0.1 | 0 / 0.1 |
| ckb_yo | 5.5 / 27 | 1.2 / 15.2 | 1.2 / 16.6 | 1.8 / 18.2 | 0 / 0.3 | 0 / 4.1 | 0 / 3.3 | 0 / 4 |
| ckb_zh | 27.6 / 25.2 | 3 / 18.9 | 2.2 / 22.1 | 4.1 / 22.1 | 0.4 / 0.9 | 1.5 / 3.4 | 15.7 / 15.7 | 14.6 / 16 |
| cs_ckb | 8.8 / 45.4 | 5.8 / 37.6 | 1.3 / 30.8 | 6 / 37.8 | 0 / 0.2 | 0.1 / 8 | 0.1 / 9.9 | 0.1 / 8.8 |
| cs_cy | 29.7 / 57.6 | 24.3 / 48 | 28.4 / 52.4 | 27.4 / 51.9 | 0.3 / 7.4 | 2.5 / 19.3 | 3.3 / 29.4 | 2.2 / 28.5 |
| cs_en | 41.2 / 66.4 | 41.2 / 66.6 | 42.3 / 67.4 | 42.3 / 67.3 | 7.9 / 28.9 | 34.7 / 61.4 | 35.9 / 62.6 | 36.2 / 62.9 |
| cs_et | 20.4 / 53.9 | 22.8 / 54.9 | 22.3 / 54.5 | 23.5 / 56 | 1.8 / 14.1 | 15.2 / 45.8 | 17.9 / 51.1 | 17.6 / 51.3 |
| cs_eu | 11.8 / 47.8 | 11.9 / 45.3 | 8.9 / 44.7 | 12.6 / 47.5 | 0.2 / 7.6 | 7.3 / 40.6 | 6.3 / 44.6 | 7 / 46 |
| cs_fon | 2.5 / 21.6 | 1.2 / 14.3 | 0.7 / 11.9 | 1 / 14 | 0 / 2.3 | 0.1 / 4.3 | 0 / 5 | 0 / 1.7 |
| cs_fr | 35.3 / 60.9 | 38.7 / 62.3 | 39.5 / 62.6 | 40.1 / 63.2 | 1.2 / 10.2 | 24.5 / 51.6 | 24.2 / 54 | 25.6 / 54.6 |
| cs_ln | 15.6 / 47.8 | 1.6 / 17.6 | 1.1 / 17.1 | 2.3 / 20.9 | 0.2 / 6.2 | 0.3 / 10.6 | 0.1 / 8.9 | 0.1 / 7.2 |
| cs_mr | 10.8 / 43.1 | 3.6 / 24.9 | 1.6 / 22.9 | 4.5 / 26.8 | 0.2 / 2.7 | 3.7 / 28.8 | 3.5 / 33.8 | 2.9 / 35.2 |
| cs_ny | 10.1 / 44.7 | 7.3 / 38.3 | 3.9 / 36.2 | 7.9 / 39.3 | 0.2 / 5.3 | 0.2 / 16 | 0.3 / 13.6 | 0.1 / 12.1 |
| cs_or | 9.1 / 42.1 | 8.7 / 40.2 | 9.3 / 41.3 | 9.9 / 42.5 | 0.1 / 0.2 | 0 / 4.9 | 0.2 / 0.5 | 0.1 / 0.5 |
| cs_so | 9.1 / 42.7 | 7.7 / 39 | 7.3 / 39.3 | 7.7 / 39.6 | 0.2 / 5 | 0.4 / 14.3 | 0.2 / 13.5 | 0.1 / 10.9 |
| cs_ss | 7.3 / 45.1 | 2.4 / 26.2 | 1.2 / 25.4 | 3.2 / 30.4 | 0.2 / 6.3 | 0.2 / 20.7 | 0 / 6.3 | 0 / 7.4 |
| cs_ti | 4.3 / 24.9 | 1.4 / 12.7 | 0.3 / 9.8 | 1.6 / 16.1 | 0 / 0.3 | 0 / 0.7 | 0 / 0.7 | 0 / 0.4 |
| cs_yo | 4.5 / 25.1 | 2 / 18.4 | 1.4 / 16.3 | 2 / 17.7 | 0.1 / 4.8 | 0.1 / 11.7 | 0 / 2.1 | 0 / 7.1 |
| cs_zh | 23.6 / 23.3 | 6.4 / 31.1 | 7.3 / 31.8 | 6.8 / 32.7 | 2 / 2.3 | 20.2 / 19.9 | 19.7 / 20 | 21.6 / 20.5 |
| cy_ckb | 10.5 / 48.3 | 8 / 43.1 | 5.2 / 41.6 | 8.4 / 42.7 | 0 / 0.2 | 0 / 5.6 | 0 / 5.2 | 0 / 2.1 |
| cy_cs | 27.2 / 54.1 | 28.9 / 55.6 | 29.8 / 56.6 | 29.7 / 56.4 | 0.3 / 4.7 | 16.1 / 41.8 | 16 / 43.4 | 12.8 / 43.4 |
| cy_en | 60 / 77.1 | 57.6 / 75.6 | 60.7 / 77.7 | 60.8 / 77.7 | 5.4 / 24.3 | 43.9 / 65.6 | 48.5 / 69.3 | 47.7 / 68.7 |
| cy_et | 22.3 / 55 | 24.2 / 56.4 | 25 / 57.5 | 25.8 / 57.9 | 0.1 / 4.8 | 5.6 / 31.3 | 7.6 / 40.8 | 7.5 / 42.2 |
| cy_eu | 14.4 / 51.6 | 17.1 / 54.9 | 19 / 56.7 | 18.5 / 56.3 | 0 / 4.8 | 0.6 / 16.8 | 2.8 / 36.2 | 3.3 / 37 |
| cy_fon | 2.2 / 18.6 | 1.3 / 14.9 | 0.6 / 11.7 | 1.3 / 14.9 | 0 / 0.9 | 0.5 / 7.1 | 0 / 1.7 | 0 / 5.5 |
| cy_fr | 40.9 / 64.2 | 44 / 65.6 | 45.8 / 66.7 | 45.3 / 66.7 | 0.7 / 9.3 | 27.7 / 53.8 | 28.7 / 54.9 | 27.4 / 55.7 |
| cy_ln | 17.4 / 49.1 | 2 / 18 | 1.1 / 17.5 | 2 / 20.1 | 0.1 / 3.8 | 0.3 / 11.4 | 0.1 / 8.1 | 0.1 / 8.8 |
| cy_mr | 13.9 / 47.5 | 10.2 / 41.3 | 6.4 / 39.1 | 11.1 / 42.4 | 0.1 / 1.5 | 2.8 / 26 | 2 / 29.7 | 1 / 26.5 |
| cy_ny | 12.4 / 46.5 | 6.1 / 36.2 | 5.6 / 40.4 | 9.3 / 41.9 | 0.1 / 3 | 0.8 / 13.6 | 0.2 / 11 | 0.1 / 9.1 |
| cy_or | 11.5 / 45.1 | 11.2 / 44.4 | 12.1 / 46 | 12.3 / 46.2 | 0 / 0.3 | 0 / 0.4 | 0.1 / 0.4 | 0.1 / 0.4 |
| cy_so | 10.8 / 44.8 | 8.6 / 41.3 | 8.2 / 42.4 | 9.6 / 43 | 0.1 / 3.4 | 1 / 13.3 | 0.4 / 14.5 | 0.2 / 12.9 |
| cy_ss | 9 / 46.7 | 3.2 / 31.3 | 1.4 / 28.3 | 4.1 / 34 | 0.1 / 3 | 0.2 / 12.3 | 0.4 / 11.3 | 0.2 / 11.4 |
| cy_ti | 4.8 / 25.5 | 1.5 / 14.2 | 0.4 / 12.3 | 2.1 / 17.9 | 0 / 0.3 | 0 / 0.8 | 0 / 0.5 | 0 / 0.2 |
| cy_yo | 5.4 / 26.3 | 2.1 / 17.7 | 1.7 / 18.3 | 2.2 / 19.4 | 0.1 / 2.5 | 0 / 2.9 | 0 / 4.1 | 0 / 6.4 |
| cy_zh | 30.4 / 28.1 | 5.4 / 31.5 | 5.7 / 33.1 | 5.3 / 32.9 | 0.8 / 1.4 | 8.4 / 9 | 19.5 / 20.1 | 15.3 / 19.3 |
| en_ckb | 13.2 / 52.5 | 9.1 / 44.9 | 4.5 / 42.8 | 9.4 / 44.4 | 0.1 / 0.3 | 0.3 / 10.2 | 0.1 / 13.8 | 0.1 / 13.4 |
| en_cs | 33.6 / 59.8 | 34.8 / 61.2 | 34.8 / 61.1 | 35.6 / 61.5 | 1.1 / 11.1 | 25.3 / 53.1 | 26.2 / 54.5 | 25.6 / 54.4 |
| en_cy | 51.6 / 72.5 | 41.6 / 62.4 | 45.6 / 66.3 | 44.7 / 65.6 | 0.5 / 9 | 6.4 / 29.4 | 8.2 / 38 | 4.8 / 35.4 |
| en_et | 27.1 / 59.9 | 28.9 / 61.3 | 28.1 / 61.2 | 29.8 / 61.8 | 0.9 / 11.9 | 22.2 / 54.7 | 22.8 / 55.6 | 22.2 / 55.3 |
| en_eu | 17.2 / 54.9 | 18.3 / 55.5 | 16.8 / 55.4 | 19.3 / 57.2 | 0.2 / 11.1 | 12.3 / 49.3 | 12.9 / 51 | 12.4 / 51.3 |
| en_fon | 3.1 / 23.5 | 1.6 / 15.6 | 1.3 / 14.6 | 1.6 / 15.9 | 0 / 0.3 | 0.1 / 4.2 | 0 / 1.9 | 0 / 1.6 |
| en_fr | 51.4 / 71.4 | 52.4 / 72 | 53.3 / 72.4 | 53.2 / 72.5 | 1.7 / 15.3 | 37.1 / 61.9 | 38.9 / 63.5 | 38.8 / 63.1 |
| en_ln | 19.1 / 50.8 | 2.2 / 18.3 | 1.5 / 18.2 | 2.7 / 20.8 | 0.4 / 9.3 | 0.7 / 11.3 | 0.1 / 9.2 | 0.2 / 10.7 |
| en_mr | 17.6 / 52.1 | 10.1 / 39 | 5.8 / 36.4 | 11.8 / 42 | 0.8 / 12 | 10.5 / 42.8 | 9 / 43.2 | 8.1 / 43.7 |
| en_ny | 13.6 / 48.7 | 8.2 / 38.7 | 6.6 / 43.3 | 10.6 / 43.5 | 0.4 / 8 | 0.7 / 16.4 | 0.4 / 16 | 0.4 / 17.6 |
| en_or | 15.1 / 50.4 | 12.4 / 46.8 | 13.2 / 47.4 | 13.1 / 47.8 | 0.1 / 0.2 | 0.4 / 0.9 | 0.3 / 0.8 | 0.3 / 0.8 |
| en_so | 12.6 / 47.3 | 9.4 / 41.8 | 9.9 / 43.2 | 10.2 / 43 | 0.2 / 6.9 | 0.6 / 15.2 | 0.2 / 13 | 0.2 / 13.3 |
| en_ss | 10.7 / 49.2 | 4.1 / 33.5 | 1.9 / 31.9 | 5.3 / 36 | 0.5 / 7.2 | 0.6 / 15 | 0.1 / 11.6 | 0.1 / 9.5 |
| en_ti | 6 / 27.9 | 1.9 / 15.7 | 0.6 / 14.6 | 2.5 / 18.9 | 0.1 / 0.3 | 0 / 0.9 | 0 / 0.8 | 0 / 0.6 |
| en_yo | 6 / 27.4 | 2.6 / 19.7 | 1.9 / 19.6 | 2.9 / 20.7 | 0.3 / 6.3 | 0.1 / 10.6 | 0.1 / 5.3 | 0 / 3.3 |

| | | | | | | | | |
|---|---|---|---|---|---|---|---|---|
| en_zh | 31.2 / 29.3 | 5.9 / 35.7 | 6 / 36.5 | 5.8 / 36.5 | 2.9 / 3.3 | 29.1 / 26.1 | 29.9 / 26.6 | 29.9 / 26.3 |
| et_ckb | 8.7 / 44.9 | 4.9 / 34.9 | 2.8 / 34.8 | 5 / 34.7 | 0 / 0.4 | 0.1 / 10.4 | 0 / 8.7 | 0 / 5.9 |
| et_cs | 24.1 / 51.6 | 25.3 / 52.3 | 24.9 / 51.8 | 25.3 / 52.6 | 1.8 / 14.1 | 15.8 / 42 | 18.2 / 46 | 17.4 / 46 |
| et_cy | 27.9 / 56.1 | 23.2 / 47.2 | 25.4 / 49.3 | 25.9 / 49.9 | 0.4 / 10.2 | 2.5 / 22.3 | 1.5 / 25 | 1.2 / 23.3 |
| et_en | 38.8 / 64.6 | 39.4 / 64.8 | 40.1 / 65.5 | 39.6 / 65.5 | 7.7 / 27.9 | 31.8 / 59.1 | 32.5 / 59.8 | 32.6 / 59.7 |
| et_eu | 11.4 / 47 | 13.4 / 48.5 | 13.5 / 49.1 | 14 / 50 | 0.3 / 12.2 | 5.9 / 37.8 | 5.7 / 41.2 | 3.9 / 41.1 |
| et_fon | 2.5 / 22.4 | 1 / 13.2 | 0.5 / 11.3 | 0.9 / 12.7 | 0 / 2.6 | 0.1 / 4.8 | 0 / 3.5 | 0 / 3.5 |
| et_fr | 33.4 / 59.2 | 35.1 / 59.3 | 35.7 / 59.3 | 36 / 60.1 | 1.4 / 14.2 | 23 / 50.2 | 24.8 / 52.1 | 25.3 / 52.6 |
| et_ln | 14.8 / 47.6 | 1.7 / 17.3 | 0.8 / 16.1 | 2 / 20.8 | 0.3 / 10.7 | 0.2 / 9.5 | 0.1 / 9.3 | 0.1 / 10.5 |
| et_mr | 10.8 / 42.8 | 5.4 / 29.3 | 2.4 / 25.8 | 5 / 28.5 | 0.3 / 1.6 | 0.7 / 2.7 | 3.9 / 33.5 | 2.8 / 35.3 |
| et_ny | 9.7 / 43.8 | 6.1 / 34.2 | 3 / 33.2 | 7.8 / 37.4 | 0.4 / 9.9 | 0.1 / 7 | 0.1 / 11.4 | 0.1 / 7.9 |
| et_or | 9 / 41.8 | 8.2 / 39.5 | 7.7 / 39.9 | 9 / 41 | 0 / 0.3 | 0.1 / 1.8 | 0.3 / 0.5 | 0.3 / 0.5 |
| et_so | 8.6 / 42.4 | 7.6 / 37.8 | 6.6 / 38.4 | 8.1 / 39.5 | 0.3 / 9 | 0.5 / 15 | 0.4 / 14.5 | 0.1 / 12.2 |
| et_ss | 6.6 / 44.8 | 2.5 / 26.3 | 0.9 / 23.6 | 3.1 / 29.3 | 0.1 / 10.7 | 0.1 / 21.8 | 0 / 7.4 | 0 / 7.8 |
| et_ti | 4 / 24.1 | 1.2 / 11.6 | 0.3 / 8.9 | 1.3 / 13.8 | 0 / 0.3 | 0 / 0.8 | 0 / 0.7 | 0 / 1.2 |
| et_yo | 4.1 / 24.6 | 2 / 17.4 | 1 / 15 | 1.6 / 16.2 | 0.2 / 7.3 | 0 / 4.1 | 0 / 5.3 | 0 / 3.2 |
| et_zh | 22.9 / 22.6 | 6.3 / 30.9 | 6.1 / 31.7 | 5.7 / 31.6 | 2.1 / 2.8 | 21.2 / 20.1 | 18.9 / 21.3 | 22.7 / 22.8 |
| eu_ckb | 7.3 / 43.7 | 4.6 / 34.2 | 1 / 27.8 | 4.3 / 32.4 | 0 / 0.2 | 0.1 / 8.3 | 0 / 8.6 | 0 / 5.3 |
| eu_cs | 19.2 / 46.9 | 16.7 / 42.2 | 14.4 / 41.9 | 18.2 / 44.4 | 0.9 / 12.2 | 13 / 40.6 | 13.8 / 41.6 | 13.2 / 41.2 |
| eu_cy | 24 / 53.1 | 22 / 46.1 | 25.6 / 50.5 | 24.5 / 49.7 | 0.2 / 10.6 | 1.2 / 18.7 | 1.5 / 22.7 | 0.9 / 21.6 |
| eu_en | 34.4 / 61.2 | 32.6 / 58.6 | 34.1 / 59.8 | 34.4 / 60.4 | 4.6 / 23.3 | 25.5 / 53.8 | 24.7 / 52.9 | 26.2 / 54.4 |
| eu_et | 16.8 / 49.6 | 16.4 / 47.7 | 17.9 / 49.4 | 18.7 / 50.2 | 0.1 / 12.2 | 7.1 / 35.2 | 9.9 / 43.9 | 9.6 / 43.6 |
| eu_fon | 2.5 / 20.7 | 1 / 13.4 | 0.4 / 10.1 | 1 / 13 | 0.3 / 9.1 | 0.1 / 9.1 | 0 / 2.8 | 0 / 1.6 |
| eu_fr | 29.4 / 56.2 | 28.8 / 53 | 30.3 / 54 | 30.3 / 54.4 | 1.2 / 14 | 20.3 / 48.3 | 21.3 / 49.4 | 21.1 / 49.5 |
| eu_ln | 14.5 / 47 | 1.6 / 17 | 0.6 / 15.8 | 1.9 / 19 | 0.1 / 12.5 | 0.4 / 16.5 | 0.1 / 8.9 | 0.1 / 8.5 |
| eu_mr | 9.6 / 42.6 | 2.8 / 22.6 | 1.4 / 21 | 3.7 / 24.6 | 0.1 / 0.6 | 3.3 / 31 | 2.7 / 31.1 | 2.4 / 32.1 |
| eu_ny | 9.3 / 44.2 | 4.3 / 30.7 | 2.8 / 31.9 | 6.4 / 35.1 | 0.4 / 11.2 | 0.2 / 12.8 | 0.1 / 11.5 | 0.1 / 11.3 |
| eu_or | 7.9 / 40.3 | 8.1 / 39.6 | 7.4 / 40 | 8.6 / 41 | 0 / 0.3 | 0 / 3.3 | 0.2 / 0.5 | 0.1 / 0.5 |
| eu_so | 8.3 / 41.7 | 6.4 / 35.4 | 4.8 / 35.8 | 7.1 / 37.8 | 0.2 / 11.4 | 0.4 / 12.8 | 0.1 / 12 | 0.1 / 10.9 |
| eu_ss | 5.9 / 43.7 | 1.9 / 25 | 0.9 / 22.6 | 2.8 / 29.4 | 0.1 / 11.3 | 0.1 / 8.1 | 0 / 6 | 0 / 8.7 |
| eu_ti | 3.8 / 24.1 | 0.8 / 11 | 0.3 / 9.3 | 1.5 / 15 | 0 / 0.3 | 0 / 1.9 | 0 / 0.6 | 0 / 0.7 |
| eu_yo | 5 / 26.3 | 1.7 / 16.8 | 1 / 15.8 | 1.7 / 16.8 | 0.1 / 7.6 | 0.1 / 7.5 | 0 / 2.8 | 0 / 4.2 |
| eu_zh | 23.2 / 22.4 | 5.6 / 22.6 | 2.3 / 20.3 | 5.9 / 24.9 | 1.8 / 2.6 | 17.5 / 17 | 20.2 / 21 | 14.3 / 19.4 |
| fon_ckb | 2.8 / 28.6 | 0.9 / 20.6 | 0.3 / 16.8 | 1.6 / 24 | 0 / 0.2 | 0 / 0.2 | 0 / 2 | 0 / 1.5 |
| fon_cs | 7.6 / 27.8 | 2.8 / 20.2 | 2.1 / 19.3 | 4.3 / 22.3 | 0 / 1.3 | 0.5 / 13 | 0.8 / 13.9 | 0.3 / 13.5 |
| fon_cy | 8.8 / 31.5 | 4 / 21.8 | 2.9 / 21.2 | 6 / 25.6 | 0 / 2.2 | 0 / 7.3 | 0.1 / 10.4 | 0.1 / 7.1 |
| fon_en | 12.9 / 35.4 | 3.9 / 25 | 2.2 / 23.4 | 6.4 / 27.8 | 0.1 / 7.8 | 0.8 / 12.5 | 1.4 / 17.7 | 1.9 / 18.3 |
| fon_et | 6.7 / 30.6 | 2.2 / 22.3 | 1.2 / 20.4 | 3.4 / 24.9 | 0 / 1.3 | 0.7 / 15.5 | 0.4 / 15.8 | 0.6 / 16.1 |
| fon_eu | 4.6 / 31.3 | 1.4 / 22.5 | 0.9 / 19.8 | 2.9 / 27.2 | 0 / 2 | 0.5 / 17.6 | 0.1 / 12.4 | 0.2 / 14.3 |
| fon_fr | 10.6 / 33.2 | 5.3 / 24.4 | 4.4 / 24.3 | 7.6 / 26.8 | 0.1 / 3.7 | 0.9 / 11.2 | 1.3 / 17.1 | 1.3 / 16.4 |
| fon_ln | 8.2 / 36.8 | 0.6 / 11 | 0.2 / 7.6 | 0.5 / 11.8 | 0.1 / 3 | 0.1 / 6.6 | 0 / 5 | 0 / 1.5 |
| fon_mr | 3.8 / 27 | 0.8 / 16.5 | 0.2 / 13.4 | 1.5 / 18.7 | 0 / 0.4 | 0.3 / 8.3 | 0.1 / 8.7 | 0 / 7 |
| fon_ny | 5.6 / 33.3 | 1.3 / 20.6 | 0.7 / 18.7 | 3 / 26.5 | 0 / 1.9 | 0.1 / 12.8 | 0 / 4.4 | 0 / 2.7 |
| fon_or | 2.7 / 25.1 | 1.1 / 18.5 | 0.7 / 17 | 2 / 22 | 0 / 0.2 | 0 / 0.2 | 0 / 0.4 | 0 / 0.3 |
| fon_so | 4.3 / 31.2 | 1.6 / 23 | 0.8 / 19.6 | 2.4 / 26.3 | 0 / 2 | 0.1 / 5.6 | 0 / 5.5 | 0 / 4.1 |
| fon_ss | 3 / 32.5 | 0.8 / 16.7 | 0.3 / 14 | 1.8 / 23.2 | 0 / 2.4 | 0 / 4.5 | 0 / 3.9 | 0 / 1.8 |
| fon_ti | 1.7 / 15.5 | 0.3 / 6.5 | 0.1 / 5.5 | 0.7 / 10.2 | 0 / 0.2 | 0 / 0.2 | 0 / 0.1 | 0 / 0.1 |
| fon_yo | 3.4 / 21.3 | 0.7 / 12.3 | 0.4 / 10.8 | 1.1 / 14.9 | 0 / 2 | 0 / 3.2 | 0 / 3.7 | 0 / 1 |
| fon_zh | 11.8 / 12.9 | 1.9 / 8.7 | 0.5 / 6.2 | 2.2 / 10.2 | 0.1 / 0.6 | 1.6 / 4.3 | 1.8 / 5.1 | 0.7 / 3.5 |
| fr_ckb | 9.2 / 46.9 | 6 / 39.7 | 3.3 / 38.4 | 6.1 / 39.1 | 0 / 0.3 | 0.1 / 7.8 | 0.1 / 13.8 | 0.1 / 13.9 |
| fr_cs | 26 / 53.6 | 27.3 / 54.9 | 27.8 / 55.3 | 27.8 / 55.2 | 1.8 / 14.2 | 19.3 / 46.8 | 20.9 / 49.1 | 20.4 / 49.4 |
| fr_cy | 33.7 / 60.1 | 28 / 51.6 | 31.7 / 56 | 30.7 / 55 | 0.3 / 9.5 | 4.1 / 27.8 | 4.2 / 32.8 | 3.7 / 34.1 |
| fr_en | 47.9 / 70.1 | 46.1 / 69.6 | 47.3 / 70.3 | 47.1 / 70.1 | 2.7 / 21.4 | 39.2 / 64.5 | 39.9 / 64.9 | 40.1 / 65.1 |
| fr_et | 20.9 / 54.3 | 22.2 / 55.2 | 18.5 / 54.5 | 23.6 / 56.3 | 1.7 / 15.8 | 11.1 / 44.6 | 16.9 / 50.6 | 15.8 / 51 |
| fr_eu | 14 / 51.2 | 13.1 / 50.8 | 11.6 / 51.3 | 16.1 / 53.5 | 0.3 / 9.9 | 10.2 / 47.7 | 9.5 / 48.8 | 9.6 / 49.4 |
| fr_fon | 2.5 / 22.6 | 1.1 / 14.7 | 0.7 / 12.3 | 1.2 / 14.5 | 0.2 / 2.4 | 0 / 1.7 | 0 / 3.8 | 0 / 1.8 |
| fr_ln | 16.8 / 49.3 | 1.7 / 17.9 | 1.2 / 17.3 | 2.1 / 20.2 | 0.2 / 7.3 | 0.1 / 8.7 | 0 / 6.3 | 0.2 / 10.6 |
| fr_mr | 12 / 45.5 | 7.1 / 34.8 | 2.2 / 27 | 6.7 / 33.2 | 0.5 / 8.8 | 5.5 / 35.9 | 5.3 / 38.3 | 4.5 / 38.4 |
| fr_ny | 10.8 / 45.5 | 5.4 / 34.9 | 4.1 / 38.3 | 7.8 / 40 | 0.2 / 6.6 | 0.5 / 16.1 | 0.2 / 13.8 | 0.2 / 13.8 |
| fr_or | 10.2 / 43.8 | 9.3 / 41.7 | 10.1 / 43.5 | 10.1 / 43.6 | 0 / 0.2 | 0.1 / 1.1 | 0.1 / 0.6 | 0.2 / 0.6 |
| fr_so | 9.7 / 43.8 | 7.5 / 39.5 | 6 / 39.9 | 7.7 / 40.1 | 0.1 / 5 | 0.5 / 15.9 | 0.3 / 15 | 0.1 / 11.1 |
| fr_ss | 7.9 / 45.5 | 2.4 / 28.5 | 1 / 26.5 | 3.4 / 31.9 | 0.1 / 6.9 | 0.1 / 10.3 | 0.1 / 10.2 | 0.1 / 8.9 |
| fr_ti | 4.6 / 25.3 | 1 / 12.9 | 0.5 / 11.7 | 1.8 / 16.9 | 0 / 0.3 | 0 / 1.1 | 0 / 0.8 | 0 / 0.5 |
| fr_yo | 4.9 / 26.2 | 2.1 / 18.7 | 1.3 / 17.6 | 1.9 / 18.8 | 0.1 / 4.6 | 0.1 / 6.6 | 0 / 3.8 | 0 / 10.2 |
| fr_zh | 25.5 / 24.4 | 7.4 / 33 | 7.2 / 34.1 | 6.8 / 33.6 | 4 / 5.5 | 16.2 / 18.1 | 22.8 / 23.6 | 19.3 / 21.8 |
| ln_ckb | 6.6 / 40 | 1.1 / 22.1 | 0.3 / 18 | 1.4 / 22.8 | 0 / 0.2 | 0 / 3 | 0 / 3.6 | 0 / 3.8 |
| ln_cs | 15.6 / 40.7 | 3.6 / 22.1 | 1.4 / 18.1 | 4.7 / 23.6 | 0.2 / 4.7 | 2.6 / 19.2 | 2.5 / 21.2 | 1.9 / 20.4 |
| ln_cy | 20.2 / 45.8 | 4.9 / 23.2 | 2.6 / 21 | 6.4 / 26.6 | 0.1 / 5.2 | 0.1 / 5.8 | 0.1 / 7.8 | 0.1 / 6.7 |
| ln_en | 28.9 / 52.1 | 5.5 / 27.2 | 1.7 / 23.4 | 6.8 / 29.5 | 0.3 / 11.1 | 6.5 / 27.2 | 6.6 / 25.5 | 6.7 / 29.9 |
| ln_et | 12.8 / 42.6 | 3.2 / 24.8 | 1.3 / 20 | 3.7 / 26.3 | 0.2 / 5.2 | 0.3 / 12.9 | 2.3 / 23.7 | 1.9 / 22.9 |
| ln_eu | 9.3 / 42.1 | 2.1 / 25.3 | 1.3 / 22 | 3.2 / 28.3 | 0.1 / 5.9 | 0.7 / 17.7 | 1.4 / 25.3 | 0.6 / 22.3 |
| ln_fon | 2.6 / 22.6 | 1.1 / 13.1 | 0.3 / 8.4 | 0.8 / 13.3 | 0 / 0.5 | 0.1 / 5.4 | 0 / 3 | 0 / 1.8 |
| ln_fr | 23.8 / 48.9 | 6.5 / 25.5 | 2.8 / 24 | 7.6 / 28 | 0.4 / 9.9 | 4 / 23.1 | 6.5 / 30.9 | 4.1 / 28 |
| ln_mr | 8.1 / 37.1 | 1.2 / 16.8 | 0.3 / 12.8 | 1.5 / 18.7 | 0.1 / 0.7 | 0.6 / 16.4 | 0.1 / 10.1 | 0.2 / 13.6 |
| ln_ny | 9.4 / 42.1 | 1.6 / 20.5 | 0.6 / 17.3 | 2.6 / 25.9 | 0.2 / 6.4 | 0.1 / 8.2 | 0.1 / 9.6 | 0.1 / 8.9 |
| ln_or | 6.8 / 36.6 | 1.4 / 18.9 | 0.5 / 15.9 | 1.8 / 21.7 | 0 / 0.3 | 0 / 0.3 | 0 / 0.4 | 0 / 0.4 |
| ln_so | 8.6 / 40 | 2.1 / 24 | 0.7 / 19.4 | 2.6 / 26.6 | 0.1 / 5.9 | 0.4 / 12.9 | 0.1 / 9.8 | 0.1 / 8.9 |
| ln_ss | 7.1 / 43 | 1.2 / 18.5 | 0.4 / 13.6 | 2 / 22.5 | 0.1 / 5.2 | 0.1 / 9.5 | 0.1 / 8.3 | 0 / 7.2 |
| ln_ti | 3.4 / 20.6 | 0.3 / 4.5 | 0.1 / 5.7 | 0.6 / 10 | 0 / 0.3 | 0 / 0.2 | 0 / 0.2 | 0 / 0.2 |
| ln_yo | 4.3 / 24.5 | 0.8 / 12.9 | 0.4 / 11 | 1.1 / 14.9 | 0.1 / 4 | 0 / 1.4 | 0.1 / 7 | 0.1 / 7.4 |
| ln_zh | 20 / 19.9 | 1.7 / 8.9 | 0.3 / 4.3 | 2.2 / 10.7 | 0.3 / 1.2 | 1.5 / 5.1 | 4.2 / 8.6 | 2.9 / 7.9 |

| | | | | | | | | |
|---|---|---|---|---|---|---|---|---|
| mr_ckb | 8.1 / 45.2 | 4.4 / 35.5 | 1.1 / 28.5 | 4.9 / 34.8 | 0.1 / 0.3 | 0 / 8.8 | 0 / 8.7 | 0 / 8 |
| mr_cs | 21.8 / 49.3 | 9.5 / 31.4 | 3.4 / 29.2 | 14.4 / 37.7 | 0.2 / 0.8 | 11.7 / 36.7 | 12.5 / 38.7 | 10.8 / 38.9 |
| mr_cy | 28.1 / 56.1 | 22.3 / 47 | 25.6 / 51.2 | 26 / 51.1 | 0.1 / 1 | 2.1 / 21.1 | 1.3 / 23.6 | 0.9 / 21.2 |
| mr_en | 40.3 / 65.8 | 35 / 61.2 | 35.4 / 61.3 | 36.9 / 62.4 | 1.5 / 7.6 | 24.8 / 53.7 | 27.8 / 56.4 | 27.9 / 56.3 |
| mr_et | 18 / 51.3 | 7.5 / 34 | 7.6 / 41 | 15.6 / 45.7 | 0.2 / 1.7 | 10 / 41.6 | 8.7 / 43 | 9.7 / 44.2 |
| mr_eu | 11.3 / 48.3 | 6.3 / 35.8 | 3.7 / 35.6 | 9.6 / 42 | 0 / 0.8 | 4.4 / 36.5 | 3.4 / 39 | 2.3 / 37.6 |
| mr_fon | 2.3 / 22.2 | 0.9 / 13.2 | 0.3 / 8.9 | 0.8 / 12.6 | 0 / 0.2 | 0 / 3.6 | 0 / 5.1 | 0 / 1.2 |
| mr_fr | 32.2 / 58.7 | 18.5 / 39.8 | 10.7 / 39.2 | 24.7 / 47.7 | 0.5 / 1.4 | 17.2 / 44.8 | 14.3 / 45 | 18.2 / 47.3 |
| mr_ln | 15.3 / 47.8 | 1 / 14 | 0.3 / 13.5 | 1.7 / 19 | 0 / 0.3 | 0 / 6.5 | 0.1 / 8.2 | 0.1 / 6.9 |
| mr_ny | 9.9 / 44.7 | 2.9 / 28.3 | 2.9 / 34.6 | 7.1 / 38 | 0.1 / 1.3 | 0.2 / 11.6 | 0.1 / 10.1 | 0 / 7.8 |
| mr_or | 11 / 45.6 | 9.2 / 42.3 | 9.7 / 43.2 | 9.5 / 43.4 | 0.1 / 0.3 | 0.2 / 0.7 | 0.2 / 1.2 | 0 / 1 |
| mr_so | 8.8 / 42.8 | 6.2 / 36.7 | 3.5 / 36.3 | 7 / 39.1 | 0.1 / 1 | 0.2 / 10 | 0.1 / 7.9 | 0 / 7.3 |
| mr_ss | 7.1 / 44.9 | 1.8 / 24.7 | 0.6 / 20.8 | 3.2 / 30.8 | 0.1 / 0.6 | 0.1 / 11.7 | 0 / 6.6 | 0 / 5.8 |
| mr_ti | 4.5 / 25.2 | 0.7 / 10.6 | 0.4 / 11.1 | 1.7 / 16.2 | 0 / 0.2 | 0 / 1.4 | 0 / 0.8 | 0 / 0.6 |
| mr_yo | 5.1 / 26.5 | 1.8 / 17.5 | 0.9 / 16.4 | 1.7 / 18.1 | 0.1 / 0.4 | 0.1 / 7.1 | 0 / 3.1 | 0 / 3.9 |
| mr_zh | 25.4 / 24 | 4.8 / 19.3 | 1 / 13.6 | 5.6 / 23.3 | 0.1 / 0.4 | 17.7 / 17 | 19.9 / 19.4 | 14.8 / 18.9 |
| ny_ckb | 6.6 / 39 | 4.5 / 33.4 | 1.4 / 29.5 | 5.2 / 35.5 | 0 / 0.2 | 0 / 0.5 | 0 / 1.6 | 0 / 2.9 |
| ny_cs | 15.2 / 40.3 | 14.2 / 37.4 | 15.2 / 39.3 | 15.3 / 39.2 | 0.2 / 4.2 | 0.6 / 15.9 | 3.5 / 23.3 | 3.1 / 23.2 |
| ny_cy | 20.7 / 46 | 17.4 / 39.1 | 20.3 / 42.2 | 20 / 42.7 | 0.2 / 8.6 | 0.1 / 7.8 | 0.1 / 6.3 |
| ny_en | 29.2 / 52.9 | 24.2 / 47.2 | 24 / 49.7 | 27.2 / 50.5 | 0.6 / 12.2 | 8.4 / 27.7 | 12.4 / 35.5 | 13.9 / 37.1 |
| ny_et | 12.7 / 42.6 | 11.8 / 38.6 | 10.9 / 40.2 | 13.5 / 41.9 | 0.1 / 4.7 | 1.5 / 20.7 | 1.6 / 24.5 | 2 / 26.7 |
| ny_eu | 8.9 / 41.1 | 8.6 / 39 | 5.9 / 36.9 | 9.8 / 42.7 | 0 / 4.5 | 0.6 / 20.4 | 0.8 / 25.4 | 0.6 / 23 |
| ny_fon | 2.5 / 21.9 | 1.3 / 14.6 | 0.6 / 12.1 | 1.4 / 15.6 | 0 / 0.3 | 0 / 4.4 | 0 / 1.4 | 0 / 3.7 |
| ny_fr | 22.5 / 47.4 | 19.8 / 42.1 | 22.4 / 45.2 | 23.8 / 46.3 | 0.2 / 5.5 | 8.6 / 31.6 | 5.6 / 32.2 | 4.5 / 32.2 |
| ny_ln | 13.6 / 44.7 | 1 / 13.7 | 0.5 / 13 | 1.8 / 19.2 | 0.1 / 5.2 | 0.5 / 13.4 | 0.1 / 10.8 | 0.1 / 9.4 |
| ny_mr | 7.4 / 36.1 | 4.3 / 27.3 | 1.4 / 21.6 | 5.3 / 29.9 | 0 / 0.5 | 0.3 / 13.6 | 0.5 / 19.6 | 0.3 / 15.8 |
| ny_or | 6.2 / 35.9 | 5.3 / 33.9 | 6.6 / 35.4 | 6.7 / 36.4 | 0 / 0.3 | 0.1 / 0.2 | 0 / 0.3 | 0.1 / 0.5 |
| ny_so | 8.2 / 39.6 | 6.2 / 35.3 | 4.5 / 35 | 6.9 / 37.1 | 0.1 / 5.5 | 0.4 / 12.8 | 0.2 / 12.3 | 0.1 / 9.5 |
| ny_ss | 7.3 / 42.5 | 2.5 / 27.2 | 0.9 / 22.1 | 4.2 / 32.2 | 0.1 / 4 | 0.1 / 10.5 | 0.2 / 13.5 | 0.1 / 9.5 |
| ny_ti | 3.3 / 19.4 | 0.8 / 10.8 | 0.3 / 9.3 | 1.7 / 15.4 | 0 / 0.3 | 0 / 0.3 | 0 / 0.2 | 0 / 0.1 |
| ny_yo | 4 / 23.3 | 2 / 17.7 | 1.4 / 16.6 | 2.3 / 19 | 0.1 / 3.5 | 0.1 / 4.1 | 0 / 3.8 | 0 / 4.4 |
| ny_zh | 18.8 / 18.9 | 5 / 20.7 | 1.3 / 14.9 | 4.7 / 21.9 | 0.3 / 1.1 | 3.9 / 6.7 | 6.2 / 11.3 | 6 / 11.5 |
| or_ckb | 8.1 / 45.1 | 6 / 38.9 | 5.7 / 40.4 | 7.5 / 42 | 0 / 0.2 | 0 / 0.2 | 0 / 0.3 | 0 / 0.3 |
| or_cs | 21.3 / 49.1 | 20.3 / 47.5 | 21.6 / 47.9 | 21.7 / 48.9 | 0 / 0.7 | 8.1 / 32.7 | 9.5 / 36.9 | 10.6 / 36.7 |
| or_cy | 28.6 / 56.4 | 22.4 / 48 | 27.4 / 52.1 | 26.3 / 51.6 | 0.1 / 0.9 | 0.5 / 8.6 | 0.4 / 5.6 | 0.8 / 10.5 |
| or_en | 41.6 / 66.2 | 35 / 61 | 36.3 / 61.7 | 36.1 / 61.7 | 1 / 8.6 | 18.9 / 47.3 | 20.6 / 49.1 | 22.5 / 50.7 |
| or_et | 17.1 / 50.7 | 17.6 / 49.6 | 18.2 / 50.4 | 18.7 / 51.1 | 0.1 / 0.9 | 6.8 / 35 | 9.1 / 41.1 | 7.9 / 38.3 |
| or_eu | 11.9 / 48.8 | 13.3 / 49.9 | 14.1 / 51.1 | 13.9 / 51.4 | 0 / 0.5 | 2.1 / 24.3 | 1.9 / 28.8 | 2.2 / 36.6 |
| or_fon | 2.3 / 21.4 | 1 / 13.5 | 0.5 / 10.7 | 1 / 13.9 | 0 / 0.2 | 0 / 0.5 | 0 / 2.3 | 0 / 1.5 |
| or_fr | 32.3 / 58.7 | 29.1 / 54.1 | 31.3 / 55.7 | 32.6 / 57 | 0.1 / 1 | 12.8 / 39.4 | 13.8 / 42.6 | 16.5 / 44.6 |
| or_ln | 15.9 / 47.8 | 1.4 / 14.8 | 0.7 / 14.7 | 1.6 / 16.3 | 0 / 0.4 | 0.3 / 5.1 | 0.1 / 6.4 | 0.1 / 7.3 |
| or_mr | 12.8 / 47 | 9.1 / 42.1 | 9.9 / 42.8 | 10.3 / 43.2 | 0.2 / 4.2 | 4.5 / 27.6 | 2.4 / 34.6 | 6.4 / 40.1 |
| or_ny | 10.8 / 45.7 | 5.3 / 34.9 | 5.4 / 38.7 | 8 / 39.6 | 0.1 / 0.7 | 0.1 / 10 | 0.1 / 5.8 | 0.1 / 7.7 |
| or_so | 8.8 / 42.5 | 6.3 / 36.8 | 7.2 / 38.5 | 7.5 / 40 | 0.1 / 0.5 | 0.3 / 2.5 | 0 / 6.7 | 0.1 / 6.9 |
| or_ss | 7.1 / 44.6 | 2.3 / 26.6 | 1.5 / 26.3 | 3.4 / 32.2 | 0 / 0.4 | 0 / 3.7 | 0 / 4.7 | 0.1 / 7.6 |
| or_ti | 4.4 / 25.3 | 0.8 / 10.5 | 0.5 / 12.6 | 1.8 / 16.2 | 0 / 0.2 | 0 / 0.3 | 0 / 0.3 | 0 / 0.2 |
| or_yo | 5.6 / 27.3 | 1.6 / 17.2 | 1.4 / 17.4 | 1.9 / 18.7 | 0 / 0.4 | 0 / 3.3 | 0 / 2.9 | 0 / 2.8 |
| or_zh | 26 / 24.3 | 5.2 / 26.7 | 4.2 / 26.8 | 6.6 / 29.5 | 0.4 / 0.6 | 8.6 / 8.4 | 18.2 / 16.7 | 18 / 17.9 |
| so_ckb | 6.3 / 40.7 | 3.5 / 30 | 2 / 31.4 | 4.5 / 33.7 | 0 / 0.5 | 0.1 / 5.9 | 0 / 1.2 | 0 / 2.6 |
| so_cs | 15.6 / 41 | 14.6 / 40.1 | 16.4 / 41.9 | 16.4 / 42.2 | 0.1 / 3.4 | 5.6 / 24.5 | 6.5 / 29.2 | 5.8 / 31.1 |
| so_cy | 21.2 / 47 | 20.1 / 43 | 23.1 / 46 | 22.6 / 46.1 | 0.2 / 4.4 | 0.7 / 11.5 | 0.2 / 10.6 | 0.2 / 9.6 |
| so_en | 30.2 / 54.4 | 27.8 / 51.7 | 30.7 / 54.5 | 31.2 / 55 | 1 / 11.4 | 17.5 / 40.5 | 21.1 / 45.2 | 19.5 / 45.7 |
| so_et | 13.1 / 43.3 | 14 / 43 | 13.9 / 43.4 | 15 / 44.5 | 0.3 / 7.2 | 2.5 / 22 | 3.7 / 30.8 | 2.9 / 30.4 |
| so_eu | 9.1 / 42.4 | 9.8 / 42.2 | 9.4 / 42.1 | 11.4 / 45.4 | 0.1 / 4.8 | 0.7 / 18.3 | 0.5 / 21.5 | 0.1 / 13.3 |
| so_fon | 2.1 / 20 | 1 / 13.6 | 0.5 / 10.6 | 0.9 / 13.8 | 0 / 0.6 | 0.1 / 7.4 | 0 / 5.7 | 0 / 1.8 |
| so_fr | 24.5 / 49.8 | 24 / 47.8 | 26.4 / 49.7 | 26.1 / 49.7 | 0.4 / 7.1 | 13.8 / 38.9 | 11.3 / 39.6 | 11.3 / 40.2 |
| so_ln | 15.3 / 45.9 | 1.4 / 15.7 | 0.6 / 13.5 | 1.8 / 18.9 | 0.1 / 4.8 | 0.2 / 11.8 | 0.1 / 9.4 | 0.1 / 8.7 |
| so_mr | 8.7 / 38.8 | 4.8 / 28.9 | 1.3 / 21.5 | 5.1 / 28.5 | 0.2 / 6.1 | 1.8 / 23.7 | 1.1 / 24 | 0.7 / 22.6 |
| so_ny | 10.5 / 43.4 | 4.7 / 31.9 | 2.5 / 31.9 | 7 / 37.2 | 0.2 / 5 | 0.5 / 17.3 | 0.2 / 12.8 | 0.1 / 10.6 |
| so_or | 7 / 37.3 | 6.2 / 35 | 7.1 / 36.9 | 7.5 / 37.8 | 0.1 / 0.3 | 0.1 / 0.2 | 0.1 / 0.4 | 0.1 / 0.4 |
| so_ss | 6.6 / 42.4 | 2 / 25.5 | 0.8 / 21.4 | 3.5 / 31.2 | 0 / 3.6 | 0 / 12.9 | 0.2 / 12.3 | 0 / 6.5 |
| so_ti | 3.6 / 22.5 | 1 / 11.3 | 0.3 / 9.6 | 1.4 / 15.2 | 0 / 0.3 | 0 / 0.4 | 0 / 0.3 | 0 / 0.2 |
| so_yo | 6.1 / 27 | 1.5 / 16.2 | 1 / 15.7 | 1.8 / 17.9 | 0.1 / 3.2 | 0.1 / 11.9 | 0.1 / 9.2 | 0 / 6 |
| so_zh | 20.3 / 20.5 | 3.6 / 21.4 | 4.3 / 22.9 | 3.5 / 23 | 0.6 / 1.5 | 8.5 / 12.4 | 10.1 / 14.6 | 10.2 / 14.4 |
| ss_ckb | 6.2 / 39.2 | 4 / 32.4 | 1.8 / 30.2 | 4.5 / 33.3 | 0 / 0.2 | 0 / 3 | 0.1 / 1.7 | 0 / 1.9 |
| ss_cs | 14.9 / 39.1 | 11.1 / 33.3 | 13 / 36 | 12.8 / 36.2 | 0.1 / 3.6 | 0.7 / 13.5 | 1.7 / 20.4 | 1.3 / 20 |
| ss_cy | 20.1 / 45.4 | 15.8 / 36.8 | 16.7 / 39.1 | 18.8 / 40.7 | 0.1 / 4.5 | 0.4 / 11 | 0.1 / 8.6 | 0.1 / 6.9 |
| ss_en | 29.2 / 52.3 | 21.6 / 43.8 | 22.9 / 47 | 25 / 47.7 | 0.4 / 11 | 7 / 25.1 | 8.4 / 29.3 | 9.2 / 32.5 |
| ss_et | 12.1 / 41.6 | 9.6 / 35.5 | 8.8 / 37.2 | 11.3 / 39.1 | 0.1 / 4 | 0.2 / 14.9 | 1.2 / 20.7 | 1 / 20.3 |
| ss_eu | 8.2 / 39.7 | 7.7 / 36.8 | 7.1 / 37.4 | 8.9 / 40.5 | 0 / 4.9 | 0.1 / 10.7 | 0.4 / 20.9 | 0.3 / 18.7 |
| ss_fon | 2.7 / 22 | 1.4 / 14.8 | 0.6 / 11.5 | 1.4 / 14.9 | 0 / 0.3 | 0 / 3.4 | 0 / 3.9 | 0 / 1.5 |
| ss_fr | 20.7 / 45.4 | 18.3 / 40.1 | 21.3 / 43.1 | 20.9 / 43.3 | 0.2 / 5.9 | 4.4 / 25.5 | 4.1 / 27.3 | 2.5 / 26.8 |
| ss_ln | 12.1 / 42.8 | 1 / 14.4 | 0.5 / 11.6 | 1.5 / 19 | 0.1 / 6.4 | 0.1 / 10.3 | 0.1 / 9.9 | 0.1 / 8.7 |
| ss_mr | 6.7 / 34.7 | 3.4 / 24.7 | 1.3 / 20.4 | 4.4 / 27.4 | 0.1 / 0.9 | 0.4 / 13.5 | 0.3 / 16.6 | 0.2 / 13.3 |
| ss_ny | 7.6 / 38.4 | 5.1 / 31.9 | 3.2 / 31.8 | 7.7 / 36.9 | 0.2 / 6.8 | 0.1 / 9.2 | 0.1 / 9.6 | 0.1 / 9.6 |
| ss_or | 6.1 / 34.7 | 4.9 / 31.1 | 4.9 / 32 | 6.2 / 34 | 0.1 / 0.3 | 0 / 0.2 | 0 / 0.4 | 0 / 0.3 |
| ss_so | 7.7 / 38.2 | 6 / 33.6 | 4.2 / 33 | 6.9 / 36.2 | 0.1 / 6 | 0.3 / 11.1 | 0.1 / 9.9 | 0.1 / 8.8 |
| ss_ti | 2.6 / 18.3 | 0.8 / 11 | 0.4 / 10 | 1.5 / 15 | 0 / 0.3 | 0 / 0.3 | 0 / 0.2 | 0 / 0.1 |
| ss_yo | 3.6 / 21.7 | 1.8 / 16.7 | 1.2 / 15.9 | 2.2 / 18.3 | 0.1 / 4.2 | 0 / 3.1 | 0.1 / 7.6 | 0 / 3.4 |
| ss_zh | 19.9 / 20 | 4.1 / 17.8 | 4 / 19.3 | 4.2 / 19.5 | 0.1 / 0.7 | 2.5 / 5.3 | 3.4 / 8.3 | 3 / 7.9 |
| ti_ckb | 5.7 / 39.5 | 3.4 / 33 | 1.3 / 26.9 | 4.4 / 36.3 | 0 / 0.2 | 0 / 1.9 | 0 / 0.3 | 0 / 0.1 |

| | | | | | | | | |
|---|---|---|---|---|---|---|---|---|
| ti_cs | 14.5 / 40.9 | 9.1 / 32.8 | 11.1 / 34.6 | 11.8 / 37.2 | 0.1 / 1.7 | 0.8 / 11.5 | 2.4 / 23.2 | 2.9 / 25.6 |
| ti_cy | 18.5 / 45.8 | 12.5 / 34.9 | 15.5 / 37.2 | 16.1 / 40.1 | 0 / 1.8 | 0.3 / 14.4 | 0.1 / 5.6 | 0.1 / 7.8 |
| ti_en | 26.4 / 52.4 | 15.2 / 41.5 | 17.2 / 45 | 20.6 / 47.1 | 0.6 / 10.2 | 8 / 29.9 | 9.4 / 34.6 | 9.3 / 35 |
| ti_et | 12 / 42.8 | 7.5 / 34.9 | 7.4 / 34.9 | 9.3 / 37.6 | 0.1 / 3.5 | 0.3 / 10.5 | 1.6 / 25.8 | 2.2 / 26.7 |
| ti_eu | 8.1 / 41.7 | 5.8 / 36.1 | 4.5 / 34.4 | 7.6 / 40.4 | 0 / 2.3 | 0.8 / 24.9 | 1.3 / 25.2 | 0.5 / 22.9 |
| ti_fon | 1.9 / 20.8 | 0.8 / 14.2 | 0.3 / 8.8 | 0.9 / 14.6 | 0 / 0.2 | 0 / 0.3 | 0 / 1.4 | 0 / 3 |
| ti_fr | 22.1 / 49 | 15.6 / 39.6 | 19.5 / 42.6 | 20.2 / 44.4 | 0.2 / 3.6 | 4.7 / 26.2 | 6.1 / 31.5 | 3.9 / 31 |
| ti_ln | 13 / 44.3 | 0.5 / 13.5 | 0.2 / 8.8 | 0.2 / 0.9 | 0 / 0.4 | 0.3 / 10.1 | 0 / 3.9 | 0.1 / 6.5 |
| ti_mr | 8.2 / 38.6 | 3.4 / 26.2 | 1.9 / 23.8 | 4.6 / 30.3 | 0 / 1.2 | 0.7 / 18.8 | 0.6 / 21.1 | 0.3 / 18.4 |
| ti_ny | 8.4 / 41.2 | 3.4 / 30.7 | 1 / 22.7 | 5.3 / 34.7 | 0 / 0.4 | 0 / 0.6 | 0 / 2.5 | 0 / 1.1 |
| ti_or | 6.7 / 38 | 4.5 / 31.2 | 5.1 / 32 | 5.5 / 34.7 | 0 / 0.2 | 0 / 0.2 | 0 / 0.7 | 0 / 0.3 |
| ti_so | 7.2 / 39.3 | 4.1 / 32.6 | 2.3 / 28 | 5.1 / 35.6 | 0.1 / 0.4 | 0.1 / 2.5 | 0 / 3.5 | 0 / 3.8 |
| ti_ss | 4.8 / 40.5 | 1.8 / 25.8 | 0.5 / 17.1 | 2.5 / 29.7 | 0.1 / 0.4 | 0 / 0.9 | 0 / 0.8 | 0 / 1.7 |
| ti_yo | 4.3 / 24.4 | 1.4 / 16.7 | 0.6 / 12.9 | 1.6 / 17.7 | 0 / 0.6 | 0.2 / 7.3 | 0 / 1.1 | 0 / 4.9 |
| ti_zh | 22.2 / 20.9 | 2.6 / 16.9 | 3 / 18.5 | 4.5 / 21.4 | 0.3 / 0.7 | 1.5 / 3.5 | 3.9 / 5.7 | 5.2 / 9.4 |
| yo_ckb | 5.5 / 36.6 | 2.4 / 27.5 | 1.4 / 27.3 | 3.5 / 31.1 | 0 / 0.2 | 0 / 2 | 0 / 3.2 | 0 / 2.8 |
| yo_cs | 12.7 / 36.6 | 7.3 / 27.1 | 9.9 / 31.4 | 9.7 / 31.7 | 0.1 / 3.3 | 0.5 / 18.1 | 1.6 / 15.1 | 0.6 / 16.7 |
| yo_cy | 16.9 / 42.4 | 10.3 / 30.1 | 12.7 / 34.7 | 13.9 / 35.5 | 0.1 / 3.7 | 0.4 / 13.5 | 0.1 / 9.8 | 0.1 / 6.9 |
| yo_en | 23.9 / 47.9 | 11.8 / 34.9 | 12.7 / 40.3 | 17.8 / 40.7 | 0.4 / 12.5 | 5.2 / 23.5 | 4.2 / 24.3 | 6.5 / 26.1 |
| yo_et | 10.8 / 38.8 | 5.3 / 29 | 7.4 / 33.7 | 8.5 / 34.3 | 0.1 / 4.5 | 1.7 / 18.9 | 1.7 / 20.1 | 1.3 / 20.8 |
| yo_eu | 7.2 / 38.8 | 3.2 / 29.6 | 4.9 / 33.9 | 6.7 / 37 | 0 / 3.9 | 0.3 / 19.4 | 0.4 / 18.2 | 0.3 / 17.2 |
| yo_fon | 1.8 / 19.7 | 1 / 13.7 | 0.5 / 11.2 | 1.1 / 14.9 | 0 / 0.4 | 0.1 / 5.2 | 0 / 4.1 | 0 / 3.1 |
| yo_fr | 19.1 / 44.1 | 11.5 / 31.5 | 16.2 / 38 | 15.9 / 38.4 | 0.3 / 7 | 3.2 / 21.8 | 1.5 / 20.2 | 1.9 / 22.6 |
| yo_ln | 12.4 / 42.8 | 0.5 / 11.4 | 0.4 / 11.1 | 1 / 16.2 | 0.1 / 4.3 | 0.3 / 9.1 | 0.1 / 8 | 0.1 / 7.7 |
| yo_mr | 6.9 / 34.9 | 1.7 / 20.9 | 1 / 21.1 | 3.8 / 27.2 | 0.1 / 2.5 | 0.3 / 13.7 | 0.2 / 12.9 | 0.1 / 10.3 |
| yo_ny | 8.9 / 40.2 | 2.4 / 25.5 | 2 / 28.4 | 6 / 33.9 | 0.1 / 3.9 | 0.5 / 9.4 | 0.1 / 6.2 | 0 / 5.3 |
| yo_or | 5.2 / 33 | 2.9 / 26.2 | 2.9 / 28.2 | 4 / 30.5 | 0 / 0.3 | 0 / 0.3 | 0 / 0.2 | 0 / 0.2 |
| **xx2en** | **35.5 / 59.6** | 29.7 / 54.4 | 30.9 / 55.4 | 31.9 / 56.4 | 2.0 / 13.3 | 20.5 / 44.1 | 22.3 / 46.9 | 22.4 / 47.6 |
| **en2xx** | **20.7 / 50.1** | 17.3 / 44.1 | 17.8 / 44.7 | 18.6 / 45.7 | 0.4 / 5.7 | 8.1 / 26.7 | 8.7 / 29.0 | 8.7 / 28.8 |
| **Mean** | **28.2 / 54.9** | 23.5 / 49.2 | 24.4 / 50.0 | 25.3 / 51.1 | 1.2 / 9.6 | 14.3 / 35.5 | 15.6 / 38.0 | 15.6 / 38.2 |
| **xx2yy** | **13.7 / 40.5** | 8.8 / 31.2 | 8.4 / 30.9 | 10.1 / 34.0 | 0.3 / 4.1 | 4.0 / 16.1 | 4.4 / 17.3 | 4.2 / 17.1 |

Table 18: Evaluation scores on the recently introduced NTREX test set (depicted as `<bleu>` / `<chrf>`) for the MT models and language models described in Section 4.1 and Section 4.2 compared against unsupervised baselines [9]. Note that LM-8B is evaluated on a 50% split of the NTREX data and is not comparable to the MT-model evaluations.

| | Baziotis et al. [9] | MT-3B | MT-7.2B | MT-10.7B | LM-8B | | | |
|---|---|---|---|---|---|---|---|---|
| | | | | | 0-shot | 1-shot | 5-shot | 10-shot |
| af_en | - / - | 57.1 / 75.4 | 58.5 / 76.5 | 58.4 / 76.4 | 7.6 / 30.6 | 49 / 69.9 | 49.2 / 69.6 | 51.8 / 71.9 |
| am_en | - / - | 24.3 / 50.3 | 27.2 / 52.5 | 27.2 / 52.7 | 1 / 10.9 | 13.4 / 37.9 | 14.6 / 40.4 | 13.8 / 40.3 |
| ar_en | - / - | 36 / 61.9 | 37.9 / 63.1 | 37.8 / 63.1 | 1.1 / 10 | 23.7 / 51.6 | 28.6 / 53.9 | 28.8 / 54.1 |
| az_en | - / - | 26.3 / 52.7 | 29.1 / 54.9 | 30.1 / 55.8 | 4.7 / 26.3 | 15.5 / 40.8 | 20.6 / 47 | 21.8 / 48.6 |
| ba_en | - / - | 8 / 28.2 | 9 / 28.4 | 9.7 / 28.8 | 0.5 / 7.4 | 11.1 / 37 | 9.5 / 34.4 | 13.2 / 38.7 |
| be_en | - / - | 31.5 / 57.6 | 32.8 / 58.5 | 35.3 / 59.7 | 4 / 19.4 | 18.4 / 40.7 | 29.1 / 56 | 30.7 / 56.3 |
| bem_en | - / - | 10 / 32.6 | 12.1 / 33.9 | 13.3 / 34.7 | 0.1 / 5.3 | 4.8 / 20.7 | 2.7 / 21.7 | 2.6 / 21.5 |
| bg_en | - / - | 38.2 / 63.3 | 39.3 / 64 | 39.4 / 64.1 | 4.3 / 25.4 | 31.6 / 58.3 | 31.6 / 58.7 | 32.6 / 58.3 |
| bn_en | - / - | 35.4 / 60.7 | 37.5 / 61.9 | 38 / 62.3 | 0.8 / 6 | 23.5 / 51.4 | 22.9 / 50.1 | 24.4 / 51.9 |
| bo_en | - / - | 12.1 / 37.8 | 13.3 / 38.8 | 14.5 / 40 | 0.1 / 2.7 | 5.9 / 28.4 | 2.9 / 25.8 | 4.3 / 27.8 |
| bs_en | - / - | 40.7 / 65.1 | 42.6 / 66.2 | 41.9 / 66 | 4.6 / 24.2 | 27.2 / 52.2 | 33.6 / 59.4 | 35.3 / 60.4 |
| ca_en | - / - | 41.6 / 66 | 43.4 / 66.9 | 43.1 / 66.8 | 5.4 / 22.9 | 31.3 / 56.6 | 34.5 / 61 | 36.1 / 61.4 |
| ckb_en | - / - | 28.3 / 53.4 | 33.9 / 58 | 33.5 / 57.9 | 1.1 / 11.4 | 7.8 / 27.4 | 18.8 / 43.6 | 20.2 / 45.3 |
| cs_en | - / - | 39.4 / 64.8 | 41.1 / 65.7 | 41.4 / 65.8 | 4.3 / 25.4 | 26 / 51.5 | 33.5 / 59.3 | 34.1 / 59.7 |
| cy_en | - / - | 43.2 / 66.6 | 45 / 68.1 | 45.2 / 68.2 | 3.9 / 24.2 | 35.2 / 58.8 | 37.5 / 61.1 | 37.2 / 60.7 |
| da_en | - / - | 42.5 / 65 | 43.7 / 65.8 | 43.7 / 65.8 | 6.3 / 28.6 | 36.1 / 60.2 | 37.6 / 61.2 | 37.5 / 61.2 |
| de_en | - / - | 38.9 / 64.6 | 39.8 / 65.4 | 40.2 / 65.5 | 4.3 / 26.4 | 31.6 / 59.4 | 33 / 59.1 | 34 / 60.4 |
| dv_en | - / - | 22.2 / 48.8 | 24.6 / 50.5 | 25 / 50.8 | 0.6 / 6.8 | 7.9 / 29.6 | 11.1 / 36.9 | 12.6 / 39.5 |
| dz_en | - / - | 6.4 / 28.2 | 6.4 / 27.9 | 6.7 / 28.7 | 0 / 1.6 | 0.9 / 13.9 | 1.1 / 17.5 | 0.9 / 15 |
| ee_en | - / - | 13.2 / 35.1 | 16.3 / 38.4 | 16.7 / 38.5 | 0.3 / 7.9 | 3.2 / 22.1 | 1 / 15.2 | 2.8 / 19.6 |
| el_en | - / - | 43 / 65.8 | 43.3 / 66.1 | 44.6 / 66.7 | 5.2 / 23.8 | 33.9 / 58.7 | 31.7 / 58 | 32.3 / 59 |
| en_GB_en | - / - | 89.3 / 94.4 | 90.3 / 94.6 | 90.5 / 94.9 | 7.9 / 26 | 97.5 / 98.7 | 99.4 / 99.8 | 99.4 / 99.7 |
| en_IN_en | - / - | 87.1 / 93.6 | 86.4 / 93 | 86.9 / 93.8 | 10.1 / 27 | 35.1 / 45.3 | 93.7 / 98.3 | 95.6 / 98.7 |
| en_af | - / - | 42.3 / 66.7 | 44.5 / 68.4 | 44.6 / 68.3 | 2.9 / 16.5 | 35 / 61.4 | 37.7 / 63.3 | 37.8 / 63.4 |
| en_am | - / - | 3.8 / 23 | 4.5 / 23.8 | 4.2 / 24.3 | 0.1 / 0.7 | 0.2 / 0.8 | 0 / 0.6 | 0 / 0.4 |
| en_ar | - / - | 24.7 / 54.2 | 24.6 / 54.3 | 24.9 / 54.7 | 0.4 / 2.9 | 6.7 / 38.1 | 15.4 / 45.3 | 12.1 / 44 |
| en_az | - / - | 14.6 / 46.7 | 15.2 / 47.5 | 15.5 / 48.2 | 0.4 / 8.6 | 10.4 / 41.6 | 10.6 / 42.9 | 7.4 / 40.9 |
| en_ba | - / - | 3.3 / 24.3 | 4 / 26.3 | 5.2 / 28.9 | 0 / 0.4 | 0.3 / 13.4 | 0.1 / 9.8 | 0.1 / 10.5 |
| en_be | - / - | 21.9 / 44.8 | 24.2 / 47.3 | 26 / 50.2 | 0.7 / 6 | 20.6 / 49.4 | 21.6 / 50.3 | 21.9 / 50.6 |
| en_bem | - / - | 2.4 / 4 | 3.9 / 19.5 | 2.1 / 3.4 | 0.1 / 6.7 | 0.1 / 8.7 | 0.2 / 10.6 | 0.1 / 5 |
| en_bg | - / - | 34.1 / 59.5 | 34.8 / 59.9 | 34.6 / 59.7 | 1.4 / 8.8 | 21.1 / 48.7 | 27.3 / 53.4 | 29.1 / 55.1 |
| en_bn | - / - | 13.8 / 48.5 | 14 / 49.2 | 14.6 / 49.8 | 0.4 / 7.4 | 8.6 / 40.8 | 7.4 / 41.5 | 5.8 / 42 |
| en_bo | - / - | 12 / 25.2 | 14.7 / 29.1 | 14 / 28.6 | 0 / 0.9 | 0 / 1.8 | 0 / 1.6 | 0 / 1.1 |
| en_bs | - / - | 29.4 / 58.6 | 30.3 / 59.4 | 30.2 / 59.3 | 0.7 / 10.5 | 23.6 / 53 | 24.9 / 54.2 | 25.3 / 54.3 |
| en_ca | - / - | 39.9 / 63 | 40.9 / 63.7 | 40.5 / 63.4 | 2.3 / 15.1 | 30.1 / 55.8 | 31.6 / 57.7 | 32.1 / 58.1 |

| | | | | | | | | |
|---|---|---|---|---|---|---|---|---|
| en_ckb | - / - | 8.5 / 42.4 | 9.2 / 42.5 | 9.2 / 42.2 | 0 / 0.2 | 0.1 / 10.7 | 0.2 / 10.9 | 0.1 / 9.3 |
| en_cs | - / - | 32.4 / 59.4 | 32.9 / 59.6 | 32.6 / 59.7 | 1 / 10.8 | 22.2 / 50.2 | 22.7 / 51.2 | 20.9 / 51.5 |
| en_cy | - / - | 29.6 / 56.4 | 33 / 59.7 | 33.3 / 59.8 | 0.3 / 9.3 | 3.4 / 25.2 | 2.6 / 24.2 | 3 / 29.1 |
| en_da | - / - | 40.4 / 63.9 | 40.9 / 64.3 | 40.7 / 64.1 | 2 / 13.5 | 34.8 / 59.2 | 35.5 / 59.9 | 36.6 / 60.6 |
| en_de | - / - | 34.5 / 61.8 | 35.2 / 62.1 | 34.7 / 61.8 | 2 / 16.2 | 21.3 / 49.9 | 23.5 / 53.1 | 22.1 / 52.7 |
| en_dv | - / - | 3.5 / 40.7 | 4.7 / 44.4 | 4.7 / 44.7 | 0 / 0.2 | 0.3 / 0.4 | 0.2 / 0.4 | 0.1 / 0.4 |
| en_dz | - / - | 22 / 35.8 | 22 / 35.5 | 22.7 / 37.2 | 0 / 0.1 | 0 / 2.1 | 0 / 0.3 | 0 / 1.1 |
| en_ee | - / - | 6.1 / 29.1 | 7.4 / 31.6 | 7.2 / 30.5 | 0.2 / 6.2 | 0.4 / 13.5 | 0.2 / 6.9 | 0.1 / 4.5 |
| en_el | - / - | 37.4 / 61.5 | 37.4 / 61.6 | 37.6 / 61.6 | 0.9 / 3.7 | 26.7 / 50.8 | 28.2 / 52.4 | 28 / 52.6 |
| en_en_GB | - / - | 6.5 / 9.1 | 34.4 / 48.4 | 5.8 / 7.1 | 8.3 / 21.8 | 99.1 / 99.7 | 99.4 / 99.8 | 99.5 / 99.8 |
| en_en_IN | - / - | 5.7 / 8.7 | 21.2 / 28 | 4.1 / 5.5 | 4.4 / 15.4 | 91.2 / 97.4 | 92.3 / 98 | 92.1 / 97.9 |
| en_es | - / - | 42 / 65.4 | 42.3 / 65.6 | 42.2 / 65.5 | 1.4 / 15.4 | 28.6 / 53.7 | 34.1 / 59 | 33.5 / 58.5 |
| en_es_MX | - / - | 2.4 / 5.7 | 7.6 / 28.1 | 1.7 / 4 | 0.8 / 9.7 | 34.1 / 58.8 | 36.9 / 60.7 | 36.5 / 60.9 |
| en_et | - / - | 26.4 / 59 | 26.7 / 59.1 | 27.3 / 59.4 | 0.5 / 10.2 | 11.2 / 42.8 | 20.2 / 52.9 | 21.8 / 53.7 |
| en_eu | - / - | 14.8 / 51.3 | 15 / 51.2 | 15.4 / 52.3 | 0.2 / 9.4 | 9.9 / 45.6 | 9.8 / 45.7 | 7.4 / 45.7 |
| en_fa | - / - | 19.1 / 45 | 19.5 / 44.9 | 19.5 / 44.9 | 0.5 / 3.1 | 14.3 / 40.7 | 14.9 / 42.4 | 14.8 / 43 |
| en_fa_AF | - / - | 0.3 / 0.9 | 0.2 / 0.8 | 0.2 / 0.8 | 0 / 0.2 | 12.4 / 39.6 | 13.5 / 41.1 | 11.4 / 40.1 |
| en_ff | - / - | 4.8 / 27.6 | 5.3 / 28.1 | 5.1 / 28.8 | 0.2 / 5.5 | 0.1 / 7.9 | 0.1 / 5.5 | 0.1 / 4.3 |
| en_fi | - / - | 21.5 / 57.1 | 21.7 / 57.4 | 22 / 57.3 | 0.5 / 11 | 14.5 / 47.5 | 16.4 / 50.5 | 17.2 / 50.4 |
| en_fil | - / - | 25.3 / 55.4 | 25.8 / 55.6 | 25.3 / 55.4 | 1.6 / 14.6 | 26.5 / 54 | 30 / 57.6 | 28.6 / 55.8 |
| en_fj | - / - | 6.4 / 27.5 | 7.3 / 28.6 | 7.7 / 29 | 0.3 / 6.8 | 0 / 4.6 | 0.2 / 7.3 | 0.1 / 5 |
| en_fo | - / - | 16.4 / 37.9 | 18.6 / 40.8 | 16.6 / 38.2 | 0.4 / 9 | 5.6 / 29.1 | 7.5 / 32.5 | 7.2 / 32.7 |
| en_fr | - / - | 38.3 / 62.9 | 39.2 / 63.5 | 39.2 / 63.5 | 0.8 / 12.7 | 25.8 / 54 | 30.2 / 56.2 | 26.7 / 56.6 |
| en_fr_CA | - / - | 2.1 / 5.7 | 4.2 / 27.8 | 2.5 / 7.6 | 0.6 / 8.6 | 29.6 / 55.8 | 29.7 / 56.5 | 30 / 57.6 |
| en_ga | - / - | 26.8 / 54.9 | 27.5 / 55.5 | 27.1 / 55.2 | 0.4 / 10.2 | 1.2 / 16.2 | 0.9 / 15.4 | 0.8 / 15.9 |
| en_gl | - / - | 39.9 / 64.5 | 40.6 / 65 | 40.5 / 64.8 | 1.3 / 13.2 | 31.3 / 57.5 | 32.6 / 58.6 | 30.6 / 58.1 |
| en_gu | - / - | 14.9 / 47.5 | 15.3 / 48 | 15.4 / 48.2 | 0.1 / 0.3 | 0.4 / 1.2 | 0.1 / 1.7 | 0.1 / 1.3 |
| en_ha | - / - | 21 / 49 | 21.8 / 50.1 | 22.8 / 51.1 | 0.2 / 4.2 | 0.4 / 11.4 | 0.3 / 9.9 | 0.3 / 9 |
| en_hi | - / - | 25.5 / 50.2 | 25.7 / 50.2 | 25.9 / 50.6 | 0.9 / 7.8 | 16.7 / 42.9 | 15.6 / 42.8 | 12.6 / 42.8 |
| en_hmn | - / - | 16.6 / 42.1 | 17.5 / 42.9 | 17.7 / 43.3 | 0.2 / 6.1 | 1.1 / 12.4 | 0.6 / 10.3 | 0.5 / 9.4 |
| en_hr | - / - | 34.9 / 62 | 35.5 / 62.7 | 35.6 / 62.8 | 3.7 / 21.3 | 25.7 / 54 | 26.7 / 55.3 | 25.1 / 55.3 |
| en_hu | - / - | 21 / 50.8 | 21.4 / 51.2 | 21.7 / 51.4 | 0.6 / 10.8 | 10.6 / 36.9 | 11.1 / 42.3 | 12.3 / 42.5 |
| en_hy | - / - | 15.7 / 46.5 | 18.5 / 51 | 19.1 / 51.7 | 0.1 / 0.5 | 0.4 / 0.8 | 0.5 / 1 | 0.5 / 1.2 |
| en_id | - / - | 38.7 / 65.2 | 39.6 / 65.8 | 39.8 / 65.8 | 0.7 / 9 | 32.8 / 60.5 | 35 / 62.7 | 34.8 / 62.7 |
| en_ig | - / - | 13.5 / 37.9 | 16.4 / 43.2 | 16.8 / 42 | 0.2 / 5.8 | 0.2 / 9.7 | 0.1 / 5.3 | 0.2 / 5.5 |
| en_is | - / - | 25.1 / 52.8 | 25.3 / 53 | 25.3 / 53.3 | 2.6 / 16.8 | 22.5 / 48.9 | 26.3 / 52.3 | 26 / 51.9 |
| en_it | - / - | 38.8 / 64 | 39.2 / 64.1 | 39 / 64.1 | 1.2 / 14.7 | 28.7 / 55.8 | 29.7 / 56.7 | 30.2 / 57 |
| en_iw | - / - | 24.3 / 53 | 22.6 / 48.5 | 25.4 / 54.2 | 0.5 / 2.6 | 14 / 42.3 | 15.9 / 44.6 | 15.7 / 44.8 |
| en_ja | - / - | 6.3 / 33.6 | 6.8 / 34.3 | 5.9 / 33.9 | 0 / 3.2 | 0.1 / 26.5 | 0.1 / 26.5 | 0.2 / 24.8 |
| en_ka | - / - | 2.7 / 15.3 | 1.6 / 11.9 | 3.1 / 15.5 | 0.6 / 6.4 | 10.2 / 42.1 | 11.5 / 44.8 | 9.7 / 46 |
| en_kk | - / - | 12.3 / 45.5 | 13.2 / 47.1 | 13.7 / 47.3 | 0.3 / 5.9 | 8.4 / 41.2 | 7.7 / 40.8 | 6.7 / 41.2 |
| en_km | - / - | 6 / 40.8 | 6 / 41.3 | 4.8 / 43.6 | 0.1 / 1.1 | 1.6 / 27.4 | 0.7 / 32.3 | 0.3 / 29.3 |
| en_kn | - / - | 10.8 / 48.4 | 12.1 / 50 | 11.9 / 49.8 | 0.1 / 0.5 | 0.2 / 4.1 | 0.1 / 4.9 | 0.1 / 4.9 |
| en_ko | - / - | 10.9 / 31.7 | 11.6 / 32.4 | 11.9 / 32.7 | 0.2 / 2.8 | 5.6 / 22.1 | 3.5 / 22.8 | 2.9 / 22.8 |
| en_ku | - / - | 0.2 / 0.5 | 0.2 / 0.6 | 0.3 / 0.8 | 0.1 / 6.4 | 0.6 / 15.8 | 0.5 / 16.3 | 0.3 / 14.3 |
| en_ky | - / - | 9 / 42.1 | 10.6 / 45 | 10.4 / 44.2 | 0.1 / 1.1 | 1.3 / 22.7 | 0.9 / 24.5 | 0.8 / 24.9 |
| en_lb | - / - | 8.5 / 29.5 | 8.9 / 29.8 | 9.6 / 31 | 0.4 / 10 | 3.5 / 32.4 | 4.1 / 32.7 | 4.6 / 33.9 |
| en_lo | - / - | 11.1 / 38 | 11.1 / 38.4 | 11.7 / 39.1 | 0.2 / 1.4 | 1.3 / 19.4 | 2.7 / 31.6 | 1.9 / 32.6 |
| en_lt | - / - | 24.3 / 56.1 | 26 / 56.7 | 25 / 56.3 | 0.6 / 11.8 | 15.3 / 46.8 | 17.8 / 50 | 17.4 / 50.3 |
| en_lv | - / - | 22.6 / 54.1 | 22.9 / 54.1 | 23.4 / 54.6 | 1 / 10.2 | 17.6 / 48.1 | 18.8 / 49.4 | 19.1 / 50.6 |
| en_mey | - / - | 0.1 / 0.6 | 0.1 / 0.5 | 0.1 / 0.6 | 0 / 0.2 | 2.9 / 29.9 | 0.7 / 21 | 0.8 / 23.2 |
| en_mg | - / - | 16.8 / 48.5 | 17.7 / 50.1 | 17.6 / 49.6 | 0.4 / 6.7 | 1.3 / 19.4 | 0.6 / 17.1 | 0.3 / 14 |
| en_mi | - / - | 20.9 / 46.4 | 19.7 / 45.8 | 21.5 / 47.1 | 0.5 / 7.9 | 0.8 / 13.6 | 0.4 / 11.2 | 0.4 / 10 |
| en_mk | - / - | 37 / 63.9 | 37.6 / 64.1 | 37.6 / 64.1 | 1 / 4 | 27.2 / 55.3 | 28.9 / 57.1 | 30.4 / 58.5 |
| en_ml | - / - | 7.1 / 44.2 | 8 / 44.4 | 8.3 / 45.3 | 0.3 / 8.8 | 5.4 / 37 | 4.6 / 39.7 | 2.1 / 37.9 |
| en_mn | - / - | 10.7 / 41.6 | 12 / 44 | 12.4 / 45 | 0.1 / 0.5 | 0.6 / 1.6 | 0.4 / 3.5 | 0.4 / 4.1 |
| en_mr | - / - | 7.5 / 36 | 7.3 / 34.2 | 8.4 / 37.3 | 1.2 / 14.1 | 7.8 / 39.6 | 7.4 / 40 | 6.8 / 40.3 |
| en_ms | - / - | 33.6 / 61.8 | 35.1 / 62.8 | 35.3 / 62.9 | 1.8 / 14.2 | 29.1 / 56.6 | 31.6 / 62 | 33 / 62.2 |
| en_mt | - / - | 45.7 / 66.6 | 47.1 / 67.3 | 47.6 / 67.5 | 0.4 / 10 | 3.7 / 25.6 | 3.7 / 34.2 | 3.3 / 33.7 |
| en_my | - / - | 1.7 / 16.7 | 1.5 / 16.2 | 1.4 / 16.1 | 0 / 0.5 | 0 / 5.4 | 0 / 5.1 | 0 / 6.4 |
| en_nd | - / - | 1.3 / 3 | 1.8 / 18.2 | 1 / 2.2 | 0.1 / 2.5 | 0.3 / 20.2 | 0.2 / 13.3 | 0.2 / 13 |
| en_ne | - / - | 7.5 / 35.9 | 7.9 / 37 | 8.1 / 37.8 | 0.1 / 0.5 | 1.4 / 23.9 | 1.7 / 30.3 | 1 / 26.5 |
| en_nl | - / - | 36.7 / 62.9 | 37.5 / 63.4 | 37.4 / 63.3 | 2 / 15.7 | 25 / 51.7 | 28.8 / 56.6 | 29.8 / 57.8 |
| en_nn | - / - | 34.6 / 60.9 | 35.8 / 61.6 | 35.8 / 61.4 | 4.5 / 14.4 | 24.9 / 53.2 | 29.9 / 56.1 | 30.6 / 57 |
| en_no | - / - | 38.7 / 63.6 | 39 / 63.7 | 38.9 / 63.6 | 1.9 / 12.7 | 33.1 / 59 | 34.9 / 60.3 | 34.8 / 60.3 |
| en_nso | - / - | 7.9 / 31.2 | 9.1 / 33.5 | 9.2 / 33.7 | 0.3 / 4.6 | 0.9 / 13.8 | 0.5 / 13.5 | 0.4 / 12.4 |
| en_ny | - / - | 12 / 41.6 | 14.8 / 45.5 | 14.4 / 44.6 | 0.7 / 8 | 1.1 / 18.3 | 0.7 / 17.8 | 0.7 / 18.3 |
| en_om | - / - | 1.1 / 23.2 | 1.8 / 30.4 | 2 / 30.3 | 0.1 / 6.5 | 0.1 / 9.7 | 0.1 / 7.3 | 0.1 / 7.6 |
| en_pa | - / - | 18.2 / 44.1 | 18.9 / 45.2 | 19.2 / 45.6 | 0.1 / 0.3 | 0.6 / 2.6 | 0.2 / 2.6 | 0.2 / 1.4 |
| en_pl | - / - | 27.6 / 55.5 | 28.1 / 55.8 | 28.2 / 55.9 | 1.4 / 12.2 | 18.1 / 45.3 | 15.2 / 46.7 | 14.9 / 45.4 |
| en_ps | - / - | 10.5 / 33.6 | 10.8 / 34.3 | 10.8 / 34.4 | 0.1 / 0.5 | 0.5 / 10.3 | 0.3 / 8.6 | 0.3 / 7.6 |
| en_pt | - / - | 41.7 / 64.9 | 42.3 / 65.3 | 42.3 / 65.3 | 3.3 / 18.3 | 30 / 55.8 | 33.4 / 58.8 | 34 / 59.7 |
| en_pt_PT | - / - | 2.7 / 5.9 | 8.9 / 34.5 | 3 / 18 | 2.9 / 13.4 | 23.9 / 49.3 | 31.7 / 57.4 | 32.7 / 57.9 |
| en_ro | - / - | 34.5 / 59 | 35.3 / 59.4 | 35 / 59.2 | 4.4 / 19.8 | 28.1 / 52.8 | 30.2 / 55 | 30.8 / 55.4 |
| en_ru | - / - | 30.7 / 55.8 | 31.9 / 56.7 | 31.8 / 56.7 | 1 / 2.9 | 18 / 41.8 | 22.3 / 48.5 | 23 / 48.9 |
| en_rw | - / - | 3.2 / 18.6 | 0.3 / 8.3 | 0.9 / 12 | 1 / 9.5 | 1.1 / 18 | 0.8 / 17.5 | 0.5 / 16.5 |
| en_sd | - / - | 10 / 35.8 | 11.4 / 37.1 | 11.2 / 37.4 | 0.1 / 0.6 | 0.3 / 8.7 | 0.2 / 8.8 | 0.3 / 10.9 |
| en_shi | - / - | 0.3 / 0.7 | 0.2 / 0.7 | 0.2 / 0.7 | 0 / 0.3 | 0.1 / 4.3 | 0 / 3.4 | 0 / 3.3 |
| en_si | - / - | 11 / 42.6 | 10.9 / 42.2 | 11.4 / 44.2 | 0.1 / 0.5 | 0.4 / 1.1 | 0.6 / 1 | 0.4 / 1.2 |
| en_sk | - / - | 33.9 / 60.3 | 34.2 / 60.4 | 34.3 / 60.7 | 2 / 14.3 | 26.4 / 53.1 | 25.6 / 53.2 | 27.5 / 54.1 |
| en_sl | - / - | 32.4 / 59.6 | 33.2 / 60 | 33.4 / 60.1 | 1.1 / 12.4 | 21.1 / 50.5 | 23.9 / 52.5 | 24.9 / 53 |

| | | | | | | | | |
|---|---|---|---|---|---|---|---|---|
| en_sm | - / - | 23.4 / 48.7 | 25.4 / 51.2 | 27 / 51.9 | 0.4 / 7.5 | 1 / 16.6 | 0.5 / 12.6 | 0.4 / 11.8 |
| en_sn | - / - | 8.9 / 38.5 | 10.6 / 41.4 | 11.4 / 43.5 | 0.3 / 6 | 0.5 / 15.3 | 0.5 / 16.8 | 0.5 / 17 |
| en_so | - / - | 14.2 / 46.9 | 14.5 / 48 | 14.8 / 48.1 | 0.3 / 7.6 | 2.1 / 14.7 | 0.3 / 12.5 | 0.4 / 14.3 |
| en_sq | - / - | 31.8 / 57.9 | 32.3 / 58.3 | 31.9 / 58 | 1.9 / 11.8 | 30.4 / 56.3 | 31.4 / 56.8 | 31.9 / 57 |
| en_sr | - / - | 21.3 / 45 | 21.7 / 45.4 | 21.8 / 45.5 | 0.2 / 1.3 | 4.2 / 21.3 | 14.9 / 39.3 | 9.3 / 34.3 |
| en_sr_Latn | - / - | 0.9 / 3.6 | 0.9 / 19 | 0.8 / 2.5 | 1.1 / 5 | 21.4 / 50.1 | 22.9 / 51.1 | 23 / 51.5 |
| en_ss | - / - | 5.7 / 35.4 | 6.6 / 36.7 | 6.9 / 38 | 0.2 / 6.7 | 0.1 / 6.1 | 0.2 / 12.9 | 0.2 / 11.4 |
| en_sv | - / - | 43.3 / 67.5 | 43.7 / 67.6 | 43.2 / 67.3 | 2 / 13.7 | 35 / 60.4 | 36 / 62.5 | 35.8 / 61.8 |
| en_sw | - / - | 31.7 / 57.2 | 30.9 / 56.6 | 29 / 54.7 | 0.8 / 9.5 | 2.4 / 23 | 1.7 / 25 | 2.8 / 31.2 |
| en_ta | - / - | 7.8 / 45.1 | 8.3 / 45.3 | 8.4 / 45.8 | 0.4 / 11.7 | 2.1 / 34.1 | 4.6 / 40.3 | 2.6 / 40.1 |
| en_te | - / - | 9.2 / 44.3 | 9.8 / 45.1 | 10 / 45 | 0.5 / 9.6 | 5.4 / 35.8 | 5.4 / 39.8 | 3.6 / 38.5 |
| en_tg | - / - | 3.2 / 21 | 5.5 / 24.4 | 3.4 / 19.5 | 0.1 / 0.5 | 0.8 / 15.3 | 0.8 / 19.7 | 0.7 / 20.6 |
| en_th | - / - | 6.6 / 40.9 | 5.6 / 40.5 | 5.8 / 39.4 | 0.2 / 10.1 | 11.8 / 47.4 | 9 / 48.3 | 12.6 / 49.1 |
| en_ti | - / - | 2.3 / 15.7 | 3.3 / 19.6 | 3.9 / 20.4 | 0.2 / 0.5 | 0 / 1.1 | 0 / 0.9 | 0 / 1.1 |
| en_tk | - / - | 12.2 / 42 | 14.2 / 46.2 | 13.8 / 45.6 | 0.2 / 6.7 | 1.1 / 18.3 | 0.5 / 19.2 | 0.6 / 19.4 |
| en_tn | - / - | 16.3 / 36.9 | 19.9 / 41.5 | 19.1 / 40.9 | 0.5 / 7.3 | 0.3 / 9.1 | 0.3 / 10.5 | 0.4 / 11.1 |
| en_to | - / - | 3.9 / 25.5 | 3.3 / 26.4 | 4.9 / 27.8 | 0.1 / 3.6 | 0.3 / 11.2 | 0.2 / 9.3 | 0.2 / 8.3 |
| en_tr | - / - | 25 / 55.1 | 25.4 / 55.4 | 25.9 / 55.7 | 0.8 / 11.6 | 11.6 / 43.4 | 11.2 / 44.4 | 10.4 / 44.7 |
| en_tt | - / - | 5 / 27.3 | 8 / 32.4 | 9.7 / 36 | 0.1 / 0.5 | 0.7 / 19.7 | 0.4 / 22.2 | 0.4 / 20.7 |
| en_ty | - / - | 0.6 / 2.4 | 1.3 / 12.8 | 0.4 / 2.1 | 0.4 / 9.2 | 0.6 / 14.1 | 0.4 / 12.6 | 0.4 / 11.3 |
| en_ug | - / - | 4.7 / 29.9 | 5.4 / 31.3 | 5.8 / 31.5 | 0.1 / 0.3 | 0.1 / 1.4 | 0.2 / 2.3 | 0 / 1.2 |
| en_uk | - / - | 24.3 / 51.4 | 25.5 / 52.6 | 25.6 / 52.5 | 1.4 / 5.6 | 18.5 / 45.9 | 20.4 / 48.2 | 18.1 / 46.9 |
| en_ur | - / - | 2.3 / 16.3 | 2.3 / 16.3 | 2.4 / 16.4 | 0.2 / 1.4 | 7.1 / 32 | 4.3 / 31.5 | 2.5 / 28.2 |
| en_uz | - / - | 0.7 / 1.5 | 0.7 / 1.4 | 0.6 / 1.3 | 0.1 / 7.3 | 1.3 / 26.8 | 1.8 / 31.9 | 2 / 33.9 |
| en_ve | - / - | 7 / 31.1 | 7.8 / 32 | 8.5 / 32.9 | 0.2 / 4.2 | 0.3 / 11.9 | 0.1 / 8.1 | 0.1 / 8.1 |
| en_vi | - / - | 0.1 / 12.5 | 0.1 / 12.4 | 0.1 / 12.4 | 1.4 / 9.2 | 34.4 / 53.4 | 34.7 / 53.8 | 34.6 / 54 |
| en_wo | - / - | 0.4 / 9.8 | 0.5 / 10 | 0.5 / 12.1 | 0.3 / 6.4 | 0.4 / 11.4 | 0.2 / 8.9 | 0 / 5.3 |
| en_xh | - / - | 10.3 / 45.1 | 10.9 / 47.1 | 11.3 / 47.5 | 0.2 / 6.8 | 0.3 / 14 | 0.2 / 12.5 | 0.2 / 13.7 |
| en_yo | - / - | 5.4 / 20.9 | 5.5 / 21.6 | 5.7 / 21.2 | 0.2 / 4.2 | 0.1 / 4.6 | 0 / 6.1 | 0.2 / 10.5 |
| en_yue | - / - | 0 / 1.3 | 0 / 1.2 | 0 / 1.2 | 0 / 1.9 | 0.1 / 13.2 | 0 / 14.6 | 0 / 15.7 |
| en_zh | - / - | 3.8 / 29.7 | 3.9 / 30.1 | 3.9 / 30.1 | 1.9 / 2.7 | 19.5 / 21.2 | 23.6 / 21.3 | 23.1 / 23.2 |
| en_zh_Hant | - / - | 1 / 8.2 | 0.9 / 8.8 | 0.7 / 5.3 | 0.4 / 4 | 6 / 23.1 | 6.5 / 22.9 | 3.8 / 22.7 |
| en_zu | - / - | 2.7 / 24.6 | 2.6 / 25.2 | 2.6 / 24.7 | 0.5 / 7.3 | 0.5 / 15.2 | 0.2 / 13.9 | 0.2 / 12.6 |
| es_MX_en | - / - | 50.4 / 71.7 | 51.9 / 72.7 | 52.3 / 72.7 | 6.1 / 24 | 40.5 / 64.9 | 41.3 / 65.4 | 41.1 / 65.3 |
| es_en | - / - | 41.1 / 66.4 | 42.4 / 67.1 | 42.9 / 67.3 | 5.9 / 26.2 | 33.1 / 60 | 35.2 / 61.9 | 35.8 / 62 |
| et_en | - / - | 35.1 / 61.8 | 36.8 / 62.8 | 36.8 / 62.8 | 8.6 / 31.2 | 25.5 / 53.8 | 30.3 / 56.7 | 30.3 / 56.6 |
| eu_en | - / - | 31.8 / 56.8 | 34 / 58.3 | 34 / 58.2 | 5.5 / 25.7 | 22.4 / 48.8 | 23.4 / 49.6 | 23.8 / 50.1 |
| fa_AF_en | - / - | 32.3 / 58.6 | 33 / 59 | 34.1 / 59.8 | 2.4 / 14.8 | 18.2 / 42.1 | 24.6 / 51.1 | 25.6 / 52 |
| fa_en | - / - | 32.2 / 58.6 | 34 / 59.7 | 33.7 / 59.5 | 3.7 / 18.7 | 16.3 / 37.6 | 24.1 / 50.2 | 25.4 / 51.6 |
| ff_en | - / - | 14.4 / 35.8 | 14.2 / 36 | 15.8 / 37.5 | 0.3 / 9.6 | 3.3 / 20.5 | 4 / 18.7 | 2.4 / 18.8 |
| fi_en | - / - | 30.3 / 57.8 | 31.7 / 58.8 | 31.5 / 58.6 | 4.2 / 24.6 | 24.3 / 52.5 | 26.4 / 52.8 | 28.1 / 55.1 |
| fil_en | - / - | 45.7 / 66.9 | 48.2 / 68.5 | 48.1 / 68.7 | 4 / 19.3 | 38.3 / 61.7 | 36.2 / 59.5 | 39.9 / 62.6 |
| fj_en | - / - | 6.6 / 26.6 | 8.4 / 28.5 | 8.2 / 28.4 | 0.5 / 10.3 | 4.7 / 18.7 | 6.2 / 27.1 | 3.7 / 21.7 |
| fo_en | - / - | 32.9 / 56.7 | 37 / 59.8 | 38.6 / 61 | 8 / 28.7 | 28.6 / 52.8 | 33.2 / 57.1 | 33.6 / 56.9 |
| fr_CA_en | - / - | 43.2 / 66.8 | 44.1 / 67.4 | 43.8 / 67.4 | 5 / 25.8 | 26.6 / 56 | 37.3 / 62.8 | 36.7 / 62.5 |
| fr_en | - / - | 38.8 / 63.5 | 39.5 / 64.1 | 39.2 / 63.9 | 4.5 / 26.1 | 31.1 / 57.6 | 32.6 / 57.8 | 32.6 / 58.8 |
| ga_en | - / - | 38.7 / 64.1 | 41 / 65.7 | 41.6 / 65.7 | 4.2 / 23 | 24.6 / 48.7 | 29.2 / 53.5 | 31.5 / 55.3 |
| gl_en | - / - | 43.7 / 66.8 | 45.3 / 67.9 | 45.1 / 67.8 | 6 / 19.5 | 32.9 / 59.7 | 35.3 / 60.3 | 36.9 / 62.3 |
| gu_en | - / - | 34.9 / 62 | 38.3 / 64 | 38 / 64.1 | 2.2 / 16.6 | 21.1 / 48 | 24.2 / 52.1 | 25.4 / 53.6 |
| ha_en | - / - | 35 / 56.9 | 39.2 / 59.9 | 38.7 / 60 | 1.1 / 14.7 | 12.5 / 33 | 21 / 43.6 | 18.9 / 43.3 |
| hi_en | - / - | 35.7 / 61.8 | 38.7 / 63.4 | 38.2 / 63.1 | 2.7 / 14.7 | 20 / 44.9 | 24.6 / 52.3 | 26.7 / 54.3 |
| hmn_en | - / - | 23.4 / 47.5 | 25.5 / 49.6 | 27 / 50.2 | 1 / 12.3 | 10 / 30.5 | 13.4 / 38.2 | 14.3 / 37.7 |
| hr_en | - / - | 41.3 / 65.5 | 42.4 / 66.5 | 42.3 / 66.4 | 6.6 / 26.9 | 29.9 / 55.6 | 31.7 / 57.6 | 35.2 / 60.6 |
| hu_en | - / - | 27.8 / 55.6 | 27.6 / 55.8 | 28.1 / 55.8 | 3.5 / 25.6 | 20.1 / 47 | 21.6 / 49 | 21.3 / 49.2 |
| hy_en | - / - | 30.7 / 55.8 | 34.8 / 59.3 | 34.1 / 58.9 | 3.2 / 18.7 | 17.8 / 44.4 | 22.9 / 48.8 | 23.7 / 50.7 |
| id_en | - / - | 43 / 66.4 | 45.1 / 67.8 | 45 / 67.8 | 3.8 / 18.4 | 34.8 / 59.9 | 37.3 / 61.5 | 38 / 61.9 |
| ig_en | - / - | 27.5 / 51.1 | 31.9 / 55.3 | 32.7 / 55.8 | 1.1 / 14.4 | 9.4 / 28.1 | 11.4 / 33.4 | 11.8 / 33.1 |
| is_en | - / - | 35.3 / 60.5 | 37.3 / 61.8 | 37.7 / 61.8 | 9.1 / 31.5 | 29 / 54.9 | 30.5 / 55.3 | 33 / 57.9 |
| it_en | - / - | 42.3 / 66.2 | 43.2 / 66.8 | 43.2 / 66.9 | 7.5 / 28.6 | 36.8 / 61.5 | 36 / 60.5 | 38.5 / 62.4 |
| iw_en | - / - | 40.5 / 63.8 | 42.9 / 65.7 | 42.9 / 65.6 | 4.3 / 19.1 | 29.2 / 55.7 | 31 / 56.9 | 31.1 / 57.4 |
| ja_en | - / - | 27.1 / 54.5 | 28.7 / 55.1 | 29.1 / 55.5 | 0.3 / 1.2 | 15 / 40.9 | 19.2 / 46.6 | 18.7 / 45.9 |
| ka_en | - / - | 28.8 / 53.9 | 31.8 / 56.1 | 32 / 56.3 | 2.6 / 14.1 | 20.1 / 45 | 26.3 / 52.4 | 25.3 / 52.6 |
| kk_en | - / - | 24.1 / 52.1 | 26.4 / 53.6 | 25.7 / 53.3 | 3.7 / 18.6 | 16.4 / 42.9 | 18.3 / 44.3 | 18 / 44 |
| km_en | - / - | 33.6 / 59.6 | 37.8 / 61.4 | 37.3 / 61.7 | 1.9 / 16.1 | 21.1 / 47.2 | 26.4 / 52.2 | 27.2 / 53.5 |
| kn_en | - / - | 29.2 / 56.8 | 31.4 / 57.9 | 31.1 / 58.3 | 1.3 / 11.5 | 13.7 / 38.9 | 18.8 / 45.8 | 20.4 / 48.2 |
| ko_en | - / - | 30.9 / 57.1 | 33.1 / 58.6 | 33.2 / 58.8 | 1.1 / 4.5 | 19 / 45.8 | 18.7 / 45.2 | 19 / 47 |
| ku_en | - / - | 27.6 / 52.5 | 30.5 / 54.4 | 30.7 / 54.9 | 2.9 / 18.3 | 17.2 / 41.7 | 16.5 / 43.2 | 20.2 / 44.2 |
| ky_en | - / - | 23.8 / 50.5 | 27.5 / 53.6 | 26.2 / 52.6 | 2.6 / 16.1 | 15.4 / 41.5 | 15.9 / 41.1 | 17 / 43.2 |
| lb_en | - / - | 21.9 / 49.8 | 24.4 / 51.6 | 26 / 52 | 4.2 / 23.8 | 29.5 / 53.3 | 32.3 / 56.6 | 30.6 / 56.1 |
| lo_en | - / - | 34.9 / 58.2 | 37.5 / 59.7 | 37.9 / 60.9 | 1.2 / 11.8 | 23.4 / 49 | 28.5 / 53.9 | 28.1 / 53.5 |
| lt_en | - / - | 32 / 59.6 | 33.9 / 60.6 | 34.1 / 60.6 | 4.9 / 25.7 | 24.3 / 52.3 | 25.7 / 52.9 | 26.4 / 52.9 |
| lv_en | - / - | 34.6 / 61.2 | 36.2 / 62.3 | 36.6 / 62.3 | 3.4 / 22.7 | 26.1 / 53.5 | 29.5 / 55.4 | 30.6 / 57.1 |
| mey_en | - / - | 18 / 45.5 | 19.7 / 47 | 20.2 / 46.9 | 0.3 / 4.2 | 4.5 / 22.6 | 11.8 / 38.8 | 10.4 / 38.8 |
| mg_en | - / - | 25.8 / 50.8 | 28.3 / 52.7 | 28.7 / 52.9 | 1.7 / 18.6 | 14 / 37.6 | 14.8 / 39.1 | 16 / 39.5 |
| mi_en | - / - | 22.1 / 45.3 | 20.5 / 44.1 | 22.5 / 46.3 | 2.2 / 19.2 | 10.4 / 35.6 | 10.8 / 35.7 | 17.7 / 41.1 |
| mk_en | - / - | 44.6 / 67.4 | 46.3 / 68.5 | 46.4 / 68.4 | 8.5 / 28.5 | 29.3 / 55.2 | 36.9 / 61.3 | 37.7 / 62.6 |
| ml_en | - / - | 30.3 / 56.8 | 31.4 / 57.1 | 32.4 / 58.2 | 0.8 / 5.7 | 11.5 / 34.3 | 17.8 / 45.8 | 18.8 / 46.5 |
| mn_en | - / - | 21.8 / 48.4 | 24 / 50.4 | 24.1 / 50.2 | 1.3 / 10.1 | 12.9 / 38.5 | 13.6 / 38.4 | 14.3 / 40.8 |
| mr_en | - / - | 32.5 / 59.3 | 34 / 60.2 | 34.9 / 60.9 | 1.7 / 9.8 | 12.7 / 35.2 | 22.6 / 50.7 | 23.4 / 50.4 |
| ms_en | - / - | 40.4 / 63.7 | 42.7 / 65.4 | 42.7 / 65.3 | 3.9 / 20.7 | 31.3 / 55.8 | 36 / 60.1 | 36.4 / 60.3 |
| mt_en | - / - | 50.9 / 72.9 | 52.4 / 73.9 | 53.5 / 74.3 | 5.6 / 25 | 40.3 / 63.2 | 40.1 / 62.5 | 41.5 / 65.1 |

| | | | | | | | | |
|---|---|---|---|---|---|---|---|---|
| my_en | - / - | 14.3 / 40.3 | 15.7 / 40.9 | 18.1 / 44.5 | 0.2 / 5.7 | 2.8 / 20.2 | 7.5 / 32.2 | 9.7 / 34.2 |
| nd_en | - / - | 18.8 / 43.1 | 21.6 / 45.6 | 21.8 / 45.8 | 0.2 / 8 | 8 / 27.2 | 9 / 32.6 | 9.1 / 32.1 |
| ne_en | - / - | 35.9 / 62 | 38.1 / 63.3 | 38.6 / 63.9 | 1.4 / 9 | 15.6 / 40.3 | 22.3 / 48.8 | 23.5 / 50.9 |
| nl_en | - / - | 41.2 / 66.1 | 42.5 / 66.7 | 42.5 / 66.6 | 8.3 / 31.4 | 35.1 / 60.3 | 36.5 / 61.4 | 36.6 / 62.3 |
| nn_en | - / - | 46 / 67.3 | 47.1 / 68.1 | 47 / 68.1 | 5.4 / 26.9 | 39 / 63.2 | 40.5 / 63.9 | 41 / 63.3 |
| no_en | - / - | 44 / 67 | 45.1 / 67.7 | 45 / 67.7 | 6.1 / 25.6 | 38.8 / 61.7 | 38.8 / 63.5 | 39.7 / 63.5 |
| nso_en | - / - | 23.6 / 47.6 | 27.5 / 51.3 | 28.1 / 51.7 | 0.6 / 10.9 | 3.7 / 26.1 | 10.1 / 31.4 | 11.1 / 34.9 |
| ny_en | - / - | 28.1 / 49.5 | 31.6 / 52.9 | 31.9 / 53.2 | 1.1 / 13 | 15.3 / 36.4 | 17.6 / 38.8 | 15.4 / 39.4 |
| om_en | - / - | 8.4 / 29.2 | 11.2 / 33.6 | 11.2 / 33.6 | 0.5 / 9.8 | 1.9 / 21.2 | 4.6 / 24.3 | 5.2 / 23.2 |
| pa_en | - / - | 34.6 / 60.8 | 38.5 / 62.9 | 38.2 / 63.1 | 2.4 / 16.1 | 16.1 / 42.3 | 24 / 51 | 25.4 / 51.8 |
| pl_en | - / - | 31.9 / 58.9 | 32.6 / 59.2 | 33.3 / 59.7 | 9.3 / 34.9 | 24.4 / 52.1 | 27.4 / 54.4 | 27.3 / 53.8 |
| ps_en | - / - | 22.6 / 49.5 | 23.9 / 50.5 | 24.5 / 50.8 | 1.9 / 14.2 | 14.1 / 38.8 | 15.6 / 40.3 | 16.3 / 42.1 |
| pt_PT_en | - / - | 44.7 / 67.4 | 45.7 / 68.2 | 45.9 / 68.2 | 5.2 / 23.6 | 35.2 / 60.3 | 36 / 60.8 | 36.8 / 61.4 |
| pt_en | - / - | 43.1 / 65.7 | 43.9 / 66.2 | 44.1 / 66.2 | 6.4 / 24.9 | 34.4 / 59.3 | 35.8 / 60.8 | 36.8 / 61.1 |
| ro_en | - / - | 37.7 / 63.9 | 38.5 / 64.6 | 39 / 64.7 | 6.1 / 28.5 | 22.8 / 46.9 | 32.3 / 58.6 | 33.4 / 59.5 |
| ru_en | - / - | 30.8 / 57.9 | 32.3 / 58.8 | 32.1 / 58.7 | 3.7 / 19.2 | 24 / 50.2 | 26.1 / 53.4 | 25.3 / 52.9 |
| rw_en | - / - | 27.6 / 52.4 | 30.9 / 55.2 | 32.3 / 55.8 | 1.9 / 18.2 | 15.8 / 37.9 | 17.2 / 42.3 | 16.3 / 42.3 |
| sd_en | - / - | 30.6 / 54.2 | 34 / 57 | 34.1 / 56.9 | 1.5 / 12.1 | 15.8 / 40.8 | 14.6 / 40.8 | 13.3 / 42.1 |
| shi_en | - / - | 3.3 / 23.3 | 4.4 / 25.8 | 5.8 / 27.6 | 0.1 / 3.1 | 1.5 / 15.9 | 2.9 / 24 | 2.5 / 23.5 |
| si_en | - / - | 31.7 / 58 | 34 / 59.4 | 34.3 / 60 | 1.1 / 10.1 | 16.2 / 44.1 | 18.1 / 45.5 | 17.3 / 43.8 |
| sk_en | - / - | 40.7 / 65.6 | 42.5 / 66.6 | 42.5 / 66.5 | 8.7 / 31.5 | 26.6 / 52.2 | 33.1 / 58.4 | 35.1 / 61 |
| sl_en | - / - | 36.5 / 62.4 | 38.2 / 63.4 | 38.3 / 63.3 | 6.5 / 28.7 | 29.7 / 56.3 | 32.2 / 58.2 | 30.8 / 56.6 |
| sm_en | - / - | 25.6 / 50.5 | 28.9 / 53.6 | 28.5 / 53.5 | 0.6 / 24.1 | 6 / 24.1 | 14.2 / 39.9 | 14 / 37.7 |
| sn_en | - / - | 28.6 / 50.6 | 33.5 / 54.4 | 34.5 / 55 | 0.8 / 10.9 | 8.6 / 25.6 | 16.6 / 38.8 | 15.1 / 37.7 |
| so_en | - / - | 36.3 / 59.1 | 40.7 / 62.8 | 41.1 / 63.3 | 1.9 / 13.1 | 20.5 / 42.3 | 20.3 / 43.8 | 21.3 / 49 |
| sq_en | - / - | 41.9 / 65.8 | 43.5 / 66.8 | 43.5 / 66.9 | 12.6 / 35.4 | 35.4 / 60.4 | 35.6 / 60 | 36.9 / 62 |
| sr_Latn_en | - / - | 37 / 62.3 | 39.4 / 63.9 | 39 / 63.8 | 4.1 / 22.8 | 26.7 / 53.1 | 31.1 / 56.9 | 30.7 / 56 |
| sr_en | - / - | 37.3 / 62 | 39.2 / 63 | 39.6 / 63.5 | 5.4 / 24 | 27.7 / 53.6 | 31.3 / 57 | 33 / 58.6 |
| ss_en | - / - | 23.9 / 46.4 | 28.4 / 50.6 | 29.6 / 51.7 | 0.4 / 10 | 6.3 / 22.8 | 8.5 / 31.5 | 6.7 / 31.4 |
| sv_en | - / - | 45.5 / 68.1 | 46.9 / 69 | 46.8 / 68.9 | 8.2 / 31.9 | 40.7 / 64.3 | 41.7 / 64.9 | 42 / 65 |
| sw_en | - / - | 41.8 / 62.7 | 44.1 / 64.5 | 44.2 / 64.5 | 3.8 / 17.9 | 31 / 54.3 | 32.2 / 55.1 | 33.3 / 56.6 |
| ta_en | - / - | 27.3 / 53.5 | 28.4 / 54 | 29.4 / 54.9 | 1.3 / 7.4 | 15.9 / 42 | 15.9 / 43.5 | 18.2 / 45.6 |
| te_en | - / - | 27.5 / 53.7 | 28.6 / 54.4 | 29.6 / 55.1 | 1 / 4.8 | 12.5 / 36.6 | 18 / 44.4 | 19.2 / 46.4 |
| tg_en | - / - | 9.6 / 32.2 | 19.9 / 43.8 | 16.6 / 39 | 2.3 / 17.1 | 16.7 / 42 | 20.4 / 46.8 | 20.6 / 47.5 |
| th_en | - / - | 37.2 / 60.3 | 39.7 / 61.9 | 39.9 / 62.4 | 3.9 / 12 | 20 / 44.3 | 27.8 / 54.5 | 30.2 / 56.3 |
| ti_en | - / - | 17.5 / 42.2 | 22.2 / 46.7 | 24.1 / 48.4 | 0.7 / 10.2 | 8 / 29.9 | 8 / 31.9 | 9.4 / 31.8 |
| tk_en | - / - | 28.5 / 54.8 | 31.6 / 57 | 31 / 56.5 | 2.1 / 18.3 | 4.4 / 26.3 | 15.7 / 40.3 | 16 / 44 |
| tn_en | - / - | 24.2 / 48 | 29 / 52.8 | 30.9 / 53.8 | 0.3 / 9.2 | 7.6 / 29.1 | 9.6 / 32 | 10.8 / 34.9 |
| to_en | - / - | 15.9 / 39.9 | 21.2 / 45 | 21.4 / 44.7 | 0.3 / 6.8 | 7.2 / 25.8 | 7.6 / 26.3 | 3.2 / 26.5 |
| tr_en | - / - | 32.5 / 58.7 | 34.3 / 59.9 | 34.6 / 60 | 4.3 / 23.6 | 14.7 / 36.6 | 21.3 / 48.3 | 22.4 / 49.4 |
| tt_en | - / - | 14.4 / 36 | 14.4 / 34.9 | 19.3 / 41.7 | 1.1 / 9.7 | 9.5 / 38 | 16.8 / 42.6 | 16.9 / 43.7 |
| ty_en | - / - | 5.5 / 28.9 | 5.5 / 28.6 | 6.3 / 29.5 | 0.5 / 10.3 | 6 / 27.7 | 5.3 / 27 | 3.5 / 26.3 |
| ug_en | - / - | 9.1 / 32.2 | 14.7 / 39.9 | 13.4 / 35.6 | 1.2 / 13.8 | 10.5 / 35.6 | 12.6 / 39.4 | 11.6 / 37.7 |
| uk_en | - / - | 34 / 59.4 | 35.2 / 60.1 | 35.1 / 60.1 | 6.6 / 26.8 | 26.7 / 54.1 | 27.4 / 53.6 | 28.1 / 54.1 |
| ur_en | - / - | 3.4 / 18.9 | 3.8 / 19 | 3.7 / 19.1 | 4.3 / 20 | 22.2 / 49.7 | 23.9 / 50.5 | 25.6 / 53 |
| uz_en | - / - | 24.5 / 52.2 | 27.2 / 54.5 | 26.9 / 54.4 | 2.8 / 17.8 | 16.4 / 44.4 | 16.5 / 42.7 | 16.9 / 42.8 |
| ve_en | - / - | 19.1 / 41.9 | 21.2 / 44.6 | 24.3 / 46.7 | 0.2 / 9.3 | 5.4 / 21.9 | 4.7 / 25 | 3.3 / 22.2 |
| vi_en | - / - | 0.4 / 16 | 0.3 / 15.9 | 0.3 / 15.9 | 5.5 / 24.9 | 13.8 / 34.5 | 27.1 / 52.8 | 30.6 / 55.4 |
| wo_en | - / - | 2.1 / 18.4 | 2.4 / 19.3 | 2.8 / 19.4 | 0.1 / 7.3 | 2.2 / 20.9 | 2.2 / 20.3 | 2.2 / 20.2 |
| xh_en | - / - | 29.9 / 52.5 | 34.2 / 56.1 | 34.1 / 56.2 | 1 / 14.8 | 13.7 / 33.4 | 16.2 / 37.8 | 14.5 / 39.6 |
| yo_en | - / - | 13.3 / 37 | 14.5 / 37.4 | 20.9 / 43.5 | 0.4 / 11.1 | 3.8 / 23.9 | 3.4 / 21.5 | 4.6 / 23 |
| yue_en | - / - | 26.7 / 54.4 | 29.1 / 56.2 | 29.3 / 56.2 | 0.3 / 1.8 | 16.3 / 44.1 | 18.1 / 46 | 17.1 / 45.9 |
| zh_Hant_en | - / - | 27.1 / 54 | 29.1 / 55.5 | 29.1 / 55.6 | 1.6 / 7.7 | 10 / 29.8 | 19.2 / 46.4 | 19.5 / 46.6 |
| zh_en | - / - | 24.3 / 53.4 | 26.8 / 54.9 | 26.3 / 54.6 | 0.4 / 3 | 17.8 / 45.4 | 18.8 / 46.4 | 19.1 / 46.6 |
| zu_en | - / - | 6.5 / 24.3 | 7.5 / 25 | 7.5 / 25.1 | 0.9 / 13.2 | 13.1 / 32.6 | 17 / 38.6 | 16.7 / 40.4 |
| | | | | | | | | |
| **xx2en** | - / - | 30.6 / 54.5 | 32.7 / 56.2 | **33.6 / 57.6** | 3.2 / 17.3 | 20.4 / 43.8 | 23.8 / 48.2 | 24.4 / 49.0 |
| **en2xx** | - / - | 16.5 / 39.6 | 17.6 / **41.9** | **17.9 / 41.9** | 0.8 / 7.3 | 11.7 / 31.2 | 12.6 / 32.4 | 12.3 / 32.3 |
| **Average** | 19.8 / - | 23.5 / 47.0 | 25.1 / 49.0 | **25.7 / 49.7** | 2.0 / 12.3 | 16.0 / 37.4 | 18.1 / 40.2 | 18.3 / 40.6 |

## A.11 Datasheet

### Datasheet - `MADLAD-400`

### Motivation

1. For what purpose was the dataset created? (Was there a specific task in mind? Was there a specific gap that needed to be filled? Please provide a description.) *We create `MADLAD-400` as a general purpose monolingual document level dataset covering 419 languages for the purpose of providing general training data for multilingual NLP tasks such as MT and language modeling. One of the goals of the Language Inclusivity Moonshot (LIM)* [12] *is to scale our language support to 1,000 languages and to support speech recognition for 97% of the global population. A core objective associated with this goal is to open source data and models. Our expectation is that releasng `MADLAD-400` will foster progress on the language research, especially on medium and low resource languages. An estimate of over 1B people globally speak languages that are not covered by mainstream models at Google or externally.*

2. Who created this dataset and on behalf of which entity? *Sneha Kudugunta[†], Isaac Caswell[◇], Biao Zhang[†], Xavier Garcia[†], Derrick Xin[†], Aditya Kusupati[◇], Romi Stella[†], Ankur Bapna[†], Orhan Firat[†] ([†]Google DeepMind, [◇]Google Research)*

3. Who funded the creation of the dataset? *Google Research and Google DeepMind*

4. Any other comments? *None*

### Composition

1. What do the instances that comprise the dataset represent? *Each instance is a preprocessed web-crawled document whose language that we annotated using a LangID model described by [12]. For the sentence level version, we used a sentence-splitter to split the documents into sentences and then deduplicated the resulting dataset.*

2. How many instances are there in total? *`MADLAD-400` has 4.0B documents (100B sentences, or 2.8T tokens) total across 419 languages with the median language containing 1.7k documents (73k sentences of 1.2M tokens.)*

3. Does the dataset contain all possible instances or is it a sample (not necessarily random) of instances from a larger set? (If the dataset is a sample, then what is the larger set? Is the sample representative of the larger set (e.g., geographic coverage)? If so, please describe how this representativeness was validated/verified. If it is not representative of the larger set, please describe why not (e.g., to cover a more diverse range of instances, because instances were withheld or unavailable).) *`MADLAD-400` are created from CommonCrawl documents that have been annotated by language, filtered and preprocessed. To maintain high precision, we filtered out data aggressively, and may not have captured every document of a given language in CommonCrawl. Moreover, there may also be languages in CommonCrawl that we may not have mined.*

4. What data does each instance consist of? *Each instance is raw text in either document form for the document level data, or in sentence form for the sentence level data.*

5. Is there a label or target associated with each instance? If so, please provide a description. *No.*

6. Is any information missing from individual instances? *No.*

7. Are relationships between individual instances made explicit? *No.*

8. Are there recommended data splits (e.g., training, development/validation, testing)? *No.*

9. Are there any errors, sources of noise, or redundancies in the dataset? *While we have taken extensive care to audit and filter `MADLAD-400`, there may still be documents annotated with the wrong language or documents of low quality.*

10. Is the dataset self-contained, or does it link to or otherwise rely on external resources (e.g., websites, tweets, other datasets)? (If it links to or relies on external resources, a) are there guarantees that they will exist, and remain constant, over time; b) are there official archival versions of the complete dataset (i.e., including the external resources as they existed at the time the dataset was created); c) are there any restrictions (e.g., licenses, fees) associated with any of the external resources that might apply to a future user? Please provide descriptions of all external resources and any restrictions associated with them, as well as links or other access points, as appropriate.) *Yes*

11. Does the dataset contain data that might be considered confidential (e.g., data that is protected by legal privilege or by doctor-patient confidentiality, data that includes the content of individuals' non-public communications)? (If so, please provide a description.) *Given that `MADLAD-400` is a general web-crawled dataset it is possible that documents in the dataset may contain such information.*

---

[12] https://blog.google/technology/ai/ways-ai-is-scaling-helpful/

12. Does the dataset contain data that, if viewed directly, might be offensive, insulting, threatening, or might otherwise cause anxiety? (If so, please describe why.) *Given that* `MADLAD-400` *is a general web-crawled dataset, even after filtering, it is possible that there are documents containing offensive content, etc.*

13. Does the dataset relate to people? *It is likely that some documents in* `MADLAD-400` *contain sentences referring to and describing people.*

14. Does the dataset identify any subpopulations (e.g., by age, gender)? *It is likely that some documents in* `MADLAD-400` *contain sentences referring to and describing people of certain subpopulations.*

15. Is it possible to identify individuals (i.e., one or more natural persons), either directly or indirectly (i.e., in combination with other data) from the dataset? *Yes, it is possible that their names are mentioned in certain documents.*

16. Does the dataset contain data that might be considered sensitive in any way (e.g., data that reveals racial or ethnic origins, sexual orientations, religious beliefs, political opinions or union memberships, or locations; financial or health data; biometric or genetic data; forms of government identification, such as social security numbers; criminal history)? *Given that* `MADLAD-400` *is a general web-crawled dataset, even after filtering, it is possible that there are documents containing sensitive data.*

17. Any other comments? *None.*

### Collection

1. How was the data associated with each instance acquired? *Each instance was acquired by performing transformations on the documents in all available snapshots of CommonCrawl as of August 20, 2022.*

2. What mechanisms or procedures were used to collect the data (e.g., hardware apparatus or sensor, manual human curation, software program, software API)? *We annotated the CommonCrawl data using a LangID model trained using the procedure described by [12]. Then, we manually inspected the data and then filtered or preprocessed the documents to create* `MADLAD-400`.

3. If the dataset is a sample from a larger set, what was the sampling strategy? `MADLAD-400` *is a subset of CommonCrawl documents determined using LangID annotations and filtering/preprocessing steps.*

4. Who was involved in the data collection process (e.g., students, crowdworkers, contractors) and how were they compensated (e.g., how much were crowdworkers paid)? *For the audit, the authors inspected the dataset. In some cases native speaker volunteers provided advice on the quality of the dataset.*

5. Over what timeframe was the data collected? (Does this timeframe match the creation timeframe of the data associated with the instances (e.g., recent crawl of old news articles)? If not, please describe the timeframe in which the data associated with the instances was created.) *We do not annotate timestamps. The version of CommonCrawl that we used has webcrawls ranging from 2008 to August 2022.*

6. Were any ethical review processes conducted (e.g., by an institutional review board)? *No.*

7. Does the dataset relate to people? *It is likely that some documents in* `MADLAD-400` *contain sentences referring to and describing people.*

8. Did you collect the data from the individuals in question directly, or obtain it via third parties or other sources (e.g., websites)? *We collected this data via webpages crawled by CommonCrawl.*

9. Were the individuals in question notified about the data collection? *No.*

10. Did the individuals in question consent to the collection and use of their data? *No.*

11. If consent was obtained, were the consenting individuals provided with a mechanism to revoke their consent in the future or for certain uses? *No.*

12. Has an analysis of the potential impact of the dataset and its use on data subjects (e.g., a data protection impact analysis) been conducted? *No.*

13. Any other comments? *None.*

### Preprocessing/cleaning/labeling

1. Was any preprocessing/cleaning/labeling of the data done (e.g., discretization or bucketing, tokenization, part-of-speech tagging, SIFT feature extraction, removal of instances, processing of missing values)? (If so, please provide a description. If not, you may skip the remainder of the questions in this section.) *Various types of preprocessing were done: deduplication of 3 sentence spans, filtering substrings according to various heuristics associated with low quality, Virama encoding correction, converting Zawgyi encoding to Unicode encoding for Myanmar script characters and a Chinese pornographic content filter heuristic. In addition, 79 annotated language datasets were removed on inspection due to low quality or mislabeling.*

2. Was the "raw" data saved in addition to the preprocessed/cleaned/labeled data (e.g., to support unanticipated future uses)? *No, this raw data is hosted by CommonCrawl.*

3. Is the software used to preprocess/clean/label the instances available? *As of June 13, 2023, no.*

4. Any other comments? *None.*

### Uses

1. Has the dataset been used for any tasks already? (If so, please provide a description.) `MADLAD-400` *has been used for MT and language modeling.*

2. Is there a repository that links to any or all papers or systems that use the dataset? *No*

3. What (other) tasks could the dataset be used for? *This dataset could be used as a general training dataset for any of the languages in* `MADLAD-400`.

4. Is there anything about the composition of the dataset or the way it was collected and prepro-cessed/cleaned/labeled that might impact future uses? (For example, is there anything that a future user might need to know to avoid uses that could result in unfair treatment of individuals or groups (e.g., stereotyping, quality of service issues) or other undesirable harms (e.g., financial harms, legal risks) If so, please provide a description. Is there anything a future user could do to mitigate these undesirable harms?) *While steps have been taken to clean* `MADLAD-400`, *content containing sensitive content about individuals or groups could affect the performance of some downstream NLP tasks. Moreover, while building applications for (a) given language(s), we urge practitioners to assess the suitability of* `MADLAD-400` *for their usecase.*

5. Are there tasks for which the dataset should not be used? (If so, please provide a description.) *N/A.*

6. Any other comments? *None.*

### Distribution

1. How will the dataset will be distributed (e.g., tarball on website, API, GitHub)? `MADLAD-400` *is made available through a GCP bucket.*

2. When will the dataset be distributed? *June 2023*

3. Will the dataset be distributed under a copyright or other intellectual property (IP) license, and/or under applicable terms of use (ToU)? (If so, please describe this license and/or ToU, and provide a link or other access point to, or otherwise reproduce, any relevant licensing terms or ToU, as well as any fees associated with these restrictions.) *AI2 has made a version of this data available under the ODC-BY license. Users are also bound by the CommonCrawl terms of use in respect of the content contained in the dataset.*

4. Have any third parties imposed IP-based or other restrictions on the data associated with the instances? *Users are bound by the CommonCrawl terms of use in respect of the content contained in the dataset.*

5. Any other comments? *None.*

### Maintenance

1. Who is supporting/hosting/maintaining the dataset? *An external organization, AI2 is hosting the dataset.*

2. How can the owner/curator/manager of the dataset be contacted (e.g., email address)? *Sneha Kudugunta* `snehakudugunta@google.com` *or Isaac Caswell (*`icaswell@google.com`*) for questions about the dataset contents, or Dirk Groeneveld* `dirkg@allenai.org` *for questions related to the hosting of the dataset.*

3. Is there an erratum? (If so, please provide a link or other access point.) *No*

4. Will the dataset be updated (e.g., to correct labeling errors, add new instances, delete instances')? (If so, please describe how often, by whom, and how updates will be communicated to users (e.g., mailing list, GitHub)?) *There are no such plans, but major issues may be corrected when reported through email or the Github page.*

5. If the dataset relates to people, are there applicable limits on the retention of the data associated with the instances (e.g., were individuals in question told that their data would be retained for a fixed period of time and then deleted)? (If so, please describe these limits and explain how they will be enforced.) *N/A*

6. Will older versions of the dataset continue to be supported/hosted/maintained? (If so, please describe how. If not, please describe how its obsolescence will be communicated to users.) *No*

7. If others want to extend/augment/build on/contribute to the dataset, is there a mechanism for them to do so? (If so, please provide a description. Will these contributions be validated/verified? If so, please describe how. If not, why not? Is there a process for communicating/distributing these contributions to other users? If so, please provide a description.) *A relatively unprocessed version of* `MADLAD-400`, `MADLAD-400`-`noisy` *is made available for others to build upon using superior cleaning/preprocessing techniques for their specific usecases.*

8. Any other comments? *None*

## A.12 Model Card

**Model Card**

**Model Details**

- Person or organization developing model: *Google DeepMind and Google Research*
- Model Date: *June 13, 2023*
- Model Types: *Machine Translation and Language Modeling Models.*
- Information about training algorithms, parameters, fairness constraints or other applied approaches, and features: *Provided in the paper.*
- Paper: *Kudugunta et al,* `MADLAD-400`*: Monolingual And Document-Level Large Audited Dataset, Under Review, 2023*
- License: *ODC-BY*
- Contact: *snehakudugunta@google.com*

**Intended Use**

- Primary intended uses: *Machine Translation and multilingual NLP tasks on over 400 languages.*
- Primary intended users: *Research community.*
- Out-of-scope use cases: *These models are trained on general domain data and are therefore not meant to work on domain-specific models out-of-the box. Moreover, these research models have not been assessed for production usecases.*

**Factors**

- *The translation quality of this model varies based on language, as seen in the paper, and likely varies on domain, though we have not assessed this.*

**Metrics**

- *We use SacreBLEU and chrF, two widely used machine translation evaluation metrics for our evaluations.*

**Ethical Considerations**

- *We trained these models with* `MADLAD-400` *and publicly available data to create baseline models that support NLP for over 400 languages, with a focus on languages underrepresented in large-scale corpora. Given that these models were trained with web-crawled datasets that may contain sensitive, offensive or otherwise low-quality content despite extensive preprocessing, it is still possible that these issues to the underlying training data may cause differences in model performance and toxic (or otherwise problematic) output for certain domains. Moreover, large models are dual use technologies that have specific risks associated with their use and development. We point the reader to surveys such as those written by Weidinger et al. [62] or Bommasani et al. [10] for a more detailed discussion of these risks, and to Liebling et al. [41] for a thorough discussion of the risks of machine translation systems.*

**Training Data**

- *For both the machine translation and language model,* `MADLAD-400` *is used. For the machine translation model, a combination of parallel datasources covering 157 languages is also used. Further details are described in the paper.*

**Evaluation Data**

- *For evaluation, we used WMT, NTREX, Flores-200 and Gatones datasets as described in Section 4.3.*

**Caveats and Recommendations**

- *We note that we evaluate on only 204 of the languages supported by these models and on machine translation and few-shot machine translation tasks. Users must consider use of this model carefully for their own usecase.*