# OpenReview forum: "MADLAD-400: A Multilingual And Document-Level Large Audited Dataset"
_NeurIPS.cc/2023/Track/Datasets_and_Benchmarks — NeurIPS 2023 Datasets and Benchmarks Poster_

### Official Review · Reviewer_MbPM · 2023-07-13
**Great data efforts to multilingual language model;**

**Rating:** 7
**Confidence:** 4
**Correctness:** Yes

**Strengths:**

1. The paper introduces a multilingual language modeling dataset spanning 419 languages. The paper also provides detailed descriptions of the data processing pipeline, including quality filters, language identification, which can be helpful to further dataset development.
2. The dataset is manually audited on the language level, to apply additional filters, merge language codes or completely remove the languages whose datasets are very noisy. Optional notes are added for some languages, e.g. if dataset is 100% Bible. These efforts improve the average quality of the dataset and also provide more information for downstream users interested in some languages.

**Additional Feedback:**

None

**Clarity:**

It is not clear why the dataset is called "monolingual" while it spans 419 languages. Maybe the authors use "monolingual" to show it is not parallel data. However, the usage contradicts the meanings of monolingual and multilingual in most cases (such as multilingual language model).


**Documentation:**

Yes

**Opportunities For Improvement:**

1. The trained language model is only evaluated on machine translation, while more benchmarks can be evaluated, such as XTREME (https://sites.research.google/xtreme)

**Relation To Prior Work:**

Yes

**Summary And Contributions:**

The paper introduces a general domain multilingual language modeling dataset based on crawled web pages. The dataset is manually audited to discard 79 languages from the preliminary dataset. To validate the efficacy of the dataset, a 10.7B-parameter multilingual machine translation model is trained and shown to be competitive with models that are significantly larger.

---

> ### Author Response · Authors · 2023-08-12
> **Review Response**
>
> We thank Reviewer MbPM for the positive review. We address the points raised below:
>
> >...The trained language model is only evaluated on machine translation, while more benchmarks can be evaluated, such as XTREME...
>
> For many of the ~200 new longtail languages introduced in MADLAD-400, the only high-quality benchmarks available are for MT. XTREME covers only 44 languages, a majority of which are not among the newly introduced languages, or tail languages at all. Therefore, we have chosen to focus on few-shot MT evaluation for the purpose of evaluating a baseline LM trained on MADLAD-400 that may be further improved upon by the community. While we have covered only 204 of the 419 languages in MADLAD-400 for evaluation, we believe that this is a reasonable evaluation of the capabilities of a baseline LM trained on MADLAD-400.
>
> There are also some tasks like WikiANN that cover 100s of languages - however, for multilingual LMs that are not instruction-tuned, few-shot classification results have been shown to be poor for models at a similar size to 8B ([Asai et al](https://arxiv.org/pdf/2305.14857.pdf)), and therefore not a useful indicator of model quality. We did some initial experiments on this dataset with our trained baseline model, and obtained similar results.  However, instruction tuned models show significant improvement upon pre-trained LMs. Instruction-tuning models for tail languages is an open question that is out of the scope of this paper, and we hope that the research community will build upon the MT models (for instruction generation) and LM model that we have released. In addition, in the new version of the draft, we have further clarified these limitations in Section 6.
>
> > It is not clear why the dataset is called "monolingual"...
>
> While we do use the word monolingual to refer to how the data is non parallel we realize that this may be confusing to those unfamiliar with MT literature. We will switch the M in MADLAD to be Multilingual to minimize this confusion.

---

### Official Review · Reviewer_5Zvf · 2023-07-19
**An interesting and clearly described dataset paper despite having marginal modeling improvement over prior works**

**Rating:** 8
**Confidence:** 5

**Strengths:**

- The explanation of the dataset construction with all the filtering processes is explained in a very detailed manner
- The datasets (MADLAD-400 and the multiway parallel corpus) and the models (MT models and the 8B decoder-only model) developed in this work will be very useful resources for the research community


**Additional Feedback:**

- Line 167 refers to section A.5, but section A.5 does not provide the detailed information
- Line 183 should refer to A.A instead of A.5
- Line 211 and 218 should refer to A.8 instead of A.7

**Clarity:**

The paper is well written with minor problems when several reference to the sections in the appendix

**Correctness:**

The dataset constructed is constructed in a sound way. Nevertheless, the evaluation especially for the decoder-only model is inadequate.

**Documentation:**

The licensing is clearly stated in the datasheet and the model card. Furthermore, the paper provides enough detail for the reproducibility of the benchmark.

**Ethics:**

There is no ethical concern

**Limitations:**

The paper specifies the limitation regarding the dataset filtering quality and the limited evaluation for both the MT and the decoder-only models accurately. Additionally, the paper further specify the skewness of the data on a specific domain such as religion.


**Opportunities For Improvement:**

- Despite the effort on curating the dataset, the advantage of using the clean compared to noisy data is never shown. Hence it is unclear whether using curated data brings any fruitful benefit compared to using larger noisy data.
- The 8B decoder-only LM is not well evaluated, for instance, the zero-shot and few-shot evaluation is only conducted on MT task, while the evaluation on common monolingual and multilingual benchmark such as MMLU, WinoGrande, RACE, XCOPA, XWINOGRAD, TYDIQA, etc, is left unexplored. This raises the question regarding the importance of having the 8B decoder model within this work.

**Relation To Prior Work:**

- The paper incorporates various knowledge from multiple prior works during the construction of the dataset
- The paper introduces a new multilingual dataset with in-depth detail on the dataset curation covering, which covers more languages compared to most of the prior works.

**Summary And Contributions:**

The paper introduces MADLAD-400, a multilingual dataset consisting of 419 languages which collected from CommonCrawl with thorough filtering & auditing processes. The paper meticulously provide detailed description on the dataset construction pipeline showing various obstacles and how to solve them in developing an extensively audited multilingual dataset. The filtering process is done through multiple automatic filters including data deduplication, low-quality content filtering, pornographic content filteing, language filtering, script-specific filtering, and toxicity filtering. Additionally, the subset of language data is also further manually audited to ensure the quality of the dataset and provide additional manually curated filtering to further reduce the noise. Two partitions of resulting data is released, i.e., noisy parititon (5T tokens / 150B sentences / 7.8B documents) and clean partition (2.8T tokens / 100B sentences / 4.0B documents). In addition to MADLAD-400, the paper also instroduced a large parallel data covering 156 languages, with 4.1B sentence pairs and 4124 language pairs which is also filtered and manually audited similar to the MADLAD-400 dataset. The paper further create a multiway parallel data by using n-gram matching method resulting in 20742 language pairs with a total of 11.9B sentences. The resulting datasets are used to build various sizes of machine translation (MT) models and an 8B decoder-only LM. The MT models are trained using MT and MASS pre-training objective, while the decoder-only LM is trained using the UL2 pre-training objective. The models are evaluated in various MT benchmarks, i.e., WMT, FLORES-200, NTREX, and GATONES. The resulting MT model with 10.7B parameters shows competitive performance compared to the state-of-the-art MT models on each benchmark. While the 8B decoder-only LM shows an acceptable few-shot performance over all benchmarks.

---

> ### Author Response · Authors · 2023-08-12
> **Review Response**
>
> We thank Reviewer 5Zvf for the positive review, and for pointing out the issues with references to the Appendix. We have corrected these issues in the new version, and note that in this version, we provided all required information in Section 3.1. We address the other points raised below:
>
> >...the advantage of using the clean compared to noisy data is never shown…
>
> Given the expense of ablating the quality of even models of the scale of ~2B (for which we used 256 v3 TPUs), we have followed best practices recommended in literature training large scale NLP models that have been experimented with.
>
> For example, [Lee et al](https://arxiv.org/pdf/2107.06499.pdf) show the importance of deduplicating data, while the [NLLB paper](https://arxiv.org/pdf/2207.04672.pdf) discusses the importance of filtering source-target pairs with unmatched toxic term frequencies. Similarly, [Banon et al](https://aclanthology.org/2020.acl-main.417.pdf) discuss the importance of avoiding source-target copies for MT. Some of the heuristics such as filtering out documents with the word “lorem ipsum” are used by widely used datasets such as [C4](https://arxiv.org/pdf/1910.10683.pdf).
>
> The remainder of the heuristics (script filters, document consistency, list case, length heuristics, technical characters, regex filters, chinese porn blocklist, virama correction, zawygi encoding) are primarily to filter out/correct documents that are not linguistic content and prevent “representation washing” - ie, ensure that the documents labeled as, say Yoruba, is actually plausibly Yoruba content.  [Kreutzer et al](https://arxiv.org/pdf/2103.12028.pdf) discuss the harms of low-quality data in the context of multilinguality, and show that for languages in both the audit and the FloReS benchmark, there is a significant positive correlation between data quality scores and spBLEU (ρ = 0.44, 0.041).
>
> >The 8B decoder-only LM... common monolingual and multilingual benchmark such as MMLU, WinoGrande, RACE, XCOPA, XWINOGRAD, TYDIQA, etc, is left unexplored…
>
> For many of the ~200 new longtail languages introduced in MADLAD-400, the only high-quality benchmarks available are for MT. Most of the benchmarks mentioned above cover ~10 languages, and aggregated benchmarks such as XTREME and BUFFET cover 44 and 55 languages respectively. Moreover a majority of these languages are not among the newly introduced languages, or tail languages at all. Therefore, we have chosen to focus on only few-shot MT evaluation for the purpose of evaluating a baseline LM trained on MADLAD-400 that may be further improved upon by the community. While we have covered only 204 of the 419 languages in MADLAD-400 for evaluation, we believe that this is a reasonable evaluation of the capabilities of a baseline LM trained on MADLAD-400.
>
> There are also some tasks like WikiANN that cover 100s of languages - however, for multilingual LMs that are not instruction-tuned, few-shot classification results have been shown to be poor for models at a similar size to 8B ([Asai et al](https://arxiv.org/pdf/2305.14857.pdf)), and therefore not a useful indicator of model quality. We did some initial experiments on this dataset with our trained baseline model, and obtained similar results.  However, instruction tuned models show significant improvement upon pre-trained LMs. Instruction-tuning models for tail languages is an open question that is out of the scope of this paper, and we hope that the research community will build upon the MT models (for instruction generation) and LM model that we have released. In addition, in the new version of the draft, we have further clarified these limitations in Section 6.

---

### Official Review · Reviewer_4BCh · 2023-07-27
**Useful to the broader community**

**Rating:** 10
**Confidence:** 5
**Correctness:** Yes
**Clarity:** Yes

**Strengths:**

More than doubling the number of languages in datasets that can be used for multilingual models.

**Additional Feedback:**

N/A

**Documentation:**

Yes

**Limitations:**

Yes

**Opportunities For Improvement:**

Better assessment of the languages. 20 documents is quite small.

Explaining why the "garbage collection" problem is not a problem for this LangID system.

**Relation To Prior Work:**

Yes

**Summary And Contributions:**

This is a very interesting resource that will be a huge boost to the overall community. In general, most multilingual models have approximately 100 languages. NLLB out of Meta has 200. There are a few corpora that have more languages, but are all based on the bible. This dataset has over 400 languages – in essence doubling the scope of data for multilingual models. In addition, the authors train some very large models on the data that are incredibly strong baselines.

The biggest downside of the paper is that the 20 documents used to evaluate if a language is noisy is very small. Furthermore, this is done by people who do not speak the language. I also worry that the LangID system will not be accurate. It is a common problem for low-resource languages to serve as a “garbage collector” and assign probability mass to noisy data. With an initial LangID of almost 500 categories, this could be even more of an issue than is normally observed. This is not addressed in the paper.

However, that being said, increasing the coverage of languages to 419 is an impressive feat in multilinguality and will be very useful to the broader research community.

---

> ### Author Response · Authors · 2023-08-12
> **Review Response**
>
> We thank Reviewer 4BCh for their positive feedback. We address the main point raised below:
>
> >I also worry that the LangID system will not be accurate.
>
> On inspecting MADLAD-400-noisy, we indeed found that the LangID model used many language categories as a “garbage collector”. This is why we applied an extensive set of filters to the initial data (Section 2.3), and did several rounds of this filtering (Section 2.4). We agree that the assessment of 20 docs per language by non-native speakers is not enough to give a detailed picture of the quality of the data, and that there is certainly some variation in the overall data quality across languages - for instance, we definitely cannot distinguish Udmurt and Komi. However, as reported by [Kreutzer et al](https://arxiv.org/pdf/2103.12028.pdf) in Quality at a Glance, manual review of 20 docs per language is enough to catch many sorts of overt noise, and ensure that all the text is at least linguistic content, and plausibly in language. Moreover, when unsure, we found native speakers who volunteered to differentiate between closely related languages and diagnose suspected issues (Eg., different languages that use the Arabic writing system).
>
> While this level of scrutiny is beyond what is applied to the majority of web-mined corpora, we acknowledge that our self-audit has its own limitations. To this end, we released the audit notes in Appendix A4, and released the samples used for the audit (in the dataset url field). We also encourage users of this dataset to audit the data for their specific use-case and report any issues that we will post on the dataset Github page.

---

### Author Response · Authors · 2023-08-12
**Overall Response**

Thank you to all the reviewers for the positive feedback. We appreciate all the constructive comments and suggestions raised, and have expanded upon the Limitations Section (Section 6) of the revised draft:

**LangID Accuracy**: We agree that a LangID system for tail languages is likely to be inaccurate, so we decided to do 2 rounds of self-auditing (partial and full), consulted native speakers when possible and did several rounds of filtering. As a result of this, we dropped 79 of 419 languages and reduced the corpus size from 5T to 2.8T. It is possible that issues still remain, so we release the samples used for the audit and encourage users to report issues with an erratum.

**Evaluation of LM on only few-shot MT**: For many of the ~200 new longtail languages introduced in MADLAD-400, the only high-quality benchmarks available are for MT. Aggregated benchmarks such as XTREME and BUFFET cover only 40-50 languages of which most are not tail languages, that are the focus of MADLAD-400. We hope that the baseline models we release spur work on instruction tuning for tail languages and creating higher language coverage benchmarks.

**A lack of ablations on data filtering**: Given the expense of ablating the quality of even models of the scale of ~2B, we have drawn upon the wealth of literature that discusses and experimentally demonstrates on popular language dataset preprocessing heuristics such as [duplication](https://arxiv.org/pdf/2107.06499.pdf), [toxicity filtering](https://arxiv.org/pdf/2207.04672.pdf), and ensuring that datasets of a given languages are both high quality and [in the correct language](https://arxiv.org/pdf/2103.12028.pdf).

Finally, note that the dataset url field now has a Google Drive link to the samples used for the self-audit and a GCP bucket link containing the full dataset.

---

### Decision · Program_Chairs · 2023-09-22

**Decision:**

Accept (Poster)

**Comment:**

This paper presents MADLAD-400 which consists of 419 languages.  It is collected from CommonCrawl via filtering and auditing processes, and it is doubled the size of NLLB.

Although there are some concerns such as LangID accuracy and only evaluation on few-shot MT, reviewers are all positive to the submission. I also think such large multilingual dataset is a very useful resource to the broader research community.